# Channel-Imposed Fusion: A Simple yet Effective Method for Medical Time Series Classification

## Abstract

**A simple model with good features beats a complex model with poor features.** Medical time series (MedTS), such as EEG and ECG, are critical for clinical diagnosis but face two main challenges: generic model architectures fail to fully leverage physiological priors, and the inherently low signal-to-noise ratio (SNR) limits feature representation. To address these issues, we propose a simple data-centric framework that effectively combines physiological priors with deep learning. First, **Channel-Imposed Fusion (CIF)** is inspired by physiological priors and is based on the idea that, ideally, information from other channels can enhance the signal-to-noise ratio (SNR) of the current channel. By linearly fusing signals across channels, CIF effectively enhances feature representations. To improve efficiency on multi-channel data, we adopt global fusion coefficients and reorder the channels according to functional regions, achieving physiologically meaningful fusion with minimal parameters. Secondly, we build a simplified HM-BiTCN on top of TCNs, aiming to capture both forward and backward temporal dependencies with the simplest possible design. Experimental results demonstrate that the combination of CIF and HM-BiTCN achieves state-of-the-art performance across multiple MedTS benchmarks, showing that competitive results can be obtained without complex model design. **We hope this work encourages the community to reconsider the core of medical time series classification: should it be driven primarily by data-centric strategies, model-centric design, or a combination of both?** Code is available at : https://anonymous.4open.science/r/CIF-8F76.

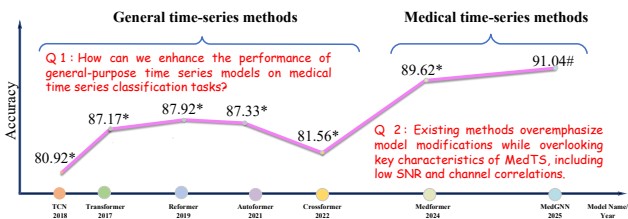

*Figure 1.* The results of various methods on the TDBrain dataset (EEG) are presented, where ∗ indicates results reported by *Medformer (Wang et al., 2024a)*, and # indicates results reported by *MedGNN (Fan et al., 2025)*. In addition, we highlight two main motivations of this work (Q1 and Q2).

## 1. Introduction

Medical time series (MedTS) data, such as electroencephalogram (EEG) and electrocardiogram (ECG) signals, are widely used in clinical settings to monitor patient health and play a crucial role in diagnosing neurological and cardiovascular diseases (Arif et al., 2024; Xiao et al., 2023; Zhu et al., 2025; Wang et al., 2024b; 2025b). Accurate classification of these signals enables early anomaly detection, personalized treatment, and optimized therapy planning, ultimately improving patient outcomes and healthcare efficiency (Liu et al., 2024a; Tian et al., 2023). With advances in deep learning, CNN-based models like EEGNet (Lawhern et al., 2018) can automatically extract informative features from raw signals, significantly improving performance.

In recent years, Transformer models (Vaswani et al., 2017), originally inspired by the self-attention mechanism (Bahdanau et al., 2014), have achieved remarkable progress in time series modeling, particularly in capturing long-range dependencies and global contextual information (Liu et al., 2021; Zhou et al., 2021). By mapping sequential data into high-dimensional token embeddings, Transformers are able to implicitly model complex temporal dependencies. Despite their success across a wide range of time series tasks, applying Transformer architectures to MedTS classification still faces several challenges, which can be summarized as follows: **(1) Misalignment between domain-specific knowledge and generic architectures.** Mainstream time series models, such as Autoformer (Wu et al., 2021), Cross-

---

[1]Anonymous Institution, Anonymous City, Anonymous Region, Anonymous Country. Correspondence to: Anonymous Author <anon.email@domain.com>.

Preliminary work. Under review by the International Conference on Machine Learning (ICML). Do not distribute.

former (Zhang & Yan, 2022), and Reformer (Kitaev et al., 2019), have demonstrated strong performance in general domains such as weather forecasting and finance. However, as illustrated in Fig. 1, these approaches fail to achieve comparable effectiveness in MedTS classification tasks. This raises the urgent question of how to enhance the applicability of general-purpose models in medical scenarios. Moreover, MedTS often encode critical physiological characteristics—for example, conduction delays across ECG leads (Auricchio et al., 2014) and rhythmic synchrony in EEG signals (Palva & Palva, 2014; Fries, 2015)—which inherently reflect *channel-level relationships*. Unfortunately, such physiological dependencies are rarely considered in generic time series modeling frameworks. **(2) Overemphasis on model optimization while neglecting the inherent low SNR of MedTS.** Unlike general-purpose time series tasks, MedTS are inherently characterized by low signal-to-noise ratio (SNR) conditions (Del Rio et al., 2011; Sraitih et al., 2022; Sharma, 2017; Mohd Apandi et al., 2020; Jia et al., 2024), where noise and artifacts can easily overshadow critical physiological features. In such environments, complex Transformer architectures do not always perform reliably in extracting effective representations, while simpler models (e.g., TCNs (Bai et al., 2018)) may suffer even greater performance degradation. Recent Transformer-based methods designed for MedTS, such as MedGNN (Fan et al., 2025) and Medformer (Wang et al., 2024a), primarily focus on architectural innovations but fall short of addressing the core issue of low SNR. This brings up a critical question: Should advances in MedTS classification come from increasingly complex model architectures, or from more principled strategies focused on data processing and representation?

To address the aforementioned limitations, we depart from the traditional *model-centric* paradigm that relies on increasingly complex architectures to capture temporal dependencies, and instead propose a **data-centric** approach grounded in the physiological properties of medical time series. Following this principle, we introduce the *Channel-Imposed Fusion (CIF)* method, which encodes prior causal structures into feature representations. Specifically, CIF constructs new features through a linear combination of signals from different channels:

$$x_{\text{new}} = ax + by, \tag{1}$$

where $x$ and $y$ denote signals from two distinct channels, and $a$ and $b$ are coefficients predefined based on domain knowledge. When $a$ and $b$ take fixed values, they are not learned directly from patient data, but instead derived from two domain-specific prior hypotheses: **(1) Physiological Coupling Hypothesis.** For ECG signals, when two leads are highly correlated (e.g., P-wave polarity and morphology are consistent (Platonov, 2012)), setting $a = b = 1$ achieves in-phase summation, thereby enhancing target signal components and improving the SNR. **(2) Noise Suppression**

**Hypothesis.** In EEG recordings, ocular artifacts such as blinks often appear highly correlated in frontal electrodes Fp1 and Fp2 (Croft & Barry, 2000). To suppress such noise, we set $a = 1, b = -1$, applying a differential fusion strategy to cancel common-mode interference. Here, the coefficients $a$ and $b$ serve as symbolic encodings of interpretable physiological principles, rather than exact data-driven estimates. When treated as learnable parameters, they can be fine-tuned under symbolic constraints imposed by prior knowledge (e.g., enforcing $a > 0, b > 0$ under coupling, and $a > 0, b < 0$ under noise suppression). This design maintains the interpretability of directional relationships (e.g., signal enhancement or cancellation) while allowing the model to adaptively adjust the magnitude of each coefficient based on the training data.

**To emphasize the importance of data-centric approaches**, we deliberately designed a simple yet effective model—the Hidden-layer Mixed Bidirectional Temporal Convolutional Network (HM-BiTCN)—to demonstrate that excellent performance does not necessarily require model complexity. The combination of CIF and HM-BiTCN not only outperforms Transformer-based methods on multiple medical datasets but also achieves new state-of-the-art (SOTA) results on general time series classification benchmarks. More importantly, the CIF method is not limited to the HM-BiTCN architecture itself; it exhibits strong transferability and can be seamlessly integrated into existing Transformer architectures, enhancing their adaptability to MedTS data. Our main contributions are:

- **Proposal of Channel-Imposed Fusion (CIF).** We introduce CIF to model inter-channel relationships in medical time series, particularly suitable for signals with well-defined physiological structures such as EEG and ECG.

- **Design of HM-BiTCN based on CIF.** By integrating CIF into HM-BiTCN, our method consistently outperforms existing SOTA models across multiple publicly available medical and non-medical time series classification datasets.

- **Methodological transferability.** CIF is architecture-agnostic and can be seamlessly integrated into mainstream models such as Transformers, compensating for the limitations of traditional positional encodings in modeling channel-level correlations, and highlighting the paradigm shift from a *model-centric* to a *data-centric* perspective.

## 2. Related Work

**Medical Time Series Classification.** Medical time series analysis diverges fundamentally from general time series

forecasting (Wu et al., 2022a; Lu et al., 2024) by prioritizing pathological signature decoding over temporal extrapolation, with modalities like EEG (Tang et al., 2021; Yang et al., 2023; Qu et al., 2020), ECG (Xiao et al., 2023; Wang et al., 2023; Kiyasseh et al., 2021), and EMG [(Xiong et al., 2021; Dai et al., 2022)] encoding distinct clinical semantics. Early methods were dominated by compact CNNs such as EEGNet (Lawhern et al., 2018), which employs depthwise separable convolutions to efficiently extract spatio–temporal features while providing preliminary interpretability via feature-map visualization. Subsequently, temporal convolutional networks (TCNs) (Bai et al., 2018; Lin et al., 2019) leveraging dilated causal convolutions achieved parallelizable computation and extended receptive fields, surpassing LSTM-based approaches (Zhou et al., 2016; Shen & Lee, 2016; Hochreiter & Schmidhuber, 1997) on multiple medical signal classification benchmarks. Hybrid architectures such as EEG-Conformer (Song et al., 2022) combined convolutional front-ends with Transformer self-attention to capture both local and global dependencies and enabled attention-based interpretability. More recently, fine-grained Transformer models such as Medformer (Wang et al., 2024a) introduced cross-channel tokenization and dual-stage self-attention, setting new SOTA accuracy on several public datasets. The latest MedGNN (Fan et al., 2025) further augments attention mechanisms with multi-resolution graph learning to jointly model spatial multi-scale channel dependencies and temporal dynamics.

**Model-centric Transformer-based time series methods.** In time series analysis, Transformer-based models learn complex dependencies through diversified architectural designs: the vanilla Transformer (Vaswani et al., 2017) first introduced multi-head self-attention and sinusoidal positional encoding to model temporal correlations globally; Informer (Zhou et al., 2021) employs ProbSparse attention to select key time steps and compress sequence length, thereby reducing the computational cost of long-range dependencies; Reformer (Kitaev et al., 2019) incorporates Locality-Sensitive Hashing (LSH) to reduce attention complexity to $\mathcal{O}(L \log L)$, making it suitable for ultra-long sequences; Autoformer (Wu et al., 2021) proposes an Auto-Correlation mechanism that aggregates periodic subsequences to enhance the implicit capture of cyclic patterns; FEDformer (Zhou et al., 2022) performs seasonal–trend decomposition in the frequency domain and uses compressed Fourier coefficients to enable cross-frequency attention interactions; Crossformer (Zhang & Yan, 2022) designs a two-stage attention mechanism across time and feature dimensions to implicitly fuse multivariate spatiotemporal couplings; iTransformer (Liu et al., 2024b) innovatively treats time steps as channel dimensions and applies standard attention to implicitly learn nonlinear inter-variable relationships; PatchTST (Nie et al., 2023) segments continuous

time steps into patch-based tokens and uses a combination of local and global attention to capture multi-scale temporal patterns; Medformer (Wang et al., 2024a) introduces multi-granularity patch embeddings and cross-channel attention for medical signals, implicitly modeling the heterogeneous couplings of physiological metrics; and MedGNN (Fan et al., 2025) combines graph attention with frequency-differential networks to incorporate medical topological priors into implicit spatiotemporal dependency learning.

## 3. Method

### 3.1. Channel-Imposed Fusion

SNR Improvement via Linear Channel Fusion. Consider two observed signals $x_1 = s_1 + \epsilon_1$ and $x_2 = s_2 + \epsilon_2$, with zero-mean, mutually uncorrelated signal and noise components. The CIF module performs a linear combination $y = ax_1 + bx_2 = as_1 + bs_2 + a\epsilon_1 + b\epsilon_2$, whose output SNR can be expressed as $\text{SNR}_{\text{out}} = \text{Var}(as_1 + bs_2)/\text{Var}(a\epsilon_1 + b\epsilon_2)$. In the simple case where the signals and noises have equal variances $\sigma_s^2$ and $\sigma_\epsilon^2$, and correlations $\rho$ and $\gamma$, this reduces to $\text{SNR}_{\text{out}} = \text{SNR}_{\text{in}} \cdot (a^2 + b^2 + 2ab\rho)/(a^2 + b^2 + 2ab\gamma)$. Maximizing this ratio yields two canonical modes: *cooperative mode* ($\rho > \gamma$) with $a = b$, which amplifies correlated signals, and *differential mode* ($\rho < \gamma$) with $a = -b$, which suppresses correlated noise. Details in Appendix A. **Given the high complexity of medical time series, we emphasize that the design of CIF is merely inspired by the observation that, under ideal conditions, a linear combination of two signals can enhance the SNR. We do not claim that CIF can reliably improve SNR in real-world scenarios.**

As shown in Eq. 1, a linear combination of two channels can incorporate physiological priors to construct more meaningful feature representations. For multi-channel data with $N$ channels, we select two subsets $X$ and $Y$, each containing $n \le N$ channels. Each corresponding channel pair $X_i$ and $Y_i$ is linearly fused to produce

$$X_i^{\text{new}} = a_i X_i + b_i Y_i, \quad i = 1, 2, \ldots, n, \qquad (2)$$

where $a_i$ and $b_i$ are the linear combination coefficients for the $i$-th pair of channels. Consequently, this multi-channel fusion scheme requires $n$ coefficients $a_i$ and $n$ coefficients $b_i$ in total. By leveraging physiological priors through such linear combinations across multiple channels, the model can enhance its capability to capture physiological patterns .

However, despite the fact that designing independent coefficients $(a_i, b_i)$ for each channel pair can maximize physiological interpretability, this approach still faces several practical challenges in real-world applications. The total number of parameters scales linearly with the number of channels, reaching $2n$, which can make parameter manage-

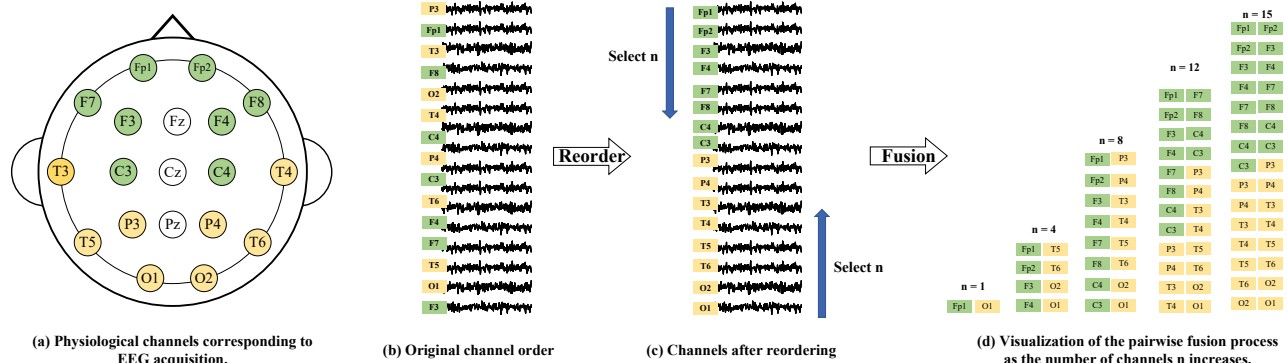

(a) Physiological channels corresponding to EEG acquisition.

(b) Original channel order

(c) Channels after reordering

(d) Visualization of the pairwise fusion process as the number of channels n increases.

*Figure 2.* Reorders the channels , placing electrodes from the same functional region adjacently along the input dimension to create an "functional region fusion" input layout. Panel (d) visualizes the pairwise channel fusion process as the number of channels $n$ varies. **It is important to emphasize that, due to the specialized nature of medical equipment, the channel order is already arranged according to regions during the data acquisition process, and the output data inherently reflects the structure shown in Figure (c).**

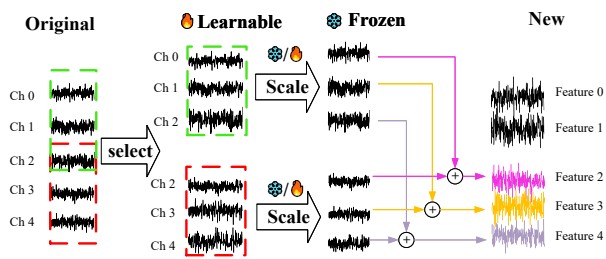

*Figure 3.* The implementation process of the CIF method.

ment and optimization cumbersome as $n$ increases. Moreover, if the coefficients are not learnable, each pair must be manually adjusted, substantially increasing the workload and introducing potential human bias. In addition, when the coefficients are uncertain or need to be searched over a range of candidate values, the number of possible configurations grows exponentially (i.e., $K^{2n}$ for $K$ candidate values per coefficient), rendering exhaustive optimization practically infeasible.

To alleviate these issues, we adopt a simplified strategy in which a single coefficient $a$ is applied uniformly to the first $n$ channels and a single coefficient $b$ is applied to the remaining $n$ channels, reducing the total number of parameters from $2n$ to 2. This substantially decreases the manual adjustment and computational cost while retaining the core advantages of linear fusion. The simplified fusion can be expressed as

$$X_i^{\text{new}} = aX_i + bY_i, \quad i = 1, \ldots, n. \tag{3}$$

Although this strategy significantly reduces the parameter space and improves practical usability, it comes with a trade-off: replacing pairwise coefficients $(a_i, b_i)$ with global coefficients $(a, b)$ reduces micro-level interpretability and may

limit the ability to selectively enhance high-signal channels. Overall, this design provides a practical compromise between physiological fidelity, parameter efficiency, and computational feasibility. For applications that require more fine-grained modeling, intermediate strategies such as groupwise coefficients or learnable parameters can be employed to balance interpretability and parameter efficiency.

To align with the use of global coefficients $(a, b)$, the subsequent stage of the CIF module reorders channels according to physiological priors, ensuring that linear fusion operates on a physiologically meaningful sequence and preserves functional relationships. First, input channels are reordered as shown in Figure 2(b) to explicitly encode the physiological priors depicted in Figure 2(a). For EEG, channels are grouped by functional regions following the international 10–20 system (Klem, 1999), with anterior regions (frontopolar and frontal) placed at the beginning and posterior regions (parietal and occipital) at the end, while preserving local spatial neighborhoods within each region as much as possible. This "anterior-to-posterior" ordering is independent of specific electrode montages, allowing application across datasets and enhancing the capture of spatial patterns related to the brain's anterior–posterior organization. For ECG, channels are grouped according to cardiac physiology: limb leads (I, II, III) and augmented leads (aVR, aVL, aVF) are placed first, precordial leads (V1–V6) in the middle, and other derived or vector leads (Vx, Vy, Vz) last. The precordial and vector leads roughly follow the thoracic spatial trajectory—from proximal to distal, right to left, and anterior to posterior—ensuring adjacent placement of leads capturing related cardiac activity, thereby enabling effective fusion across functional regions or lead groups. This process, referred to as *functional region fusion* (FRF), allows the CIF module to exploit cross-region spatial dependencies.

In Figure 2 (d), we illustrate the specific physiological chan-

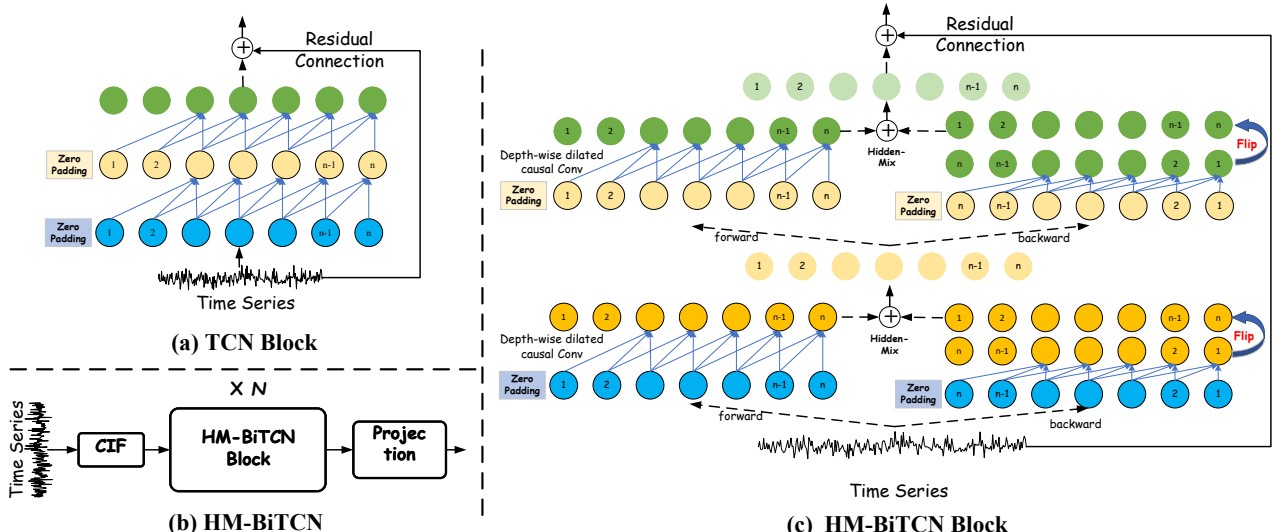

*Figure 4.* HM-BiTCN Architecture Diagram.

nel fusion relationships as the number of channels $n$ varies; for clarity, Figure 3 provides a simplified schematic of this process. Specifically, the anterior and posterior brain regions are first arranged according to physiological order, and then the first $n$ channels are fused with the last $n$ channels. When $n$ is less than half of the total number of channels, fusion occurs across different functional regions; when $n$ exceeds half, fusion occurs within the same functional region. This approach relies on only a few tunable parameters—$n$, $a$, and $b$—which control the number of channels affected by the CIF module. As $n$ increases, more channels are influenced while maintaining physiological diversity across channels. This flexible fusion strategy provides a foundation for systematically exploring different channel subsets and integrating additional physiological features, while also supporting subsequent parameter optimization.

**It is important to emphasize that the coefficients $a$ and $b$ can be configured in two ways: either as fixed values or as learnable parameters. In the learnable case, these parameters are trained end-to-end together with the selected network.**

### 3.2. HM-BiTCN Structure Design and Theoretical Analysis

To demonstrate the advantages of data-centric approaches and to show that simple models can achieve strong performance, as illustrated in Figure 4 (a), the conventional TCN only models unidirectional causal relationships in time series. In contrast, our proposed HM-BiTCN (Figure 4 (c)) extends the traditional TCN by introducing bidirectional feature mixing, allowing each layer to capture both for-ward and backward temporal dependencies. This design is complementary to CIF's enhancement along the channel dimension: CIF focuses on capturing inter-channel spatial relationships and improving the signal-to-noise ratio, while HM-BiTCN strengthens feature extraction along the temporal dimension. The combined enhancement across both channel and temporal dimensions enables CIF+HM-BiTCN to more fully exploit the information contained in the data, achieving superior performance. Importantly, when the parameters $a$ and $b$ in CIF are set to be learnable, they are optimized jointly with HM-BiTCN's structural parameters in an end-to-end training process, thereby enabling unified channel- and time-domain feature enhancement. For further details, please refer to Appendix D and Appendix E.

## 4. Experiments

**Medical Time Series Datasets.** (1) **APAVA** (Escudero et al., 2006) is an EEG dataset where each sample is assigned a binary label indicating whether the subject has Alzheimer's disease. (2) **TDBRAIN** (van Dijk et al., 2022) is an EEG dataset with a binary label assigned to each sample, indicating whether the subject has Parkinson's disease. (3) **ADFTD** (Miltiadous et al., 2023b;a) is an EEG dataset with a three-class label for each sample, categorizing the subject as Healthy, having Frontotemporal Dementia, or Alzheimer's disease. (4) **PTB** (PhysioBank, 2000) is an ECG dataset where each sample is labeled with a binary indicator of Myocardial Infarction. (5) **PTB-XL** (Wagner et al., 2020) is an ECG dataset with a five-class label for each sample, representing various heart conditions. Table 4 provides information on the processed datasets. The data processing methodology is the same as that of Medformer (Wang et al.,

*Table 1.* **Results of Subject-Independent Setup.** The results we compare include those reported by Medforme (Wang et al., 2024a) and MedGNN (Fan et al., 2025). Additionally, we have reproduced all the methods in the Table 9 to ensure a **fairer comparison**. The best result is highlighted in **bold**, and the second-best is underlined.

| Datasets | Models | Accuracy ↑ | Precision ↑ | Recall ↑ | F1 score ↑ | AUROC ↑ | AUPRC ↑ |
|---|---|---|---|---|---|---|---|
| **APAVA** (2-Classes) Reported | Autoformer | 68.64±1.82 | 68.48±2.10 | 68.77±2.27 | 68.06±1.94 | 75.94±3.61 | 74.38±4.05 |
| | Crossformer | 73.77±1.95 | 79.29±4.36 | 68.86±1.70 | 68.93±1.85 | 72.39±3.33 | 72.05±3.65 |
| | FEDformer | 74.94±2.15 | 74.59±1.50 | 73.56±3.55 | 73.51±3.39 | 83.72±1.97 | 82.94±2.37 |
| | Informer | 73.11±4.40 | 75.17±6.06 | 69.17±4.56 | 69.47±5.06 | 70.46±4.91 | 70.75±5.27 |
| | iTransformer | 74.55±1.66 | 74.77±2.10 | 71.76±1.72 | 72.30±1.79 | 85.55±1.55 | 84.39±1.57 |
| | MTST | 71.14±1.59 | 79.30±0.97 | 65.27±2.28 | 64.01±3.16 | 68.87±2.34 | 71.06±1.60 |
| | Nonformer | 71.89±3.81 | 71.80±4.58 | 69.44±3.56 | 69.74±3.84 | 70.55±2.96 | 70.78±4.08 |
| | PatchTST | 67.03±1.65 | 78.76±1.28 | 59.91±2.02 | 55.97±3.10 | 65.65±0.28 | 67.99±0.76 |
| | Reformer | 78.70±2.00 | 82.50±3.95 | 75.00±1.61 | 75.93±1.82 | 73.94±1.40 | 76.04±1.14 |
| | Transformer | 76.30±4.72 | 77.64±5.95 | 73.09±5.01 | 73.75±5.38 | 72.50±6.60 | 73.23±7.60 |
| | Medformer | 78.74±0.64 | 81.11±0.84 | 75.40±0.66 | 76.31±0.71 | 83.20±0.91 | 83.66±0.92 |
| | MedGNN | 82.60±0.35 | **87.70**±0.22 | 78.93±0.09 | 80.25±0.16 | 85.93±0.26 | - |
| | HM-BiTCN + CIF | **86.30**±1.05 | 86.16±1.09 | **85.47**±1.12 | **85.71**±1.09 | **94.26**±0.54 | **94.42**±0.49 |
| **TDBrain** (2-Classes) Reported | Autoformer | 87.33±3.79 | 88.06±3.56 | 87.33±3.79 | 87.26±3.84 | 93.81±2.26 | 93.32±2.42 |
| | Crossformer | 81.56±2.19 | 81.97±2.25 | 81.56±2.20 | 81.50±2.20 | 91.20±1.78 | 91.51±1.71 |
| | FEDformer | 78.13±1.98 | 78.52±1.91 | 78.13±1.98 | 78.04±2.01 | 86.56±1.86 | 86.48±1.99 |
| | Informer | 89.02±2.50 | 89.43±2.14 | 89.02±2.50 | 88.98±2.54 | 96.64±0.68 | 96.75±0.63 |
| | iTransformer | 74.67±1.06 | 74.71±1.06 | 74.67±1.06 | 74.65±1.06 | 83.37±1.14 | 83.73±1.27 |
| | MTST | 76.96±3.76 | 77.24±3.59 | 76.96±3.76 | 76.88±3.83 | 85.27±4.46 | 82.81±5.64 |
| | Nonformer | 87.88±2.48 | 88.86±1.84 | 87.88±2.48 | 87.78±2.56 | 97.05±0.68 | 96.99±0.68 |
| | PatchTST | 79.25±3.79 | 79.60±4.09 | 79.25±3.79 | 79.20±3.77 | 87.95±4.96 | 86.36±6.67 |
| | Reformer | 87.92±2.01 | 88.64±1.40 | 87.92±2.01 | 87.85±2.08 | 96.30±0.54 | 96.40±0.45 |
| | Transformer | 87.17±1.67 | 87.99±1.68 | 87.17±1.67 | 87.10±1.68 | 96.28±0.92 | 96.34±0.81 |
| | Medformer | 89.62±0.81 | 89.68±0.78 | 89.62±0.81 | 89.62±0.81 | 96.41±0.35 | 96.51±0.33 |
| | MedGNN | 91.04±0.09 | 91.15±0.12 | 91.04±0.20 | 91.04±0.08 | 96.74±0.04 | - |
| | HM-BiTCN + CIF | **93.13**±1.41 | **93.33**±1.37 | **93.13**±1.41 | **93.12**±1.42 | **98.62**±0.66 | **98.68**±0.63 |
| **ADFTD** (3-Classes) Reported | Autoformer | 45.25±1.48 | 43.67±1.94 | 42.96±2.03 | 42.59±1.85 | 61.02±1.82 | 43.10±2.30 |
| | Crossformer | 50.45±2.31 | 45.57±1.63 | 45.88±1.82 | 45.50±1.70 | 66.45±2.03 | 48.33±2.05 |
| | FEDformer | 46.30±0.59 | 46.05±0.76 | 44.22±1.38 | 43.91±1.37 | 62.62±1.75 | 46.11±1.44 |
| | Informer | 48.45±1.96 | 46.54±1.68 | 46.06±1.84 | 45.74±1.38 | 65.87±1.27 | 47.60±1.30 |
| | iTransformer | 52.60±1.59 | 46.79±1.27 | 47.28±1.29 | 46.79±1.13 | 67.26±1.16 | 49.53±1.21 |
| | MTST | 45.60±2.03 | 44.70±1.33 | 45.05±1.30 | 44.31±1.74 | 62.50±0.81 | 45.16±0.85 |
| | Nonformer | 49.95±1.05 | 47.71±0.97 | 47.46±1.50 | 46.96±1.35 | 66.23±1.37 | 47.33±1.78 |
| | PatchTST | 44.37±0.95 | 42.40±1.13 | 42.06±1.48 | 41.97±1.37 | 60.08±1.50 | 42.49±1.79 |
| | Reformer | 50.78±1.17 | 49.64±1.49 | 49.89±1.67 | 47.94±0.69 | 69.17±1.58 | 51.73±1.94 |
| | Transformer | 50.47±2.14 | 49.13±1.83 | 48.01±1.53 | 48.09±1.59 | 67.93±1.59 | 48.33±2.02 |
| | Medformer | 53.27±1.54 | 51.02±1.57 | 50.71±1.55 | 50.65±1.51 | 70.93±1.19 | 51.21±1.32 |
| | MedGNN | 56.12±0.11 | 55.07±0.09 | 55.47±0.34 | 55.00±0.24 | 74.68±0.33 | - |
| | HM-BiTCN + CIF | **58.56**±0.93 | **55.65**±0.81 | **55.86**±0.79 | **55.42**±0.82 | **76.07**±0.59 | **59.75**±0.67 |
| **PTB** (2-Classes) Reported | Autoformer | 73.35±2.10 | 72.11±2.89 | 63.24±3.17 | 63.69±3.84 | 78.54±3.48 | 74.25±3.53 |
| | Crossformer | 80.17±3.79 | 85.04±1.83 | 71.25±6.29 | 72.75±7.19 | 88.55±3.45 | 87.31±3.25 |
| | FEDformer | 76.05±2.54 | 77.58±3.61 | 66.10±3.55 | 67.14±4.37 | 85.93±4.31 | 82.59±5.42 |
| | Informer | 78.69±1.68 | 82.87±1.02 | 69.19±2.90 | 70.84±3.47 | 92.09±0.53 | 90.02±0.60 |
| | iTransformer | 83.89±0.71 | 88.25±1.18 | 76.39±1.01 | 79.06±1.06 | 91.18±1.16 | 90.93±0.98 |
| | MTST | 76.59±1.90 | 79.88±1.90 | 66.31±2.95 | 67.38±3.71 | 86.86±2.75 | 83.75±2.84 |
| | Nonformer | 78.66±0.49 | 82.77±0.86 | 69.12±0.87 | 70.90±1.00 | 89.37±2.51 | 86.67±2.38 |
| | PatchTST | 74.74±1.62 | 76.94±1.51 | 63.89±2.71 | 64.36±3.38 | 88.79±0.91 | 83.39±0.96 |
| | Reformer | 77.96±2.13 | 81.72±1.61 | 68.20±3.35 | 69.65±3.88 | 91.13±0.74 | 88.42±1.30 |
| | Transformer | 77.37±1.02 | 81.84±0.66 | 67.14±1.80 | 68.47±2.19 | 90.08±1.76 | 87.22±1.68 |
| | Medformer | 83.50±2.01 | 85.19±0.94 | 77.11±3.39 | 79.18±3.31 | 92.81±1.48 | 90.32±1.54 |
| | MedGNN | 84.53±0.28 | 87.35±0.45 | 77.90±0.66 | 80.40±0.62 | 93.31±0.46 | - |
| | HM-BiTCN + CIF | **88.29**±1.45 | **90.66**±1.48 | **83.21**±2.02 | **85.59**±1.96 | **94.28**±0.93 | **93.78**±1.11 |
| **PTB-XL** (5-Classes) Reported | Autoformer | 61.68±2.72 | 51.60±1.64 | 49.10±1.52 | 48.85±2.27 | 82.04±1.44 | 51.93±1.71 |
| | Crossformer | 73.30±2.16 | 65.06±0.35 | **61.23**±0.33 | 62.59±0.14 | 90.02±0.06 | 67.43±0.22 |
| | FEDformer | 57.20±9.47 | 52.38±6.09 | 49.04±7.26 | 47.89±8.44 | 82.13±4.17 | 52.31±7.03 |
| | Informer | 71.43±0.32 | 62.64±0.60 | 59.12±0.47 | 60.44±0.43 | 88.65±0.09 | 64.76±0.17 |
| | iTransformer | 69.28±0.22 | 59.59±0.45 | 54.62±0.18 | 56.20±0.19 | 86.71±0.10 | 60.27±0.21 |
| | MTST | 72.14±0.27 | 63.84±0.72 | 60.01±0.81 | 61.43±0.38 | 88.97±0.33 | 65.83±0.51 |
| | Nonformer | 70.56±0.55 | 61.57±0.66 | 57.75±0.72 | 59.10±0.66 | 88.32±0.36 | 63.40±0.79 |
| | PatchTST | 73.23±0.25 | 65.70±0.64 | 60.82±0.76 | **62.61**±0.34 | 89.74±0.19 | 67.32±0.22 |
| | Reformer | 71.72±0.43 | 63.12±1.02 | 59.20±0.75 | 60.69±0.18 | 88.80±0.24 | 64.72±0.47 |
| | Transformer | 70.59±0.44 | 61.57±0.65 | 57.62±0.35 | 59.05±0.25 | 88.21±0.16 | 63.36±0.29 |
| | Medformer | 72.87±0.23 | 64.14±0.42 | 60.60±0.46 | 62.02±0.37 | 89.66±0.13 | 66.39±0.22 |
| | MedGNN | **73.87**±0.18 | **66.26**±0.29 | 61.13±0.29 | 62.54±0.20 | 90.21±0.15 | - |
| | HM-BiTCN + CIF | 73.73±0.30 | 65.41±0.67 | 60.70±1.08 | 61.89±0.91 | **90.53**±0.22 | **67.75**±0.75 |

2024a) and MedGNN (Fan et al., 2025).

**Baselines.** We compare with 12 state-of-the-art time series transformer methods: Autoformer (Wu et al., 2021), Crossformer (Zhang & Yan, 2022), FEDformer (Zhou et al., 2022), Informer (Zhou et al., 2021), iTransformer (Liu et al., 2024b), MTST (Zhang et al., 2024), Nonformer (Liu et al., 2022), PatchTST (Nie et al., 2023), Reformer (Kitaev et al., 2019), vanilla Transformer (Vaswani et al., 2017), Medformer (Wang et al., 2024a), MedGNN (Fan et al., 2025).

**Implementation.** We evaluate the models using six metrics: accuracy, precision (macro-averaged), recall (macro-averaged), F1 score (macro-averaged), AUROC (macro-averaged), and AUPRC (macro-averaged). The training process is repeated with five random seeds (41–45) on fixed training, validation, and test sets, and the mean and standard deviation of the models are computed. All experiments were conducted on an NVIDIA RTX 3090 GPU and implemented using PyTorch version 1.11.0 (Paszke et al., 2017). Following the methodology of Medformer (Wang et al., 2024a) and MedGNN (Fan et al., 2025), we adopt a **Subject-Independent Split** dataset partitioning strategy, where each subject appears in only one of the training, validation, or test sets. This partitioning strategy simulates real-world diagnostic scenarios while introducing challenges related to inter-subject variability. The model is trained to minimize the supervised loss function:

$$\mathcal{L} = \frac{1}{N} \sum_{i=1}^{N} \ell(\hat{y}_i, y_i),$$

where $\ell(\hat{y}_i, y_i)$ quantifies the discrepancy between the predicted labels $\hat{y}_i$ and the ground-truth labels $y_i$. Specifically, the encoder $f_\theta : \mathbb{R}^{T \times C} \to \mathbb{R}^d$ maps each input $x_i$ to a latent representation $h_i = f_\theta(x_i)$, which is then passed to the classifier $g_\phi$ to produce the final prediction $\hat{y}_i = g_\phi(h_i)$.

### 4.1. Results of Subject-Independent

Following Medformer (Wang et al., 2024a), in this setup, the training, validation, and test sets are split based on subjects. All subjects and their corresponding samples are assigned to the training, validation, and test sets according to a predetermined ratio or subject IDs, ensuring that samples from the same subject appear in only one of these sets. This simulates real-world MedTS-based disease diagnosis, aiming to train a model on subjects with known labels and then test it on unseen subjects to determine whether they have a specific disease. All five datasets are evaluated using this setup.

Table 1 presents the results **reported** by various methods in the subject-independent setting, while Table 9 shows the results of our **reproduction** of these methods. Our method achieves the highest average scores across six metrics on four out of the five datasets. On PTB-XL, our method tops

AUROC and AUPRC and ranks second in Accuracy versus reported results, and ranks first in Accuracy, Precision, AUROC, AUPRC versus our reproduced results.

#### 4.1.1. EFFICIENCY ANALYSIS

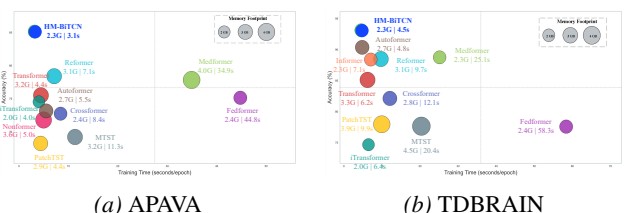

*(a)* APAVA          *(b)* TDBRAIN

*Figure 5.* Effectiveness and efficiency on two datasets .

We evaluate the model efficiency in terms of accuracy, training speed, and memory footprint using two datasets: APAVA and TDBRAIN. In Figure 5, a marker closer to the upper-left corner indicates higher accuracy and faster training speed, while a smaller marker area corresponds to lower memory usage. The results show that HM-BiTCN achieves the best overall performance among all baseline methods, demonstrating its high efficiency and reliability across different application scenarios.

### 4.2. Ablation Study

**(1) Effectiveness of CIF:** Table 2 demonstrates the excellent performance of combining HM-BiTCN with CIF, confirming the compatibility of the HM-BiTCN with CIF. See Appendix J.1 for details. Appendix F presents ablation studies on the HM-BiTCN architecture, the performance improvements from integrating CIF into its components, and comparisons with the vanilla TCN structure.

**(2) Hyperparameter Transfer and Adaptation:** We evaluate the transferability of key hyperparameters (e.g., $a$, $b$, $n$) from HM-BiTCN to other models. If transferred settings underperform, we further fine-tune them for adaptation. See Appendices J.2 and J.3. Figure 6 illustrates the outstanding performance of CIF when combined with other models.

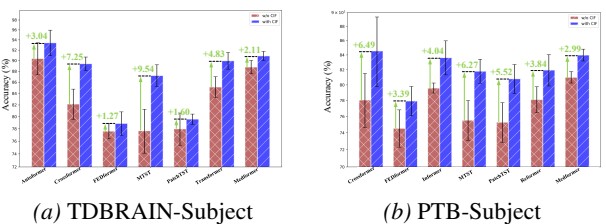

*(a)* TDBRAIN-Subject      *(b)* PTB-Subject

*Figure 6.* The improvements achieved by various baselines when combined with the CIF method.

**(3) Exploring alternative fusion strategies:** In Appendix G, we systematically evaluate several fusion approaches, including the fusion using only the Fp1 and Fp2

*Table 2.* Exploring the Integration of HM-BiTCN Structure with CIF.

| Datasets | APAVA | | ADFTD | | PTB | | TDBRAIN | | PTB-XL | |
|---|---|---|---|---|---|---|---|---|---|---|
| Metrics | Accuracy | F1 Score | Accuracy | F1 Score | Accuracy | F1 Score | Accuracy | F1 Score | Accuracy | F1 Score |
| w/ CIF | $86.30_{\pm1.05}$ | $85.71_{\pm1.09}$ | $58.56_{\pm0.93}$ | $55.42_{\pm0.82}$ | $88.29_{\pm1.45}$ | $85.59_{\pm1.96}$ | $93.13_{\pm1.41}$ | $93.12_{\pm1.42}$ | $73.73_{\pm0.30}$ | $61.89_{\pm0.91}$ |
| w/o CIF | $82.49_{\pm1.40}$ | $81.60_{\pm1.39}$ | $52.05_{\pm2.22}$ | $49.48_{\pm2.70}$ | $81.87_{\pm1.87}$ | $75.84_{\pm3.20}$ | $84.90_{\pm2.60}$ | $84.76_{\pm2.74}$ | $72.92_{\pm0.88}$ | $61.49_{\pm0.82}$ |
| **Improvement** | **+3.81%** | **+4.11%** | **+6.51%** | **+5.94%** | **+6.42%** | **+9.75%** | **+8.23%** | **+8.36%** | **+0.81%** | **+0.40%** |

*Table 3.* Performance on the HAR and UCI-HAR non-medical time series datasets. * denotes the results reported by Medformer.

| Dataset / Metric | | Crossformer * (Zhang & Yan, 2022) | Reformer * (Kitaev et al., 2019) | Transformer * (Vaswani et al., 2017) | TCN * (Bai et al., 2018) | ModernTCN * (Luo & Wang, 2024) | Mamba * (Gu & Dao, 2023) | Medformer * (Wang et al., 2024a) | HM-BiTCN (This work) | HM-BiTCN + CIF (This work) |
|---|---|---|---|---|---|---|---|---|---|---|
| **FLAAP** | Accuracy | $75.84_{\pm0.52}$ | $71.65_{\pm1.27}$ | $74.96_{\pm1.25}$ | $66.48_{\pm1.66}$ | $74.80_{\pm0.96}$ | $64.87_{\pm2.78}$ | $76.44_{\pm0.64}$ | $76.08_{\pm0.81}$ | $\mathbf{76.82_{\pm1.32}}$ |
| *(10 Classes)* | F1 Score | $75.52_{\pm0.66}$ | $71.14_{\pm1.45}$ | $74.49_{\pm1.39}$ | $65.29_{\pm1.74}$ | $74.35_{\pm0.85}$ | $64.14_{\pm2.70}$ | $76.25_{\pm0.65}$ | $75.54_{\pm0.94}$ | $\mathbf{76.39_{\pm1.18}}$ |
| **UCI-HAR** | Accuracy | $89.74_{\pm1.08}$ | $88.44_{\pm2.02}$ | $88.86_{\pm1.65}$ | $93.08_{\pm0.95}$ | $91.44_{\pm1.01}$ | $87.78_{\pm1.10}$ | $91.65_{\pm0.74}$ | $93.72_{\pm0.73}$ | $\mathbf{93.78_{\pm0.32}}$ |
| *(6 Classes)* | F1 Score | $89.70_{\pm1.10}$ | $88.34_{\pm1.98}$ | $88.80_{\pm1.67}$ | $93.19_{\pm0.88}$ | $91.47_{\pm0.98}$ | $87.72_{\pm1.10}$ | $91.61_{\pm0.75}$ | $93.69_{\pm0.76}$ | $\mathbf{93.74_{\pm0.34}}$ |

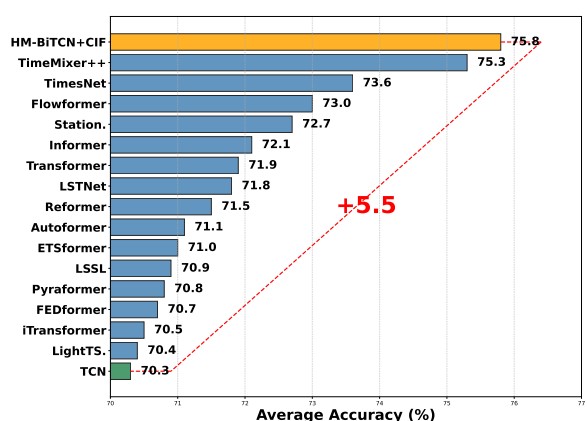

*Figure 7.* Average accuracy of various methods on the UEA dataset. More details in Appendix H.

channels mentioned in the Introduction, the fusion of a reduced set of left-right symmetric channels, and our further exploration using canonical correlation analysis (CCA) for fusion. The experimental results demonstrate that CIF exhibits strong scalability, maintaining performance advantages across different fusion strategies.

**(4)Results on general time series classification tasks**

To evaluate the performance of our method on general time series, we follow the design of Medformer (Wang et al., 2024a) and test it on two human activity recognition (HAR) datasets: FLAAP(13,123 samples, 10 classes) (Kumar & Suresh, 2022) and UCI-HAR(10,299 samples, 6 classes) (Anguita et al., 2013). Additionally, to conduct a more comprehensive evaluation, following TimeMixer++ (Wang et al., 2025a), we used 10 multivariate datasets from the UEA Time Series Classification Archive (2018) for the assessment of classification tasks.

As shown in Table 3, and Fig. 7, the combination of HM-BiTCN and CIF consistently outperforms other architectures in general time series classification, achieving a 5.5% im-

provement over the original TCN and surpassing current SOTA methods. Although CIF was originally designed for MedTS, its integration with HM-BiTCN significantly outperforms Transformer-based models in both medical and general time series classification tasks, demonstrating the effectiveness of our data-centric approach.

The results in Tables 3, 8, 2 and Figure 6 show that CIF achieves significant improvements in MedTS classification, while the gains on non-medical data are relatively limited. This observation further demonstrates that a data-driven perspective is particularly effective for MedTS classification with physiological characteristics.

**More Discussions in Appendix L**

## 5. Conclusion

In this work, we propose a simple yet effective method for medical time series classification, *Channel-Imposed Fusion (CIF)*, which is inspired by the physiological relationships between channels and the idea that, under ideal conditions, the linear combination of signals can enhance the signal-to-noise ratio (SNR). Combined with the simple HM-BiTCN architecture, CIF outperforms existing state-of-the-art methods across multiple medical datasets and performs strongly on general time series classification tasks, demonstrating that data-centric design allows simple models to outperform more complex architectures. More importantly, CIF exemplifies the shift from the traditional *model-centric* paradigm to a *data-centric* perspective, where structured representations grounded in physiological priors are both efficient and scalable for medical time series classification. CIF also exhibits strong transferability and can be seamlessly integrated into mainstream models such as Transformers, enhancing their applicability in medical scenarios. **We hope this work encourages the community to reconsider the core of medical time series classification: should it be driven primarily by *data-centric strategies* or by *model-centric design* or both?**

## Impact Statement

It is important to clarify that CIF is inspired by the theoretical observation that, under ideal conditions, the linear combination of two channels can enhance the signal-to-noise ratio (SNR). However, due to the inherent complexity of medical time series, we *do not claim* that CIF reliably improves SNR in real-world scenarios.

This work aims to advance the field of Machine Learning. While there may be various potential societal implications, we do not believe that any specific consequences need to be highlighted in this context.

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

# Appendix

## A. Ideal-Case Explanation of SNR Enhancement Using Another Channel.

Consider the linear combination of the two observed signals:

$$y = ax_1 + bx_2, \tag{4}$$

where $a$ and $b$ are real coefficients. The observed signals are given by

$$x_1 = s_1 + \epsilon_1, \quad x_2 = s_2 + \epsilon_2, \tag{5}$$

with zero-mean signal and noise components:

$$\mathbb{E}[s_i] = \mathbb{E}[\epsilon_i] = 0, \quad i = 1, 2,$$

and mutually uncorrelated signal and noise components: $\mathrm{Cov}(s_i, \epsilon_j) = 0$.

The power of a zero-mean random signal is given by its variance:

$$P_s = \mathrm{Var}[s] = \mathbb{E}[(s - \mathbb{E}[s])^2] = \mathbb{E}[s^2]. \tag{6}$$

This is why, for zero-mean signals, the SNR can be expressed as a ratio of variances (or mean-square values) (Kay, 1993).

For the linear combination of signals:

$$\begin{aligned} \mathrm{Var}(as_1 + bs_2) &= a^2 \mathrm{Var}(s_1) + b^2 \mathrm{Var}(s_2) + 2ab\, \mathrm{Cov}(s_1, s_2) \\ &= a^2\sigma_s^2 + b^2\sigma_s^2 + 2ab(\rho\sigma_s^2) \\ &= \sigma_s^2(a^2 + b^2 + 2ab\rho), \end{aligned} \tag{7}$$

where $\rho = \mathrm{Corr}(s_1, s_2)$.

Similarly, the noise power of the linear combination is:

$$\begin{aligned} \mathrm{Var}(a\epsilon_1 + b\epsilon_2) &= a^2 \mathrm{Var}(\epsilon_1) + b^2 \mathrm{Var}(\epsilon_2) + 2ab\, \mathrm{Cov}(\epsilon_1, \epsilon_2) \\ &= a^2\sigma_\epsilon^2 + b^2\sigma_\epsilon^2 + 2ab(\gamma\sigma_\epsilon^2) \\ &= \sigma_\epsilon^2(a^2 + b^2 + 2ab\gamma), \end{aligned} \tag{8}$$

where $\gamma = \mathrm{Corr}(\epsilon_1, \epsilon_2)$.

Using the definition of SNR as the ratio of signal power to noise power:

$$\mathrm{SNR_{out}} = \frac{\mathrm{Var}(as_1 + bs_2)}{\mathrm{Var}(a\epsilon_1 + b\epsilon_2)} = \frac{\sigma_s^2(a^2 + b^2 + 2ab\rho)}{\sigma_\epsilon^2(a^2 + b^2 + 2ab\gamma)} = \mathrm{SNR_{in}} \cdot \frac{a^2 + b^2 + 2ab\rho}{a^2 + b^2 + 2ab\gamma}, \tag{9}$$

where $\mathrm{SNR_{in}} = \sigma_s^2/\sigma_\epsilon^2$.

¿ *Remark:* The zero-mean property ensures that the variance equals the mean-square value, which is why SNR can be expressed as a ratio of variances (Kay, 1993; Haykin, 2002).

For SNR improvement relative to individual channels:

$$\frac{a^2 + b^2 + 2ab\rho}{a^2 + b^2 + 2ab\gamma} > 1 \quad \Rightarrow \quad 2ab(\rho - \gamma) > 0. \tag{10}$$

- **Difference Mode** ($ab < 0$): $\rho < \gamma$ — suppress correlated noise while possibly attenuating some correlated signal.

- **Cooperative Mode** ($ab > 0$): $\rho > \gamma$ — amplify correlated signals relative to less-correlated noise.

**Optimization via the ratio $k = a/b$ (explicit choice of $a, b$)**

Because the output SNR

$$\text{SNR}_{\text{out}} = \text{SNR}_{\text{in}} \cdot \frac{a^2 + b^2 + 2ab\rho}{a^2 + b^2 + 2ab\gamma} \tag{11}$$

is homogeneous of degree zero in $(a, b)$ (i.e. invariant under common scaling $(a, b) \mapsto (ca, cb)$, $c \neq 0$), only the ratio $k = a/b$ matters. Assume $b \neq 0$ and set

$$k = \frac{a}{b}. \tag{12}$$

Define

$$F(k) \;=\; \frac{a^2 + b^2 + 2ab\rho}{a^2 + b^2 + 2ab\gamma} \;=\; \frac{k^2 + 1 + 2k\rho}{k^2 + 1 + 2k\gamma}, \tag{13}$$

so that

$$\text{SNR}_{\text{out}} = \text{SNR}_{\text{in}} \cdot F(k). \tag{14}$$

We therefore study extrema of $F(k)$ to determine favorable relative weights $a : b$.

Let

$$N(k) = k^2 + 1 + 2k\rho, \qquad D(k) = k^2 + 1 + 2k\gamma, \tag{15}$$

so $F(k) = N(k)/D(k)$. Then

$$N'(k) = 2k + 2\rho, \qquad D'(k) = 2k + 2\gamma. \tag{16}$$

Using the quotient rule,

$$F'(k) = \frac{N'(k)D(k) - N(k)D'(k)}{D(k)^2}. \tag{17}$$

The numerator simplifies and factorizes as

$$N'(k)D(k) - N(k)D'(k) = 2(\gamma - \rho)(k - 1)(k + 1), \tag{18}$$

so we obtain

$$F'(k) = \frac{2(\gamma - \rho)(k - 1)(k + 1)}{(k^2 + 1 + 2k\gamma)^2}. \tag{19}$$

**Critical points and their nature**

From (19), the critical points satisfy

$$F'(k) = 0 \iff k = -1, \; k = +1 \quad \text{or} \quad \rho = \gamma. \tag{20}$$

- If $\rho > \gamma$ (so $\gamma - \rho < 0$), then $(k - 1)(k + 1) < 0$ for $k \in (-1, 1)$ and $(k - 1)(k + 1) > 0$ for $k > 1$ or $k < -1$. Hence $F'(k) > 0$ on $(-1, 1)$ and $F'(k) < 0$ outside, so

$$k = 1 \text{ is a local maximum}, \qquad k = -1 \text{ is a local minimum}. \tag{21}$$

- If $\rho < \gamma$ (so $\gamma - \rho > 0$), the inequalities reverse: $F'(k) < 0$ on $(-1, 1)$ and $F'(k) > 0$ for $|k| > 1$, thus

$$k = -1 \text{ is a local maximum}, \qquad k = 1 \text{ is a local minimum}. \tag{22}$$

- If $\rho = \gamma$, then from (13) we have

$$F(k) \equiv 1, \quad \forall k \in \mathbb{R}, \tag{23}$$

so no choice of $k$ changes the SNR.

**Interpretation in terms of $a, b$ and explicit choice**

Recall from (12) that $k = a/b$. The critical ratios $k = \pm 1$ correspond to equal-magnitude weights:

$$k = +1 \iff a = b, \qquad k = -1 \iff a = -b. \tag{24}$$

Since $\text{SNR}_{\text{out}}$ in (11) is invariant to a common scaling of $(a, b)$, we may fix a convenient normalization:

**Fix $b = 1$.** Then $k = a$ and the recommended choices are

$$\text{Cooperative mode } (\rho > \gamma): \quad (a, b) = (1, 1), \tag{25}$$

$$\text{Differential mode } (\rho < \gamma): \quad (a, b) = (-1, 1). \tag{26}$$

**Fix $b = -1$.** Then $k = a$ and the recommended choices are

$$\text{Cooperative mode } (\rho > \gamma): \quad (a, b) = (-1, -1), \tag{27}$$

$$\text{Differential mode } (\rho < \gamma): \quad (a, b) = (1, -1). \tag{28}$$

**Degenerate and neutral cases**

- If $\rho = \gamma$, then by (23) we have
$$\text{SNR}_{\text{out}} = \text{SNR}_{\text{in}}, \quad \forall (a, b) \neq (0, 0), \tag{29}$$
i.e. the linear combination cannot improve SNR (except for singular noise-cancellation cases).

- The denominator in (13) vanishes when
$$k^2 + 1 + 2k\gamma = 0, \tag{30}$$
which corresponds to zero output noise variance. This is a nongeneric, degenerate configuration (perfect noise cancellation).

From (21)–(22) and (24), we conclude:

- If $\rho > \gamma$: the SNR is maximized (among equal-variance combinations) by equal-phase combining $a = b$ (cooperative mode).

- If $\rho < \gamma$: the SNR is maximized by equal-magnitude opposite-phase combining $a = -b$ (differential mode).

- If $\rho = \gamma$: $F(k) \equiv 1$ and no linear combining improves SNR, up to degenerate cases.

**Remarks on the choice of $(a, b)$ ranges.** If the goal is to find the SNR extremum points, it is sufficient to consider $(a, b)$ combinations where $a$ and $b$ are either of the same sign (both positive or both negative) or of opposite signs. For example, exploring $(a, b)$ within the rectangle $[-1, 1] \times [-1, 1]$ already includes the critical ratios $k = \pm 1$ and thus captures the SNR maxima and minima. While this rectangle does not cover all possible $k$ values (i.e., all real ratios $a/b$), it is enough to determine the locations of the extremal points, thanks to the homogeneity of SNR in $(a, b)$.

# B. Data Preprocessing

*Table 4.* **The information of processed datasets.** The table shows the number of subjects, samples, classes, channels, sampling rate, sample timestamps, modality of MedTS, and file size. Here, **#-Timestamps** indicates the number of timestamps per sample. All data processing procedures follow those of Medformer (Wang et al., 2024a).

| Datasets | #-Subject | #-Sample | #-Class | #-Channel | #-Timestamps | Sampling Rate | Modality | File Size |
|---|---|---|---|---|---|---|---|---|
| APAVA | 23 | 5,967 | 2 | 16 | 256 | 256Hz | EEG | 186MB |
| ADFTD | 88 | 69,752 | 3 | 19 | 256 | 256Hz | EEG | 2.52GB |
| TDBrain | 72 | 6,240 | 2 | 33 | 256 | 256Hz | EEG | 571MB |
| PTB | 198 | 64,356 | 2 | 15 | 300 | 250Hz | ECG | 2.15GB |
| PTB-XL | 17,596 | 191,400 | 5 | 12 | 250 | 250Hz | ECG | 4.28GB |

*Table 5.* Datasets and mapping details of UEA dataset (Bagnall et al., 2018).

| Dataset | Sample Numbers(train set,test set) | Variable Number | Series Length |
|---|---|---|---|
| EthanolConcentration | (261, 263) | 3 | 1751 |
| FaceDetection | (5890, 3524) | 144 | 62 |
| Handwriting | (150, 850) | 3 | 152 |
| Heartbeat | (204, 205) | 61 | 405 |
| JapaneseVowels | (270, 370) | 12 | 29 |
| PEMSSF | (267, 173) | 963 | 144 |
| SelfRegulationSCP1 | (268, 293) | 6 | 896 |
| SelfRegulationSCP2 | (200, 180) | 7 | 1152 |
| SpokenArabicDigits | (6599, 2199) | 13 | 93 |
| UWaveGestureLibrary | (120, 320) | 3 | 315 |

# C. Implementation Details

We implement our method and all the baselines based on the Time-Series-Library project[1] from Tsinghua University (Wu et al., 2023), which integrates all methods under the same framework and training techniques to ensure a relatively fair comparison. The 12 baseline time series transformer methods are Autoformer (Wu et al., 2021), Crossformer (Zhang & Yan, 2022), FEDformer (Zhou et al., 2022), Informer (Zhou et al., 2021), iTransformer (Liu et al., 2024b), MTST (Zhang et al., 2024), Nonformer (Liu et al., 2022), PatchTST (Nie et al., 2023), Reformer (Kitaev et al., 2019), vanilla Transformer (Vaswani et al., 2017), Medformer (Wang et al., 2024a), and MedGNN (Fan et al., 2025).

**Autoformer** Autoformer (Wu et al., 2021) employs an auto-correlation mechanism to replace self-attention for time series forecasting. Additionally, they use a time series decomposition block to separate the time series into trend-cyclical and seasonal components for improved learning. The raw source code is available at `https://github.com/thuml/Autoformer`.

**Crossformer** Crossformer (Zhang & Yan, 2022) designs a single-channel patching approach for token embedding. They utilize two-stage self-attention to leverage both temporal features and channel correlations. A router mechanism is proposed to reduce time and space complexity during the cross-dimension stage. The raw code is available at `https://github.com/Thinklab-SJTU/Crossformer`.

**FEDformer** FEDformer (Zhou et al., 2022) leverages frequency domain information using the Fourier transform. They introduce frequency-enhanced blocks and frequency-enhanced attention, which are computed in the frequency domain. A novel time series decomposition method replaces the layer norm module in the transformer architecture to improve learning. The raw code is available at `https://github.com/MAZiqing/FEDformer`.

**Informer** Informer (Zhou et al., 2021) is the first paper to employ a one-forward procedure instead of an autoregressive method in time series forecasting tasks. They introduce ProbSparse self-attention to reduce complexity and memory usage. The raw code is available at `https://github.com/zhouhaoyi/Informer2020`.

---

[1] `https://github.com/thuml/Time-Series-Library`

**iTransformer** iTransformer (Liu et al., 2024b) questions the conventional approach of embedding attention tokens in time series forecasting tasks and proposes an inverted approach by embedding the whole series of channels into a token. They also invert the dimension of other transformer modules, such as the layer norm and feed-forward networks. The raw code is available at `https://github.com/thuml/iTransformer`.

**MTST** MTST (Zhang et al., 2024) uses the same token embedding method as Crossformer and PatchTST. It highlights the importance of different patching lengths in forecasting tasks and designs a method that can take different sizes of patch tokens as input simultaneously. The raw code is available at `https://github.com/networkslab/MTST`.

**Nonformer** Nonformer (Liu et al., 2022) analyzes the impact of non-stationarity in time series forecasting tasks and its significant effect on results. They design a de-stationary attention module and incorporate normalization and denormalization steps before and after training to alleviate the over-stationarization problem. The raw code is available at `https://github.com/thuml/Nonstationary_Transformers`.

**PatchTST** PatchTST (Nie et al., 2023) embeds a sequence of single-channel timestamps as a patch token to replace the attention token used in the vanilla transformer. This approach enlarges the receptive field and enhances forecasting ability. The raw code is available at `https://github.com/yuqinie98/PatchTST`.

**Reformer** Reformer (Kitaev et al., 2019) replaces dot-product attention with locality-sensitive hashing. They also use a reversible residual layer instead of standard residuals. The raw code is available at `https://github.com/lucidrains/reformer-pytorch`.

**Transformer** Transformer (Vaswani et al., 2017), commonly known as the vanilla transformer, is introduced in the well-known paper "Attention is All You Need." It can also be applied to time series by embedding each timestamp of all channels as an attention token. The PyTorch version of the code is available at `https://github.com/jadore801120/attention-is-all-you-need-pytorch`.

**Medformer** Medformer (Wang et al., 2024a) uses cross-channel patch embedding to model spatiotemporal dependencies. The raw code is available at `https://github.com/DL4mHealth/Medformer`

**MedGNN** MedGNN (Fan et al., 2025) employs multi-resolution spatiotemporal graph learning to extract dynamic features across multiple time scales. The raw code is available at `https://github.com/aikunyi/MedGNN`.

### C.1. Evaluation metrics

For all methods, the optimizer used is Adam, with a learning rate of 1e-4. The batch size is set to {32,32,128,128,128} for the datasets APAVA, TDBrain, ADFD, PTB, and PTB-XL, respectively. Training is conducted for 100 epochs, with early stopping triggered after 10 epochs without improvement in the F1 score on the validation set. We save the model with the best F1 score on the validation set and evaluate it on the test set. We employ six evaluation metrics: accuracy, precision (macro-averaged), recall (macro-averaged), F1 score (macro-averaged), AUROC (macro-averaged), and AUPRC (macro-averaged). Both subject-dependent and subject-independent setups are implemented for different datasets. Each experiment is run with 5 random seeds (41-45) and fixed training, validation, and test sets to compute the average results and standard deviations.

To comprehensively and fairly evaluate the performance of each model in the classification task, we select five evaluation metrics: Accuracy, Precision, Recall, F1 score, and AUROC. The definitions and specific calculation formulas for each metric are presented below:

Accuracy measures the proportion of correct predictions out of the total number of predictions. It's calculated as:

$$\text{Accuracy} = \frac{\text{Number of correct predictions}}{\text{Total number of predictions}}. \tag{31}$$

This metric is useful when the classes are balanced but may be misleading in cases of class imbalance.

Precision focuses on the quality of positive predictions and measures the proportion of correctly predicted positive instances out of all instances predicted as positive. It's especially useful when false positives need to be minimized. The formula is:

$$\text{Precision} = \frac{\text{True Positives}}{\text{True Positives} + \text{False Positives}}. \tag{32}$$

Recall measures the proportion of actual positive instances that were correctly identified. It's important when false negatives

are costly. The formula is:

$$\text{Recall} = \frac{\text{True Positives}}{\text{True Positives} + \text{False Negatives}}. \tag{33}$$

It shows how well the model captures all relevant instances.

The F1 score is the harmonic mean of precision and recall, balancing the two when one is more important than the other. It's particularly useful when dealing with imbalanced datasets, as it accounts for both false positives and false negatives. The formula is:

$$\text{F1 Score} = 2 \times \frac{\text{Precision} \times \text{Recall}}{\text{Precision} + \text{Recall}}. \tag{34}$$

It gives a single metric that reflects both precision and recall performance.

The Area Under the Receiver Operating Characteristic Curve (AUROC) measures the ability of a model to distinguish between classes, defined as

$$\text{AUROC} = \int_0^1 \text{TPR}(\text{FPR}) \, d(\text{FPR}), \tag{35}$$

where

$$\text{TPR} = \frac{TP}{TP + FN}, \qquad \text{FPR} = \frac{FP}{FP + TN}.$$

The Area Under the Precision–Recall Curve (AUPRC) summarizes the trade-off between precision and recall across different thresholds, defined as

$$\text{AUPRC} = \int_0^1 \text{Precision}(\text{Recall}) \, d(\text{Recall}), \tag{36}$$

where

$$\text{Precision} = \frac{TP}{TP + FP}, \qquad \text{Recall} = \frac{TP}{TP + FN}.$$

# D. HM-BiTCN Structure Design and Theoretical Analysis

Modeling short-term and long-term dependencies in time series data is challenging. Traditional CNNs excel at capturing local features but have limited receptive fields, hindering long-range dependency learning. Transformer-based methods effectively model long-term dependencies, but their complex design lacks interpretability, which is a key issue in medical time-series classification.To address these limitations, TCNs use causal convolutions for explicit temporal modeling and dilated convolutions to expand the receptive field, overcoming the constraints of traditional CNNs. Building on the advantages of TCN, we propose the HM-BiTCN, which combines the benefits of dilated convolutions, bidirectional causal convolution, and residual connections. This approach allows for better capture of temporal dependencies while preserving causality.

## D.1. Dilated Convolution

Dilated convolution expands the receptive field without significantly increasing computational cost (Yu & Koltun, 2015). For a 1D input sequence $x = [x_1, x_2, \ldots, x_T]$, its output is defined as $y(t) = \sum_{i=0}^{k-1} x(t + i \cdot d) \cdot w(i)$, where $t$ is the current time step, $k$ is the kernel size, $d$ is the dilation factor, and $w(i)$ is the weight at the $i$-th position in the kernel. Increasing $d$ effectively enlarges the receptive field, enabling the network to capture longer-term temporal dependencies. When stacking multiple dilated convolutional layers, the receptive field grows progressively. For the $l$-th layer, the receptive field $r_l$ can be expressed as $r_l = k + (k-1) \sum_{j=1}^{l-1} d_j$, where $d_j$ is the dilation factor of the $j$-th layer. By gradually increasing $d_j$, the network captures temporal dependencies across both global and local scales, offering an effective way to model long-term dependencies in time series.

## D.2. Bidirectional Causal Convolution Structure

In addition to dilated convolutions, HM-BiTCN introduces a *bidirectional causal convolution structure*, inspired by prior bidirectional temporal modeling approaches (Hanson et al., 2018; Hu et al., 2024; Yin et al., 2025). Unlike traditional TCNs that use only forward causal convolutions, our architecture applies causal convolutions in both forward and backward directions, enabling the model to capture dependencies from both past and future contexts while strictly preserving causality. The *forward causal convolution* processes the input sequence $x(t)$ in chronological order, producing output $y_{\text{forward}}(t) = \sum_{i=0}^{k-1} x(t - i \cdot d) \cdot w_{\text{forward}}(i)$, which depends only on current and past inputs. For the *backward causal convolution*, we first reverse the input sequence as $x_{\text{flip}}(t) = x(T - t)$, and then apply a causal convolution over this flipped sequence. This ensures that the model captures future-directed dependencies without introducing information leakage. The output is given by $y_{\text{backward}}(t) = \sum_{i=0}^{k-1} x(T - (t - i \cdot d)) \cdot w_{\text{backward}}(i)$. These two operations are implemented using separate convolutional layers (convforward and convbackward), and their outputs are summed to form the final bidirectional result: $y_{\text{bi}}(t) = y_{\text{forward}}(t) + \text{flip}(y_{\text{backward}}(t))$. By integrating both directions under strict causality constraints, HM-BiTCN achieves superior temporal dependency modeling compared to unidirectional causal approaches.

## D.3. Multi-Scale Feature Learning and Residual Connections

To further improve the model's capacity to capture dependencies at different temporal scales, HM-BiTCN incorporates *multi-scale feature learning* and *residual connections*. Multi-scale Feature Learning: In HM-BiTCN, we employ a hierarchy of dilation factors that decrease layer by layer to capture temporal dependencies at multiple scales. Lower layers use larger dilation factors to expand the receptive field, aggregating long-range information and smoothing short-term noise in highly redundant medical time series; higher layers use smaller dilation factors to focus on local dependencies and capture fine-grained features. This coarse-to-fine, global-to-local design enables the network to extract broad patterns in its initial layers and refine precise details in its later layers, thereby enhancing adaptability across a wide range of time series tasks. Residual connections: Residual connections (He et al., 2016) are introduced between the dilated convolutional layers to facilitate the efficient flow of information through the network. The residual connection is defined as $y = F(x) + x$, where $F(x)$ is the convolutional output, and $x$ is the input. This design alleviates the vanishing gradient problem and improves the overall stability of the network during training.

# E. Pseudocode of CIF Method and Key Components of HM-BiTCN

---

**Algorithm 1** CIF Module: Channel-Imposed Fusion with Physiological Ordering

---

**Require:** Encoded features $X \in \mathbb{R}^{B \times T \times C}$, fusion length $n$, fusion weights $a, b$, mode flag $t$
**Ensure:** Fused features $X' \in \mathbb{R}^{B \times T \times C}$
 1: **if** channels of $X$ follow a physiological ordering **then**
 2:     Reorder channels of $X$ according to physiological topology
 3: **end if**
 4: Extract front segment $F \leftarrow X_{[:,:,1:n]}$
 5: Extract back segment $B \leftarrow X_{[:,:,C-n+1:C]}$
 6: Compute fused representation $Y \leftarrow a \cdot F + b \cdot B$
 7: Initialize output $X' \leftarrow X$
 8: **if** $t > 0$ **then**
 9:     Replace front channels of $X'$ with $Y$
10: **else**
11:     Replace back channels of $X'$ with $Y$
12: **end if**
13: **Output:** $X'$

---

**Algorithm 2** Model with CCA-based Feature Interaction

---

**Require:** Encoded features $X \in \mathbb{R}^{B \times L \times D}$, number of CCA components $n$
**Ensure:** Updated encoded features $X'$
 1: Extract front features $F \leftarrow X_{[:,:,1:n]}$
 2: Extract back features $B \leftarrow X_{[:,:,D-n+1:D]}$
 3: Flatten features:
 4:     $\tilde{F} \leftarrow \mathrm{reshape}(F, [B \cdot L, n])$
 5:     $\tilde{B} \leftarrow \mathrm{reshape}(B, [B \cdot L, n])$
 6: Compute canonical correlations without gradient update
 7:     $(U_1, V_1, \rho_1) \leftarrow \mathrm{CCA}(\tilde{F}, \tilde{B})$
 8:     $(U_2, V_2, \rho_2) \leftarrow \mathrm{CCA}(\tilde{B}, \tilde{F})$
 9: Initialize output $X' \leftarrow X$
10: Update front channels of $X'$:
11:     $X'_{[:,:,1:n]} \leftarrow F \odot \rho_1 + B$
12: Update back channels of $X'$:
13:     $X'_{[:,:,D-n+1:D]} \leftarrow F + B \odot \rho_2$
14: **Output:** $X'$

---

---

**Algorithm 3** Canonical Correlation Analysis (CCA)

---

**Require:** Centered data matrices $X \in \mathbb{R}^{N \times D_1}, Y \in \mathbb{R}^{N \times D_2}$, number of components $k$, regularization coefficient $\lambda$
**Ensure:** Canonical projections $X_c, Y_c$ and correlation scores $\rho$
 1: Compute covariance matrices:
 2:     $C_{xx} \leftarrow \frac{1}{N-1} X^\top X + \lambda I$
 3:     $C_{yy} \leftarrow \frac{1}{N-1} Y^\top Y + \lambda I$
 4:     $C_{xy} \leftarrow \frac{1}{N-1} X^\top Y$
 5: Compute whitening transforms:
 6:     $W_x \leftarrow C_{xx}^{-1/2}$
 7:     $W_y \leftarrow C_{yy}^{-1/2}$
 8: Form correlation matrix:
 9:     $T \leftarrow W_x C_{xy} W_y^\top$
10: Compute singular value decomposition:
11:     $(U, \Sigma, V^\top) \leftarrow \mathrm{SVD}(T)$
12: Select top $k$ components:
13:     $A \leftarrow U_{[:,1:k]}$
14:     $B \leftarrow V_{[:,1:k]}$
15:     $\rho \leftarrow \mathrm{diag}(\Sigma_{1:k})$
16: Compute canonical variables:
17:     $X_c \leftarrow X W_x A$
18:     $Y_c \leftarrow Y W_y B$
19: **Output:** $X_c, Y_c, \rho$

---

---

**Algorithm 4** BidirectionalCausalConv

---

**Require:** Input $x \in \mathbb{R}^{B \times C \times T}$, kernel size $k$, dilations $d_f, d_b$
 1: Compute $p_f \leftarrow (k-1) \cdot d_f$
 2: Compute $p_b \leftarrow (k-1) \cdot d_b$
 3: $x_f \leftarrow \mathrm{PadLeft}(x, p_f)$
 4: $x_b \leftarrow \mathrm{PadLeft}(\mathrm{Flip}(x), p_b)$
 5: $y_f \leftarrow \mathrm{Conv1D}(x_f, \mathrm{dilation} = d_f)$
 6: $y_b \leftarrow \mathrm{Flip}(\mathrm{Conv1D}(x_b, \mathrm{dilation} = d_b))$
 7: **Output:** $y_f + y_b$

---

---

**Algorithm 5** BidirectionalDilatedConvBlock

---

**Require:** Input $x$, channels $C_{in}, C_{out}$, kernel size $k$, dilation $d$
 1: **if** $C_{in} \neq C_{out}$ or final layer **then**
 2:     $res \leftarrow \mathrm{Conv1D}(x, \mathrm{kernel} = 1)$
 3: **else**
 4:     $res \leftarrow x$
 5: **end if**
 6: $x \leftarrow \mathrm{GELU}(x)$
 7: $x \leftarrow \mathrm{BidirectionalCausalConv}(x, k, d, d)$
 8: $x \leftarrow \mathrm{GELU}(x)$
 9: $x \leftarrow \mathrm{BidirectionalCausalConv}(x, k, d, d)$
10: **Output:** $x + res$

---

# F. Ablation Experiments of the HM-BiTCN Structure

*Table 6.* The ablation experiments of the HM-BiTCN structure, where "Forward" indicates using only the forward part, and "Backward" indicates using only the backward part.

| Datasets | Models | CIF | Forward (vanilla TCN) | Backward | Accuracy ↑ | Precision ↑ | Recall ↑ | F1 score ↑ | AUROC ↑ | AUPRC ↑ |
|---|---|---|---|---|---|---|---|---|---|---|
| APAVA (2-Classes) | HM-BiTCN | | ✓ | | $82.31_{\pm2.34}$ | $83.29_{\pm2.50}$ | $80.39_{\pm2.65}$ | $81.02_{\pm2.63}$ | $91.50_{\pm1.80}$ | $91.66_{\pm1.82}$ |
| | HM-BiTCN | | | ✓ | $79.45_{\pm3.51}$ | $80.69_{\pm3.04}$ | $77.14_{\pm4.47}$ | $77.58_{\pm4.75}$ | $87.95_{\pm3.82}$ | $88.41_{\pm3.74}$ |
| | HM-BiTCN | | ✓ | ✓ | $82.49_{\pm1.40}$ | $82.38_{\pm1.79}$ | $81.20_{\pm1.32}$ | $81.60_{\pm1.39}$ | $91.10_{\pm1.63}$ | $91.30_{\pm1.71}$ |
| APAVA (2-Classes) | HM-BiTCN | ✓ | ✓ | | $80.43_{\pm5.60}$ | $80.46_{\pm5.23}$ | $79.56_{\pm5.98}$ | $79.50_{\pm5.97}$ | $89.23_{\pm4.44}$ | $89.62_{\pm4.25}$ |
| | HM-BiTCN | ✓ | | ✓ | $79.39_{\pm3.44}$ | $79.49_{\pm3.76}$ | $78.09_{\pm2.94}$ | $78.35_{\pm3.33}$ | $87.62_{\pm3.09}$ | $88.11_{\pm2.91}$ |
| | HM-BiTCN | ✓ | ✓ | ✓ | $85.16_{\pm1.55}$ | $84.76_{\pm1.62}$ | $85.33_{\pm1.27}$ | $84.82_{\pm1.49}$ | $94.06_{\pm1.07}$ | $94.21_{\pm0.99}$ |
| ADFTD (3-Classes) | HM-BiTCN | | ✓ | | $53.32_{\pm1.35}$ | $52.01_{\pm1.54}$ | $51.46_{\pm2.19}$ | $51.21_{\pm1.99}$ | $70.78_{\pm1.78}$ | $53.16_{\pm2.20}$ |
| | HM-BiTCN | | | ✓ | $52.80_{\pm1.18}$ | $50.16_{\pm0.77}$ | $49.23_{\pm1.22}$ | $49.24_{\pm1.02}$ | $68.65_{\pm0.71}$ | $49.95_{\pm0.97}$ |
| | HM-BiTCN | | ✓ | ✓ | $52.05_{\pm2.22}$ | $50.45_{\pm3.00}$ | $50.40_{\pm2.55}$ | $49.48_{\pm2.70}$ | $69.43_{\pm2.84}$ | $50.99_{\pm3.15}$ |
| ADFTD (3-Classes) | HM-BiTCN | ✓ | ✓ | | $56.06_{\pm0.47}$ | $53.21_{\pm1.03}$ | $53.54_{\pm1.36}$ | $52.82_{\pm1.33}$ | $72.93_{\pm0.88}$ | $55.71_{\pm1.03}$ |
| | HM-BiTCN | ✓ | | ✓ | $56.54_{\pm1.33}$ | $54.28_{\pm0.96}$ | $54.63_{\pm1.06}$ | $53.91_{\pm1.11}$ | $73.46_{\pm1.17}$ | $56.12_{\pm1.61}$ |
| | HM-BiTCN | ✓ | ✓ | ✓ | $58.56_{\pm0.93}$ | $55.65_{\pm0.81}$ | $55.86_{\pm0.79}$ | $55.42_{\pm0.82}$ | $76.07_{\pm0.59}$ | $59.75_{\pm0.67}$ |
| TDBrain (2-Classes) | HM-BiTCN | | ✓ | | $87.23_{\pm2.87}$ | $87.75_{\pm2.48}$ | $87.23_{\pm2.87}$ | $87.17_{\pm2.93}$ | $95.55_{\pm1.69}$ | $95.73_{\pm1.60}$ |
| | HM-BiTCN | | | ✓ | $86.92_{\pm3.46}$ | $87.41_{\pm3.17}$ | $86.92_{\pm3.46}$ | $86.86_{\pm3.51}$ | $95.28_{\pm1.78}$ | $95.42_{\pm1.70}$ |
| | HM-BiTCN | | ✓ | ✓ | $84.90_{\pm2.60}$ | $86.02_{\pm2.00}$ | $84.90_{\pm2.60}$ | $84.76_{\pm2.74}$ | $93.94_{\pm1.92}$ | $94.20_{\pm1.85}$ |
| TDBrain (2-Classes) | HM-BiTCN | ✓ | ✓ | | $93.29_{\pm1.73}$ | $93.34_{\pm1.73}$ | $93.29_{\pm1.73}$ | $93.29_{\pm1.73}$ | $98.50_{\pm0.63}$ | $98.56_{\pm0.60}$ |
| | HM-BiTCN | ✓ | | ✓ | $93.69_{\pm1.52}$ | $93.83_{\pm1.42}$ | $93.69_{\pm1.52}$ | $93.68_{\pm1.53}$ | $98.56_{\pm0.67}$ | $98.59_{\pm0.64}$ |
| | HM-BiTCN | ✓ | ✓ | ✓ | $93.13_{\pm1.41}$ | $93.33_{\pm1.37}$ | $93.13_{\pm1.41}$ | $93.12_{\pm1.42}$ | $98.62_{\pm0.66}$ | $98.68_{\pm0.63}$ |
| PTB (2-Classes) | HM-BiTCN | | ✓ | | $82.56_{\pm1.74}$ | $86.16_{\pm1.51}$ | $74.91_{\pm2.88}$ | $77.24_{\pm2.92}$ | $95.69_{\pm0.64}$ | $94.56_{\pm0.76}$ |
| | HM-BiTCN | | | ✓ | $81.07_{\pm4.24}$ | $85.36_{\pm2.71}$ | $72.50_{\pm6.59}$ | $74.33_{\pm6.71}$ | $92.83_{\pm2.38}$ | $91.28_{\pm2.79}$ |
| | HM-BiTCN | | ✓ | ✓ | $81.87_{\pm1.87}$ | $86.50_{\pm1.24}$ | $73.49_{\pm2.90}$ | $75.84_{\pm3.20}$ | $94.20_{\pm0.29}$ | $93.04_{\pm0.45}$ |
| PTB (2-Classes) | HM-BiTCN | ✓ | ✓ | | $87.33_{\pm1.41}$ | $90.26_{\pm1.24}$ | $81.64_{\pm2.04}$ | $84.19_{\pm1.97}$ | $96.21_{\pm1.30}$ | $95.67_{\pm1.52}$ |
| | HM-BiTCN | ✓ | | ✓ | $84.35_{\pm2.28}$ | $87.42_{\pm2.07}$ | $77.54_{\pm3.30}$ | $79.98_{\pm3.41}$ | $91.25_{\pm1.92}$ | $90.42_{\pm2.28}$ |
| | HM-BiTCN | ✓ | ✓ | ✓ | $88.29_{\pm1.45}$ | $90.66_{\pm1.48}$ | $83.21_{\pm2.02}$ | $85.59_{\pm1.96}$ | $94.28_{\pm0.93}$ | $93.78_{\pm1.11}$ |
| FLAAP (10-Classes) | HM-BiTCN | | ✓ | | $70.81_{\pm2.31}$ | $72.58_{\pm1.33}$ | $69.81_{\pm2.79}$ | $70.07_{\pm2.24}$ | $95.89_{\pm0.28}$ | $76.90_{\pm1.21}$ |
| | HM-BiTCN | | | ✓ | $70.29_{\pm2.04}$ | $72.77_{\pm2.09}$ | $68.86_{\pm2.39}$ | $69.56_{\pm1.98}$ | $95.61_{\pm0.26}$ | $76.56_{\pm1.56}$ |
| | HM-BiTCN | | ✓ | ✓ | $76.08_{\pm0.81}$ | $76.05_{\pm0.83}$ | $75.95_{\pm0.84}$ | $75.54_{\pm0.94}$ | $96.49_{\pm0.10}$ | $81.19_{\pm0.65}$ |
| FLAAP (10-Classes) | HM-BiTCN | ✓ | ✓ | | $72.30_{\pm1.50}$ | $72.98_{\pm1.61}$ | $71.65_{\pm1.35}$ | $71.54_{\pm1.50}$ | $95.92_{\pm0.55}$ | $77.87_{\pm2.27}$ |
| | HM-BiTCN | ✓ | | ✓ | $72.81_{\pm1.04}$ | $74.05_{\pm0.80}$ | $71.86_{\pm1.25}$ | $72.12_{\pm1.17}$ | $96.20_{\pm0.21}$ | $79.22_{\pm1.25}$ |
| | HM-BiTCN | ✓ | ✓ | ✓ | $76.82_{\pm1.32}$ | $77.38_{\pm0.85}$ | $76.52_{\pm1.24}$ | $76.39_{\pm1.18}$ | $96.48_{\pm0.06}$ | $81.77_{\pm0.81}$ |
| UCI-HAR (6-Classes) | HM-BiTCN | | ✓ | | $91.94_{\pm0.98}$ | $92.36_{\pm0.90}$ | $92.02_{\pm0.96}$ | $91.98_{\pm0.93}$ | $99.30_{\pm0.08}$ | $97.31_{\pm0.47}$ |
| | HM-BiTCN | | | ✓ | $93.03_{\pm0.62}$ | $93.28_{\pm0.63}$ | $93.12_{\pm0.60}$ | $93.05_{\pm0.62}$ | $99.36_{\pm0.19}$ | $97.72_{\pm0.46}$ |
| | HM-BiTCN | | ✓ | ✓ | $93.72_{\pm0.73}$ | $94.02_{\pm0.72}$ | $93.75_{\pm0.70}$ | $93.69_{\pm0.76}$ | $99.60_{\pm0.09}$ | $98.31_{\pm0.40}$ |
| UCI-HAR (6-Classes) | HM-BiTCN | ✓ | ✓ | | $92.18_{\pm0.45}$ | $92.42_{\pm0.47}$ | $92.21_{\pm0.44}$ | $92.14_{\pm0.44}$ | $99.17_{\pm0.11}$ | $97.04_{\pm0.15}$ |
| | HM-BiTCN | ✓ | | ✓ | $92.62_{\pm0.90}$ | $92.88_{\pm0.86}$ | $92.68_{\pm0.88}$ | $92.63_{\pm0.89}$ | $99.25_{\pm0.18}$ | $97.15_{\pm0.57}$ |
| | HM-BiTCN | ✓ | ✓ | ✓ | $93.78_{\pm0.32}$ | $94.08_{\pm0.26}$ | $93.79_{\pm0.32}$ | $93.74_{\pm0.34}$ | $99.34_{\pm0.19}$ | $97.60_{\pm0.46}$ |

As described in Section 2 of the Methods, HM-BiTCN is constructed by adding a backward branch to the vanilla TCN architecture; therefore, using only the Forward part is essentially equivalent to the vanilla TCN structure.

From the table 6, it can be observed that when both the Forward and Backward parts of the HM-BiTCN structure are used simultaneously, the performance drops significantly compared to using only one of them individually. We speculate that this is mainly due to the presence of substantial noise within medical time-series data. When both parts of the structure are applied at the same time, it is akin to capturing noise from two different directions simultaneously. Instead of enhancing the representation, this leads to noise accumulation, which ultimately results in degraded performance.

However, after processing the data with CIF , the combination of the Forward and Backward parts of the HM-BiTCN structure eventually outperforms the use of either part alone. This result strongly demonstrates the feature-capturing capability of the HM-BiTCN when both directions are utilized together. It indicates that once noise interference is effectively reduced, the bidirectional structure of HM-BiTCN can better leverage its strengths, thereby improving overall performance.

Further observations show that using only the Forward part of HM-BiTCN outperforms the Backward part. This is closely related to the inherent unidirectionality of medical time-series signals such as EEG and ECG, where information typically propagates forward in time (e.g., neural signal transmission in EEG or atrial-to-ventricular activation in ECG). Such

characteristics enable the Forward structure to capture key features and temporal evolution more effectively, yielding better performance. This finding not only deepens the understanding of medical signal processing but also provides insights for optimizing HM-BiTCN in related applications.

To evaluate the performance of our method on general time series, we follow the design of Medformer (Wang et al., 2024a) and test it on two human activity recognition (HAR) datasets: FLAAP(13,123 samples, 10 classes) (Kumar & Suresh, 2022) and UCI-HAR(10,299 samples, 6 classes) (Anguita et al., 2013).

Additionally, on the non-medical datasets FLAAP and UCI-HAR, we observed that integrating the bidirectional structure significantly improves performance. This indicates that in high-SNR scenarios, bidirectional modeling can more effectively capture both forward and backward feature information, enhancing overall model performance. In contrast, CIF provides relatively limited gains on these high-SNR datasets. This observation further highlights the design advantage of CIF: it is specifically tailored for low-SNR medical time series, fusing inter-channel physiological information to enhance signal quality and discriminative power, while its marginal benefit is smaller for low-noise non-medical data. Overall, these findings not only reveal the differential adaptability of model architectures under varying data characteristics but also underscore the unique value of CIF in complex medical scenarios.

## G. Further exploration of physiological structures

The parameters $(a, b, t, n)$ in CIF are explicit hyperparameters that can be directly set and adjusted based on experience. For example, Figure. 8 shows the corresponding locations of EEG channels on the human brain, we have adjusted the AFAVA dataset, which comprises 16 channels: C3, C4, F3, F4, F7, F8, Fp1, Fp2, O1, O2, P3, P4, T3, T4, T5, and T6. For the first six channels, we performed pairwise fusion as follows:

$$C3_{new} = a \cdot C3 + b \cdot C4,$$
$$F3_{new} = a \cdot F3 + b \cdot F4,$$
$$F7_{new} = a \cdot F7 + b \cdot F8.$$

Here, the C3, F3, and F7 channels correspond to C4, F4, and F8, respectively, exhibiting left–right physiological symmetry, and they also belong to the same functional region. We refer to this type of fusion as *Left–Right Physiological Symmetry Fusion* (LR-PSF). In contrast, the channel fusion strategy introduced earlier in this paper, controlled by the hyperparameter $n$, reflects fusion across different functional regions, and is therefore termed *Functional Region Fusion* (FRF).

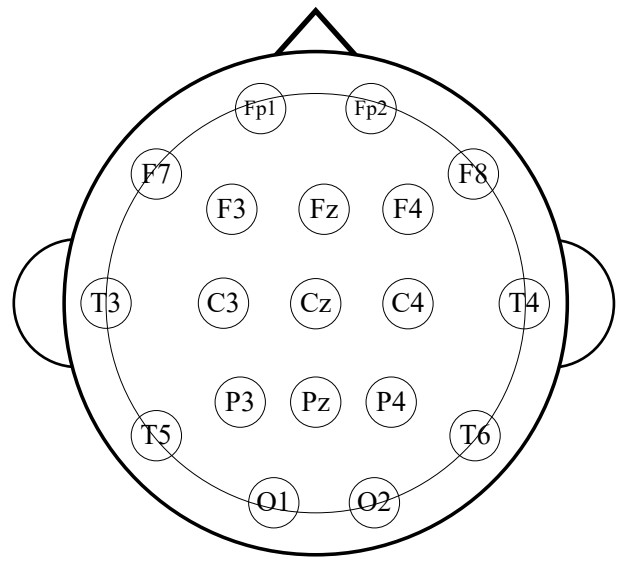

*Figure 8.* Physiological Placement Diagram of EEG Channels.

*Table 7.* **Results of Subject-Independent Setup. APAVA Dataset**

| Datasets | Models | Accuracy ↑ | Precision ↑ | Recall ↑ | F1 score ↑ | AUROC ↑ | AUPRC ↑ |
|---|---|---|---|---|---|---|---|
| **APAVA** | HM-BiTCN | $82.49_{\pm1.40}$ | $82.38_{\pm1.79}$ | $81.20_{\pm1.32}$ | $81.60_{\pm1.39}$ | $91.10_{\pm1.63}$ | $91.30_{\pm1.71}$ |
| (2-Classes) | HM-BiTCN + CIF (-Fp1 + Fp2) | $82.70_{\pm1.90}$ | $83.34_{\pm1.96}$ | $80.73_{\pm2.14}$ | $81.47_{\pm2.13}$ | $91.24_{\pm2.25}$ | $91.62_{\pm2.08}$ |
| | HM-BiTCN + CIF (FRF) | $86.30_{\pm1.05}$ | $86.16_{\pm1.09}$ | $85.47_{\pm1.12}$ | $85.71_{\pm1.09}$ | $94.26_{\pm0.54}$ | $94.42_{\pm0.49}$ |
| | HM-BiTCN + CIF (LR-PSF) | $86.23_{\pm2.09}$ | $85.82_{\pm2.14}$ | $\mathbf{86.04}_{\pm2.06}$ | $85.83_{\pm2.12}$ | $94.59_{\pm1.08}$ | $94.64_{\pm1.08}$ |
| | HM-BiTCN + CIF (CCA) | $\mathbf{87.91}_{\pm1.60}$ | $\mathbf{89.81}_{\pm0.93}$ | $85.85_{\pm2.12}$ | $\mathbf{86.93}_{\pm1.91}$ | $\mathbf{96.97}_{\pm0.52}$ | $\mathbf{96.91}_{\pm0.49}$ |

Building on these two types of physiologically motivated fusion, we further introduce a data-driven fusion mechanism based on Canonical Correlation Analysis (CCA) (Hotelling, 1992) (For detailed pseudocode, see Code 2 3.). Specifically, for each input sample we first select the first $n$ channels and the last $n$ channels, denoted as

$$\mathbf{X}^{front} \in \mathbb{R}^{B \times L \times n}, \qquad \mathbf{X}^{back} \in \mathbb{R}^{B \times L \times n},$$

where $B$ is the batch size, $L$ is the sequence length, and $n$ is the number of paired channels. We then reshape them into two matrices

$$\mathbf{F} \in \mathbb{R}^{(BL) \times n}, \qquad \mathbf{B} \in \mathbb{R}^{(BL) \times n},$$

by stacking all temporal positions and samples along the first dimension.

We apply a CCA module to these two matrices and obtain their canonical projections

$$\mathbf{F}_c, \mathbf{B}_c \in \mathbb{R}^{(BL) \times n}.$$

For each canonical component $i \in \{1, \ldots, n\}$, we compute the Pearson correlation coefficient between the corresponding canonical variables $\mathbf{F}_c^{(i)}$ and $\mathbf{B}_c^{(i)}$, yielding a correlation score vector

$$\boldsymbol{\rho} = (\rho_1, \ldots, \rho_n)^\top \in \mathbb{R}^n.$$

These scores quantify the statistical dependence between the paired channels and are used as adaptive, channel-wise fusion weights.

Concretely, we broadcast $\rho$ back to the original tensor shape and construct two fused channel groups:

$$\widetilde{\mathbf{X}}^{\text{front}} = \mathbf{X}^{\text{front}} \odot \rho + \mathbf{X}^{\text{back}}, \qquad \widetilde{\mathbf{X}}^{\text{back}} = \mathbf{X}^{\text{front}} + \mathbf{X}^{\text{back}} \odot \rho',$$

where $\odot$ denotes element-wise (channel-wise) multiplication and $\rho'$ is another correlation score vector obtained by swapping the roles of $\mathbf{F}$ and $\mathbf{B}$ in CCA. The fused tensors $\widetilde{\mathbf{X}}^{\text{front}}$ and $\widetilde{\mathbf{X}}^{\text{back}}$ are then written back to the first and last $n$ channels of the encoder input, respectively.

In this way, the proposed CCA-based fusion adaptively strengthens or attenuates each paired channel according to its learned cross-channel correlation, providing a controllable and data-driven mechanism that complements the physiologically defined LR-PSF and FRF fusion strategies.

We consider that a full theoretical and experimental treatment of CCA would require a more extensive discussion. Therefore, here we present it solely as an exploratory method and do not include its results in the main text.

We also examined the relationship between the `Fp1` and `Fp2` channels, as introduced in the Introduction. For this purpose, we conducted experiments using the following fusion approach:

$$\texttt{Fp1}_{\text{new}} = -1 \cdot \texttt{Fp1} + 1 \cdot \texttt{Fp2},$$

which we refer to as CIF $(-\texttt{Fp1} + \texttt{Fp2})$.

The results in the table 7 reveal that explicit fusion leveraging the prior knowledge of channels can more effectively integrate channel features, thereby yielding more accurate classification outcomes. Many previous methods, especially various general time series models, are unable to incorporate such medical prior knowledge in a "controllable" manner.

## H. Results on General Time Series

*Table 8.* Full results for the classification task. ∗. in the Transformers indicates the name of ∗former. We report the classification accuracy (%) as the result.

| Datasets / Models | RNN | | | TCN | Transformers | | | | | | | | | | | MLP | | | CNN | Time Mixer++ | HM-BiTCN | HM-BiTCN +CIF |
|---|---|---|---|---|---|---|---|---|---|---|---|---|---|---|---|---|---|---|---|---|---|---|
| | LSTM (1997) | LSTNet (2018) | LSSL (2022) | TCN (2019) | Trans. (2017) | Re. (2019) | In. (2021) | Pyra. (2021) | Auto. (2021) | Station. (2022) | FED. (2022) | ETS. (2022) | Flow. (2022b) | iTrans. (2024b) | DLinear (2023) | LightTS. (2022) | TiDE (2023) | TimesNet (2022a) | (2025a) | (Ours) | (Ours) |
| EthanolConcentration | 32.3 | 39.9 | 31.1 | 28.9 | 32.7 | 31.9 | 31.6 | 30.8 | 31.6 | 32.7 | 28.1 | 31.2 | 33.8 | 28.1 | 32.6 | 29.7 | 27.1 | 35.7 | 39.9 | 31.9 | 32.3 |
| FaceDetection | 57.7 | 65.7 | 66.7 | 52.8 | 67.3 | 68.6 | 67.0 | 65.7 | 68.4 | 68.0 | 66.0 | 66.3 | 67.6 | 66.3 | 68.0 | 67.5 | 65.3 | 68.6 | 71.8 | 66.8 | 67.2 |
| Handwriting | 15.2 | 25.8 | 24.6 | 53.3 | 32.0 | 27.4 | 32.8 | 29.4 | 36.7 | 31.6 | 28.0 | 32.5 | 33.8 | 24.2 | 27.0 | 26.1 | 23.2 | 32.1 | 26.5 | 49.5 | 51.2 |
| Heartbeat | 72.2 | 77.1 | 72.7 | 75.6 | 76.1 | 77.1 | 80.5 | 75.6 | 74.6 | 73.7 | 73.7 | 71.2 | 77.6 | 75.6 | 75.1 | 75.1 | 74.6 | 78.0 | 79.1 | 74.6 | 77.5 |
| JapaneseVowels | 79.7 | 98.1 | 98.4 | 98.9 | 98.7 | 97.8 | 98.9 | 98.4 | 96.2 | 99.2 | 98.4 | 95.9 | 98.9 | 96.6 | 96.2 | 96.2 | 95.6 | 98.4 | 97.9 | 97.8 | 98.3 |
| PEMS-SF | 39.9 | 86.7 | 86.1 | 68.8 | 82.1 | 82.7 | 81.5 | 83.2 | 82.7 | 87.3 | 80.9 | 86.0 | 83.8 | 87.9 | 75.1 | 88.4 | 86.9 | 89.6 | 91.0 | 82.6 | 86.1 |
| SelfRegulationSCP1 | 68.9 | 84.0 | 90.8 | 84.6 | 92.2 | 90.4 | 90.1 | 88.1 | 84.0 | 89.4 | 88.7 | 89.6 | 92.5 | 90.2 | 87.3 | 89.8 | 89.2 | 91.8 | 93.1 | 89.7 | 91.1 |
| SelfRegulationSCP2 | 46.6 | 52.8 | 52.2 | 55.6 | 53.9 | 56.7 | 53.3 | 53.3 | 50.6 | 57.2 | 54.4 | 55.0 | 56.1 | 54.4 | 50.5 | 51.1 | 53.4 | 57.2 | 65.6 | 61.6 | 62.2 |
| SpokenArabicDigits | 31.9 | 100.0 | 100.0 | 95.6 | 98.4 | 97.0 | 100.0 | 99.6 | 100.0 | 100.0 | 100.0 | 100.0 | 98.8 | 96.0 | 81.4 | 100.0 | 95.0 | 99.0 | 99.8 | 99.5 | 99.6 |
| UWaveGestureLibrary | 41.2 | 87.8 | 85.9 | 88.4 | 85.6 | 85.6 | 85.6 | 83.4 | 85.9 | 87.5 | 85.3 | 85.0 | 86.6 | 85.9 | 82.1 | 80.3 | 84.9 | 85.3 | 88.2 | 92.1 | 92.8 |
| Average Accuracy | 48.6 | 71.8 | 70.9 | 70.3 | 71.9 | 71.5 | 72.1 | 70.8 | 71.1 | 72.7 | 70.7 | 71.0 | 73.0 | 70.5 | 67.5 | 70.4 | 69.5 | 73.6 | 75.3 | 74.6 | **75.8** |

We compared our method with various approaches on the general time series UEA dataset. The results of these methods were provided by TimeMixer++ (Wang et al., 2025a). Experimental results show that CIF can enhance the performance of HM-BiTCN on general time series classification tasks. Moreover, the combination of HM-BiTCN with CIF achieves SOTA performance.

## I. Reproduced Results of Existing Methods

### I.1. Results of Subject-Independent

Table 1 presents the results of various methods reported under the subject-independent setup, while Table 9 shows the results of our reproduction of these methods. Our approach achieves the highest scores on six metrics across four out of the five datasets. For the PTB-XL dataset, compared to the results reported by other methods, our method achieves the highest AUROC, with Accuracy ranking second.

*Table 9.* **Results of Subject-Independent Setup.** The training, validation, and test sets are distributed based on subjects according to a predetermined ratio/IDs. Results of the APAVA, TDBrain, ADFTD, PTB, and PTB-XL datasets under this setup are presented here. Unfortunately, MedGNN has only released the training parameters for the APAVA and ADFTD datasets.

| Datasets | Models | Accuracy ↑ | Precision ↑ | Recall ↑ | F1 score ↑ | AUROC ↑ | AUPRC ↑ |
|---|---|---|---|---|---|---|---|
| **APAVA** (2-Classes) Reproduced | Autoformer | 73.18±7.33 | 73.87±6.72 | 73.01±6.10 | 72.40±7.03 | 81.64±7.24 | 81.10±7.75 |
| | Crossformer | 72.76±2.04 | 79.64±2.45 | 67.41±2.62 | 66.88±3.61 | 71.81±4.06 | 71.64±3.74 |
| | FEDformer | 75.16±1.67 | 74.98±0.69 | 73.34±2.97 | 73.50±2.90 | 83.89±1.54 | 83.27±1.62 |
| | Informer | 72.20±2.78 | 73.92±4.80 | 68.48±2.51 | 68.74±2.70 | 70.14±3.43 | 70.84±3.80 |
| | iTransformer | 74.55±1.66 | 74.77±2.10 | 71.76±1.72 | 72.30±1.79 | 85.59±1.55 | 84.39±1.57 |
| | MTST | 69.24±1.24 | 75.87±2.80 | 63.28±1.81 | 61.62±2.75 | 66.09±3.27 | 68.08±2.93 |
| | Nonformer | 71.81±4.20 | 71.31±4.40 | 70.15±3.38 | 70.38±3.74 | 71.54±2.73 | 72.79±2.50 |
| | PatchTST | 68.27±2.11 | 78.56±1.88 | 61.53±2.60 | 58.52±4.07 | 64.61±2.18 | 67.14±2.06 |
| | Reformer | 78.42±2.85 | 80.89±4.52 | 75.20±2.28 | 76.09±2.54 | 75.48±2.79 | 77.52±2.64 |
| | Transformer | 75.53±4.28 | 76.90±5.05 | 72.14±4.87 | 72.64±5.44 | 72.30±6.04 | 73.04±7.15 |
| | Medformer | 77.85±2.42 | 80.31±3.21 | 74.38±2.49 | 75.21±2.67 | 80.85±3.80 | 81.62±3.24 |
| | MedGNN | 77.40±5.77 | 82.77±4.46 | 73.24±7.06 | 73.29±9.01 | 81.31±2.94 | 82.80±2.91 |
| | **HM-BiTCN + CIF** | **85.16±1.55** | **84.76±1.62** | **85.33±1.27** | **84.82±1.49** | **94.06±1.07** | **94.21±0.99** |
| **TDBrain** (2-Classes) Reproduced | Autoformer | 90.38±3.03 | 91.16±2.42 | 90.38±3.03 | 90.31±3.09 | 95.83±2.14 | 95.43±2.31 |
| | Crossformer | 82.15±2.60 | 82.81±2.11 | 82.15±2.60 | 82.04±2.70 | 91.20±2.23 | 91.47±2.16 |
| | FEDformer | 77.60±1.23 | 78.25±1.52 | 77.60±1.23 | 77.48±1.19 | 86.31±1.23 | 86.48±1.36 |
| | Informer | 88.42±2.99 | 89.01±2.45 | 88.42±2.99 | 88.36±3.05 | 96.54±0.90 | 96.66±0.85 |
| | iTransformer | 74.69±1.02 | 74.76±1.04 | 74.69±1.02 | 74.67±1.02 | 83.35±1.24 | 83.65±1.41 |
| | MTST | 77.67±3.58 | 78.97±4.37 | 77.67±3.58 | 77.45±3.55 | 86.47±4.84 | 84.99±6.43 |
| | Nonformer | 88.10±2.39 | 88.76±1.74 | 88.10±2.39 | 88.04±2.47 | 96.56±0.91 | 96.36±1.21 |
| | PatchTST | 77.98±2.64 | 79.30±3.73 | 77.98±2.64 | 77.76±2.65 | 86.67±4.03 | 84.93±5.47 |
| | Reformer | 88.50±2.30 | 89.01±1.40 | 88.50±2.30 | 88.45±2.23 | 96.10±0.63 | 96.19±0.55 |
| | Transformer | 85.13±1.86 | 86.39±1.56 | 85.13±1.86 | 84.99±1.93 | 95.61±1.05 | 95.63±0.91 |
| | Medformer | 88.77±1.24 | 88.91±1.11 | 88.77±1.24 | 88.76±1.25 | 96.38±0.34 | 96.44±0.30 |
| | MedGNN | - | - | - | - | - | - |
| | **HM-BiTCN + CIF** | **93.13±1.41** | **93.33±1.37** | **93.13±1.41** | **93.12±1.42** | **98.62±0.66** | **98.68±0.63** |
| **ADFTD** (3-Classes) Reproduced | Autoformer | 46.90±2.89 | 45.59±2.37 | 44.91±2.23 | 44.34±2.52 | 63.49±2.44 | 45.63±2.29 |
| | Crossformer | 50.18±1.97 | 45.97±1.84 | 46.30±1.73 | 45.90±1.84 | 66.68±1.67 | 48.65±1.89 |
| | FEDformer | 45.75±0.78 | 45.71±1.29 | 44.27±1.28 | 43.51±1.00 | 62.64±1.64 | 45.88±1.35 |
| | Informer | 48.42±1.99 | 46.94±1.60 | 46.41±0.99 | 45.76±0.43 | 65.99±1.14 | 47.49±1.07 |
| | iTransformer | 52.85±1.36 | 46.97±1.05 | 47.31±1.03 | 46.84±0.78 | 67.46±0.96 | 49.90±0.89 |
| | MTST | 45.77±1.70 | 44.39±1.73 | 43.70±1.82 | 43.36±1.98 | 61.38±1.57 | 44.01±1.60 |
| | Nonformer | 50.81±1.06 | 48.71±1.40 | 48.55±1.47 | 48.36±1.38 | 66.95±1.54 | 48.08±1.82 |
| | PatchTST | 43.32±0.53 | 41.95±0.38 | 41.45±1.26 | 40.75±1.62 | 60.21±0.29 | 42.49±0.57 |
| | Reformer | 51.28±2.60 | 49.68±2.75 | 49.64±2.02 | 48.45±2.06 | 69.20±2.53 | 51.74±3.24 |
| | Transformer | 50.53±0.94 | 49.31±0.87 | 48.57±1.23 | 48.42±1.28 | 67.98±0.90 | 49.07±1.35 |
| | Medformer | 53.70±1.18 | 51.51±1.32 | 50.49±1.48 | 50.35±1.53 | 70.48±1.17 | 50.91±1.13 |
| | MedGNN | 50.22±3.21 | 48.65±3.72 | 47.50±4.57 | 47.33±4.40 | 67.18±4.39 | 48.84±4.11 |
| | **HM-BiTCN + CIF** | **58.56±0.93** | **55.65±0.81** | **55.86±0.79** | **55.42±0.82** | **76.07±0.59** | **59.75±0.67** |
| **PTB** (2-Classes) Reproduced | Autoformer | 71.99±2.74 | 69.60±3.85 | 61.50±4.23 | 61.43±5.07 | 74.29±1.89 | 70.26±2.00 |
| | Crossformer | 78.06±3.44 | 81.53±3.13 | 68.62±5.63 | 69.76±6.53 | 88.31±2.07 | 85.81±2.43 |
| | FEDformer | 74.54±2.27 | 77.99±4.10 | 63.14±3.29 | 63.28±4.36 | 84.63±4.27 | 80.91±5.55 |
| | Informer | 79.59±0.65 | 83.33±0.77 | 70.58±0.95 | 72.58±1.08 | 92.77±0.48 | 90.89±0.57 |
| | iTransformer | 83.43±1.19 | 88.06±1.47 | 75.64±1.55 | 78.29±1.70 | 91.38±1.41 | 91.08±1.30 |
| | MTST | 75.53±2.45 | 78.72±1.87 | 64.78±4.06 | 65.30±4.81 | 87.76±4.09 | 83.60±3.92 |
| | Nonformer | 78.93±1.46 | 82.48±1.53 | 69.68±2.15 | 71.50±2.56 | 90.54±0.59 | 87.78±1.46 |
| | PatchTST | 75.28±2.44 | 77.05±2.44 | 64.86±4.05 | 65.41±5.28 | 88.11±2.59 | 82.65±2.87 |
| | Reformer | 78.11±1.65 | 82.70±0.80 | 68.17±2.68 | 69.68±3.34 | 90.77±1.56 | 88.14±1.20 |
| | Transformer | 76.43±1.98 | 81.25±1.15 | 65.64±3.29 | 66.44±4.39 | 90.21±1.24 | 87.28±1.49 |
| | Medformer | 80.99±0.75 | 83.01±0.72 | 73.35±1.16 | 75.47±1.21 | 93.10±1.18 | 90.69±1.04 |
| | MedGNN | - | - | - | - | - | - |
| | **HM-BiTCN + CIF** | **88.29±1.45** | **90.66±1.48** | **83.21±2.02** | **85.59±1.96** | **94.28±0.93** | **93.78±1.11** |
| **PTB-XL** (5-Classes) Reproduced | Autoformer | 60.37±1.72 | 49.83±0.50 | 49.17±0.72 | 48.43±0.47 | 81.35±0.56 | 50.83±0.72 |
| | Crossformer | 73.15±0.17 | 64.62±0.31 | **61.28±0.45** | **62.49±0.34** | 89.94±0.15 | 67.14±0.29 |
| | FEDformer | 56.01±10.27 | 51.61±5.58 | 48.77±7.64 | 47.23±8.76 | 82.01±4.20 | 52.23±7.14 |
| | Informer | 71.28±0.27 | 62.28±0.53 | 59.18±0.54 | 60.38±0.38 | 88.57±0.09 | 64.57±0.18 |
| | iTransformer | 69.18±0.33 | 59.44±0.48 | 54.84±0.25 | 56.44±0.23 | 86.65±0.17 | 60.27±0.44 |
| | MTST | 72.14±0.29 | 64.01±0.61 | 59.31±0.27 | 61.07±0.32 | 88.82±0.19 | 65.52±0.52 |
| | Nonformer | 70.51±0.79 | 61.42±1.03 | 58.15±0.57 | 59.31±0.63 | 88.20±0.43 | 63.33±0.77 |
| | PatchTST | 73.13±0.20 | 65.38±0.54 | 60.61±0.56 | 62.39±0.27 | 89.66±0.16 | 66.97±0.24 |
| | Reformer | 71.24±0.46 | 61.78±0.79 | 59.67±0.99 | 60.51±0.65 | 88.67±0.21 | 64.29±0.29 |
| | Transformer | 70.46±0.42 | 61.56±0.58 | 57.86±0.65 | 59.16±0.46 | 88.18±0.14 | 63.25±0.27 |
| | Medformer | 72.94±0.15 | 64.39±0.32 | 59.98±0.49 | 61.61±0.33 | 89.67±0.10 | 66.25±0.25 |
| | MedGNN | - | - | - | - | - | - |
| | **HM-BiTCN + CIF** | **73.73±0.30** | **65.41±0.67** | 60.70±1.08 | 61.89±0.91 | **90.53±0.22** | **67.75±0.75** |

In comparison with our reproduced results, our method ranks first in Accuracy, AUROC, and AUPRC, and second in Precision and Recall. Additionally, it is noteworthy that under the subject-independent setup, the F1 score for ADFTD is 55.42%.

## J. Ablation Study

### J.1. Module Study of CIF

We conduct experiments by integrating our Channel-Imposed Fusion principle with other methods.

In the CIF structure, there are four different hyperparameters. First, $t$ is the switch for forward and backward feature fusion. When $t = 1$, it replaces the forward features with the fused features; when $t = -1$, it replaces the backward features with the fused features. The second hyperparameter, $n$, determines the number of selected channels. Additionally, there are two scaling factors associated with the channels: $a$, the scaling factor for the first $n$ channels, and $b$, the scaling factor for the remaining $n$ channels. Both $a$ and $b$ can either be learnable or fixed parameters. We use 🔥 to represent learnable parameters, and ❄ to represent non-learnable parameters.

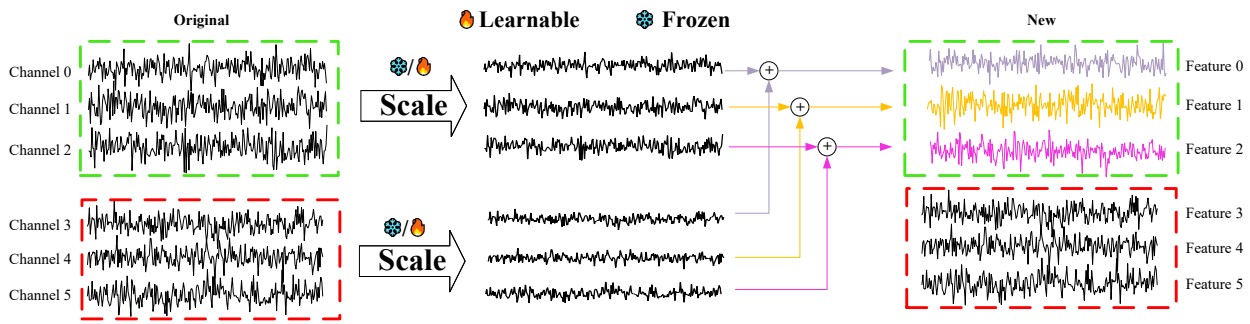

*Figure 9.* The case of $X_i^{new} = X_{fused}$ is marked by setting $t = 1$.

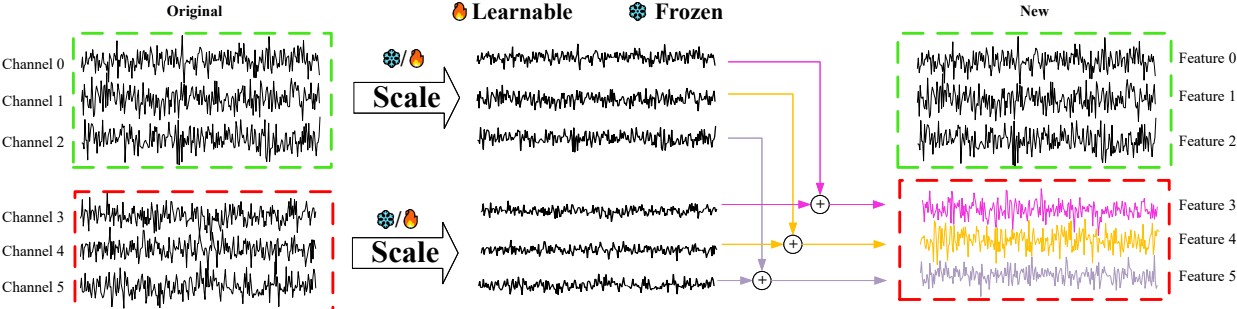

*Figure 10.* The case of $X_j^{new} = X_{fused}$ is marked by setting $t = -1$.

Following the theoretical analysis of the parameters $a$ and $b$ in Appendix A, we perform an evenly spaced grid search over the range $[-1, 1] \times [-1, 1]$ and report the overall performance trends as these hyperparameters vary

The ablation study(Table 10 11 12) reveals that the performance of the CIF architecture is highly sensitive to its hyperparameters, with optimal configurations varying significantly across datasets. For the APAVA dataset (16 channels), the best results (86.30% accuracy, 94.26% AUROC) are achieved using backward fusion ($t = -1$) on 9 selected channels ($n = 9$), where the first 7 channels apply fixed reinforcement weights ($a = -0.8$, learnable) and the rest use fixed suppression weights ($b = -0.6$, learnable). Conversely, the ADFTD dataset (19 channels) performs best (58.56% accuracy, 76.07% AUROC) with forward fusion ($t = 1$) over 10 selected channels ($n = 10$) using dynamically learned weights for negative suppression ($a = -0.19$) and positive enhancement ($b = 0.27$). These differences, also observed in PTB (APAVA: 78–85% accuracy vs. ADFTD: 50–58%), emphasize the critical role of dataset-specific tuning. In particular, the number of selected channels ($n$) and fusion direction ($t$) are pivotal, while adaptive weighting ($a/b$) further improves robustness. Overall, the results

highlight the necessity of careful hyperparameter optimization—especially in channel selection and fusion strategy—for maximizing CIF performance across diverse medical time series tasks.

Overall, the number of selected channels $n$ and the fusion direction $t$ are the key factors influencing performance, while the adaptive weights $a/b$ further enhance the robustness and generalization ability of the model. Taken together, these results indicate that, for diverse medical time-series tasks, fully leveraging the advantages of CIF requires careful and systematic optimization of hyperparameters such as channel selection and fusion strategy.

*Table 10.* Ablation study of CIF structure hyperparameters on ADFTD dataset (19 Channels). We use 🔥 to represent learnable parameters, and ❄ to represent non-learnable parameters.

| t | n | a | b | Accuracy | Precision | Recall | F1 score | AUROC | AUPRC |
|---|---|---|---|---|---|---|---|---|---|
| - | - | - | - | 52.05±2.22 | 50.45±3.00 | 50.40±2.55 | 49.48±2.70 | 69.43±2.84 | 50.99±3.15 |
| 1 | 10 | -0.20 ❄ | -0.30 ❄ | 57.33±0.95 | 54.06±1.47 | 54.44±1.33 | 53.73±1.47 | 74.82±1.36 | 58.24±1.78 |
| 1 | 10 | -0.25 ❄ | -0.40 ❄ | 57.08±1.46 | 53.31±0.98 | 53.24±1.14 | 52.87±0.98 | 73.77±1.28 | 56.86±1.47 |
| 1 | 11 | -0.20 ❄ | -0.30 ❄ | 56.68±1.30 | 53.04±2.00 | 53.45±2.03 | 52.84±2.30 | 73.81±1.55 | 57.34±1.46 |
| 1 | 10 | -0.19 🔥 | -0.27 🔥 | **58.56±0.93** | **55.65±0.81** | **55.86±0.79** | **55.42±0.82** | **76.07±0.59** | **59.75±0.67** |
| 1 | 7 | 0.20 🔥 | -0.60 🔥 | 56.36±0.58 | 53.91±1.10 | 54.34±1.03 | 53.78±0.91 | 74.19±0.69 | 56.62±1.28 |
| 1 | 9 | -0.20 🔥 | -0.60 🔥 | 55.20±1.50 | 53.05±1.14 | 53.21±1.06 | 52.76±1.02 | 73.09±1.13 | 55.50±1.71 |
| 1 | 10 | -0.20 🔥 | -0.20 🔥 | 57.11±0.92 | 55.04±1.43 | 55.53±1.57 | 54.58±1.29 | 75.13±1.33 | 58.47±1.43 |
| 1 | 11 | -0.20 🔥 | -0.30 🔥 | 57.83±1.16 | 54.32±1.22 | 54.51±1.18 | 54.01±1.21 | 74.74±1.12 | 57.96±1.08 |
| 1 | 11 | -0.18 🔥 | -0.25 🔥 | 57.00±1.32 | 53.00±2.28 | 53.17±1.94 | 52.32±2.19 | 73.97±1.89 | 57.00±1.93 |
| 1 | 7 | -1.00 ❄ | -1.00 ❄ | 51.29±1.14 | 49.72±1.46 | 49.95±1.65 | 49.35±1.44 | 69.13±0.82 | 51.01±1.19 |
| 1 | 7 | -1.00 🔥 | -1.00 🔥 | 50.46±1.57 | 48.50±1.19 | 48.68±1.29 | 48.20±1.38 | 68.18±1.08 | 49.73±1.24 |
| 1 | 7 | -1.00 ❄ | -0.80 ❄ | 50.62±2.38 | 49.34±2.17 | 49.49±2.18 | 48.97±2.34 | 68.55±1.78 | 50.52±2.40 |
| 1 | 7 | -1.00 🔥 | -0.80 🔥 | 50.51±2.57 | 49.04±2.31 | 49.26±2.60 | 48.73±2.66 | 68.32±2.13 | 50.38±2.53 |
| 1 | 7 | -1.00 ❄ | -0.60 ❄ | 51.51±1.56 | 50.15±1.36 | 50.35±1.03 | 49.80±1.24 | 69.03±0.92 | 51.09±1.48 |
| 1 | 7 | -1.00 🔥 | -0.60 🔥 | 51.09±2.05 | 49.84±1.97 | 50.24±2.16 | 49.33±1.84 | 69.04±1.83 | 51.15±2.55 |
| 1 | 7 | -1.00 ❄ | -0.40 ❄ | 51.17±1.27 | 48.47±1.03 | 48.45±1.36 | 47.83±1.38 | 68.20±0.79 | 49.55±1.06 |
| 1 | 7 | -1.00 ❄ | -0.40 ❄ | 53.59±1.01 | 52.16±2.37 | 52.16±3.11 | 51.01±3.24 | 70.77±2.29 | 53.17±3.59 |
| 1 | 7 | -1.00 🔥 | -0.40 🔥 | 51.27±0.91 | 48.14±0.91 | 48.28±1.15 | 47.66±1.18 | 67.95±1.05 | 48.92±1.71 |
| 1 | 7 | -1.00 🔥 | -0.40 🔥 | 53.03±1.70 | 52.41±2.70 | 51.95±3.09 | 50.35±3.43 | 70.69±2.34 | 52.99±3.58 |
| 1 | 7 | -1.00 ❄ | -0.20 ❄ | 51.11±0.61 | 48.09±0.51 | 47.88±0.59 | 47.40±0.59 | 67.73±0.67 | 48.74±0.89 |
| 1 | 7 | -1.00 ❄ | -0.20 ❄ | 54.26±0.58 | 52.53±1.78 | 51.58±2.01 | 51.31±2.32 | 70.87±1.43 | 53.25±2.30 |
| 1 | 7 | -1.00 🔥 | -0.20 🔥 | 50.21±2.53 | 47.75±2.54 | 47.84±2.62 | 47.38±2.51 | 67.34±2.67 | 48.72±3.07 |
| 1 | 7 | -1.00 🔥 | -0.20 🔥 | 54.91±1.15 | 53.99±1.35 | 53.54±2.01 | 53.16±1.56 | 72.41±1.23 | 55.67±1.70 |
| 1 | 8 | -1.00 ❄ | -1.00 ❄ | 51.22±0.94 | 49.68±1.50 | 49.67±2.01 | 49.18±1.59 | 68.92±1.00 | 51.02±1.05 |
| 1 | 8 | -1.00 🔥 | -1.00 🔥 | 52.00±1.48 | 50.10±1.12 | 49.87±1.43 | 49.56±1.17 | 69.38±0.97 | 51.20±1.26 |
| 1 | 8 | -1.00 ❄ | -0.80 ❄ | 52.20±1.50 | 49.88±1.71 | 49.90±1.85 | 49.69±1.75 | 69.44±1.36 | 51.39±1.69 |
| 1 | 8 | -1.00 🔥 | -0.80 🔥 | 51.19±2.06 | 47.57±1.30 | 47.61±1.79 | 47.18±1.62 | 67.52±1.78 | 48.94±1.69 |
| 1 | 8 | -1.00 ❄ | -0.60 ❄ | 51.67±1.42 | 47.79±0.92 | 47.97±1.22 | 47.49±1.21 | 67.89±1.31 | 49.35±1.62 |
| 1 | 8 | -1.00 🔥 | -0.60 🔥 | 51.15±1.78 | 47.52±0.70 | 47.29±1.10 | 46.62±0.71 | 67.69±1.50 | 48.90±1.31 |
| 1 | 8 | -1.00 ❄ | -0.40 ❄ | 52.58±1.40 | 49.21±2.48 | 49.35±1.80 | 48.50±2.27 | 68.95±1.57 | 50.26±2.08 |
| 1 | 8 | -1.00 ❄ | -0.40 ❄ | 52.40±0.54 | 50.11±2.26 | 49.92±2.13 | 48.87±2.64 | 69.13±1.85 | 50.62±2.47 |
| 1 | 8 | -1.00 🔥 | -0.40 🔥 | 52.61±1.20 | 49.38±2.01 | 49.07±1.57 | 48.32±1.97 | 68.92±1.50 | 50.01±1.66 |
| 1 | 8 | -1.00 🔥 | -0.40 🔥 | 52.47±1.46 | 49.68±1.96 | 49.59±1.70 | 48.30±2.21 | 68.60±1.70 | 49.58±2.21 |
| 1 | 8 | -1.00 ❄ | -0.20 ❄ | 51.83±1.52 | 48.65±2.57 | 48.79±2.05 | 48.16±2.45 | 67.83±1.86 | 49.21±2.17 |
| 1 | 8 | -1.00 ❄ | -0.20 ❄ | 52.14±1.35 | 50.70±2.37 | 50.25±2.35 | 49.75±2.07 | 69.64±1.73 | 51.02±2.65 |
| 1 | 8 | -1.00 🔥 | -0.20 🔥 | 52.70±1.14 | 50.10±1.76 | 49.66±1.42 | 48.99±1.71 | 69.22±1.06 | 50.20±1.09 |
| 1 | 8 | -1.00 🔥 | -0.20 🔥 | 52.64±0.73 | 49.69±2.78 | 50.02±1.94 | 48.95±2.76 | 69.15±1.81 | 50.69±2.16 |
| 1 | 9 | -1.00 ❄ | -1.00 ❄ | 53.35±1.53 | 49.43±0.75 | 49.65±0.54 | 49.05±1.03 | 69.05±0.60 | 51.18±0.44 |
| 1 | 9 | -1.00 🔥 | -1.00 🔥 | 53.42±2.39 | 50.34±1.98 | 50.23±1.85 | 49.47±2.21 | 69.43±1.40 | 51.76±1.87 |

| t | n | a | b | Accuracy | Precision | Recall | F1 score | AUROC | AUPRC |
|---|---|---|---|---|---|---|---|---|---|
| 1 | 9 | -1.00 ❄ | -0.80 ❄ | 52.60±1.52 | 50.27±0.83 | 50.10±0.85 | 49.49±0.70 | 69.19±0.43 | 51.16±0.76 |
| 1 | 9 | -1.00 🔥 | -0.80 🔥 | 52.41±1.31 | 49.15±0.81 | 49.14±1.22 | 48.13±1.43 | 68.53±0.45 | 50.22±0.82 |
| 1 | 9 | -1.00 ❄ | -0.60 ❄ | 52.18±0.91 | 49.49±1.11 | 49.40±1.37 | 48.90±0.85 | 68.75±1.16 | 50.44±1.61 |
| 1 | 9 | -1.00 🔥 | -0.60 🔥 | 51.92±1.60 | 49.03±0.83 | 48.73±0.78 | 48.13±0.88 | 68.31±0.40 | 49.95±0.46 |
| 1 | 9 | -1.00 ❄ | -0.40 ❄ | 51.34±0.70 | 49.72±0.34 | 49.90±0.42 | 49.32±0.42 | 68.67±0.31 | 50.21±0.90 |
| 1 | 9 | -1.00 ❄ | -0.40 ❄ | 51.36±2.78 | 49.89±1.37 | 49.48±1.07 | 48.83±1.52 | 69.11±1.57 | 50.22±1.88 |
| 1 | 9 | -1.00 🔥 | -0.40 🔥 | 52.24±1.09 | 49.96±0.92 | 49.42±0.99 | 49.01±0.85 | 68.50±0.91 | 49.71±1.22 |
| 1 | 9 | -1.00 🔥 | -0.40 🔥 | 51.23±2.51 | 50.34±0.95 | 49.68±0.70 | 48.94±1.37 | 69.81±1.24 | 51.01±1.47 |
| 1 | 9 | -1.00 ❄ | -0.20 ❄ | 51.30±0.83 | 49.86±0.96 | 49.61±0.99 | 49.00±0.83 | 68.90±0.99 | 50.08±1.43 |
| 1 | 9 | -1.00 ❄ | -0.20 ❄ | 50.63±2.05 | 49.20±1.98 | 48.76±1.75 | 48.08±1.47 | 68.55±1.89 | 49.46±2.35 |
| 1 | 9 | -1.00 🔥 | -0.20 🔥 | 53.22±2.54 | 51.76±2.90 | 50.85±3.02 | 50.22±3.38 | 69.92±2.50 | 51.51±3.36 |
| 1 | 9 | -1.00 🔥 | -0.20 🔥 | 51.03±2.23 | 49.86±1.36 | 49.59±1.59 | 49.03±1.66 | 69.13±1.44 | 50.35±2.08 |
| 1 | 10 | -1.00 ❄ | -1.00 ❄ | 51.76±1.87 | 49.07±2.33 | 48.94±2.07 | 48.35±2.23 | 68.73±1.56 | 50.36±2.20 |
| 1 | 10 | -1.00 🔥 | -1.00 🔥 | 49.94±1.17 | 47.35±2.22 | 47.48±1.98 | 46.89±2.20 | 67.17±0.98 | 48.61±1.84 |
| 1 | 10 | -1.00 ❄ | -0.80 ❄ | 52.94±2.14 | 50.68±2.19 | 50.17±2.00 | 49.79±1.94 | 69.46±1.79 | 51.11±2.61 |
| 1 | 10 | -1.00 🔥 | -0.80 🔥 | 52.13±2.00 | 49.49±2.73 | 49.51±2.54 | 48.81±2.72 | 68.65±1.77 | 50.42±2.08 |
| 1 | 10 | -1.00 ❄ | -0.60 ❄ | 51.87±2.46 | 49.68±2.77 | 49.99±2.84 | 49.24±2.95 | 69.18±2.01 | 51.03±2.77 |
| 1 | 10 | -1.00 🔥 | -0.60 🔥 | 51.86±1.69 | 49.10±1.32 | 49.76±1.38 | 48.69±1.49 | 69.01±1.02 | 50.41±1.35 |
| 1 | 10 | -1.00 ❄ | -0.40 ❄ | 52.66±1.03 | 47.97±1.50 | 48.38±0.97 | 47.34±1.41 | 68.12±1.15 | 49.01±1.38 |
| 1 | 10 | -1.00 ❄ | -0.40 ❄ | 52.51±2.82 | 51.49±3.07 | 51.10±3.57 | 50.33±3.43 | 70.03±3.22 | 51.80±4.02 |
| 1 | 10 | -1.00 🔥 | -0.40 🔥 | 52.41±1.03 | 48.62±1.93 | 49.14±1.60 | 47.99±1.75 | 68.41±1.13 | 49.39±1.37 |
| 1 | 10 | -1.00 🔥 | -0.40 🔥 | 51.49±1.49 | 50.82±1.60 | 50.21±1.68 | 49.51±1.59 | 69.39±1.74 | 50.83±2.24 |
| 1 | 10 | -1.00 ❄ | -0.20 ❄ | 52.93±0.91 | 50.09±2.50 | 49.23±1.64 | 48.63±1.98 | 69.06±1.84 | 50.34±2.30 |
| 1 | 10 | -1.00 ❄ | -0.20 ❄ | 54.29±2.40 | 52.85±2.43 | 51.05±1.91 | 50.86±2.26 | 71.26±2.20 | 53.12±2.65 |
| 1 | 10 | -1.00 🔥 | -0.20 🔥 | 53.17±2.38 | 49.78±3.85 | 50.04±3.26 | 49.42±3.77 | 69.42±3.45 | 50.81±4.85 |
| 1 | 10 | -1.00 🔥 | -0.20 🔥 | 52.73±3.06 | 51.96±2.78 | 51.07±2.84 | 50.37±3.15 | 70.70±2.84 | 52.46±3.47 |
| 1 | 11 | -1.00 ❄ | -1.00 ❄ | 52.58±0.58 | 48.82±1.37 | 49.03±1.32 | 48.54±1.36 | 68.70±0.98 | 50.46±1.21 |
| 1 | 11 | -1.00 🔥 | -1.00 🔥 | 51.62±1.79 | 48.69±2.60 | 48.58±2.66 | 48.48±2.49 | 68.51±1.74 | 50.69±2.33 |
| 1 | 11 | -1.00 ❄ | -0.80 ❄ | 52.30±1.30 | 49.28±1.37 | 48.77±0.61 | 48.05±1.12 | 68.63±1.10 | 50.61±1.37 |
| 1 | 11 | -1.00 🔥 | -0.80 🔥 | 53.35±2.23 | 49.09±2.26 | 49.08±2.12 | 47.91±2.25 | 68.62±1.60 | 50.38±1.64 |
| 1 | 11 | -1.00 ❄ | -0.60 ❄ | 51.63±1.71 | 49.23±1.82 | 48.99±1.43 | 48.42±1.59 | 69.21±1.40 | 51.41±1.71 |
| 1 | 11 | -1.00 🔥 | -0.60 🔥 | 52.79±2.04 | 48.88±3.26 | 49.16±2.06 | 48.29±2.78 | 68.38±2.51 | 50.14±2.64 |
| 1 | 11 | -1.00 ❄ | -0.40 ❄ | 51.21±2.03 | 48.89±1.47 | 48.68±0.96 | 47.98±1.63 | 68.61±1.26 | 50.42±1.04 |
| 1 | 11 | -1.00 🔥 | -0.40 🔥 | 52.76±1.32 | 48.29±3.35 | 48.87±2.77 | 47.83±3.61 | 68.48±2.72 | 50.29±3.06 |
| 1 | 11 | -1.00 ❄ | -0.20 ❄ | 52.17±2.07 | 49.95±1.95 | 49.34±1.79 | 48.76±2.17 | 69.64±1.89 | 51.36±1.79 |
| 1 | 11 | -1.00 ❄ | -0.20 ❄ | 52.94±1.57 | 50.48±1.48 | 50.62±1.19 | 49.65±1.34 | 69.49±1.11 | 50.60±1.77 |
| 1 | 11 | -1.00 🔥 | -0.20 🔥 | 52.86±1.18 | 49.30±2.91 | 49.10±2.10 | 48.35±2.69 | 69.00±1.91 | 50.58±2.21 |
| 1 | 11 | -1.00 🔥 | -0.20 🔥 | 52.79±1.55 | 50.06±1.70 | 50.33±1.44 | 49.25±1.58 | 69.63±1.11 | 50.74±1.63 |
| 1 | 12 | -1.00 ❄ | -1.00 ❄ | 52.23±0.70 | 49.02±1.30 | 48.85±1.39 | 48.42±1.79 | 69.42±1.28 | 51.53±1.57 |
| 1 | 12 | -1.00 🔥 | -1.00 🔥 | 52.87±0.91 | 48.84±1.15 | 48.80±1.09 | 48.17±1.61 | 69.83±1.51 | 51.81±1.39 |
| 1 | 12 | -1.00 ❄ | -0.80 ❄ | 51.77±1.81 | 49.15±1.61 | 48.85±1.60 | 48.10±1.99 | 69.28±1.63 | 51.18±1.62 |
| 1 | 12 | -1.00 🔥 | -0.80 🔥 | 51.30±1.90 | 48.55±2.06 | 48.64±2.13 | 48.15±2.30 | 68.67±1.85 | 50.49±2.12 |
| 1 | 12 | -1.00 ❄ | -0.60 ❄ | 51.09±2.09 | 48.78±1.64 | 48.58±1.66 | 48.01±1.97 | 68.73±1.96 | 50.50±2.11 |
| 1 | 12 | -1.00 🔥 | -0.60 🔥 | 50.75±2.02 | 48.16±2.49 | 47.98±2.48 | 47.38±2.42 | 68.05±2.19 | 49.60±2.52 |
| 1 | 12 | -1.00 ❄ | -0.40 ❄ | 50.24±1.48 | 47.38±1.71 | 47.13±1.68 | 46.79±1.63 | 67.87±1.12 | 49.14±1.11 |
| 1 | 12 | -1.00 🔥 | -0.40 🔥 | 49.54±2.95 | 47.32±2.66 | 47.10±2.16 | 46.00±2.48 | 67.31±1.79 | 48.73±2.04 |
| 1 | 12 | -1.00 ❄ | -0.20 ❄ | 50.92±1.56 | 48.17±1.85 | 48.04±1.63 | 47.63±1.68 | 68.12±1.42 | 49.36±1.84 |
| 1 | 12 | -1.00 ❄ | -0.20 ❄ | 51.41±2.32 | 49.46±2.48 | 49.58±2.61 | 48.42±2.26 | 68.68±2.27 | 49.71±2.82 |
| 1 | 12 | -1.00 🔥 | -0.20 🔥 | 51.05±1.58 | 48.59±2.39 | 48.22±1.78 | 47.63±1.73 | 68.61±1.48 | 49.90±1.68 |
| 1 | 12 | -1.00 🔥 | -0.20 🔥 | 52.52±0.88 | 49.16±2.35 | 49.51±2.45 | 48.47±1.92 | 69.29±2.02 | 50.12±2.34 |

*Table 11.* Ablation study of CIF structure hyperparameters on PTB dataset (15 Channels). We use 🔥 to represent learnable parameters, and ❄ to represent non-learnable parameters.

| t | n | a | b | Accuracy | Precision | Recall | F1 score | AUROC | AUPRC |
|---|---|---|---|---|---|---|---|---|---|
| - | - | - | - | 81.87±1.87 | 86.50±1.24 | 73.49±2.90 | 75.84±3.20 | 94.20±0.29 | 93.04±0.45 |
| 1 | 8 | 0.21 ❄ | -0.50 ❄ | **88.29±1.45** | **90.66±1.48** | **83.21±2.02** | **85.59±1.96** | 94.28±0.93 | **93.78±1.11** |
| 1 | 8 | 0.20 ❄ | -0.50 ❄ | 84.45±2.15 | 88.34±1.15 | 77.40±3.52 | 79.89±3.51 | 93.15±0.70 | 91.94±0.67 |
| 1 | 9 | 0.22 ❄ | -0.50 ❄ | 85.14±0.22 | 87.42±1.08 | 79.05±0.52 | 81.42±0.35 | 93.82±1.19 | 92.73±1.27 |
| 1 | 10 | 0.22 ❄ | -0.50 ❄ | 85.40±2.08 | 88.17±1.96 | 79.17±3.12 | 81.57±3.07 | **94.57±1.13** | 93.53±1.44 |
| 1 | 11 | 0.22 ❄ | -0.50 ❄ | 84.47±0.84 | 88.02±0.59 | 77.50±1.26 | 80.10±1.28 | 92.77±0.89 | 92.02±0.78 |
| 1 | 8 | 0.23 ❄ | -0.50 ❄ | 83.96±5.41 | 87.13±3.96 | 76.89±8.22 | 78.66±9.51 | 92.49±2.91 | 91.14±3.79 |
| 1 | 8 | 0.25 ❄ | -0.50 ❄ | 85.56±2.09 | 88.96±1.77 | 79.03±3.03 | 81.61±3.14 | 93.61±1.11 | 93.07±1.03 |
| -1 | 7 | -1.00 ❄ | -1.00 ❄ | 82.57±0.88 | 86.88±0.94 | 74.58±1.26 | 77.09±1.35 | 93.75±1.30 | 92.74±1.65 |
| -1 | 7 | -1.00 🔥 | -1.00 🔥 | 82.26±3.27 | 86.11±2.38 | 74.28±4.93 | 76.48±5.44 | 92.94±2.37 | 91.53±2.75 |
| -1 | 7 | -1.00 🔥 | -0.90 ❄ | 81.49±3.97 | 86.49±2.89 | 72.81±6.00 | 74.78±6.87 | 93.41±1.80 | 92.62±2.04 |
| -1 | 7 | -1.00 🔥 | -0.90 🔥 | 78.44±4.44 | 84.94±2.03 | 68.22±7.05 | 69.04±8.81 | 92.58±2.17 | 91.48±2.14 |
| -1 | 7 | -1.00 ❄ | -0.80 ❄ | 79.94±3.18 | 85.35±2.61 | 70.48±4.79 | 72.26±6.02 | 93.43±1.49 | 92.30±1.61 |
| -1 | 7 | -1.00 🔥 | -0.80 🔥 | 80.57±4.70 | 85.89±3.20 | 71.38±7.18 | 72.86±9.26 | 92.43±0.59 | 91.48±1.19 |
| -1 | 7 | -1.00 ❄ | -0.70 ❄ | 81.01±2.74 | 86.90±0.99 | 71.93±4.34 | 73.99±5.25 | 92.44±2.72 | 91.70±2.53 |
| -1 | 7 | -1.00 🔥 | -0.70 🔥 | 83.64±1.03 | 87.21±0.54 | 76.36±1.71 | 78.88±1.65 | 93.50±0.94 | 92.83±0.73 |
| -1 | 7 | -1.00 ❄ | -0.60 ❄ | 80.22±1.47 | 86.35±1.48 | 70.72±2.19 | 72.82±2.57 | 92.61±1.47 | 91.70±1.77 |
| -1 | 7 | -1.00 🔥 | -0.60 🔥 | 81.73±2.77 | 86.55±1.78 | 73.24±4.32 | 75.45±4.82 | 90.80±1.65 | 90.14±1.62 |
| -1 | 8 | -1.00 ❄ | -1.00 ❄ | 82.23±3.14 | 86.39±1.88 | 74.18±4.86 | 76.40±5.01 | 94.31±1.14 | 93.25±1.17 |
| -1 | 8 | -1.00 🔥 | -1.00 🔥 | 80.07±1.79 | 84.28±2.52 | 71.10±2.50 | 73.16±2.85 | 93.63±0.95 | 92.04±1.42 |
| -1 | 8 | -1.00 ❄ | -0.90 ❄ | 81.36±2.30 | 86.49±1.89 | 72.62±3.49 | 74.87±3.81 | 92.14±2.56 | 91.19±2.68 |
| -1 | 8 | -1.00 🔥 | -0.90 🔥 | 79.96±1.46 | 84.20±1.84 | 70.92±2.02 | 72.98±2.32 | 92.00±1.97 | 90.66±2.01 |
| -1 | 8 | -1.00 ❄ | -0.80 ❄ | 80.34±1.93 | 86.14±1.65 | 71.00±3.00 | 73.07±3.40 | 91.44±2.64 | 91.01±2.55 |
| -1 | 8 | -1.00 🔥 | -0.80 🔥 | 77.08±4.30 | 84.12±3.23 | 66.14±6.66 | 66.44±9.24 | 89.68±3.65 | 88.73±3.53 |
| -1 | 8 | -1.00 ❄ | -0.70 ❄ | 79.17±2.48 | 85.39±1.26 | 69.35±4.22 | 70.91±5.05 | 92.43±1.29 | 91.69±0.98 |
| -1 | 8 | -1.00 🔥 | -0.70 🔥 | 80.03±4.82 | 85.46±2.00 | 70.75±7.74 | 71.98±8.94 | 91.05±2.98 | 90.05±3.18 |
| -1 | 9 | -1.00 ❄ | -1.00 ❄ | 81.23±2.67 | 86.60±1.43 | 72.39±4.23 | 74.52±4.64 | 92.20±1.22 | 91.55±1.57 |
| -1 | 9 | -1.00 🔥 | -1.00 🔥 | 80.05±4.78 | 86.50±2.17 | 70.45±7.40 | 71.73±9.27 | 92.73±0.93 | 91.87±1.14 |
| -1 | 9 | -1.00 ❄ | -0.90 ❄ | 80.89±4.06 | 87.74±1.78 | 71.49±6.25 | 73.24±7.85 | 93.83±2.39 | 93.33±2.64 |
| -1 | 9 | -1.00 🔥 | -0.90 🔥 | 82.34±3.22 | 87.35±2.06 | 74.12±5.14 | 76.30±5.62 | 94.05±2.17 | 93.54±2.08 |
| -1 | 9 | -1.00 ❄ | -0.80 ❄ | 84.06±3.49 | 88.91±1.67 | 76.49±5.44 | 78.88±5.80 | 92.73±1.43 | 92.39±1.26 |
| -1 | 9 | -1.00 🔥 | -0.80 🔥 | 82.50±2.14 | 88.19±1.12 | 74.03±3.33 | 76.51±3.67 | 94.33±0.84 | 93.83±1.05 |
| -1 | 9 | -1.00 ❄ | -0.70 ❄ | 81.44±4.04 | 87.88±1.77 | 72.37±6.23 | 74.28±7.32 | 92.35±1.77 | 91.86±1.50 |
| -1 | 9 | -1.00 🔥 | -0.70 🔥 | 80.07±3.15 | 87.48±1.01 | 70.24±4.97 | 71.96±6.03 | 92.83±2.18 | 92.43±2.24 |
| -1 | 10 | -1.00 ❄ | -1.00 ❄ | 79.43±3.39 | 85.03±2.12 | 69.86±5.42 | 71.34±6.75 | 90.25±3.82 | 89.29±4.14 |
| -1 | 10 | -1.00 🔥 | -1.00 🔥 | 82.20±1.03 | 86.48±1.03 | 74.07±1.47 | 76.53±1.61 | 92.05±2.25 | 91.29±2.26 |
| -1 | 10 | -1.00 ❄ | -0.90 ❄ | 83.08±3.17 | 87.22±1.54 | 75.41±5.02 | 77.64±5.69 | 90.27±3.83 | 89.89±3.83 |
| -1 | 10 | -1.00 🔥 | -0.90 🔥 | 80.74±3.10 | 86.14±1.21 | 71.72±4.94 | 73.63±5.85 | 92.44±1.50 | 91.53±1.55 |
| -1 | 10 | -1.00 ❄ | -0.80 ❄ | 78.96±3.54 | 86.02±2.07 | 68.75±5.52 | 70.07±6.74 | 90.23±3.34 | 89.36±3.48 |
| -1 | 10 | -1.00 🔥 | -0.80 🔥 | 79.20±3.45 | 85.98±0.63 | 69.31±5.83 | 70.58±6.97 | 93.24±1.60 | 92.07±1.78 |
| -1 | 10 | -1.00 ❄ | -0.70 ❄ | 82.37±1.89 | 87.63±0.88 | 74.00±3.03 | 76.44±3.21 | 93.05±1.90 | 92.39±1.83 |
| -1 | 10 | -1.00 🔥 | -0.70 🔥 | 77.99±4.51 | 85.22±0.87 | 67.50±7.35 | 67.94±9.84 | 91.94±3.85 | 90.81±3.67 |
| -1 | 11 | -1.00 ❄ | -1.00 ❄ | 81.63±3.29 | 86.31±2.32 | 73.11±5.00 | 75.22±5.90 | 91.76±2.02 | 90.82±2.04 |
| -1 | 11 | -1.00 🔥 | -1.00 🔥 | 79.09±3.82 | 86.30±1.79 | 68.91±5.95 | 70.15±7.70 | 91.35±3.78 | 90.46±3.63 |
| -1 | 11 | -1.00 ❄ | -0.90 ❄ | 79.68±2.84 | 86.22±1.10 | 69.89±4.48 | 71.60±5.35 | 92.11±1.09 | 91.33±1.27 |
| -1 | 11 | -1.00 🔥 | -0.90 🔥 | 78.44±3.40 | 86.20±1.44 | 67.84±5.36 | 68.96±6.63 | 90.75±5.55 | 90.19±5.31 |
| -1 | 11 | -1.00 ❄ | -0.80 ❄ | 80.05±2.78 | 87.14±1.09 | 70.25±4.35 | 72.08±5.08 | 92.86±2.82 | 92.27±2.86 |
| -1 | 11 | -1.00 🔥 | -0.80 🔥 | 82.18±0.99 | 86.96±1.56 | 73.88±1.47 | 76.36±1.56 | 91.48±3.29 | 90.85±3.31 |

Continued on next page

| t | n | a | b | Accuracy | Precision | Recall | F1 score | AUROC | AUPRC |
|---|---|---|---|---|---|---|---|---|---|
| -1 | 11 | -1.00 ❄ | -0.70 ❄ | $80.82_{\pm2.07}$ | $85.72_{\pm1.44}$ | $71.94_{\pm3.23}$ | $74.08_{\pm3.62}$ | $89.25_{\pm2.75}$ | $88.51_{\pm2.83}$ |
| -1 | 11 | -1.00 🔥 | -0.70 🔥 | $78.20_{\pm2.54}$ | $84.96_{\pm1.72}$ | $67.72_{\pm3.95}$ | $69.00_{\pm5.18}$ | $90.29_{\pm0.65}$ | $89.52_{\pm0.96}$ |
| -1 | 12 | -1.00 ❄ | -1.00 ❄ | $78.27_{\pm3.04}$ | $86.34_{\pm1.19}$ | $67.54_{\pm4.80}$ | $68.65_{\pm6.04}$ | $91.52_{\pm2.68}$ | $90.92_{\pm2.48}$ |
| -1 | 12 | -1.00 🔥 | -1.00 🔥 | $82.45_{\pm1.76}$ | $87.85_{\pm0.79}$ | $74.10_{\pm3.00}$ | $76.55_{\pm3.05}$ | $92.78_{\pm1.02}$ | $92.30_{\pm0.82}$ |
| -1 | 12 | -1.00 ❄ | -0.90 ❄ | $79.49_{\pm1.84}$ | $85.59_{\pm2.16}$ | $69.67_{\pm2.57}$ | $71.57_{\pm3.17}$ | $91.80_{\pm1.78}$ | $90.84_{\pm1.96}$ |
| -1 | 12 | -1.00 🔥 | -0.90 🔥 | $82.03_{\pm1.90}$ | $88.37_{\pm0.83}$ | $73.22_{\pm3.06}$ | $75.64_{\pm3.42}$ | $93.21_{\pm1.67}$ | $92.67_{\pm1.74}$ |
| -1 | 12 | -1.00 ❄ | -0.80 ❄ | $80.02_{\pm2.82}$ | $86.84_{\pm2.62}$ | $70.21_{\pm4.03}$ | $72.15_{\pm4.65}$ | $91.42_{\pm0.95}$ | $90.99_{\pm1.27}$ |
| -1 | 12 | -1.00 🔥 | -0.80 🔥 | $81.67_{\pm3.24}$ | $87.40_{\pm2.30}$ | $72.82_{\pm4.86}$ | $75.00_{\pm5.75}$ | $92.56_{\pm1.50}$ | $91.81_{\pm1.68}$ |
| -1 | 12 | -1.00 ❄ | -0.70 ❄ | $80.32_{\pm4.95}$ | $87.36_{\pm2.58}$ | $70.62_{\pm7.57}$ | $72.03_{\pm8.91}$ | $93.41_{\pm3.07}$ | $92.78_{\pm3.34}$ |
| -1 | 12 | -1.00 🔥 | -0.70 🔥 | $79.00_{\pm1.99}$ | $87.52_{\pm0.83}$ | $68.43_{\pm3.05}$ | $70.02_{\pm3.74}$ | $92.27_{\pm2.22}$ | $92.16_{\pm1.82}$ |
| 1 | 7 | -1.00 ❄ | -1.00 ❄ | $85.02_{\pm1.35}$ | $88.50_{\pm1.01}$ | $78.27_{\pm1.99}$ | $80.88_{\pm1.99}$ | $95.79_{\pm0.74}$ | $95.11_{\pm0.72}$ |
| 1 | 7 | -1.00 🔥 | -1.00 🔥 | $84.81_{\pm1.84}$ | $88.77_{\pm1.20}$ | $77.86_{\pm3.06}$ | $80.43_{\pm2.94}$ | $96.33_{\pm1.03}$ | $95.64_{\pm1.19}$ |
| 1 | 7 | -1.00 ❄ | -0.90 ❄ | $83.37_{\pm2.41}$ | $87.74_{\pm1.09}$ | $75.71_{\pm3.86}$ | $78.15_{\pm4.02}$ | $94.66_{\pm1.34}$ | $93.86_{\pm1.44}$ |
| 1 | 7 | -1.00 🔥 | -0.90 🔥 | $83.70_{\pm1.62}$ | $87.88_{\pm1.13}$ | $76.19_{\pm2.45}$ | $78.77_{\pm2.47}$ | $95.73_{\pm0.89}$ | $95.00_{\pm0.68}$ |
| 1 | 7 | -1.00 ❄ | -0.80 ❄ | $84.10_{\pm1.64}$ | $88.14_{\pm1.54}$ | $76.78_{\pm2.28}$ | $79.40_{\pm2.41}$ | $95.46_{\pm1.68}$ | $94.73_{\pm1.86}$ |
| 1 | 7 | -1.00 🔥 | -0.80 🔥 | $83.59_{\pm2.52}$ | $88.30_{\pm1.38}$ | $75.88_{\pm3.99}$ | $78.38_{\pm4.01}$ | $93.55_{\pm1.14}$ | $93.01_{\pm1.42}$ |
| 1 | 7 | -1.00 ❄ | -0.70 ❄ | $82.68_{\pm2.56}$ | $87.55_{\pm1.12}$ | $74.56_{\pm4.03}$ | $76.94_{\pm4.37}$ | $95.07_{\pm1.61}$ | $94.34_{\pm1.52}$ |
| 1 | 7 | -1.00 🔥 | -0.70 🔥 | $86.59_{\pm1.96}$ | $89.57_{\pm1.34}$ | $80.67_{\pm3.11}$ | $83.14_{\pm2.92}$ | $95.86_{\pm0.99}$ | $95.33_{\pm1.08}$ |
| 1 | 7 | -1.00 ❄ | -0.60 ❄ | $83.42_{\pm3.53}$ | $87.76_{\pm2.07}$ | $75.78_{\pm5.45}$ | $78.08_{\pm5.59}$ | $95.23_{\pm1.66}$ | $94.35_{\pm1.82}$ |
| 1 | 7 | -1.00 🔥 | -0.60 🔥 | $84.09_{\pm2.80}$ | $87.38_{\pm1.49}$ | $77.12_{\pm4.43}$ | $79.45_{\pm4.54}$ | $93.88_{\pm1.22}$ | $93.14_{\pm1.40}$ |
| 1 | 8 | -1.00 ❄ | -1.00 ❄ | $82.84_{\pm2.54}$ | $84.33_{\pm2.58}$ | $76.30_{\pm3.90}$ | $78.32_{\pm3.72}$ | $94.66_{\pm1.53}$ | $92.85_{\pm1.97}$ |
| 1 | 8 | -1.00 🔥 | -1.00 🔥 | $81.17_{\pm1.65}$ | $84.79_{\pm1.21}$ | $72.91_{\pm2.68}$ | $75.10_{\pm2.91}$ | $94.65_{\pm1.08}$ | $92.31_{\pm2.18}$ |
| 1 | 8 | -1.00 ❄ | -0.90 ❄ | $83.41_{\pm0.89}$ | $86.62_{\pm1.18}$ | $76.22_{\pm1.40}$ | $78.67_{\pm1.40}$ | $94.87_{\pm1.00}$ | $93.35_{\pm1.07}$ |
| 1 | 8 | -1.00 🔥 | -0.90 🔥 | $81.23_{\pm2.07}$ | $84.90_{\pm1.55}$ | $73.00_{\pm3.35}$ | $75.14_{\pm3.64}$ | $94.30_{\pm1.54}$ | $92.69_{\pm2.04}$ |
| 1 | 8 | -1.00 ❄ | -0.80 ❄ | $82.75_{\pm2.09}$ | $85.86_{\pm1.82}$ | $75.41_{\pm3.40}$ | $77.66_{\pm3.50}$ | $94.49_{\pm1.21}$ | $92.78_{\pm1.69}$ |
| 1 | 8 | -1.00 🔥 | -0.80 🔥 | $83.49_{\pm2.10}$ | $85.88_{\pm2.27}$ | $76.66_{\pm2.93}$ | $78.97_{\pm3.02}$ | $95.08_{\pm2.27}$ | $93.62_{\pm2.66}$ |
| 1 | 8 | -1.00 ❄ | -0.70 ❄ | $82.14_{\pm2.38}$ | $85.38_{\pm1.66}$ | $74.44_{\pm3.79}$ | $76.63_{\pm4.11}$ | $94.74_{\pm1.66}$ | $93.38_{\pm1.67}$ |
| 1 | 8 | -1.00 🔥 | -0.70 🔥 | $83.04_{\pm1.70}$ | $85.52_{\pm1.90}$ | $76.05_{\pm2.52}$ | $78.33_{\pm2.58}$ | $94.25_{\pm1.34}$ | $92.44_{\pm1.50}$ |
| 1 | 8 | -1.00 ❄ | -0.60 ❄ | $83.87_{\pm2.61}$ | $86.85_{\pm2.41}$ | $76.95_{\pm4.01}$ | $79.26_{\pm4.08}$ | $95.20_{\pm2.86}$ | $93.94_{\pm3.31}$ |
| 1 | 8 | -1.00 🔥 | -0.60 🔥 | $84.24_{\pm2.01}$ | $87.41_{\pm1.67}$ | $77.34_{\pm2.97}$ | $79.81_{\pm3.00}$ | $94.88_{\pm0.96}$ | $93.38_{\pm1.20}$ |
| 1 | 9 | -1.00 ❄ | -1.00 ❄ | $82.58_{\pm3.05}$ | $86.28_{\pm1.69}$ | $74.89_{\pm4.91}$ | $77.07_{\pm4.91}$ | $93.56_{\pm2.06}$ | $92.48_{\pm1.79}$ |
| 1 | 9 | -1.00 🔥 | -1.00 🔥 | $81.97_{\pm2.26}$ | $85.64_{\pm1.61}$ | $73.99_{\pm3.46}$ | $76.26_{\pm3.77}$ | $93.68_{\pm1.04}$ | $92.21_{\pm1.27}$ |
| 1 | 9 | -1.00 ❄ | -0.90 ❄ | $81.24_{\pm2.88}$ | $84.95_{\pm2.58}$ | $72.97_{\pm4.37}$ | $75.05_{\pm4.74}$ | $94.07_{\pm2.15}$ | $92.79_{\pm2.34}$ |
| 1 | 9 | -1.00 🔥 | -0.90 🔥 | $83.23_{\pm1.06}$ | $86.49_{\pm2.05}$ | $75.95_{\pm1.17}$ | $78.42_{\pm1.32}$ | $93.78_{\pm1.86}$ | $92.82_{\pm2.05}$ |
| 1 | 9 | -1.00 ❄ | -0.80 ❄ | $81.84_{\pm2.15}$ | $86.93_{\pm1.44}$ | $73.30_{\pm3.27}$ | $75.65_{\pm3.53}$ | $93.56_{\pm0.88}$ | $92.60_{\pm1.15}$ |
| 1 | 9 | -1.00 🔥 | -0.80 🔥 | $82.90_{\pm1.57}$ | $86.86_{\pm1.99}$ | $75.18_{\pm2.19}$ | $77.67_{\pm2.29}$ | $94.64_{\pm0.85}$ | $93.78_{\pm0.94}$ |
| 1 | 9 | -1.00 ❄ | -0.70 ❄ | $83.52_{\pm2.03}$ | $86.44_{\pm2.85}$ | $76.45_{\pm2.50}$ | $78.89_{\pm2.76}$ | $93.94_{\pm1.29}$ | $92.76_{\pm1.71}$ |
| 1 | 9 | -1.00 🔥 | -0.70 🔥 | $79.94_{\pm3.23}$ | $84.83_{\pm2.00}$ | $70.69_{\pm5.00}$ | $72.45_{\pm5.82}$ | $93.59_{\pm1.56}$ | $92.47_{\pm1.42}$ |
| 1 | 9 | -1.00 ❄ | -0.60 ❄ | $81.79_{\pm3.50}$ | $87.48_{\pm1.74}$ | $73.12_{\pm5.52}$ | $75.20_{\pm6.04}$ | $94.39_{\pm1.57}$ | $93.35_{\pm2.09}$ |
| 1 | 9 | -1.00 🔥 | -0.60 🔥 | $82.20_{\pm1.47}$ | $87.39_{\pm2.01}$ | $73.78_{\pm2.18}$ | $76.26_{\pm2.31}$ | $94.56_{\pm3.17}$ | $93.59_{\pm3.93}$ |
| 1 | 10 | -1.00 ❄ | -1.00 ❄ | $82.13_{\pm3.26}$ | $87.02_{\pm1.61}$ | $73.82_{\pm5.11}$ | $75.97_{\pm5.88}$ | $93.67_{\pm1.96}$ | $92.80_{\pm2.01}$ |
| 1 | 10 | -1.00 🔥 | -1.00 🔥 | $82.69_{\pm4.49}$ | $86.75_{\pm2.85}$ | $74.85_{\pm7.00}$ | $76.75_{\pm7.98}$ | $94.19_{\pm0.91}$ | $93.34_{\pm0.96}$ |
| 1 | 10 | -1.00 ❄ | -0.90 ❄ | $81.80_{\pm2.20}$ | $86.99_{\pm2.63}$ | $73.17_{\pm3.06}$ | $75.57_{\pm3.48}$ | $92.92_{\pm1.69}$ | $92.09_{\pm2.04}$ |
| 1 | 10 | -1.00 🔥 | -0.90 🔥 | $86.15_{\pm1.41}$ | $88.87_{\pm0.92}$ | $80.15_{\pm2.16}$ | $82.62_{\pm2.05}$ | $95.97_{\pm0.65}$ | $95.32_{\pm0.57}$ |
| 1 | 10 | -1.00 ❄ | -0.80 ❄ | $82.52_{\pm2.43}$ | $87.71_{\pm1.90}$ | $74.19_{\pm3.65}$ | $76.64_{\pm4.01}$ | $93.67_{\pm3.19}$ | $92.93_{\pm3.20}$ |
| 1 | 10 | -1.00 🔥 | -0.80 🔥 | $82.92_{\pm1.74}$ | $87.86_{\pm1.61}$ | $74.83_{\pm2.52}$ | $77.40_{\pm2.76}$ | $93.42_{\pm3.18}$ | $92.76_{\pm2.99}$ |
| 1 | 10 | -1.00 ❄ | -0.70 ❄ | $84.52_{\pm1.47}$ | $88.52_{\pm0.74}$ | $77.38_{\pm2.32}$ | $80.01_{\pm2.38}$ | $95.45_{\pm1.12}$ | $94.74_{\pm1.04}$ |
| 1 | 10 | -1.00 🔥 | -0.70 🔥 | $83.00_{\pm1.47}$ | $87.12_{\pm0.99}$ | $75.26_{\pm2.31}$ | $77.76_{\pm2.42}$ | $94.24_{\pm1.35}$ | $93.54_{\pm1.19}$ |
| 1 | 11 | -1.00 ❄ | -1.00 ❄ | $83.24_{\pm2.22}$ | $87.34_{\pm1.29}$ | $75.60_{\pm3.43}$ | $78.05_{\pm3.68}$ | $94.73_{\pm1.95}$ | $93.82_{\pm1.85}$ |
| 1 | 11 | -1.00 🔥 | -1.00 🔥 | $82.60_{\pm0.88}$ | $87.11_{\pm0.95}$ | $74.55_{\pm1.37}$ | $77.08_{\pm1.43}$ | $92.63_{\pm1.52}$ | $91.79_{\pm1.51}$ |

| t | n | a | b | Accuracy | Precision | Recall | F1 score | AUROC | AUPRC |
|---|---|---|---|---|---|---|---|---|---|
| 1 | 11 | -1.00 ❄ | -0.90 ❄ | $80.82_{\pm2.45}$ | $85.77_{\pm1.59}$ | $71.90_{\pm3.72}$ | $74.02_{\pm4.24}$ | $93.53_{\pm1.82}$ | $92.37_{\pm1.58}$ |
| 1 | 11 | -1.00 🔥 | -0.90 🔥 | $81.41_{\pm3.01}$ | $86.76_{\pm2.08}$ | $72.68_{\pm4.66}$ | $74.78_{\pm5.44}$ | $94.61_{\pm1.16}$ | $93.24_{\pm1.68}$ |
| 1 | 11 | -1.00 ❄ | -0.80 ❄ | $82.73_{\pm2.62}$ | $87.66_{\pm2.10}$ | $74.59_{\pm3.93}$ | $77.01_{\pm4.30}$ | $94.59_{\pm0.65}$ | $93.61_{\pm0.69}$ |
| 1 | 11 | -1.00 🔥 | -0.80 🔥 | $77.98_{\pm2.41}$ | $85.65_{\pm1.32}$ | $67.28_{\pm4.08}$ | $68.40_{\pm4.92}$ | $92.95_{\pm3.27}$ | $92.12_{\pm3.26}$ |
| 1 | 11 | -1.00 ❄ | -0.70 ❄ | $83.28_{\pm3.98}$ | $87.98_{\pm2.44}$ | $75.47_{\pm6.08}$ | $77.72_{\pm6.22}$ | $94.63_{\pm1.63}$ | $93.23_{\pm2.43}$ |
| 1 | 11 | -1.00 🔥 | -0.70 🔥 | $81.95_{\pm3.19}$ | $87.27_{\pm1.44}$ | $73.43_{\pm5.07}$ | $75.57_{\pm5.85}$ | $94.49_{\pm1.31}$ | $93.62_{\pm1.35}$ |
| 1 | 12 | -1.00 ❄ | -1.00 ❄ | $82.56_{\pm2.11}$ | $85.78_{\pm2.15}$ | $75.05_{\pm3.29}$ | $77.34_{\pm3.22}$ | $92.70_{\pm3.46}$ | $91.46_{\pm4.17}$ |
| 1 | 12 | -1.00 🔥 | -1.00 🔥 | $79.36_{\pm2.38}$ | $85.74_{\pm2.69}$ | $69.50_{\pm3.66}$ | $71.23_{\pm4.45}$ | $89.66_{\pm2.53}$ | $88.86_{\pm2.93}$ |
| 1 | 12 | -1.00 ❄ | -0.90 ❄ | $79.23_{\pm3.52}$ | $86.61_{\pm1.48}$ | $69.12_{\pm5.66}$ | $70.47_{\pm6.76}$ | $92.23_{\pm2.15}$ | $91.33_{\pm2.28}$ |
| 1 | 12 | -1.00 🔥 | -0.90 🔥 | $82.68_{\pm2.77}$ | $88.63_{\pm1.29}$ | $74.20_{\pm4.25}$ | $76.64_{\pm4.70}$ | $94.50_{\pm2.22}$ | $94.05_{\pm2.22}$ |
| 1 | 12 | -1.00 ❄ | -0.80 ❄ | $80.16_{\pm4.61}$ | $84.27_{\pm5.05}$ | $71.06_{\pm6.59}$ | $72.75_{\pm7.90}$ | $92.14_{\pm2.38}$ | $90.81_{\pm3.40}$ |
| 1 | 12 | -1.00 🔥 | -0.80 🔥 | $81.78_{\pm3.53}$ | $86.61_{\pm2.21}$ | $73.41_{\pm5.56}$ | $75.44_{\pm6.14}$ | $92.98_{\pm2.44}$ | $92.10_{\pm2.42}$ |
| 1 | 12 | -1.00 ❄ | -0.70 ❄ | $81.35_{\pm2.77}$ | $87.19_{\pm2.53}$ | $72.42_{\pm4.21}$ | $74.62_{\pm4.55}$ | $94.66_{\pm1.02}$ | $93.97_{\pm1.37}$ |
| 1 | 12 | -1.00 🔥 | -0.70 🔥 | $85.25_{\pm2.74}$ | $88.51_{\pm2.37}$ | $78.74_{\pm4.12}$ | $81.17_{\pm3.99}$ | $95.07_{\pm1.05}$ | $94.48_{\pm1.36}$ |

*Table 12.* Ablation study of CIF structure hyperparameters on APAVA dataset (16 Channels). We use 🔥 to represent learnable parameters, and ❄️ to represent non-learnable parameters.

| t | n | a | b | Accuracy | Precision | Recall | F1 score | AUROC | AUPRC |
|---|---|---|---|---|---|---|---|---|---|
| - | - | - | - | $82.49_{\pm1.40}$ | $82.38_{\pm1.79}$ | $81.20_{\pm1.32}$ | $81.60_{\pm1.39}$ | $91.10_{\pm1.63}$ | $91.30_{\pm1.71}$ |
| -1 | 5 | 1 ❄️ | 1 ❄️ | $81.48_{\pm1.96}$ | $81.99_{\pm2.38}$ | $79.52_{\pm2.00}$ | $80.21_{\pm2.07}$ | $90.93_{\pm1.74}$ | $91.17_{\pm1.77}$ |
| -1 | 6 | 1 🔥 | 1 🔥 | $84.00_{\pm2.37}$ | $84.33_{\pm2.61}$ | $82.37_{\pm2.44}$ | $83.02_{\pm2.50}$ | $92.67_{\pm2.57}$ | $92.87_{\pm2.55}$ |
| -1 | 7 | 1 🔥 | 1 🔥 | $81.77_{\pm2.31}$ | $82.67_{\pm3.24}$ | $79.78_{\pm2.04}$ | $80.49_{\pm2.26}$ | $89.92_{\pm2.60}$ | $90.23_{\pm2.53}$ |
| 1 | 9 | 1 ❄️ | 1 ❄️ | $85.16_{\pm1.55}$ | $84.76_{\pm1.62}$ | $85.33_{\pm1.27}$ | $84.82_{\pm1.49}$ | $94.06_{\pm1.07}$ | $94.21_{\pm0.99}$ |
| -1 | 10 | 1 🔥 | 1 🔥 | $82.60_{\pm1.82}$ | $82.30_{\pm2.04}$ | $81.92_{\pm2.29}$ | $81.92_{\pm1.95}$ | $91.94_{\pm1.55}$ | $92.16_{\pm1.50}$ |
| -1 | 11 | 1 ❄️ | 2 ❄️ | $78.98_{\pm0.97}$ | $78.57_{\pm1.09}$ | $77.60_{\pm0.93}$ | $77.93_{\pm0.97}$ | $87.29_{\pm0.93}$ | $87.35_{\pm0.96}$ |
| 1 | 6 | -1.00 ❄️ | -1.00 ❄️ | $81.20_{\pm1.28}$ | $81.50_{\pm1.16}$ | $79.51_{\pm1.84}$ | $80.01_{\pm1.67}$ | $90.03_{\pm1.00}$ | $90.28_{\pm1.03}$ |
| 1 | 6 | -1.00 🔥 | -1.00 🔥 | $81.24_{\pm1.16}$ | $81.57_{\pm0.95}$ | $79.56_{\pm1.81}$ | $80.05_{\pm1.62}$ | $90.29_{\pm0.90}$ | $90.61_{\pm0.88}$ |
| 1 | 7 | -1.00 ❄️ | -1.00 ❄️ | $81.33_{\pm2.24}$ | $81.36_{\pm2.52}$ | $79.71_{\pm2.22}$ | $80.24_{\pm2.31}$ | $89.57_{\pm1.98}$ | $90.02_{\pm1.89}$ |
| 1 | 7 | -1.00 ❄️ | -1.00 ❄️ | $81.33_{\pm2.24}$ | $81.36_{\pm2.52}$ | $79.71_{\pm2.22}$ | $80.24_{\pm2.31}$ | $89.57_{\pm1.98}$ | $90.02_{\pm1.89}$ |
| 1 | 7 | -1.00 🔥 | -1.00 🔥 | $81.62_{\pm2.58}$ | $81.63_{\pm2.95}$ | $80.09_{\pm2.48}$ | $80.60_{\pm2.62}$ | $89.94_{\pm2.18}$ | $90.33_{\pm2.07}$ |
| 1 | 7 | -1.00 ❄️ | -0.90 ❄️ | $81.45_{\pm2.29}$ | $81.48_{\pm2.56}$ | $79.83_{\pm2.31}$ | $80.37_{\pm2.39}$ | $89.73_{\pm1.95}$ | $90.17_{\pm1.85}$ |
| 1 | 7 | -1.00 🔥 | -0.90 🔥 | $81.58_{\pm2.27}$ | $81.72_{\pm2.66}$ | $79.88_{\pm2.21}$ | $80.47_{\pm2.33}$ | $89.67_{\pm2.01}$ | $90.10_{\pm1.93}$ |
| 1 | 7 | -1.00 ❄️ | -0.80 ❄️ | $82.28_{\pm1.80}$ | $82.97_{\pm1.82}$ | $80.30_{\pm2.14}$ | $81.01_{\pm2.09}$ | $90.47_{\pm1.67}$ | $90.83_{\pm1.58}$ |
| 1 | 7 | -1.00 🔥 | -0.80 🔥 | $81.80_{\pm1.95}$ | $82.45_{\pm2.33}$ | $80.01_{\pm2.01}$ | $80.59_{\pm2.07}$ | $89.90_{\pm1.61}$ | $90.35_{\pm1.53}$ |
| 1 | 7 | -1.00 ❄️ | -0.70 ❄️ | $82.17_{\pm2.06}$ | $82.87_{\pm2.44}$ | $80.41_{\pm2.08}$ | $80.99_{\pm2.16}$ | $90.51_{\pm1.76}$ | $90.91_{\pm1.64}$ |
| 1 | 7 | -1.00 🔥 | -0.70 🔥 | $82.05_{\pm1.11}$ | $83.13_{\pm1.64}$ | $79.89_{\pm1.42}$ | $80.66_{\pm1.33}$ | $90.47_{\pm1.20}$ | $90.86_{\pm1.11}$ |
| 1 | 7 | -1.00 ❄️ | -0.60 ❄️ | $82.38_{\pm1.55}$ | $83.21_{\pm2.12}$ | $80.41_{\pm1.74}$ | $81.12_{\pm1.69}$ | $90.96_{\pm1.20}$ | $91.26_{\pm1.16}$ |
| 1 | 7 | -1.00 🔥 | -0.60 🔥 | $82.19_{\pm1.49}$ | $83.02_{\pm2.15}$ | $80.23_{\pm1.61}$ | $80.93_{\pm1.60}$ | $90.81_{\pm1.26}$ | $91.15_{\pm1.18}$ |
| 1 | 7 | -1.00 ❄️ | -0.50 ❄️ | $82.94_{\pm0.93}$ | $83.86_{\pm1.07}$ | $80.95_{\pm1.45}$ | $81.68_{\pm1.27}$ | $91.39_{\pm0.77}$ | $91.69_{\pm0.71}$ |
| 1 | 7 | -1.00 🔥 | -0.50 🔥 | $82.52_{\pm1.10}$ | $83.35_{\pm1.63}$ | $80.60_{\pm1.40}$ | $81.28_{\pm1.30}$ | $91.17_{\pm0.98}$ | $91.49_{\pm0.92}$ |
| 1 | 7 | -1.00 ❄️ | -0.40 ❄️ | $82.96_{\pm0.89}$ | $83.73_{\pm1.10}$ | $81.07_{\pm1.32}$ | $81.77_{\pm1.16}$ | $91.79_{\pm0.60}$ | $92.03_{\pm0.60}$ |
| 1 | 7 | -1.00 🔥 | -0.40 🔥 | $83.02_{\pm0.79}$ | $83.82_{\pm1.14}$ | $81.08_{\pm1.05}$ | $81.82_{\pm0.95}$ | $91.85_{\pm0.60}$ | $92.09_{\pm0.60}$ |
| 1 | 7 | -1.00 ❄️ | -0.30 ❄️ | $83.00_{\pm1.71}$ | $83.51_{\pm2.29}$ | $81.31_{\pm1.43}$ | $81.95_{\pm1.61}$ | $91.78_{\pm1.64}$ | $92.02_{\pm1.63}$ |
| 1 | 7 | -1.00 🔥 | -0.30 🔥 | $82.78_{\pm1.68}$ | $83.49_{\pm2.38}$ | $81.10_{\pm1.04}$ | $81.70_{\pm1.43}$ | $91.98_{\pm1.35}$ | $92.19_{\pm1.36}$ |
| 1 | 7 | -1.00 ❄️ | -0.20 ❄️ | $83.13_{\pm1.72}$ | $83.47_{\pm2.29}$ | $81.71_{\pm1.60}$ | $82.18_{\pm1.67}$ | $92.09_{\pm1.65}$ | $92.25_{\pm1.65}$ |
| 1 | 7 | -1.00 🔥 | -0.20 🔥 | $83.16_{\pm1.93}$ | $83.61_{\pm2.47}$ | $81.55_{\pm1.79}$ | $82.14_{\pm1.91}$ | $92.13_{\pm1.74}$ | $92.30_{\pm1.72}$ |
| 1 | 7 | -1.00 ❄️ | -0.10 ❄️ | $84.39_{\pm0.90}$ | $84.68_{\pm0.94}$ | $82.92_{\pm1.22}$ | $83.48_{\pm1.06}$ | $93.08_{\pm0.92}$ | $93.20_{\pm0.98}$ |
| 1 | 7 | -1.00 🔥 | -0.10 🔥 | $84.56_{\pm0.97}$ | $84.89_{\pm0.69}$ | $83.05_{\pm1.42}$ | $83.64_{\pm1.20}$ | $93.19_{\pm0.94}$ | $93.30_{\pm1.01}$ |
| 1 | 7 | -1.00 ❄️ | 0.10 ❄️ | $84.25_{\pm1.46}$ | $84.47_{\pm1.37}$ | $82.77_{\pm1.79}$ | $83.33_{\pm1.64}$ | $92.77_{\pm1.08}$ | $92.90_{\pm1.13}$ |
| 1 | 7 | -1.00 🔥 | 0.10 🔥 | $84.39_{\pm1.55}$ | $84.56_{\pm1.49}$ | $82.96_{\pm1.83}$ | $83.50_{\pm1.71}$ | $92.84_{\pm1.18}$ | $92.95_{\pm1.22}$ |
| 1 | 7 | -1.00 ❄️ | 0.20 ❄️ | $84.02_{\pm1.48}$ | $84.15_{\pm1.52}$ | $82.56_{\pm1.61}$ | $83.12_{\pm1.60}$ | $92.51_{\pm1.21}$ | $92.66_{\pm1.20}$ |
| 1 | 7 | -1.00 🔥 | 0.20 🔥 | $83.37_{\pm1.81}$ | $83.65_{\pm2.04}$ | $81.85_{\pm1.83}$ | $82.40_{\pm1.87}$ | $92.11_{\pm1.31}$ | $92.27_{\pm1.26}$ |
| 1 | 7 | -1.00 ❄️ | 0.30 ❄️ | $82.95_{\pm2.19}$ | $82.86_{\pm2.64}$ | $81.70_{\pm1.98}$ | $82.11_{\pm2.18}$ | $91.84_{\pm1.67}$ | $92.01_{\pm1.62}$ |
| 1 | 7 | -1.00 🔥 | 0.30 🔥 | $82.66_{\pm1.88}$ | $82.62_{\pm2.43}$ | $81.40_{\pm1.56}$ | $81.79_{\pm1.81}$ | $91.34_{\pm1.88}$ | $91.43_{\pm1.94}$ |
| 1 | 7 | -1.00 ❄️ | 0.40 ❄️ | $82.33_{\pm1.91}$ | $82.35_{\pm2.29}$ | $80.91_{\pm1.87}$ | $81.38_{\pm1.97}$ | $91.20_{\pm1.98}$ | $91.32_{\pm2.00}$ |
| 1 | 7 | -1.00 🔥 | 0.40 🔥 | $83.12_{\pm1.75}$ | $83.32_{\pm2.07}$ | $81.57_{\pm1.79}$ | $82.13_{\pm1.84}$ | $91.91_{\pm1.65}$ | $92.08_{\pm1.59}$ |
| 1 | 7 | -1.00 ❄️ | 0.50 ❄️ | $82.07_{\pm1.77}$ | $82.32_{\pm2.31}$ | $80.33_{\pm1.61}$ | $80.96_{\pm1.78}$ | $90.99_{\pm1.65}$ | $91.21_{\pm1.59}$ |
| 1 | 7 | -1.00 🔥 | 0.50 🔥 | $82.35_{\pm2.42}$ | $82.57_{\pm2.49}$ | $80.63_{\pm2.64}$ | $81.24_{\pm2.63}$ | $91.04_{\pm1.86}$ | $91.27_{\pm1.79}$ |
| 1 | 7 | -1.00 ❄️ | 0.60 ❄️ | $82.03_{\pm2.11}$ | $82.19_{\pm2.65}$ | $80.39_{\pm2.01}$ | $80.97_{\pm2.16}$ | $90.74_{\pm1.94}$ | $90.95_{\pm1.87}$ |
| 1 | 7 | -1.00 🔥 | 0.60 🔥 | $82.08_{\pm2.01}$ | $82.22_{\pm2.48}$ | $80.48_{\pm1.97}$ | $81.04_{\pm2.08}$ | $90.80_{\pm1.92}$ | $91.00_{\pm1.86}$ |
| 1 | 7 | -1.00 ❄️ | 0.70 ❄️ | $81.73_{\pm2.22}$ | $81.66_{\pm2.39}$ | $80.24_{\pm2.36}$ | $80.72_{\pm2.38}$ | $90.43_{\pm1.90}$ | $90.63_{\pm1.82}$ |
| 1 | 7 | -1.00 🔥 | 0.70 🔥 | $81.79_{\pm2.20}$ | $81.87_{\pm2.48}$ | $80.13_{\pm2.25}$ | $80.70_{\pm2.32}$ | $90.50_{\pm1.90}$ | $90.71_{\pm1.84}$ |
| 1 | 7 | -1.00 ❄️ | 0.80 ❄️ | $81.37_{\pm1.94}$ | $81.69_{\pm2.61}$ | $79.56_{\pm1.75}$ | $80.19_{\pm1.94}$ | $90.28_{\pm2.07}$ | $90.50_{\pm2.00}$ |
| 1 | 7 | -1.00 🔥 | 0.80 🔥 | $81.83_{\pm2.37}$ | $81.99_{\pm2.63}$ | $80.10_{\pm2.44}$ | $80.71_{\pm2.52}$ | $90.31_{\pm1.97}$ | $90.54_{\pm1.91}$ |
| 1 | 7 | -1.00 ❄️ | 0.90 ❄️ | $81.72_{\pm1.73}$ | $82.11_{\pm2.28}$ | $79.84_{\pm1.62}$ | $80.52_{\pm1.76}$ | $90.29_{\pm1.72}$ | $90.49_{\pm1.67}$ |
| 1 | 7 | -1.00 🔥 | 0.90 🔥 | $82.07_{\pm2.30}$ | $82.18_{\pm2.41}$ | $80.39_{\pm2.48}$ | $80.98_{\pm2.48}$ | $90.38_{\pm1.84}$ | $90.59_{\pm1.78}$ |

| t | n | a | b | Accuracy | Precision | Recall | F1 score | AUROC | AUPRC |
|---|---|---|---|---|---|---|---|---|---|
| 1 | 7 | -1.00 ❄ | 1.00 ❄ | $81.29_{\pm1.73}$ | $81.08_{\pm1.65}$ | $79.84_{\pm2.05}$ | $80.27_{\pm1.93}$ | $89.88_{\pm1.38}$ | $90.09_{\pm1.36}$ |
| 1 | 7 | -1.00 🔥 | 1.00 🔥 | $81.31_{\pm2.34}$ | $82.15_{\pm2.11}$ | $79.34_{\pm3.26}$ | $79.89_{\pm2.98}$ | $89.94_{\pm1.88}$ | $90.14_{\pm1.84}$ |
| 1 | 7 | -0.90 ❄ | -1.00 ❄ | $81.34_{\pm2.29}$ | $81.35_{\pm2.56}$ | $79.74_{\pm2.33}$ | $80.26_{\pm2.41}$ | $89.54_{\pm2.01}$ | $89.96_{\pm1.97}$ |
| 1 | 7 | -0.90 🔥 | -1.00 🔥 | $81.30_{\pm2.17}$ | $81.41_{\pm2.48}$ | $79.59_{\pm2.20}$ | $80.16_{\pm2.28}$ | $89.53_{\pm1.99}$ | $89.96_{\pm1.95}$ |
| 1 | 7 | -0.90 ❄ | -0.90 ❄ | $81.58_{\pm2.33}$ | $81.58_{\pm2.82}$ | $80.05_{\pm2.21}$ | $80.56_{\pm2.36}$ | $89.96_{\pm2.13}$ | $90.33_{\pm2.04}$ |
| 1 | 7 | -0.90 🔥 | -0.90 🔥 | $81.89_{\pm2.19}$ | $82.12_{\pm2.15}$ | $80.15_{\pm2.54}$ | $80.73_{\pm2.47}$ | $90.10_{\pm2.14}$ | $90.41_{\pm2.09}$ |
| 1 | 7 | -0.90 ❄ | -0.80 ❄ | $82.00_{\pm1.62}$ | $82.67_{\pm1.83}$ | $80.05_{\pm1.93}$ | $80.73_{\pm1.86}$ | $90.12_{\pm1.47}$ | $90.49_{\pm1.41}$ |
| 1 | 7 | -0.90 🔥 | -0.80 🔥 | $81.75_{\pm1.77}$ | $82.20_{\pm2.33}$ | $80.00_{\pm1.77}$ | $80.58_{\pm1.82}$ | $89.88_{\pm1.42}$ | $90.30_{\pm1.36}$ |
| 1 | 7 | -0.90 ❄ | -0.70 ❄ | $81.31_{\pm1.74}$ | $81.92_{\pm2.21}$ | $79.29_{\pm1.77}$ | $79.99_{\pm1.84}$ | $89.40_{\pm1.63}$ | $89.88_{\pm1.52}$ |
| 1 | 7 | -0.90 🔥 | -0.70 🔥 | $81.87_{\pm2.08}$ | $82.28_{\pm2.50}$ | $80.18_{\pm2.09}$ | $80.74_{\pm2.16}$ | $90.02_{\pm1.74}$ | $90.44_{\pm1.63}$ |
| 1 | 7 | -0.90 ❄ | -0.60 ❄ | $81.98_{\pm1.47}$ | $82.62_{\pm1.97}$ | $80.09_{\pm1.50}$ | $80.75_{\pm1.54}$ | $90.37_{\pm1.07}$ | $90.73_{\pm1.03}$ |
| 1 | 7 | -0.90 🔥 | -0.60 🔥 | $81.93_{\pm1.35}$ | $82.56_{\pm2.14}$ | $80.08_{\pm0.93}$ | $80.73_{\pm1.17}$ | $90.09_{\pm1.10}$ | $90.49_{\pm1.07}$ |
| 1 | 7 | -0.90 ❄ | -0.50 ❄ | $81.72_{\pm1.50}$ | $82.60_{\pm2.15}$ | $79.70_{\pm1.32}$ | $80.40_{\pm1.44}$ | $90.11_{\pm1.68}$ | $90.46_{\pm1.61}$ |
| 1 | 7 | -0.90 🔥 | -0.50 🔥 | $81.33_{\pm1.40}$ | $81.65_{\pm2.17}$ | $79.66_{\pm1.26}$ | $80.19_{\pm1.32}$ | $90.08_{\pm1.36}$ | $90.45_{\pm1.21}$ |
| 1 | 7 | -0.90 ❄ | -0.40 ❄ | $82.35_{\pm0.75}$ | $82.80_{\pm1.60}$ | $80.67_{\pm0.25}$ | $81.25_{\pm0.47}$ | $91.06_{\pm0.76}$ | $91.35_{\pm0.71}$ |
| 1 | 7 | -0.90 🔥 | -0.40 🔥 | $82.45_{\pm0.46}$ | $82.95_{\pm1.18}$ | $80.68_{\pm0.54}$ | $81.31_{\pm0.41}$ | $91.13_{\pm0.59}$ | $91.40_{\pm0.55}$ |
| 1 | 7 | -0.90 ❄ | -0.30 ❄ | $81.38_{\pm1.83}$ | $81.35_{\pm2.47}$ | $80.09_{\pm1.34}$ | $80.46_{\pm1.64}$ | $90.22_{\pm1.37}$ | $90.48_{\pm1.47}$ |
| 1 | 7 | -0.90 🔥 | -0.30 🔥 | $81.31_{\pm1.88}$ | $81.24_{\pm2.53}$ | $80.15_{\pm1.24}$ | $80.44_{\pm1.63}$ | $90.21_{\pm1.42}$ | $90.47_{\pm1.52}$ |
| 1 | 7 | -0.90 ❄ | -0.20 ❄ | $82.01_{\pm1.85}$ | $81.92_{\pm2.42}$ | $80.77_{\pm1.62}$ | $81.13_{\pm1.79}$ | $90.86_{\pm1.71}$ | $91.07_{\pm1.77}$ |
| 1 | 7 | -0.90 🔥 | -0.20 🔥 | $82.42_{\pm1.76}$ | $82.42_{\pm2.22}$ | $80.98_{\pm1.67}$ | $81.47_{\pm1.77}$ | $91.10_{\pm1.67}$ | $91.28_{\pm1.72}$ |
| 1 | 7 | -0.90 ❄ | -0.10 ❄ | $83.94_{\pm0.79}$ | $84.11_{\pm0.90}$ | $82.51_{\pm1.01}$ | $83.04_{\pm0.89}$ | $92.50_{\pm0.78}$ | $92.65_{\pm0.83}$ |
| 1 | 7 | -0.90 🔥 | -0.10 🔥 | $82.92_{\pm1.67}$ | $83.22_{\pm1.78}$ | $81.44_{\pm2.00}$ | $81.92_{\pm1.87}$ | $91.66_{\pm1.60}$ | $91.82_{\pm1.62}$ |
| 1 | 7 | -0.90 ❄ | 0.10 ❄ | $83.61_{\pm1.65}$ | $83.89_{\pm1.99}$ | $82.02_{\pm1.54}$ | $82.64_{\pm1.67}$ | $92.12_{\pm1.49}$ | $92.28_{\pm1.51}$ |
| 1 | 7 | -0.90 🔥 | 0.10 🔥 | $83.75_{\pm2.23}$ | $84.27_{\pm2.21}$ | $81.95_{\pm2.53}$ | $82.66_{\pm2.49}$ | $92.21_{\pm2.01}$ | $92.36_{\pm1.99}$ |
| 1 | 7 | -0.90 ❄ | 0.20 ❄ | $83.37_{\pm1.46}$ | $83.92_{\pm2.01}$ | $81.60_{\pm1.40}$ | $82.29_{\pm1.47}$ | $92.18_{\pm1.65}$ | $92.33_{\pm1.64}$ |
| 1 | 7 | -0.90 🔥 | 0.20 🔥 | $83.51_{\pm1.25}$ | $84.15_{\pm1.64}$ | $81.66_{\pm1.33}$ | $82.40_{\pm1.32}$ | $92.33_{\pm1.47}$ | $92.47_{\pm1.48}$ |
| 1 | 7 | -0.90 ❄ | 0.30 ❄ | $82.99_{\pm1.52}$ | $83.25_{\pm1.74}$ | $81.45_{\pm1.87}$ | $81.98_{\pm1.70}$ | $92.01_{\pm1.52}$ | $92.15_{\pm1.50}$ |
| 1 | 7 | -0.90 🔥 | 0.30 🔥 | $81.62_{\pm1.47}$ | $81.77_{\pm2.19}$ | $80.25_{\pm1.19}$ | $80.64_{\pm1.34}$ | $90.76_{\pm1.81}$ | $90.98_{\pm1.74}$ |
| 1 | 7 | -0.90 ❄ | 0.40 ❄ | $81.50_{\pm1.62}$ | $81.95_{\pm2.46}$ | $79.67_{\pm1.41}$ | $80.30_{\pm1.57}$ | $90.76_{\pm1.91}$ | $90.97_{\pm1.86}$ |
| 1 | 7 | -0.90 🔥 | 0.40 🔥 | $81.59_{\pm1.74}$ | $81.90_{\pm2.32}$ | $79.81_{\pm1.63}$ | $80.43_{\pm1.76}$ | $90.73_{\pm1.84}$ | $90.94_{\pm1.79}$ |
| 1 | 7 | -0.90 ❄ | 0.50 ❄ | $81.68_{\pm1.91}$ | $81.84_{\pm2.40}$ | $80.16_{\pm2.12}$ | $80.62_{\pm2.04}$ | $90.85_{\pm1.90}$ | $91.04_{\pm1.84}$ |
| 1 | 7 | -0.90 🔥 | 0.50 🔥 | $81.71_{\pm1.71}$ | $82.05_{\pm2.40}$ | $79.99_{\pm1.72}$ | $80.56_{\pm1.76}$ | $90.82_{\pm1.98}$ | $91.01_{\pm1.93}$ |
| 1 | 7 | -0.90 ❄ | 0.60 ❄ | $81.45_{\pm2.07}$ | $81.58_{\pm2.65}$ | $79.96_{\pm2.13}$ | $80.41_{\pm2.16}$ | $90.34_{\pm1.90}$ | $90.54_{\pm1.86}$ |
| 1 | 7 | -0.90 🔥 | 0.60 🔥 | $81.52_{\pm1.84}$ | $81.68_{\pm2.44}$ | $80.00_{\pm1.89}$ | $80.47_{\pm1.90}$ | $90.34_{\pm1.84}$ | $90.55_{\pm1.79}$ |
| 1 | 7 | -0.90 ❄ | 0.70 ❄ | $81.44_{\pm2.60}$ | $81.43_{\pm2.92}$ | $79.84_{\pm2.66}$ | $80.37_{\pm2.75}$ | $89.97_{\pm2.15}$ | $90.19_{\pm2.10}$ |
| 1 | 7 | -0.90 🔥 | 0.70 🔥 | $81.02_{\pm1.46}$ | $81.49_{\pm2.75}$ | $79.19_{\pm1.13}$ | $79.80_{\pm1.29}$ | $90.21_{\pm2.15}$ | $90.38_{\pm2.09}$ |
| 1 | 7 | -0.90 ❄ | 0.80 ❄ | $81.05_{\pm1.88}$ | $81.28_{\pm2.59}$ | $79.29_{\pm1.71}$ | $79.88_{\pm1.87}$ | $89.82_{\pm2.03}$ | $89.96_{\pm2.01}$ |
| 1 | 7 | -0.90 🔥 | 0.80 🔥 | $81.41_{\pm2.61}$ | $81.44_{\pm2.87}$ | $79.75_{\pm2.70}$ | $80.31_{\pm2.77}$ | $89.82_{\pm2.10}$ | $90.00_{\pm2.05}$ |
| 1 | 7 | -0.90 ❄ | 0.90 ❄ | $81.23_{\pm1.87}$ | $81.02_{\pm1.80}$ | $79.77_{\pm2.18}$ | $80.21_{\pm2.08}$ | $89.72_{\pm1.71}$ | $89.91_{\pm1.67}$ |
| 1 | 7 | -0.90 🔥 | 0.90 🔥 | $81.59_{\pm2.37}$ | $81.53_{\pm2.33}$ | $80.01_{\pm2.69}$ | $80.52_{\pm2.60}$ | $89.95_{\pm1.92}$ | $90.17_{\pm1.87}$ |
| 1 | 7 | -0.90 ❄ | 1.00 ❄ | $81.26_{\pm2.06}$ | $81.47_{\pm2.19}$ | $79.39_{\pm2.20}$ | $80.03_{\pm2.23}$ | $89.73_{\pm1.76}$ | $89.91_{\pm1.74}$ |
| 1 | 7 | -0.90 🔥 | 1.00 🔥 | $80.78_{\pm1.60}$ | $80.71_{\pm1.59}$ | $79.13_{\pm1.81}$ | $79.65_{\pm1.76}$ | $89.51_{\pm1.56}$ | $89.63_{\pm1.48}$ |
| 1 | 7 | -0.80 ❄ | -1.00 ❄ | $81.24_{\pm2.14}$ | $81.34_{\pm2.24}$ | $79.59_{\pm2.40}$ | $80.11_{\pm2.36}$ | $89.28_{\pm1.74}$ | $89.65_{\pm1.80}$ |
| 1 | 7 | -0.80 🔥 | -1.00 🔥 | $81.24_{\pm1.98}$ | $81.67_{\pm2.43}$ | $79.34_{\pm2.05}$ | $79.98_{\pm2.11}$ | $89.23_{\pm1.77}$ | $89.63_{\pm1.83}$ |
| 1 | 7 | -0.80 ❄ | -0.90 ❄ | $81.44_{\pm2.14}$ | $81.48_{\pm2.46}$ | $79.88_{\pm2.12}$ | $80.38_{\pm2.22}$ | $89.56_{\pm1.82}$ | $89.93_{\pm1.83}$ |
| 1 | 7 | -0.80 🔥 | -0.90 🔥 | $81.12_{\pm1.93}$ | $81.32_{\pm2.23}$ | $79.44_{\pm1.93}$ | $79.97_{\pm2.01}$ | $89.23_{\pm1.52}$ | $89.66_{\pm1.56}$ |
| 1 | 7 | -0.80 ❄ | -0.80 ❄ | $81.22_{\pm1.66}$ | $81.38_{\pm2.25}$ | $79.53_{\pm1.40}$ | $80.09_{\pm1.59}$ | $89.32_{\pm1.30}$ | $89.77_{\pm1.30}$ |
| 1 | 7 | -0.80 🔥 | -0.80 🔥 | $80.96_{\pm1.29}$ | $81.22_{\pm2.05}$ | $79.24_{\pm1.08}$ | $79.79_{\pm1.18}$ | $88.96_{\pm1.01}$ | $89.44_{\pm1.05}$ |
| 1 | 7 | -0.80 ❄ | -0.70 ❄ | $81.73_{\pm1.63}$ | $82.03_{\pm2.08}$ | $80.00_{\pm1.63}$ | $80.59_{\pm1.66}$ | $89.71_{\pm1.26}$ | $90.08_{\pm1.25}$ |
| 1 | 7 | -0.80 🔥 | -0.70 🔥 | $81.93_{\pm1.59}$ | $82.45_{\pm2.15}$ | $80.08_{\pm1.61}$ | $80.73_{\pm1.64}$ | $89.96_{\pm1.34}$ | $90.30_{\pm1.33}$ |

| t | n | a | b | Accuracy | Precision | Recall | F1 score | AUROC | AUPRC |
|---|---|---|---|---|---|---|---|---|---|
| 1 | 7 | -0.80 ❄ | -0.60 ❄ | 81.73±1.42 | 81.72±1.72 | 80.22±1.34 | 80.71±1.42 | 89.55±1.25 | 89.95±1.21 |
| 1 | 7 | -0.80 🔥 | -0.60 🔥 | 81.52±1.45 | 81.87±2.22 | 79.83±1.16 | 80.39±1.31 | 89.60±1.05 | 90.00±1.04 |
| 1 | 7 | -0.80 ❄ | -0.50 ❄ | 81.40±1.46 | 81.51±1.70 | 79.76±1.51 | 80.29±1.52 | 89.54±1.56 | 89.88±1.47 |
| 1 | 7 | -0.80 🔥 | -0.50 🔥 | 80.99±1.82 | 81.09±1.84 | 79.30±2.07 | 79.83±2.03 | 89.14±1.94 | 89.44±1.89 |
| 1 | 7 | -0.80 ❄ | -0.40 ❄ | 81.09±1.65 | 81.21±1.75 | 79.40±1.80 | 79.94±1.79 | 89.26±1.86 | 89.49±1.78 |
| 1 | 7 | -0.80 🔥 | -0.40 🔥 | 80.98±1.50 | 81.04±1.59 | 79.32±1.68 | 79.84±1.65 | 89.21±1.65 | 89.48±1.66 |
| 1 | 7 | -0.80 ❄ | -0.30 ❄ | 82.03±1.41 | 82.48±2.02 | 80.30±1.15 | 80.90±1.31 | 90.40±1.25 | 90.68±1.21 |
| 1 | 7 | -0.80 🔥 | -0.30 🔥 | 82.08±1.29 | 82.60±1.95 | 80.30±0.89 | 80.93±1.12 | 90.39±1.17 | 90.69±1.14 |
| 1 | 7 | -0.80 ❄ | -0.20 ❄ | 81.93±1.35 | 82.02±1.85 | 80.40±0.99 | 80.91±1.21 | 90.31±1.25 | 90.60±1.27 |
| 1 | 7 | -0.80 🔥 | -0.20 🔥 | 81.33±1.84 | 81.22±2.32 | 79.94±1.59 | 80.36±1.77 | 89.96±1.68 | 90.25±1.72 |
| 1 | 7 | -0.80 ❄ | -0.10 ❄ | 82.59±1.87 | 82.58±2.35 | 81.24±1.57 | 81.68±1.79 | 91.01±1.85 | 91.20±1.87 |
| 1 | 7 | -0.80 🔥 | -0.10 🔥 | 81.73±1.76 | 81.47±2.19 | 80.55±1.54 | 80.88±1.72 | 90.39±1.72 | 90.60±1.76 |
| 1 | 7 | -0.80 ❄ | 0.10 ❄ | 82.89±1.77 | 83.56±2.02 | 81.01±1.90 | 81.72±1.91 | 91.67±1.74 | 91.83±1.74 |
| 1 | 7 | -0.80 🔥 | 0.10 🔥 | 82.25±1.79 | 82.78±2.06 | 80.35±1.89 | 81.04±1.91 | 91.08±1.69 | 91.28±1.69 |
| 1 | 7 | -0.80 ❄ | 0.20 ❄ | 82.03±2.50 | 82.65±2.68 | 80.14±2.73 | 80.78±2.76 | 90.83±2.46 | 91.04±2.40 |
| 1 | 7 | -0.80 🔥 | 0.20 🔥 | 82.70±1.52 | 83.16±2.03 | 80.95±1.45 | 81.59±1.52 | 91.58±1.60 | 91.73±1.62 |
| 1 | 7 | -0.80 ❄ | 0.30 ❄ | 82.07±1.68 | 82.51±2.06 | 80.25±1.76 | 80.89±1.80 | 90.43±2.29 | 90.65±2.22 |
| 1 | 7 | -0.80 🔥 | 0.30 🔥 | 81.27±1.65 | 81.43±2.11 | 79.82±1.66 | 80.23±1.70 | 90.11±2.22 | 90.33±2.14 |
| 1 | 7 | -0.80 ❄ | 0.40 ❄ | 81.23±1.54 | 81.37±2.17 | 79.65±1.41 | 80.15±1.51 | 90.19±1.80 | 90.43±1.75 |
| 1 | 7 | -0.80 🔥 | 0.40 🔥 | 82.29±1.25 | 82.70±1.85 | 80.50±1.06 | 81.16±1.18 | 90.92±1.67 | 91.11±1.64 |
| 1 | 7 | -0.80 ❄ | 0.50 ❄ | 80.92±1.92 | 81.08±2.68 | 79.51±1.75 | 79.89±1.85 | 89.95±2.06 | 90.15±2.03 |
| 1 | 7 | -0.80 🔥 | 0.50 🔥 | 80.94±1.93 | 80.81±2.53 | 79.72±1.83 | 80.01±1.91 | 89.87±2.00 | 90.07±1.99 |
| 1 | 7 | -0.80 ❄ | 0.60 ❄ | 80.92±1.66 | 80.55±1.65 | 79.75±1.98 | 80.02±1.85 | 89.67±1.66 | 89.82±1.68 |
| 1 | 7 | -0.80 🔥 | 0.60 🔥 | 80.99±1.90 | 80.75±2.03 | 79.70±2.09 | 80.04±2.04 | 89.65±1.59 | 89.87±1.57 |
| 1 | 8 | -1.00 ❄ | -1.00 ❄ | 84.12±1.05 | 84.18±1.59 | 82.90±0.95 | 83.32±1.01 | 92.68±0.93 | 92.78±0.89 |
| 1 | 8 | -1.00 🔥 | -1.00 🔥 | 84.91±0.96 | 85.15±1.53 | 83.57±0.62 | 84.09±0.85 | 93.08±0.98 | 93.16±0.87 |
| 1 | 8 | -1.00 ❄ | -0.90 ❄ | 84.32±1.62 | 84.10±1.90 | 83.36±1.39 | 83.64±1.57 | 92.63±1.28 | 92.79±1.21 |
| 1 | 8 | -1.00 🔥 | -0.90 🔥 | 84.89±1.03 | 85.03±1.45 | 83.63±0.85 | 84.11±0.98 | 92.88±1.08 | 93.06±1.05 |
| 1 | 8 | -1.00 ❄ | -0.80 ❄ | 85.27±0.85 | 85.49±1.00 | 84.10±1.06 | 84.51±0.90 | 93.30±0.76 | 93.44±0.74 |
| 1 | 8 | -1.00 🔥 | -0.80 🔥 | 85.10±0.87 | 85.61±1.22 | 83.72±1.26 | 84.24±1.03 | 93.38±0.90 | 93.52±0.86 |
| 1 | 8 | -1.00 ❄ | -0.70 ❄ | 84.91±0.91 | 85.51±1.65 | 83.42±0.99 | 84.00±0.91 | 93.67±1.08 | 93.80±1.07 |
| 1 | 8 | -1.00 🔥 | -0.70 🔥 | 84.56±0.64 | 85.52±1.80 | 82.89±0.74 | 83.55±0.52 | 93.70±1.03 | 93.81±1.03 |
| 1 | 8 | -1.00 ❄ | -0.60 ❄ | 84.40±0.69 | 85.27±1.55 | 82.70±0.97 | 83.37±0.77 | 93.56±1.25 | 93.70±1.19 |
| 1 | 8 | -1.00 🔥 | -0.60 🔥 | 84.98±0.66 | 85.65±1.36 | 83.45±1.01 | 84.05±0.77 | 93.82±1.08 | 93.95±1.03 |
| 1 | 8 | -1.00 ❄ | -0.50 ❄ | 85.21±0.92 | 85.91±1.28 | 83.61±1.16 | 84.28±1.02 | 93.70±1.11 | 93.83±1.07 |
| 1 | 8 | -1.00 🔥 | -0.50 🔥 | 85.34±0.93 | 85.35±0.76 | 84.18±1.28 | 84.60±1.09 | 93.49±0.94 | 93.63±0.90 |
| 1 | 8 | -1.00 ❄ | -0.40 ❄ | 84.98±0.90 | 85.77±1.28 | 83.22±1.02 | 83.97±0.97 | 93.80±0.85 | 93.92±0.84 |
| 1 | 8 | -1.00 🔥 | -0.40 🔥 | 84.47±1.51 | 85.36±1.72 | 82.54±1.64 | 83.37±1.65 | 93.27±1.55 | 93.45±1.46 |
| 1 | 8 | -1.00 ❄ | -0.30 ❄ | 84.25±1.35 | 85.00±1.36 | 82.38±1.61 | 83.16±1.53 | 93.02±1.34 | 93.17±1.26 |
| 1 | 8 | -1.00 🔥 | -0.30 🔥 | 84.56±0.86 | 85.30±0.89 | 82.75±1.15 | 83.51±1.02 | 93.59±0.72 | 93.72±0.69 |
| 1 | 8 | -1.00 ❄ | -0.20 ❄ | 83.35±2.14 | 83.95±2.93 | 81.69±1.57 | 82.34±1.96 | 92.57±2.06 | 92.73±1.95 |
| 1 | 8 | -1.00 🔥 | -0.20 🔥 | 83.87±1.29 | 84.91±1.72 | 81.79±1.22 | 82.67±1.33 | 93.04±1.41 | 93.18±1.31 |
| 1 | 8 | -1.00 ❄ | -0.10 ❄ | 83.47±2.14 | 83.80±2.53 | 81.93±2.24 | 82.49±2.21 | 92.22±1.91 | 92.39±1.80 |
| 1 | 8 | -1.00 🔥 | -0.10 🔥 | 83.58±2.04 | 83.92±2.43 | 81.99±2.02 | 82.60±2.09 | 92.37±1.95 | 92.53±1.84 |
| 1 | 8 | -1.00 ❄ | 0.10 ❄ | 83.59±1.46 | 83.74±1.51 | 82.09±1.67 | 82.65±1.60 | 92.21±1.25 | 92.36±1.22 |
| 1 | 8 | -1.00 🔥 | 0.10 🔥 | 82.91±2.16 | 83.13±2.21 | 81.23±2.38 | 81.85±2.36 | 91.17±2.32 | 91.37±2.17 |
| 1 | 8 | -1.00 ❄ | 0.20 ❄ | 81.31±2.64 | 81.25±2.71 | 79.83±2.89 | 80.27±2.87 | 90.09±2.07 | 90.27±2.07 |
| 1 | 8 | -1.00 🔥 | 0.20 🔥 | 80.77±2.03 | 80.77±2.39 | 79.25±2.13 | 79.69±2.16 | 89.65±2.12 | 89.73±2.13 |
| 1 | 8 | -1.00 ❄ | 0.30 ❄ | 81.29±2.45 | 81.41±2.65 | 79.66±2.56 | 80.17±2.60 | 89.46±2.36 | 89.71±2.24 |
| 1 | 8 | -1.00 🔥 | 0.30 🔥 | 81.30±2.50 | 81.36±2.75 | 79.72±2.57 | 80.21±2.63 | 89.43±2.52 | 89.69±2.40 |

| t | n | a | b | Accuracy | Precision | Recall | F1 score | AUROC | AUPRC |
|---|---|---|---|---|---|---|---|---|---|
| 1 | 8 | -1.00 ❄ | 0.40 ❄ | 80.31±2.06 | 80.88±2.68 | 78.15±1.95 | 78.87±2.09 | 89.10±2.26 | 89.35±2.17 |
| 1 | 8 | -1.00 🔥 | 0.40 🔥 | 80.17±2.20 | 80.59±2.73 | 78.06±2.13 | 78.76±2.28 | 88.60±2.62 | 88.90±2.50 |
| 1 | 8 | -1.00 ❄ | 0.50 ❄ | 79.04±2.24 | 79.87±3.14 | 76.60±2.07 | 77.34±2.24 | 87.44±2.78 | 87.79±2.67 |
| 1 | 8 | -1.00 🔥 | 0.50 🔥 | 79.40±2.11 | 79.94±2.83 | 77.16±1.89 | 77.86±2.08 | 88.00±2.29 | 88.35±2.21 |
| 1 | 8 | -1.00 ❄ | 0.60 ❄ | 78.32±2.40 | 78.88±2.95 | 76.20±2.09 | 76.76±2.29 | 86.79±2.20 | 87.16±2.11 |
| 1 | 8 | -1.00 🔥 | 0.60 🔥 | 78.53±2.37 | 79.03±2.94 | 76.44±2.10 | 77.01±2.29 | 87.11±2.10 | 87.48±2.06 |
| 1 | 8 | -1.00 ❄ | 0.70 ❄ | 78.16±2.00 | 78.58±2.40 | 75.89±1.95 | 76.51±2.07 | 86.46±2.07 | 86.89±2.01 |
| 1 | 8 | -1.00 🔥 | 0.70 🔥 | 77.83±1.94 | 78.63±2.46 | 75.29±1.89 | 75.96±2.01 | 85.92±2.25 | 86.34±2.19 |
| 1 | 8 | -1.00 ❄ | 0.80 ❄ | 77.23±1.36 | 77.84±1.99 | 74.72±1.32 | 75.35±1.38 | 85.34±1.85 | 85.69±1.77 |
| 1 | 8 | -1.00 🔥 | 0.80 🔥 | 77.23±1.63 | 78.03±2.45 | 74.63±1.34 | 75.29±1.49 | 85.38±2.03 | 85.82±2.03 |
| 1 | 8 | -1.00 ❄ | 0.90 ❄ | 76.79±1.74 | 77.49±2.37 | 74.23±1.77 | 74.82±1.83 | 84.53±2.01 | 85.01±2.00 |
| 1 | 8 | -1.00 🔥 | 0.90 🔥 | 76.52±1.27 | 76.86±1.96 | 74.11±1.03 | 74.68±1.13 | 84.70±1.39 | 85.11±1.30 |
| 1 | 8 | -1.00 ❄ | 1.00 ❄ | 77.04±1.46 | 77.68±2.25 | 74.54±0.97 | 75.16±1.16 | 84.76±1.19 | 85.27±1.21 |
| 1 | 8 | -1.00 🔥 | 1.00 🔥 | 76.90±1.48 | 77.73±2.41 | 74.32±0.83 | 74.94±1.05 | 84.52±1.16 | 85.02±1.15 |
| 1 | 8 | -0.90 ❄ | -1.00 ❄ | 83.35±1.73 | 83.28±1.97 | 82.35±1.80 | 82.59±1.79 | 92.15±0.89 | 92.26±0.86 |
| 1 | 8 | -0.90 🔥 | -1.00 🔥 | 83.07±1.89 | 82.95±2.03 | 82.08±2.08 | 82.30±2.01 | 91.80±1.12 | 91.91±1.09 |
| 1 | 8 | -0.90 ❄ | -0.90 ❄ | 83.58±1.84 | 83.36±2.19 | 82.69±1.73 | 82.88±1.84 | 92.20±1.08 | 92.31±1.04 |
| 1 | 8 | -0.90 🔥 | -0.90 🔥 | 84.43±1.40 | 84.28±1.65 | 83.42±1.39 | 83.72±1.43 | 92.62±0.87 | 92.67±0.85 |
| 1 | 8 | -0.90 ❄ | -0.80 ❄ | 83.86±2.07 | 83.70±2.40 | 82.92±1.89 | 83.16±2.04 | 92.31±1.21 | 92.44±1.21 |
| 1 | 8 | -0.90 🔥 | -0.80 🔥 | 83.70±1.81 | 83.76±2.48 | 82.63±1.49 | 82.94±1.69 | 92.52±1.24 | 92.64±1.27 |
| 1 | 8 | -0.90 ❄ | -0.70 ❄ | 84.36±0.98 | 84.79±1.91 | 82.93±0.80 | 83.47±0.86 | 92.78±1.16 | 92.93±1.16 |
| 1 | 8 | -0.90 🔥 | -0.70 🔥 | 83.97±1.05 | 84.46±2.13 | 82.56±0.81 | 83.06±0.88 | 92.77±1.07 | 92.92±1.07 |
| 1 | 8 | -0.90 ❄ | -0.60 ❄ | 84.74±1.29 | 84.95±1.72 | 83.49±1.01 | 83.94±1.21 | 93.28±0.90 | 93.38±0.89 |
| 1 | 8 | -0.90 🔥 | -0.60 🔥 | 84.58±1.37 | 85.19±2.20 | 83.14±0.87 | 83.70±1.18 | 93.33±1.18 | 93.43±1.16 |
| 1 | 8 | -0.90 ❄ | -0.50 ❄ | 84.64±1.22 | 84.72±1.40 | 83.44±1.35 | 83.86±1.29 | 93.01±1.12 | 93.17±1.07 |
| 1 | 8 | -0.90 🔥 | -0.50 🔥 | 84.72±1.08 | 85.00±1.39 | 83.29±1.05 | 83.86±1.10 | 93.01±1.17 | 93.17±1.12 |
| 1 | 8 | -0.90 ❄ | -0.40 ❄ | 84.75±1.32 | 84.76±1.42 | 83.51±1.45 | 83.97±1.41 | 93.01±1.02 | 93.14±1.02 |
| 1 | 8 | -0.90 🔥 | -0.40 🔥 | 84.57±0.89 | 84.90±1.15 | 83.05±0.96 | 83.66±0.94 | 93.08±0.92 | 93.20±0.93 |
| 1 | 8 | -0.90 ❄ | -0.30 ❄ | 84.92±1.28 | 85.22±1.41 | 83.47±1.46 | 84.05±1.40 | 93.35±1.01 | 93.45±1.02 |
| 1 | 8 | -0.90 🔥 | -0.30 🔥 | 84.25±1.82 | 84.32±2.00 | 82.91±1.89 | 83.41±1.90 | 92.82±1.33 | 92.92±1.37 |
| 1 | 8 | -0.90 ❄ | -0.20 ❄ | 84.44±1.35 | 84.82±1.20 | 82.87±1.72 | 83.49±1.56 | 92.75±1.31 | 92.89±1.26 |
| 1 | 8 | -0.90 🔥 | -0.20 🔥 | 83.97±1.12 | 84.59±1.12 | 82.14±1.29 | 82.89±1.26 | 92.55±1.22 | 92.71±1.14 |
| 1 | 8 | -0.90 ❄ | -0.10 ❄ | 83.90±0.94 | 84.18±0.92 | 82.35±1.26 | 82.94±1.10 | 92.53±0.92 | 92.69±0.88 |
| 1 | 8 | -0.90 🔥 | -0.10 🔥 | 83.61±0.93 | 84.06±1.27 | 81.91±1.05 | 82.57±1.00 | 92.44±1.04 | 92.60±1.03 |
| 1 | 8 | -0.90 ❄ | 0.10 ❄ | 82.14±2.78 | 82.44±3.33 | 80.65±2.35 | 81.15±2.65 | 91.02±2.35 | 91.26±2.25 |
| 1 | 8 | -0.90 🔥 | 0.10 🔥 | 82.38±1.98 | 82.59±2.42 | 80.78±1.74 | 81.35±1.94 | 91.09±2.04 | 91.31±1.97 |
| 1 | 8 | -0.90 ❄ | 0.20 ❄ | 82.18±2.34 | 82.46±2.89 | 80.66±2.11 | 81.16±2.29 | 90.44±2.32 | 90.70±2.24 |
| 1 | 8 | -0.90 🔥 | 0.20 🔥 | 81.24±2.99 | 81.32±3.43 | 79.81±2.77 | 80.24±2.98 | 89.57±2.76 | 89.77±2.69 |
| 1 | 8 | -0.90 ❄ | 0.30 ❄ | 80.75±2.19 | 80.96±2.76 | 79.06±2.09 | 79.59±2.20 | 88.97±2.63 | 89.23±2.54 |
| 1 | 8 | -0.90 🔥 | 0.30 🔥 | 80.66±2.40 | 80.85±2.84 | 78.97±2.38 | 79.49±2.46 | 89.08±2.65 | 89.35±2.56 |
| 1 | 8 | -0.90 ❄ | 0.40 ❄ | 79.64±2.51 | 79.80±3.08 | 77.79±2.29 | 78.35±2.48 | 88.31±2.32 | 88.62±2.25 |
| 1 | 8 | -0.90 🔥 | 0.40 🔥 | 79.76±2.79 | 79.68±3.28 | 78.15±2.63 | 78.62±2.80 | 88.16±2.37 | 88.48±2.32 |
| 1 | 8 | -0.90 ❄ | 0.50 ❄ | 78.55±2.61 | 78.46±3.01 | 76.71±2.53 | 77.22±2.67 | 86.46±2.46 | 86.85±2.44 |
| 1 | 8 | -0.90 🔥 | 0.50 🔥 | 78.46±2.48 | 78.51±2.87 | 76.57±2.47 | 77.08±2.57 | 86.58±2.23 | 86.96±2.21 |
| 1 | 8 | -0.90 ❄ | 0.60 ❄ | 77.90±1.95 | 77.77±2.17 | 76.02±2.07 | 76.51±2.11 | 85.68±2.56 | 86.09±2.56 |
| 1 | 8 | -0.90 🔥 | 0.60 🔥 | 78.36±1.87 | 78.18±2.15 | 76.55±1.86 | 77.05±1.93 | 86.32±1.81 | 86.69±1.86 |
| 1 | 8 | -0.90 ❄ | 0.70 ❄ | 78.03±1.81 | 78.65±2.39 | 75.67±1.81 | 76.29±1.89 | 85.96±2.14 | 86.36±2.11 |
| 1 | 8 | -0.90 🔥 | 0.70 🔥 | 77.76±1.79 | 78.52±2.30 | 75.11±1.76 | 75.83±1.87 | 85.64±2.09 | 86.08±2.07 |
| 1 | 8 | -0.90 ❄ | 0.80 ❄ | 77.64±2.23 | 77.93±2.83 | 75.36±2.06 | 75.98±2.20 | 85.53±1.89 | 85.99±1.91 |
| 1 | 8 | -0.90 🔥 | 0.80 🔥 | 77.41±1.95 | 78.00±2.28 | 74.82±2.10 | 75.49±2.15 | 85.18±2.41 | 85.53±2.47 |

| t | n | a | b | Accuracy | Precision | Recall | F1 score | AUROC | AUPRC |
|---|---|---|---|---|---|---|---|---|---|
| 1 | 8 | -0.90 ❄ | 0.90 ❄ | 76.72±2.13 | 77.53±2.91 | 74.22±1.95 | 74.77±2.11 | 84.26±2.17 | 84.69±2.23 |
| 1 | 8 | -0.90 🔥 | 0.90 🔥 | 76.70±1.98 | 77.20±2.64 | 74.36±1.82 | 74.89±1.95 | 84.27±2.06 | 84.68±2.15 |
| 1 | 8 | -0.90 ❄ | 1.00 ❄ | 77.01±1.66 | 77.79±2.43 | 74.34±1.23 | 75.02±1.41 | 84.23±1.65 | 84.73±1.64 |
| 1 | 8 | -0.90 🔥 | 1.00 🔥 | 77.09±1.36 | 77.87±2.13 | 74.47±1.04 | 75.12±1.18 | 84.27±1.44 | 84.70±1.52 |
| 1 | 8 | -0.80 ❄ | -1.00 ❄ | 83.51±1.91 | 83.37±1.61 | 82.43±2.47 | 82.70±2.24 | 91.74±1.31 | 91.81±1.33 |
| 1 | 8 | -0.80 🔥 | -1.00 🔥 | 83.33±1.98 | 83.22±1.60 | 82.31±2.64 | 82.51±2.38 | 91.59±1.46 | 91.68±1.46 |
| 1 | 8 | -0.80 ❄ | -0.90 ❄ | 83.44±1.35 | 83.84±2.08 | 82.11±1.44 | 82.52±1.37 | 92.12±0.96 | 92.23±1.04 |
| 1 | 8 | -0.80 🔥 | -0.90 🔥 | 83.72±1.48 | 83.55±1.62 | 82.97±1.74 | 83.05±1.58 | 92.21±0.94 | 92.26±0.98 |
| 1 | 8 | -0.80 ❄ | -0.80 ❄ | 83.86±1.36 | 84.20±2.17 | 82.59±1.33 | 83.00±1.30 | 92.50±1.00 | 92.58±1.04 |
| 1 | 8 | -0.80 🔥 | -0.80 🔥 | 83.87±1.40 | 83.95±1.92 | 82.76±1.41 | 83.08±1.38 | 92.48±1.02 | 92.57±1.05 |
| 1 | 8 | -0.80 ❄ | -0.70 ❄ | 83.89±1.05 | 84.03±2.00 | 82.76±0.86 | 83.10±0.91 | 92.50±1.13 | 92.62±1.15 |
| 1 | 8 | -0.80 🔥 | -0.70 🔥 | 83.72±1.05 | 83.94±2.00 | 82.58±0.99 | 82.91±0.93 | 92.52±1.11 | 92.63±1.14 |
| 1 | 8 | -0.80 ❄ | -0.60 ❄ | 84.30±1.36 | 84.49±2.21 | 83.14±0.97 | 83.53±1.21 | 92.69±1.25 | 92.78±1.28 |
| 1 | 8 | -0.80 🔥 | -0.60 🔥 | 84.29±1.12 | 84.50±1.95 | 83.16±0.81 | 83.52±0.95 | 92.73±1.18 | 92.81±1.21 |
| 1 | 8 | -0.80 ❄ | -0.50 ❄ | 84.33±1.52 | 84.47±2.04 | 83.07±1.26 | 83.53±1.45 | 92.61±1.20 | 92.72±1.16 |
| 1 | 8 | -0.80 🔥 | -0.50 🔥 | 84.28±1.48 | 84.19±1.77 | 83.14±1.37 | 83.52±1.48 | 92.58±1.10 | 92.71±1.06 |
| 1 | 8 | -0.80 ❄ | -0.40 ❄ | 84.01±1.26 | 83.98±1.67 | 82.81±1.01 | 83.22±1.19 | 92.53±0.98 | 92.69±0.96 |
| 1 | 8 | -0.80 🔥 | -0.40 🔥 | 83.86±1.77 | 83.75±2.12 | 82.70±1.70 | 83.08±1.79 | 92.17±1.43 | 92.34±1.40 |
| 1 | 8 | -0.80 ❄ | -0.30 ❄ | 83.84±1.98 | 83.97±2.53 | 82.59±2.03 | 83.01±2.02 | 92.42±1.58 | 92.54±1.61 |
| 1 | 8 | -0.80 🔥 | -0.30 🔥 | 83.98±1.79 | 84.35±1.64 | 82.36±2.16 | 82.98±2.04 | 92.34±1.49 | 92.50±1.45 |
| 1 | 8 | -0.80 ❄ | -0.20 ❄ | 83.82±1.36 | 83.99±1.36 | 82.28±1.54 | 82.87±1.49 | 92.19±1.17 | 92.35±1.16 |
| 1 | 8 | -0.80 🔥 | -0.20 🔥 | 83.40±1.55 | 83.71±1.40 | 81.74±1.88 | 82.36±1.78 | 91.81±1.25 | 91.99±1.18 |
| 1 | 8 | -0.80 ❄ | -0.10 ❄ | 83.61±1.09 | 83.97±1.26 | 81.94±1.17 | 82.59±1.17 | 92.05±1.05 | 92.19±1.09 |
| 1 | 8 | -0.80 🔥 | -0.10 🔥 | 83.49±1.10 | 83.75±1.18 | 81.86±1.18 | 82.49±1.18 | 92.00±1.16 | 92.16±1.18 |
| 1 | 8 | -0.80 ❄ | 0.10 ❄ | 80.88±2.92 | 81.02±3.31 | 79.41±2.37 | 79.85±2.74 | 89.83±2.36 | 90.14±2.27 |
| 1 | 8 | -0.80 🔥 | 0.10 🔥 | 81.72±3.09 | 81.92±3.52 | 80.36±2.59 | 80.77±2.93 | 90.46±2.33 | 90.72±2.27 |
| 1 | 8 | -0.80 ❄ | 0.20 ❄ | 81.36±1.99 | 81.66±2.51 | 79.67±1.84 | 80.22±1.97 | 89.72±2.24 | 89.99±2.21 |
| 1 | 8 | -0.80 🔥 | 0.20 🔥 | 81.06±2.13 | 81.59±2.58 | 79.18±2.12 | 79.79±2.19 | 89.30±2.43 | 89.60±2.33 |
| 1 | 8 | -0.80 ❄ | 0.30 ❄ | 79.82±2.62 | 80.09±3.48 | 78.20±2.10 | 78.66±2.42 | 88.38±2.57 | 88.69±2.51 |
| 1 | 8 | -0.80 🔥 | 0.30 🔥 | 80.84±2.17 | 81.23±2.76 | 79.13±1.84 | 79.66±2.06 | 89.12±2.10 | 89.41±2.09 |
| 1 | 8 | -0.80 ❄ | 0.40 ❄ | 79.54±2.09 | 79.91±2.97 | 77.54±1.65 | 78.17±1.91 | 87.71±2.30 | 88.05±2.27 |
| 1 | 8 | -0.80 🔥 | 0.40 🔥 | 79.50±2.29 | 79.84±3.09 | 77.50±1.94 | 78.12±2.17 | 87.79±2.39 | 88.12±2.34 |
| 1 | 8 | -0.80 ❄ | 0.50 ❄ | 78.56±2.26 | 78.97±3.48 | 76.50±1.65 | 77.11±1.97 | 86.58±2.73 | 86.88±2.66 |
| 1 | 8 | -0.80 🔥 | 0.50 🔥 | 78.18±2.26 | 78.50±3.41 | 76.17±1.67 | 76.74±1.97 | 86.59±2.73 | 86.97±2.71 |
| 1 | 8 | -0.80 ❄ | 0.60 ❄ | 77.86±1.91 | 77.81±2.33 | 75.84±1.89 | 76.39±1.99 | 85.38±2.65 | 85.79±2.67 |
| 1 | 8 | -0.80 🔥 | 0.60 🔥 | 78.25±1.87 | 78.47±2.91 | 76.18±1.47 | 76.77±1.69 | 86.13±2.27 | 86.51±2.29 |
| 1 | 9 | -1.00 ❄ | -1.00 ❄ | 86.16±1.34 | 85.62±1.38 | 86.00±1.37 | 85.77±1.37 | 94.32±1.02 | 94.47±0.99 |
| 1 | 9 | -1.00 🔥 | -1.00 🔥 | 86.07±1.14 | 85.54±1.21 | 86.20±1.51 | 85.73±1.24 | 94.53±1.28 | 94.66±1.22 |
| 1 | 9 | -1.00 ❄ | -0.90 ❄ | 85.48±2.00 | 85.21±1.97 | 85.39±1.57 | 85.08±1.92 | 94.20±1.28 | 94.35±1.23 |
| 1 | 9 | -1.00 🔥 | -0.90 🔥 | 85.70±2.10 | 85.35±2.03 | 85.65±1.80 | 85.32±2.06 | 94.25±1.28 | 94.40±1.24 |
| 1 | 9 | -1.00 ❄ | -0.80 ❄ | 85.81±1.73 | 85.47±1.72 | 85.78±1.38 | 85.44±1.67 | 94.35±1.09 | 94.48±1.05 |
| 1 | 9 | -1.00 🔥 | -0.80 🔥 | 85.37±2.62 | 85.34±2.22 | 85.34±1.87 | 84.97±2.49 | 94.12±1.18 | 94.27±1.15 |
| 1 | 9 | -1.00 ❄ | -0.70 ❄ | 82.91±4.09 | 83.81±3.21 | 83.49±2.51 | 82.57±3.81 | 93.75±1.11 | 93.92±1.09 |
| 1 | 9 | -1.00 🔥 | -0.70 🔥 | 84.01±3.68 | 84.45±2.76 | 84.01±2.30 | 83.57±3.44 | 93.89±1.01 | 94.04±0.99 |
| 1 | 9 | -1.00 ❄ | -0.60 ❄ | 85.31±1.43 | 84.94±1.55 | 85.29±1.13 | 84.93±1.35 | 94.10±0.93 | 94.26±0.87 |
| 1 | 9 | -1.00 🔥 | -0.60 🔥 | 86.26±0.76 | 85.84±0.76 | 86.17±0.49 | 85.88±0.67 | 94.49±0.44 | 94.62±0.42 |
| 1 | 9 | -1.00 ❄ | -0.50 ❄ | 86.54±1.55 | 86.33±1.88 | 86.10±1.11 | 86.08±1.46 | 94.43±0.70 | 94.57±0.67 |
| 1 | 9 | -1.00 🔥 | -0.50 🔥 | 86.39±1.06 | 86.04±1.20 | 86.14±0.66 | 85.97±0.96 | 94.53±0.50 | 94.66±0.49 |
| 1 | 9 | -1.00 ❄ | -0.40 ❄ | 85.74±1.00 | 85.57±1.09 | 85.18±1.26 | 85.20±1.07 | 94.29±0.59 | 94.43±0.57 |
| 1 | 9 | -1.00 🔥 | -0.40 🔥 | 85.77±1.17 | 85.60±1.22 | 85.19±1.35 | 85.23±1.24 | 94.19±0.58 | 94.34±0.54 |

| t | n | a | b | Accuracy | Precision | Recall | F1 score | AUROC | AUPRC |
|---|---|---|---|---|---|---|---|---|---|
| 1 | 9 | -1.00 ❄ | -0.30 ❄ | $85.56_{\pm1.36}$ | $85.57_{\pm1.26}$ | $84.83_{\pm1.99}$ | $84.93_{\pm1.62}$ | $93.99_{\pm0.82}$ | $94.14_{\pm0.78}$ |
| 1 | 9 | -1.00 🔥 | -0.30 🔥 | $85.34_{\pm1.66}$ | $85.45_{\pm1.64}$ | $84.50_{\pm2.06}$ | $84.67_{\pm1.87}$ | $93.87_{\pm0.82}$ | $94.03_{\pm0.76}$ |
| 1 | 9 | -1.00 ❄ | -0.20 ❄ | $85.31_{\pm0.85}$ | $85.05_{\pm1.04}$ | $84.51_{\pm0.82}$ | $84.70_{\pm0.84}$ | $93.58_{\pm0.71}$ | $93.75_{\pm0.67}$ |
| 1 | 9 | -1.00 🔥 | -0.20 🔥 | $85.52_{\pm1.39}$ | $85.49_{\pm1.78}$ | $84.50_{\pm1.14}$ | $84.85_{\pm1.34}$ | $93.65_{\pm0.88}$ | $93.81_{\pm0.82}$ |
| 1 | 9 | -1.00 ❄ | -0.10 ❄ | $84.65_{\pm0.96}$ | $84.43_{\pm1.24}$ | $83.79_{\pm0.92}$ | $84.00_{\pm0.94}$ | $93.34_{\pm0.74}$ | $93.51_{\pm0.72}$ |
| 1 | 9 | -1.00 🔥 | -0.10 🔥 | $85.03_{\pm1.42}$ | $85.30_{\pm2.07}$ | $83.69_{\pm1.03}$ | $84.23_{\pm1.32}$ | $93.62_{\pm0.93}$ | $93.77_{\pm0.88}$ |
| 1 | 9 | -1.00 ❄ | 0.10 ❄ | $83.35_{\pm0.99}$ | $84.57_{\pm1.70}$ | $81.21_{\pm0.99}$ | $82.07_{\pm1.03}$ | $92.63_{\pm1.25}$ | $92.79_{\pm1.15}$ |
| 1 | 9 | -1.00 🔥 | 0.10 🔥 | $83.65_{\pm1.32}$ | $84.36_{\pm1.72}$ | $81.81_{\pm1.37}$ | $82.54_{\pm1.38}$ | $92.57_{\pm1.34}$ | $92.74_{\pm1.25}$ |
| 1 | 9 | -1.00 ❄ | 0.20 ❄ | $83.90_{\pm2.14}$ | $84.40_{\pm2.06}$ | $82.13_{\pm2.45}$ | $82.83_{\pm2.39}$ | $92.65_{\pm1.78}$ | $92.81_{\pm1.66}$ |
| 1 | 9 | -1.00 🔥 | 0.20 🔥 | $83.41_{\pm1.93}$ | $83.88_{\pm1.94}$ | $81.62_{\pm2.15}$ | $82.32_{\pm2.14}$ | $92.21_{\pm1.67}$ | $92.38_{\pm1.52}$ |
| 1 | 9 | -1.00 ❄ | 0.30 ❄ | $83.16_{\pm2.07}$ | $83.70_{\pm1.92}$ | $81.31_{\pm2.47}$ | $82.01_{\pm2.38}$ | $92.20_{\pm1.86}$ | $92.34_{\pm1.73}$ |
| 1 | 9 | -1.00 🔥 | 0.30 🔥 | $83.10_{\pm1.87}$ | $83.90_{\pm1.95}$ | $81.06_{\pm2.07}$ | $81.87_{\pm2.08}$ | $92.02_{\pm1.84}$ | $92.18_{\pm1.72}$ |
| 1 | 9 | -1.00 ❄ | 0.40 ❄ | $82.84_{\pm1.76}$ | $83.51_{\pm1.33}$ | $80.89_{\pm2.31}$ | $81.60_{\pm2.16}$ | $91.59_{\pm1.89}$ | $91.73_{\pm1.73}$ |
| 1 | 9 | -1.00 🔥 | 0.40 🔥 | $81.84_{\pm1.34}$ | $82.72_{\pm1.52}$ | $79.61_{\pm1.45}$ | $80.44_{\pm1.49}$ | $90.87_{\pm1.86}$ | $91.04_{\pm1.66}$ |
| 1 | 9 | -1.00 ❄ | 0.50 ❄ | $81.91_{\pm2.08}$ | $81.99_{\pm2.27}$ | $80.27_{\pm2.18}$ | $80.84_{\pm2.20}$ | $91.05_{\pm2.02}$ | $91.20_{\pm1.78}$ |
| 1 | 9 | -1.00 🔥 | 0.50 🔥 | $81.50_{\pm1.66}$ | $81.65_{\pm1.87}$ | $79.77_{\pm1.79}$ | $80.35_{\pm1.77}$ | $90.67_{\pm1.72}$ | $90.79_{\pm1.47}$ |
| 1 | 9 | -1.00 ❄ | 0.60 ❄ | $81.38_{\pm1.09}$ | $81.89_{\pm1.47}$ | $79.36_{\pm1.06}$ | $80.08_{\pm1.11}$ | $90.67_{\pm1.42}$ | $90.73_{\pm1.33}$ |
| 1 | 9 | -1.00 🔥 | 0.60 🔥 | $81.44_{\pm0.93}$ | $82.07_{\pm1.00}$ | $79.32_{\pm1.10}$ | $80.08_{\pm1.07}$ | $90.78_{\pm1.29}$ | $90.84_{\pm1.22}$ |
| 1 | 9 | -1.00 ❄ | 0.70 ❄ | $81.09_{\pm0.90}$ | $82.04_{\pm0.97}$ | $78.79_{\pm1.28}$ | $79.57_{\pm1.15}$ | $90.33_{\pm1.26}$ | $90.35_{\pm1.10}$ |
| 1 | 9 | -1.00 🔥 | 0.70 🔥 | $81.23_{\pm0.71}$ | $81.92_{\pm0.61}$ | $79.07_{\pm1.05}$ | $79.82_{\pm0.95}$ | $90.42_{\pm1.15}$ | $90.46_{\pm1.03}$ |
| 1 | 9 | -1.00 ❄ | 0.80 ❄ | $80.52_{\pm0.98}$ | $81.47_{\pm1.32}$ | $78.10_{\pm1.03}$ | $78.92_{\pm1.06}$ | $89.87_{\pm1.24}$ | $89.80_{\pm1.24}$ |
| 1 | 9 | -1.00 🔥 | 0.80 🔥 | $80.17_{\pm1.14}$ | $81.05_{\pm1.37}$ | $77.76_{\pm1.33}$ | $78.54_{\pm1.32}$ | $89.68_{\pm1.13}$ | $89.65_{\pm1.09}$ |
| 1 | 9 | -1.00 ❄ | 0.90 ❄ | $79.92_{\pm0.64}$ | $80.92_{\pm0.83}$ | $77.36_{\pm0.64}$ | $78.19_{\pm0.68}$ | $89.39_{\pm1.07}$ | $89.30_{\pm1.05}$ |
| 1 | 9 | -1.00 🔥 | 0.90 🔥 | $79.99_{\pm0.84}$ | $80.70_{\pm1.14}$ | $77.66_{\pm0.97}$ | $78.41_{\pm0.95}$ | $89.50_{\pm1.11}$ | $89.37_{\pm1.07}$ |
| 1 | 9 | -1.00 ❄ | 1.00 ❄ | $79.27_{\pm0.64}$ | $79.94_{\pm1.08}$ | $76.90_{\pm0.67}$ | $77.62_{\pm0.66}$ | $88.76_{\pm1.02}$ | $88.54_{\pm1.14}$ |
| 1 | 9 | -1.00 🔥 | 1.00 🔥 | $78.91_{\pm1.49}$ | $80.03_{\pm1.67}$ | $76.16_{\pm1.62}$ | $76.97_{\pm1.69}$ | $88.21_{\pm2.36}$ | $88.14_{\pm2.21}$ |
| 1 | 9 | -0.90 ❄ | -1.00 ❄ | $85.58_{\pm2.16}$ | $85.25_{\pm1.89}$ | $85.67_{\pm1.94}$ | $85.22_{\pm2.14}$ | $93.97_{\pm1.34}$ | $94.13_{\pm1.32}$ |
| 1 | 9 | -0.90 🔥 | -1.00 🔥 | $85.76_{\pm2.20}$ | $85.40_{\pm2.06}$ | $85.78_{\pm2.04}$ | $85.39_{\pm2.20}$ | $94.03_{\pm1.44}$ | $94.18_{\pm1.41}$ |
| 1 | 9 | -0.90 ❄ | -0.90 ❄ | $85.81_{\pm1.11}$ | $85.23_{\pm1.13}$ | $85.83_{\pm1.19}$ | $85.46_{\pm1.15}$ | $94.12_{\pm0.98}$ | $94.29_{\pm0.94}$ |
| 1 | 9 | -0.90 🔥 | -0.90 🔥 | $85.65_{\pm1.63}$ | $85.48_{\pm1.80}$ | $85.39_{\pm1.02}$ | $85.20_{\pm1.49}$ | $94.14_{\pm0.90}$ | $94.29_{\pm0.87}$ |
| 1 | 9 | -0.90 ❄ | -0.80 ❄ | $85.39_{\pm2.84}$ | $85.26_{\pm2.48}$ | $85.48_{\pm1.93}$ | $85.04_{\pm2.68}$ | $94.04_{\pm1.05}$ | $94.18_{\pm1.05}$ |
| 1 | 9 | -0.90 🔥 | -0.80 🔥 | $85.26_{\pm3.24}$ | $85.15_{\pm2.81}$ | $85.33_{\pm2.33}$ | $84.90_{\pm3.09}$ | $94.01_{\pm1.18}$ | $94.15_{\pm1.18}$ |
| 1 | 9 | -0.90 ❄ | -0.70 ❄ | $85.23_{\pm2.83}$ | $85.34_{\pm2.47}$ | $85.06_{\pm2.08}$ | $84.78_{\pm2.67}$ | $94.23_{\pm1.11}$ | $94.38_{\pm1.07}$ |
| 1 | 9 | -0.90 🔥 | -0.70 🔥 | $85.23_{\pm2.88}$ | $85.30_{\pm2.44}$ | $85.09_{\pm2.07}$ | $84.79_{\pm2.71}$ | $94.23_{\pm1.16}$ | $94.38_{\pm1.14}$ |
| 1 | 9 | -0.90 ❄ | -0.60 ❄ | $84.00_{\pm3.17}$ | $84.01_{\pm2.61}$ | $84.68_{\pm2.39}$ | $83.76_{\pm3.04}$ | $93.89_{\pm1.28}$ | $94.04_{\pm1.22}$ |
| 1 | 9 | -0.90 🔥 | -0.60 🔥 | $85.74_{\pm2.99}$ | $85.56_{\pm2.43}$ | $85.91_{\pm2.20}$ | $85.42_{\pm2.86}$ | $94.44_{\pm1.14}$ | $94.58_{\pm1.10}$ |
| 1 | 9 | -0.90 ❄ | -0.50 ❄ | $86.01_{\pm0.83}$ | $85.89_{\pm1.17}$ | $85.53_{\pm0.97}$ | $85.50_{\pm0.82}$ | $94.34_{\pm0.45}$ | $94.49_{\pm0.42}$ |
| 1 | 9 | -0.90 🔥 | -0.50 🔥 | $84.85_{\pm2.47}$ | $85.08_{\pm2.36}$ | $84.56_{\pm1.95}$ | $84.34_{\pm2.34}$ | $94.14_{\pm0.89}$ | $94.29_{\pm0.85}$ |
| 1 | 9 | -0.90 ❄ | -0.40 ❄ | $85.45_{\pm1.78}$ | $85.68_{\pm2.13}$ | $84.67_{\pm2.06}$ | $84.80_{\pm1.87}$ | $93.88_{\pm1.21}$ | $94.04_{\pm1.16}$ |
| 1 | 9 | -0.90 🔥 | -0.40 🔥 | $85.46_{\pm1.62}$ | $85.56_{\pm1.93}$ | $84.80_{\pm1.90}$ | $84.86_{\pm1.70}$ | $93.92_{\pm1.26}$ | $94.08_{\pm1.22}$ |
| 1 | 9 | -0.90 ❄ | -0.30 ❄ | $85.59_{\pm1.58}$ | $85.46_{\pm1.68}$ | $85.02_{\pm1.71}$ | $85.04_{\pm1.63}$ | $93.95_{\pm0.86}$ | $94.11_{\pm0.81}$ |
| 1 | 9 | -0.90 🔥 | -0.30 🔥 | $85.14_{\pm1.96}$ | $84.89_{\pm2.06}$ | $84.51_{\pm2.26}$ | $84.56_{\pm2.08}$ | $93.45_{\pm1.52}$ | $93.62_{\pm1.45}$ |
| 1 | 9 | -0.90 ❄ | -0.20 ❄ | $85.00_{\pm1.14}$ | $85.29_{\pm1.28}$ | $84.03_{\pm1.91}$ | $84.25_{\pm1.49}$ | $93.49_{\pm0.63}$ | $93.66_{\pm0.59}$ |
| 1 | 9 | -0.90 🔥 | -0.20 🔥 | $84.40_{\pm2.40}$ | $84.20_{\pm2.47}$ | $83.33_{\pm2.64}$ | $83.66_{\pm2.57}$ | $92.83_{\pm1.83}$ | $93.07_{\pm1.77}$ |
| 1 | 9 | -0.90 ❄ | -0.10 ❄ | $84.89_{\pm0.93}$ | $85.20_{\pm1.54}$ | $83.66_{\pm1.03}$ | $84.09_{\pm0.95}$ | $93.41_{\pm0.79}$ | $93.58_{\pm0.75}$ |
| 1 | 9 | -0.90 🔥 | -0.10 🔥 | $84.79_{\pm1.06}$ | $84.88_{\pm1.53}$ | $83.67_{\pm0.82}$ | $84.05_{\pm0.97}$ | $93.44_{\pm0.72}$ | $93.60_{\pm0.69}$ |
| 1 | 9 | -0.90 ❄ | 0.10 ❄ | $84.21_{\pm1.85}$ | $84.73_{\pm2.37}$ | $82.51_{\pm1.67}$ | $83.22_{\pm1.85}$ | $92.75_{\pm1.64}$ | $92.94_{\pm1.55}$ |
| 1 | 9 | -0.90 🔥 | 0.10 🔥 | $84.04_{\pm1.91}$ | $84.66_{\pm2.38}$ | $82.26_{\pm1.84}$ | $83.00_{\pm1.96}$ | $92.82_{\pm1.64}$ | $93.01_{\pm1.55}$ |
| 1 | 9 | -0.90 ❄ | 0.20 ❄ | $83.82_{\pm1.99}$ | $84.27_{\pm2.17}$ | $82.10_{\pm2.16}$ | $82.78_{\pm2.16}$ | $92.62_{\pm1.74}$ | $92.80_{\pm1.64}$ |
| 1 | 9 | -0.90 🔥 | 0.20 🔥 | $83.93_{\pm2.12}$ | $84.38_{\pm2.25}$ | $82.22_{\pm2.30}$ | $82.90_{\pm2.30}$ | $92.63_{\pm1.80}$ | $92.82_{\pm1.69}$ |

| t | n | a | b | Accuracy | Precision | Recall | F1 score | AUROC | AUPRC |
|---|---|---|---|---|---|---|---|---|---|
| 1 | 9 | -0.90 ❄ | 0.30 ❄ | 83.06±1.92 | 83.52±1.95 | 81.26±2.20 | 81.94±2.16 | 92.13±1.86 | 92.29±1.73 |
| 1 | 9 | -0.90 🔥 | 0.30 🔥 | 83.91±1.34 | 84.43±1.38 | 82.20±1.62 | 82.87±1.52 | 92.59±1.48 | 92.71±1.35 |
| 1 | 9 | -0.90 ❄ | 0.40 ❄ | 82.99±1.07 | 83.76±1.19 | 81.01±1.36 | 81.77±1.24 | 91.72±1.28 | 91.82±1.25 |
| 1 | 9 | -0.90 🔥 | 0.40 🔥 | 83.45±1.14 | 84.40±1.42 | 81.39±1.19 | 82.23±1.21 | 92.28±1.39 | 92.41±1.32 |
| 1 | 9 | -0.90 ❄ | 0.50 ❄ | 82.73±1.34 | 83.39±1.40 | 80.72±1.49 | 81.49±1.48 | 91.71±1.39 | 91.77±1.37 |
| 1 | 9 | -0.90 🔥 | 0.50 🔥 | 82.71±1.31 | 83.34±1.55 | 80.75±1.39 | 81.51±1.40 | 91.80±1.40 | 91.85±1.34 |
| 1 | 9 | -0.90 ❄ | 0.60 ❄ | 82.03±1.18 | 82.84±1.20 | 79.86±1.37 | 80.66±1.34 | 91.20±1.10 | 91.20±1.06 |
| 1 | 9 | -0.90 🔥 | 0.60 🔥 | 81.97±1.26 | 82.81±1.29 | 79.79±1.48 | 80.59±1.45 | 91.15±1.22 | 91.12±1.18 |
| 1 | 9 | -0.90 ❄ | 0.70 ❄ | 81.58±1.13 | 82.53±1.21 | 79.32±1.44 | 80.12±1.35 | 90.72±1.24 | 90.70±1.19 |
| 1 | 9 | -0.90 🔥 | 0.70 🔥 | 80.92±0.64 | 81.94±0.93 | 78.50±0.63 | 79.35±0.66 | 90.35±1.14 | 90.30±1.10 |
| 1 | 9 | -0.90 ❄ | 0.80 ❄ | 79.69±1.67 | 80.50±2.38 | 77.32±1.40 | 78.08±1.57 | 88.77±1.94 | 88.78±1.85 |
| 1 | 9 | -0.90 🔥 | 0.80 🔥 | 80.03±1.27 | 80.91±1.87 | 77.60±1.08 | 78.40±1.21 | 89.27±1.66 | 89.27±1.60 |
| 1 | 9 | -0.90 ❄ | 0.90 ❄ | 79.80±0.84 | 81.00±1.11 | 77.13±0.84 | 77.99±0.89 | 89.43±1.03 | 89.35±1.03 |
| 1 | 9 | -0.90 🔥 | 0.90 🔥 | 79.82±0.94 | 81.20±1.20 | 77.06±0.97 | 77.94±1.02 | 89.43±1.15 | 89.24±1.17 |
| 1 | 9 | -0.90 ❄ | 1.00 ❄ | 79.36±0.74 | 80.15±1.22 | 76.89±0.67 | 77.66±0.71 | 88.77±1.28 | 88.57±1.26 |
| 1 | 9 | -0.90 🔥 | 1.00 🔥 | 79.23±0.91 | 80.23±0.95 | 76.59±1.04 | 77.40±1.07 | 88.79±1.17 | 88.64±1.18 |
| 1 | 9 | -0.80 ❄ | -1.00 ❄ | 85.37±1.64 | 85.90±0.60 | 84.34±2.90 | 84.54±2.30 | 93.82±1.30 | 94.00±1.21 |
| 1 | 9 | -0.80 🔥 | -1.00 🔥 | 84.46±1.74 | 85.26±1.23 | 83.67±2.36 | 83.67±2.05 | 93.72±1.05 | 93.92±1.00 |
| 1 | 9 | -0.80 ❄ | -0.90 ❄ | 85.65±1.36 | 86.08±0.56 | 84.55±2.41 | 84.86±1.86 | 93.96±0.79 | 94.13±0.76 |
| 1 | 9 | -0.80 🔥 | -0.90 🔥 | 85.95±1.70 | 86.43±0.92 | 84.76±2.68 | 85.14±2.19 | 94.10±0.89 | 94.27±0.86 |
| 1 | 9 | -0.80 ❄ | -0.80 ❄ | 85.95±1.20 | 86.05±0.99 | 85.26±1.92 | 85.34±1.51 | 94.28±0.77 | **94.44±0.72** |
| 1 | 9 | -0.80 🔥 | -0.80 🔥 | 86.00±1.20 | 86.01±1.14 | 85.32±1.74 | 85.40±1.42 | 94.20±0.76 | 94.36±0.72 |
| 1 | 9 | -0.80 ❄ | -0.70 ❄ | 85.19±2.09 | 85.57±2.04 | 84.32±1.74 | 84.50±2.02 | 94.10±0.86 | 94.26±0.83 |
| 1 | 9 | -0.80 🔥 | -0.70 🔥 | 85.41±2.39 | 85.68±2.21 | 84.70±2.05 | 84.79±2.33 | 93.98±0.91 | 94.14±0.87 |
| 1 | 9 | -0.80 ❄ | -0.60 ❄ | 86.23±1.17 | **86.35±1.35** | 85.14±1.24 | 85.55±1.22 | 94.09±0.81 | 94.27±0.75 |
| 1 | 9 | -0.80 🔥 | -0.60 🔥 | **86.30±1.05** | 86.16±1.09 | **85.47±1.12** | **85.71±1.09** | **94.26±0.54** | 94.42±0.49 |
| 1 | 9 | -0.80 ❄ | -0.50 ❄ | 85.67±2.74 | 85.40±2.42 | 85.61±2.03 | 85.30±2.60 | 94.18±1.24 | 94.34±1.20 |
| 1 | 9 | -0.80 🔥 | -0.50 🔥 | 85.67±1.39 | 85.34±1.59 | 85.20±1.02 | 85.18±1.30 | 94.07±0.66 | 94.24±0.63 |
| 1 | 9 | -0.80 ❄ | -0.40 ❄ | 86.08±1.04 | 85.69±1.09 | 85.51±1.17 | 85.57±1.09 | 93.97±0.94 | 94.14±0.89 |
| 1 | 9 | -0.80 🔥 | -0.40 🔥 | 86.18±0.77 | 85.81±0.86 | 85.74±0.91 | 85.70±0.79 | 94.14±0.70 | 94.30±0.65 |
| 1 | 9 | -0.80 ❄ | -0.30 ❄ | 85.49±1.89 | 85.45±2.18 | 84.68±2.02 | 84.87±1.94 | 93.73±1.45 | 93.87±1.37 |
| 1 | 9 | -0.80 🔥 | -0.30 🔥 | 85.02±0.76 | 84.90±0.99 | 84.06±0.84 | 84.34±0.76 | 93.43±0.63 | 93.63±0.60 |
| 1 | 9 | -0.80 ❄ | -0.20 ❄ | 84.84±2.03 | 85.20±2.56 | 83.38±1.87 | 83.98±2.02 | 93.22±1.64 | 93.44±1.57 |
| 1 | 9 | -0.80 🔥 | -0.20 🔥 | 84.28±1.67 | 84.97±2.20 | 82.55±1.54 | 83.26±1.65 | 93.04±1.47 | 93.26±1.41 |
| 1 | 9 | -0.80 ❄ | -0.10 ❄ | 84.07±1.50 | 84.56±1.98 | 82.37±1.38 | 83.07±1.51 | 92.70±1.15 | 92.96±1.05 |
| 1 | 9 | -0.80 🔥 | -0.10 🔥 | 83.83±2.37 | 84.58±3.02 | 81.99±2.32 | 82.75±2.44 | 92.33±1.96 | 92.60±1.88 |
| 1 | 9 | -0.80 ❄ | 0.10 ❄ | 84.21±1.34 | 85.39±1.96 | 82.14±1.24 | 83.03±1.33 | 92.98±1.31 | 93.20±1.20 |
| 1 | 9 | -0.80 🔥 | 0.10 🔥 | 84.29±1.48 | 85.49±1.95 | 82.23±1.52 | 83.11±1.57 | 92.95±1.33 | 93.17±1.23 |
| 1 | 9 | -0.80 ❄ | 0.20 ❄ | 84.32±1.36 | 84.94±1.39 | 82.54±1.57 | 83.28±1.51 | 92.76±1.43 | 92.97±1.32 |
| 1 | 9 | -0.80 🔥 | 0.20 🔥 | 83.83±1.93 | 84.13±2.52 | 82.35±1.71 | 82.91±1.88 | 92.37±1.75 | 92.57±1.66 |
| 1 | 9 | -0.80 ❄ | 0.30 ❄ | 83.48±1.29 | 84.23±1.50 | 81.54±1.39 | 82.32±1.39 | 92.28±1.43 | 92.47±1.33 |
| 1 | 9 | -0.80 🔥 | 0.30 🔥 | 83.44±1.09 | 84.45±1.17 | 81.33±1.28 | 82.19±1.23 | 92.35±1.32 | 92.53±1.23 |
| 1 | 9 | -0.80 ❄ | 0.40 ❄ | 82.66±1.24 | 83.38±1.47 | 80.65±1.39 | 81.41±1.36 | 91.36±1.59 | 91.50±1.43 |
| 1 | 9 | -0.80 🔥 | 0.40 🔥 | 83.05±1.19 | 83.81±1.18 | 81.04±1.42 | 81.82±1.36 | 91.86±1.36 | 91.94±1.24 |
| 1 | 9 | -0.80 ❄ | 0.50 ❄ | 82.63±1.22 | 83.53±1.38 | 80.47±1.32 | 81.31±1.34 | 91.70±1.26 | 91.70±1.33 |
| 1 | 9 | -0.80 🔥 | 0.50 🔥 | 82.35±1.09 | 83.13±1.13 | 80.25±1.30 | 81.04±1.25 | 91.58±1.31 | 91.57±1.33 |
| 1 | 9 | -0.80 ❄ | 0.60 ❄ | 81.38±1.59 | 82.08±1.55 | 79.21±1.89 | 79.97±1.84 | 90.25±2.10 | 90.37±1.94 |
| 1 | 9 | -0.80 🔥 | 0.60 🔥 | 81.75±0.95 | 82.79±1.28 | 79.46±1.20 | 80.29±1.12 | 90.97±1.32 | 91.01±1.29 |
| 1 | 10 | -1.00 ❄ | -1.00 ❄ | 82.39±1.78 | 82.06±1.69 | 81.98±1.15 | 81.83±1.61 | 91.79±0.89 | 91.99±0.84 |
| 1 | 10 | -1.00 🔥 | -1.00 🔥 | 82.10±1.92 | 81.76±1.63 | 81.89±1.15 | 81.60±1.73 | 91.81±0.77 | 92.01±0.74 |

| t | n | a | b | Accuracy | Precision | Recall | F1 score | AUROC | AUPRC |
|---|---|---|---|---|---|---|---|---|---|
| 1 | 10 | -1.00 ❄ | -0.90 ❄ | $83.10_{\pm2.42}$ | $83.35_{\pm2.48}$ | $82.35_{\pm1.46}$ | $82.41_{\pm2.15}$ | $92.36_{\pm1.11}$ | $92.55_{\pm1.08}$ |
| 1 | 10 | -1.00 🔥 | -0.90 🔥 | $82.92_{\pm1.43}$ | $83.61_{\pm1.93}$ | $81.32_{\pm1.96}$ | $81.82_{\pm1.76}$ | $91.78_{\pm1.19}$ | $92.04_{\pm1.07}$ |
| 1 | 10 | -1.00 ❄ | -0.80 ❄ | $84.00_{\pm1.41}$ | $84.69_{\pm0.61}$ | $82.39_{\pm2.37}$ | $82.93_{\pm2.02}$ | $92.73_{\pm1.12}$ | $92.93_{\pm1.00}$ |
| 1 | 10 | -1.00 🔥 | -0.80 🔥 | $83.13_{\pm1.40}$ | $83.55_{\pm1.35}$ | $81.69_{\pm2.11}$ | $82.12_{\pm1.85}$ | $91.99_{\pm0.97}$ | $92.24_{\pm0.85}$ |
| 1 | 10 | -1.00 ❄ | -0.70 ❄ | $83.02_{\pm1.22}$ | $83.67_{\pm1.74}$ | $81.51_{\pm1.96}$ | $81.96_{\pm1.56}$ | $91.92_{\pm1.09}$ | $92.20_{\pm0.97}$ |
| 1 | 10 | -1.00 🔥 | -0.70 🔥 | $83.93_{\pm1.30}$ | $84.13_{\pm1.04}$ | $82.68_{\pm2.01}$ | $83.05_{\pm1.67}$ | $92.64_{\pm1.28}$ | $92.86_{\pm1.15}$ |
| 1 | 10 | -1.00 ❄ | -0.60 ❄ | $83.09_{\pm1.47}$ | $83.24_{\pm1.73}$ | $81.98_{\pm1.92}$ | $82.24_{\pm1.66}$ | $92.03_{\pm1.14}$ | $92.31_{\pm0.99}$ |
| 1 | 10 | -1.00 🔥 | -0.60 🔥 | $83.31_{\pm1.34}$ | $83.63_{\pm1.34}$ | $82.04_{\pm2.18}$ | $82.38_{\pm1.72}$ | $91.86_{\pm1.11}$ | $92.16_{\pm0.95}$ |
| 1 | 10 | -1.00 ❄ | -0.50 ❄ | $83.63_{\pm1.38}$ | $83.31_{\pm1.76}$ | $83.30_{\pm1.35}$ | $83.12_{\pm1.32}$ | $92.94_{\pm1.13}$ | $93.05_{\pm1.07}$ |
| 1 | 10 | -1.00 🔥 | -0.50 🔥 | $83.84_{\pm1.54}$ | $83.50_{\pm1.90}$ | $83.36_{\pm1.48}$ | $83.29_{\pm1.50}$ | $92.92_{\pm1.14}$ | $93.04_{\pm1.07}$ |
| 1 | 10 | -1.00 ❄ | -0.40 ❄ | $83.44_{\pm1.22}$ | $83.16_{\pm1.60}$ | $82.60_{\pm1.28}$ | $82.76_{\pm1.23}$ | $92.10_{\pm0.94}$ | $92.32_{\pm0.86}$ |
| 1 | 10 | -1.00 🔥 | -0.40 🔥 | $83.68_{\pm1.47}$ | $83.26_{\pm1.69}$ | $82.90_{\pm1.40}$ | $83.04_{\pm1.48}$ | $92.41_{\pm1.30}$ | $92.60_{\pm1.17}$ |
| 1 | 10 | -1.00 ❄ | -0.30 ❄ | $84.14_{\pm1.54}$ | $84.13_{\pm1.65}$ | $83.02_{\pm1.94}$ | $83.35_{\pm1.70}$ | $92.59_{\pm1.25}$ | $92.85_{\pm1.10}$ |
| 1 | 10 | -1.00 🔥 | -0.30 🔥 | $84.14_{\pm1.51}$ | $84.26_{\pm1.76}$ | $82.97_{\pm1.83}$ | $83.33_{\pm1.63}$ | $92.54_{\pm1.17}$ | $92.80_{\pm1.02}$ |
| 1 | 10 | -1.00 ❄ | -0.20 ❄ | $84.49_{\pm1.39}$ | $84.53_{\pm1.51}$ | $83.19_{\pm1.53}$ | $83.67_{\pm1.48}$ | $92.77_{\pm1.18}$ | $92.96_{\pm0.97}$ |
| 1 | 10 | -1.00 🔥 | -0.20 🔥 | $84.99_{\pm1.11}$ | $84.68_{\pm1.17}$ | $84.10_{\pm1.17}$ | $84.35_{\pm1.17}$ | $93.07_{\pm1.01}$ | $93.22_{\pm0.88}$ |
| 1 | 10 | -1.00 ❄ | -0.10 ❄ | $85.12_{\pm1.28}$ | $84.99_{\pm1.30}$ | $84.03_{\pm1.45}$ | $84.40_{\pm1.38}$ | $93.20_{\pm1.06}$ | $93.40_{\pm0.90}$ |
| 1 | 10 | -1.00 🔥 | -0.10 🔥 | $85.07_{\pm1.18}$ | $84.90_{\pm1.22}$ | $84.03_{\pm1.31}$ | $84.38_{\pm1.26}$ | $93.20_{\pm0.98}$ | $93.41_{\pm0.84}$ |
| 1 | 10 | -1.00 ❄ | 0.10 ❄ | $85.20_{\pm1.27}$ | $85.52_{\pm1.67}$ | $83.79_{\pm1.30}$ | $84.36_{\pm1.31}$ | $93.83_{\pm0.98}$ | $93.95_{\pm0.84}$ |
| 1 | 10 | -1.00 🔥 | 0.10 🔥 | $85.39_{\pm0.95}$ | $85.67_{\pm1.26}$ | $84.03_{\pm1.05}$ | $84.58_{\pm1.01}$ | $93.85_{\pm0.93}$ | $93.96_{\pm0.81}$ |
| 1 | 10 | -1.00 ❄ | 0.20 ❄ | $84.61_{\pm2.08}$ | $84.76_{\pm2.42}$ | $83.28_{\pm2.06}$ | $83.79_{\pm2.13}$ | $93.05_{\pm1.55}$ | $93.15_{\pm1.51}$ |
| 1 | 10 | -1.00 🔥 | 0.20 🔥 | $83.96_{\pm1.32}$ | $84.31_{\pm1.63}$ | $82.40_{\pm1.41}$ | $83.00_{\pm1.38}$ | $92.88_{\pm1.26}$ | $92.97_{\pm1.16}$ |
| 1 | 10 | -1.00 ❄ | 0.30 ❄ | $82.95_{\pm1.59}$ | $84.10_{\pm1.90}$ | $80.95_{\pm2.19}$ | $81.65_{\pm2.02}$ | $92.40_{\pm1.45}$ | $92.51_{\pm1.41}$ |
| 1 | 10 | -1.00 🔥 | 0.30 🔥 | $82.36_{\pm1.83}$ | $83.87_{\pm2.10}$ | $80.25_{\pm2.70}$ | $80.91_{\pm2.54}$ | $92.27_{\pm1.35}$ | $92.38_{\pm1.32}$ |
| 1 | 10 | -1.00 ❄ | 0.40 ❄ | $82.33_{\pm1.92}$ | $84.13_{\pm2.36}$ | $79.84_{\pm2.27}$ | $80.75_{\pm2.23}$ | $91.95_{\pm1.82}$ | $92.11_{\pm1.74}$ |
| 1 | 10 | -1.00 🔥 | 0.40 🔥 | $82.08_{\pm1.93}$ | $84.03_{\pm2.22}$ | $79.57_{\pm2.57}$ | $80.41_{\pm2.50}$ | $91.87_{\pm1.80}$ | $92.03_{\pm1.74}$ |
| 1 | 10 | -1.00 ❄ | 0.50 ❄ | $81.20_{\pm2.56}$ | $83.46_{\pm2.24}$ | $78.47_{\pm3.40}$ | $79.26_{\pm3.51}$ | $91.14_{\pm2.18}$ | $91.32_{\pm2.13}$ |
| 1 | 10 | -1.00 🔥 | 0.50 🔥 | $82.10_{\pm2.55}$ | $84.46_{\pm1.10}$ | $79.38_{\pm3.61}$ | $80.21_{\pm3.66}$ | $92.01_{\pm1.75}$ | $92.19_{\pm1.77}$ |
| 1 | 10 | -1.00 ❄ | 0.60 ❄ | $82.56_{\pm2.79}$ | $84.15_{\pm3.28}$ | $80.11_{\pm2.92}$ | $81.05_{\pm3.03}$ | $91.53_{\pm3.05}$ | $91.68_{\pm2.95}$ |
| 1 | 10 | -1.00 🔥 | 0.60 🔥 | $82.43_{\pm2.81}$ | $83.64_{\pm3.29}$ | $80.12_{\pm2.90}$ | $81.01_{\pm3.01}$ | $90.84_{\pm3.48}$ | $91.09_{\pm3.34}$ |
| 1 | 10 | -1.00 ❄ | 0.70 ❄ | $82.85_{\pm1.77}$ | $84.55_{\pm1.60}$ | $80.32_{\pm2.11}$ | $81.30_{\pm2.12}$ | $91.51_{\pm2.11}$ | $91.75_{\pm2.01}$ |
| 1 | 10 | -1.00 🔥 | 0.70 🔥 | $82.01_{\pm2.88}$ | $83.43_{\pm3.38}$ | $79.59_{\pm2.98}$ | $80.49_{\pm3.10}$ | $90.63_{\pm3.37}$ | $90.85_{\pm3.24}$ |
| 1 | 10 | -1.00 ❄ | 0.80 ❄ | $82.50_{\pm1.48}$ | $84.59_{\pm1.19}$ | $79.75_{\pm1.79}$ | $80.80_{\pm1.81}$ | $90.85_{\pm2.07}$ | $91.15_{\pm1.98}$ |
| 1 | 10 | -1.00 🔥 | 0.80 🔥 | $82.63_{\pm1.73}$ | $84.52_{\pm1.40}$ | $79.98_{\pm2.10}$ | $80.99_{\pm2.11}$ | $90.85_{\pm2.43}$ | $91.08_{\pm2.44}$ |
| 1 | 10 | -1.00 ❄ | 0.90 ❄ | $81.82_{\pm2.02}$ | $83.59_{\pm2.26}$ | $79.11_{\pm2.16}$ | $80.11_{\pm2.26}$ | $90.07_{\pm3.10}$ | $90.44_{\pm2.88}$ |
| 1 | 10 | -1.00 🔥 | 0.90 🔥 | $81.59_{\pm1.80}$ | $83.13_{\pm2.47}$ | $79.02_{\pm1.71}$ | $79.96_{\pm1.84}$ | $89.65_{\pm3.19}$ | $90.07_{\pm2.93}$ |
| 1 | 10 | -1.00 ❄ | 1.00 ❄ | $81.13_{\pm1.69}$ | $82.88_{\pm1.79}$ | $78.36_{\pm1.88}$ | $79.32_{\pm1.95}$ | $89.14_{\pm2.90}$ | $89.58_{\pm2.66}$ |
| 1 | 10 | -1.00 🔥 | 1.00 🔥 | $81.10_{\pm1.70}$ | $82.79_{\pm1.98}$ | $78.36_{\pm1.83}$ | $79.32_{\pm1.90}$ | $89.02_{\pm3.03}$ | $89.47_{\pm2.76}$ |
| 1 | 10 | -0.90 ❄ | -1.00 ❄ | $82.59_{\pm2.68}$ | $82.56_{\pm2.82}$ | $81.84_{\pm1.90}$ | $81.91_{\pm2.44}$ | $91.85_{\pm1.39}$ | $92.04_{\pm1.33}$ |
| 1 | 10 | -0.90 🔥 | -1.00 🔥 | $83.28_{\pm2.87}$ | $83.28_{\pm2.41}$ | $82.93_{\pm1.80}$ | $82.75_{\pm2.62}$ | $92.44_{\pm0.97}$ | $92.59_{\pm1.00}$ |
| 1 | 10 | -0.90 ❄ | -0.90 ❄ | $83.91_{\pm2.02}$ | $83.97_{\pm2.62}$ | $83.04_{\pm1.41}$ | $83.23_{\pm1.82}$ | $92.80_{\pm1.18}$ | $92.94_{\pm1.19}$ |
| 1 | 10 | -0.90 🔥 | -0.90 🔥 | $83.90_{\pm1.48}$ | $84.63_{\pm1.54}$ | $82.33_{\pm2.23}$ | $82.86_{\pm1.93}$ | $92.59_{\pm1.29}$ | $92.79_{\pm1.19}$ |
| 1 | 10 | -0.90 ❄ | -0.80 ❄ | $83.69_{\pm1.20}$ | $84.52_{\pm1.09}$ | $82.10_{\pm2.17}$ | $82.61_{\pm1.76}$ | $92.59_{\pm0.84}$ | $92.78_{\pm0.76}$ |
| 1 | 10 | -0.90 🔥 | -0.80 🔥 | $83.72_{\pm1.01}$ | $83.93_{\pm0.61}$ | $82.53_{\pm1.91}$ | $82.84_{\pm1.51}$ | $92.38_{\pm1.01}$ | $92.60_{\pm0.91}$ |
| 1 | 10 | -0.90 ❄ | -0.70 ❄ | $83.52_{\pm1.21}$ | $83.79_{\pm0.44}$ | $82.42_{\pm2.31}$ | $82.64_{\pm1.83}$ | $92.53_{\pm1.04}$ | $92.74_{\pm0.93}$ |
| 1 | 10 | -0.90 🔥 | -0.70 🔥 | $83.02_{\pm2.15}$ | $83.25_{\pm2.19}$ | $82.83_{\pm1.28}$ | $82.49_{\pm1.88}$ | $92.67_{\pm0.95}$ | $92.83_{\pm0.95}$ |
| 1 | 10 | -0.90 ❄ | -0.60 ❄ | $84.05_{\pm1.60}$ | $84.14_{\pm2.22}$ | $83.16_{\pm1.74}$ | $83.34_{\pm1.63}$ | $92.85_{\pm1.20}$ | $93.01_{\pm1.16}$ |
| 1 | 10 | -0.90 🔥 | -0.60 🔥 | $83.55_{\pm1.59}$ | $83.72_{\pm2.20}$ | $82.50_{\pm1.81}$ | $82.75_{\pm1.65}$ | $92.31_{\pm1.35}$ | $92.57_{\pm1.21}$ |
| 1 | 10 | -0.90 ❄ | -0.50 ❄ | $83.41_{\pm1.23}$ | $83.06_{\pm1.42}$ | $82.63_{\pm1.44}$ | $82.74_{\pm1.31}$ | $92.32_{\pm0.92}$ | $92.53_{\pm0.92}$ |
| 1 | 10 | -0.90 🔥 | -0.50 🔥 | $84.28_{\pm1.78}$ | $84.12_{\pm2.11}$ | $83.44_{\pm2.07}$ | $83.60_{\pm1.90}$ | $92.76_{\pm1.27}$ | $92.91_{\pm1.20}$ |

| t | n | a | b | Accuracy | Precision | Recall | F1 score | AUROC | AUPRC |
|---|---|---|---|---|---|---|---|---|---|
| 1 | 10 | -0.90 ❄ | -0.40 ❄ | 83.73±1.30 | 84.51±1.75 | 82.02±1.61 | 82.66±1.45 | 92.44±1.13 | 92.73±1.01 |
| 1 | 10 | -0.90 🔥 | -0.40 🔥 | 83.58±1.75 | 83.79±2.16 | 82.18±1.77 | 82.67±1.79 | 92.26±1.36 | 92.52±1.26 |
| 1 | 10 | -0.90 ❄ | -0.30 ❄ | 83.02±1.73 | 83.49±1.89 | 81.56±2.52 | 81.98±2.15 | 92.01±1.34 | 92.28±1.26 |
| 1 | 10 | -0.90 🔥 | -0.30 🔥 | 83.30±2.09 | 83.70±2.11 | 81.87±2.82 | 82.29±2.48 | 92.21±1.43 | 92.46±1.34 |
| 1 | 10 | -0.90 ❄ | -0.20 ❄ | 84.79±1.35 | 84.91±1.27 | 83.52±1.79 | 83.98±1.56 | 92.90±1.31 | 93.16±1.15 |
| 1 | 10 | -0.90 🔥 | -0.20 🔥 | 84.93±1.34 | 85.25±1.30 | 83.45±1.62 | 84.05±1.50 | 92.89±1.25 | 93.15±1.08 |
| 1 | 10 | -0.90 ❄ | -0.10 ❄ | 84.75±1.72 | 84.97±1.26 | 83.42±2.34 | 83.88±2.08 | 93.04±1.28 | 93.26±1.20 |
| 1 | 10 | -0.90 🔥 | -0.10 🔥 | 85.28±0.92 | 85.48±0.80 | 83.97±1.31 | 84.48±1.10 | 93.47±0.87 | 93.67±0.76 |
| 1 | 10 | -0.90 ❄ | 0.10 ❄ | 85.17±1.27 | 85.83±1.30 | 83.45±1.48 | 84.20±1.42 | 93.55±1.07 | 93.71±0.95 |
| 1 | 10 | -0.90 🔥 | 0.10 🔥 | 85.26±1.33 | 86.12±1.10 | 83.45±1.76 | 84.23±1.59 | 93.62±1.02 | 93.79±0.90 |
| 1 | 10 | -0.90 ❄ | 0.20 ❄ | 84.40±1.92 | 85.22±2.03 | 82.53±2.18 | 83.31±2.12 | 93.24±1.44 | 93.38±1.36 |
| 1 | 10 | -0.90 🔥 | 0.20 🔥 | 84.64±1.99 | 85.41±2.03 | 82.81±2.26 | 83.58±2.21 | 93.42±1.53 | 93.54±1.45 |
| 1 | 10 | -0.90 ❄ | 0.30 ❄ | 83.79±2.29 | 84.16±2.57 | 82.12±2.35 | 82.79±2.41 | 92.42±2.07 | 92.59±2.04 |
| 1 | 10 | -0.90 🔥 | 0.30 🔥 | 84.09±1.92 | 84.88±2.27 | 82.26±2.09 | 83.01±2.06 | 92.83±1.68 | 92.96±1.57 |
| 1 | 10 | -0.90 ❄ | 0.40 ❄ | 83.35±2.22 | 84.75±2.58 | 81.09±2.41 | 82.00±2.42 | 92.22±2.03 | 92.41±1.94 |
| 1 | 10 | -0.90 🔥 | 0.40 🔥 | 83.19±2.04 | 84.21±2.33 | 81.16±2.40 | 81.93±2.30 | 92.01±1.82 | 92.19±1.72 |
| 1 | 10 | -0.90 ❄ | 0.50 ❄ | 82.78±2.16 | 83.87±2.50 | 80.63±2.41 | 81.45±2.40 | 91.52±2.15 | 91.72±2.05 |
| 1 | 10 | -0.90 🔥 | 0.50 🔥 | 82.80±2.00 | 84.12±2.53 | 80.49±2.06 | 81.39±2.13 | 91.62±2.16 | 91.81±2.05 |
| 1 | 10 | -0.90 ❄ | 0.60 ❄ | 83.05±2.85 | 84.44±3.08 | 80.73±3.14 | 81.63±3.19 | 91.27±3.08 | 91.47±2.97 |
| 1 | 10 | -0.90 🔥 | 0.60 🔥 | 82.81±3.02 | 84.38±3.45 | 80.40±3.21 | 81.33±3.31 | 91.19±3.06 | 91.42±2.94 |
| 1 | 10 | -0.90 ❄ | 0.70 ❄ | 81.83±2.43 | 83.46±2.89 | 79.29±2.59 | 80.21±2.69 | 90.11±3.29 | 90.38±3.17 |
| 1 | 10 | -0.90 🔥 | 0.70 🔥 | 81.98±2.43 | 83.45±2.84 | 79.49±2.55 | 80.42±2.65 | 90.19±3.28 | 90.46±3.15 |
| 1 | 10 | -0.90 ❄ | 0.80 ❄ | 82.12±1.91 | 83.59±2.32 | 79.61±1.95 | 80.57±2.05 | 90.15±2.89 | 90.52±2.76 |
| 1 | 10 | -0.90 🔥 | 0.80 🔥 | 82.57±1.51 | 84.47±1.03 | 79.92±1.91 | 80.92±1.91 | 90.53±2.37 | 90.85±2.32 |
| 1 | 10 | -0.90 ❄ | 0.90 ❄ | 81.33±2.10 | 82.93±2.05 | 78.65±2.43 | 79.58±2.46 | 89.00±2.80 | 89.45±2.63 |
| 1 | 10 | -0.90 🔥 | 0.90 🔥 | 81.22±2.00 | 82.72±2.31 | 78.57±2.11 | 79.50±2.21 | 88.90±3.00 | 89.35±2.80 |
| 1 | 10 | -0.90 ❄ | 1.00 ❄ | 80.70±2.22 | 82.20±2.75 | 78.02±2.29 | 78.93±2.39 | 88.22±3.63 | 88.72±3.36 |
| 1 | 10 | -0.90 🔥 | 1.00 🔥 | 80.84±2.14 | 82.29±2.62 | 78.19±2.19 | 79.10±2.30 | 88.31±3.29 | 88.81±3.03 |
| 1 | 10 | -0.80 ❄ | -1.00 ❄ | 82.64±3.04 | 82.99±2.51 | 82.27±1.67 | 82.05±2.71 | 92.09±0.93 | 92.24±0.97 |
| 1 | 10 | -0.80 🔥 | -1.00 🔥 | 82.47±2.87 | 83.03±2.53 | 82.02±1.49 | 81.83±2.51 | 92.11±0.90 | 92.25±0.94 |
| 1 | 10 | -0.80 ❄ | -0.90 ❄ | 82.07±2.68 | 82.37±3.01 | 81.16±1.80 | 81.28±2.39 | 91.27±1.31 | 91.53±1.19 |
| 1 | 10 | -0.80 🔥 | -0.90 🔥 | 82.43±2.88 | 82.81±3.05 | 81.66±2.12 | 81.69±2.63 | 91.91±1.28 | 92.11±1.22 |
| 1 | 10 | -0.80 ❄ | -0.80 ❄ | 83.51±1.48 | 83.71±1.00 | 82.42±2.24 | 82.65±1.93 | 92.39±1.38 | 92.59±1.28 |
| 1 | 10 | -0.80 🔥 | -0.80 🔥 | 83.77±1.48 | 84.19±1.37 | 82.44±2.02 | 82.84±1.81 | 92.45±1.33 | 92.65±1.24 |
| 1 | 10 | -0.80 ❄ | -0.70 ❄ | 83.03±1.77 | 83.59±2.51 | 82.29±1.90 | 82.26±1.80 | 92.56±0.98 | 92.73±0.99 |
| 1 | 10 | -0.80 🔥 | -0.70 🔥 | 83.02±1.81 | 83.61±2.58 | 82.29±1.82 | 82.26±1.79 | 92.59±1.03 | 92.75±1.05 |
| 1 | 10 | -0.80 ❄ | -0.60 ❄ | 84.28±1.33 | 84.09±1.49 | 83.34±1.59 | 83.58±1.43 | 92.56±1.14 | 92.74±1.10 |
| 1 | 10 | -0.80 🔥 | -0.60 🔥 | 83.80±1.20 | 84.28±1.81 | 82.55±1.97 | 82.89±1.55 | 92.78±1.16 | 92.94±1.13 |
| 1 | 10 | -0.80 ❄ | -0.50 ❄ | 83.84±1.44 | 83.72±1.74 | 82.79±1.55 | 83.09±1.50 | 92.19±1.06 | 92.40±1.09 |
| 1 | 10 | -0.80 🔥 | -0.50 🔥 | 83.87±1.89 | 83.54±1.91 | 83.09±2.22 | 83.21±2.04 | 92.40±1.36 | 92.57±1.29 |
| 1 | 10 | -0.80 ❄ | -0.40 ❄ | 83.96±1.86 | 83.83±2.24 | 83.00±1.66 | 83.25±1.80 | 92.54±1.42 | 92.73±1.41 |
| 1 | 10 | -0.80 🔥 | -0.40 🔥 | 83.96±1.79 | 83.97±2.33 | 83.00±1.53 | 83.24±1.68 | 92.50±1.40 | 92.69±1.38 |
| 1 | 10 | -0.80 ❄ | -0.30 ❄ | 83.79±1.79 | 84.27±2.40 | 82.24±1.80 | 82.82±1.83 | 92.62±1.39 | 92.83±1.36 |
| 1 | 10 | -0.80 🔥 | -0.30 🔥 | 83.91±1.84 | 84.44±2.58 | 82.34±1.79 | 82.94±1.84 | 92.46±1.59 | 92.71±1.51 |
| 1 | 10 | -0.80 ❄ | -0.20 ❄ | 84.07±1.86 | 84.76±2.30 | 82.25±1.93 | 83.00±1.97 | 92.69±1.80 | 92.93±1.69 |
| 1 | 10 | -0.80 🔥 | -0.20 🔥 | 84.39±2.20 | 84.78±2.25 | 82.77±2.45 | 83.42±2.40 | 92.80±1.70 | 93.03±1.61 |
| 1 | 10 | -0.80 ❄ | -0.10 ❄ | 84.91±1.21 | 85.39±1.19 | 83.33±1.56 | 83.97±1.40 | 93.02±1.31 | 93.28±1.16 |
| 1 | 10 | -0.80 🔥 | -0.10 🔥 | 85.03±1.17 | 85.57±0.97 | 83.41±1.55 | 84.08±1.37 | 93.22±1.11 | 93.46±0.99 |
| 1 | 10 | -0.80 ❄ | 0.10 ❄ | 85.05±1.42 | 86.16±1.20 | 83.04±1.73 | 83.93±1.66 | 93.39±1.18 | 93.60±1.09 |
| 1 | 10 | -0.80 🔥 | 0.10 🔥 | 84.81±1.82 | 85.62±1.86 | 82.99±2.13 | 83.76±2.04 | 93.32±1.34 | 93.48±1.29 |

| t | n | a | b | Accuracy | Precision | Recall | F1 score | AUROC | AUPRC |
|---|---|---|---|---|---|---|---|---|---|
| 1 | 10 | -0.80 ❄ | 0.20 ❄ | $84.86_{\pm2.12}$ | $85.58_{\pm2.18}$ | $83.14_{\pm2.48}$ | $83.85_{\pm2.37}$ | $93.29_{\pm1.67}$ | $93.43_{\pm1.60}$ |
| 1 | 10 | -0.80 🔥 | 0.20 🔥 | $84.91_{\pm1.94}$ | $85.70_{\pm2.08}$ | $83.17_{\pm2.28}$ | $83.89_{\pm2.16}$ | $93.28_{\pm1.60}$ | $93.43_{\pm1.54}$ |
| 1 | 10 | -0.80 ❄ | 0.30 ❄ | $83.83_{\pm2.15}$ | $84.81_{\pm2.52}$ | $81.92_{\pm2.43}$ | $82.67_{\pm2.38}$ | $92.32_{\pm1.75}$ | $92.47_{\pm1.68}$ |
| 1 | 10 | -0.80 🔥 | 0.30 🔥 | $84.18_{\pm2.32}$ | $85.02_{\pm2.50}$ | $82.32_{\pm2.62}$ | $83.07_{\pm2.55}$ | $92.58_{\pm1.94}$ | $92.76_{\pm1.88}$ |
| 1 | 10 | -0.80 ❄ | 0.40 ❄ | $83.80_{\pm2.71}$ | $84.93_{\pm2.96}$ | $81.71_{\pm2.94}$ | $82.56_{\pm2.96}$ | $92.10_{\pm2.30}$ | $92.30_{\pm2.22}$ |
| 1 | 10 | -0.80 🔥 | 0.40 🔥 | $82.77_{\pm2.13}$ | $84.13_{\pm2.48}$ | $80.54_{\pm2.53}$ | $81.36_{\pm2.42}$ | $91.49_{\pm1.87}$ | $91.71_{\pm1.78}$ |
| 1 | 10 | -0.80 ❄ | 0.50 ❄ | $82.92_{\pm2.35}$ | $84.52_{\pm2.66}$ | $80.48_{\pm2.54}$ | $81.44_{\pm2.61}$ | $91.33_{\pm2.48}$ | $91.54_{\pm2.42}$ |
| 1 | 10 | -0.80 🔥 | 0.50 🔥 | $82.75_{\pm2.39}$ | $84.39_{\pm2.53}$ | $80.27_{\pm2.69}$ | $81.22_{\pm2.72}$ | $91.14_{\pm2.45}$ | $91.38_{\pm2.39}$ |
| 1 | 10 | -0.80 ❄ | 0.60 ❄ | $82.82_{\pm2.70}$ | $84.30_{\pm2.92}$ | $80.42_{\pm2.94}$ | $81.35_{\pm3.00}$ | $90.82_{\pm2.86}$ | $91.08_{\pm2.77}$ |
| 1 | 10 | -0.80 🔥 | 0.60 🔥 | $82.70_{\pm2.69}$ | $84.47_{\pm2.95}$ | $80.17_{\pm2.96}$ | $81.13_{\pm3.02}$ | $90.79_{\pm2.98}$ | $91.05_{\pm2.90}$ |
| 1 | 11 | -1.00 ❄ | -1.00 ❄ | $79.22_{\pm1.88}$ | $79.78_{\pm2.26}$ | $77.02_{\pm1.89}$ | $77.66_{\pm1.98}$ | $87.90_{\pm2.48}$ | $87.61_{\pm2.56}$ |
| 1 | 11 | -1.00 🔥 | -1.00 🔥 | $78.95_{\pm2.00}$ | $79.44_{\pm2.13}$ | $76.75_{\pm2.24}$ | $77.36_{\pm2.30}$ | $87.35_{\pm2.75}$ | $87.03_{\pm2.91}$ |
| 1 | 11 | -1.00 ❄ | -0.90 ❄ | $80.32_{\pm1.48}$ | $80.55_{\pm1.47}$ | $78.51_{\pm2.08}$ | $79.03_{\pm1.80}$ | $88.71_{\pm2.44}$ | $88.42_{\pm2.46}$ |
| 1 | 11 | -1.00 🔥 | -0.90 🔥 | $79.55_{\pm1.68}$ | $80.46_{\pm1.44}$ | $77.03_{\pm2.11}$ | $77.79_{\pm2.10}$ | $87.82_{\pm2.48}$ | $87.56_{\pm2.60}$ |
| 1 | 11 | -1.00 ❄ | -0.80 ❄ | $80.14_{\pm3.22}$ | $80.91_{\pm2.61}$ | $77.92_{\pm4.12}$ | $78.49_{\pm4.09}$ | $88.18_{\pm3.57}$ | $87.92_{\pm3.74}$ |
| 1 | 11 | -1.00 🔥 | -0.80 🔥 | $80.95_{\pm2.34}$ | $81.48_{\pm1.94}$ | $78.87_{\pm2.97}$ | $79.52_{\pm2.83}$ | $89.08_{\pm2.46}$ | $88.89_{\pm2.49}$ |
| 1 | 11 | -1.00 ❄ | -0.70 ❄ | $80.50_{\pm2.96}$ | $81.16_{\pm2.38}$ | $78.33_{\pm3.71}$ | $78.94_{\pm3.70}$ | $88.26_{\pm3.57}$ | $88.11_{\pm3.71}$ |
| 1 | 11 | -1.00 🔥 | -0.70 🔥 | $80.69_{\pm2.81}$ | $81.34_{\pm2.24}$ | $78.50_{\pm3.52}$ | $79.15_{\pm3.50}$ | $88.48_{\pm3.45}$ | $88.33_{\pm3.59}$ |
| 1 | 11 | -1.00 ❄ | -0.60 ❄ | $80.80_{\pm2.83}$ | $80.97_{\pm2.55}$ | $79.02_{\pm3.42}$ | $79.52_{\pm3.37}$ | $88.66_{\pm3.59}$ | $88.57_{\pm3.74}$ |
| 1 | 11 | -1.00 🔥 | -0.60 🔥 | $80.60_{\pm2.85}$ | $80.78_{\pm2.74}$ | $78.74_{\pm3.32}$ | $79.30_{\pm3.32}$ | $88.66_{\pm3.45}$ | $88.57_{\pm3.59}$ |
| 1 | 11 | -1.00 ❄ | -0.50 ❄ | $81.15_{\pm3.63}$ | $81.83_{\pm1.84}$ | $79.62_{\pm5.33}$ | $79.70_{\pm5.11}$ | $89.42_{\pm3.83}$ | $89.32_{\pm3.93}$ |
| 1 | 11 | -1.00 🔥 | -0.50 🔥 | $81.54_{\pm2.37}$ | $81.54_{\pm1.82}$ | $80.24_{\pm3.35}$ | $80.49_{\pm2.99}$ | $89.61_{\pm3.25}$ | $89.50_{\pm3.33}$ |
| 1 | 11 | -1.00 ❄ | -0.40 ❄ | $81.54_{\pm2.67}$ | $81.72_{\pm2.38}$ | $79.89_{\pm3.50}$ | $80.35_{\pm3.16}$ | $89.57_{\pm3.11}$ | $89.58_{\pm3.15}$ |
| 1 | 11 | -1.00 🔥 | -0.40 🔥 | $81.33_{\pm3.16}$ | $81.79_{\pm2.42}$ | $79.45_{\pm4.14}$ | $79.95_{\pm3.95}$ | $89.35_{\pm3.30}$ | $89.37_{\pm3.31}$ |
| 1 | 11 | -1.00 ❄ | -0.30 ❄ | $82.12_{\pm1.82}$ | $82.48_{\pm1.78}$ | $80.38_{\pm2.26}$ | $80.96_{\pm2.13}$ | $90.11_{\pm2.71}$ | $90.25_{\pm2.84}$ |
| 1 | 11 | -1.00 🔥 | -0.30 🔥 | $81.01_{\pm3.27}$ | $82.10_{\pm2.88}$ | $78.56_{\pm3.96}$ | $79.31_{\pm4.08}$ | $89.49_{\pm3.67}$ | $89.35_{\pm4.16}$ |
| 1 | 11 | -1.00 ❄ | -0.20 ❄ | $83.20_{\pm1.45}$ | $83.60_{\pm1.84}$ | $81.46_{\pm1.37}$ | $82.14_{\pm1.48}$ | $91.53_{\pm1.64}$ | $91.71_{\pm1.66}$ |
| 1 | 11 | -1.00 🔥 | -0.20 🔥 | $83.10_{\pm1.53}$ | $83.60_{\pm1.87}$ | $81.27_{\pm1.50}$ | $81.99_{\pm1.58}$ | $91.58_{\pm1.63}$ | $91.77_{\pm1.63}$ |
| 1 | 11 | -1.00 ❄ | -0.10 ❄ | $84.37_{\pm1.54}$ | $84.57_{\pm2.01}$ | $82.99_{\pm1.33}$ | $83.53_{\pm1.50}$ | $92.48_{\pm1.38}$ | $92.56_{\pm1.54}$ |
| 1 | 11 | -1.00 🔥 | -0.10 🔥 | $84.46_{\pm1.33}$ | $84.87_{\pm1.86}$ | $82.99_{\pm1.31}$ | $83.56_{\pm1.35}$ | $92.81_{\pm1.37}$ | $92.88_{\pm1.51}$ |
| 1 | 11 | -1.00 ❄ | 0.10 ❄ | $83.73_{\pm1.69}$ | $84.85_{\pm2.98}$ | $82.21_{\pm1.05}$ | $82.73_{\pm1.36}$ | $93.46_{\pm1.44}$ | $93.53_{\pm1.44}$ |
| 1 | 11 | -1.00 🔥 | 0.10 🔥 | $82.74_{\pm2.32}$ | $84.99_{\pm2.84}$ | $80.43_{\pm2.60}$ | $81.21_{\pm2.61}$ | $93.20_{\pm1.45}$ | $93.22_{\pm1.50}$ |
| 1 | 11 | -1.00 ❄ | 0.20 ❄ | $83.62_{\pm2.29}$ | $84.44_{\pm3.21}$ | $82.16_{\pm1.83}$ | $82.66_{\pm2.13}$ | $92.80_{\pm1.99}$ | $92.81_{\pm2.07}$ |
| 1 | 11 | -1.00 🔥 | 0.20 🔥 | $83.89_{\pm2.04}$ | $84.90_{\pm2.93}$ | $82.32_{\pm1.74}$ | $82.88_{\pm1.92}$ | $93.41_{\pm1.67}$ | $93.48_{\pm1.65}$ |
| 1 | 11 | -1.00 ❄ | 0.30 ❄ | $83.82_{\pm2.04}$ | $84.80_{\pm2.93}$ | $82.24_{\pm1.38}$ | $82.82_{\pm1.79}$ | $93.31_{\pm1.81}$ | $93.37_{\pm1.80}$ |
| 1 | 11 | -1.00 🔥 | 0.30 🔥 | $82.96_{\pm1.90}$ | $84.00_{\pm2.84}$ | $81.31_{\pm1.82}$ | $81.85_{\pm1.84}$ | $92.42_{\pm2.02}$ | $92.52_{\pm1.95}$ |
| 1 | 11 | -1.00 ❄ | 0.40 ❄ | $83.59_{\pm2.03}$ | $84.92_{\pm3.01}$ | $81.71_{\pm1.68}$ | $82.43_{\pm1.93}$ | $93.09_{\pm2.10}$ | $93.15_{\pm2.08}$ |
| 1 | 11 | -1.00 🔥 | 0.40 🔥 | $83.72_{\pm2.15}$ | $85.07_{\pm3.14}$ | $81.84_{\pm1.68}$ | $82.57_{\pm2.00}$ | $93.11_{\pm2.17}$ | $93.17_{\pm2.14}$ |
| 1 | 11 | -1.00 ❄ | 0.50 ❄ | $81.58_{\pm2.82}$ | $83.88_{\pm2.85}$ | $79.04_{\pm3.47}$ | $79.77_{\pm3.58}$ | $92.37_{\pm1.95}$ | $92.40_{\pm1.99}$ |
| 1 | 11 | -1.00 🔥 | 0.50 🔥 | $81.38_{\pm2.67}$ | $83.85_{\pm2.80}$ | $78.72_{\pm3.23}$ | $79.50_{\pm3.40}$ | $92.17_{\pm1.82}$ | $92.21_{\pm1.85}$ |
| 1 | 11 | -1.00 ❄ | 0.60 ❄ | $81.27_{\pm2.91}$ | $83.59_{\pm3.45}$ | $78.66_{\pm3.11}$ | $79.46_{\pm3.33}$ | $91.70_{\pm2.11}$ | $91.73_{\pm2.11}$ |
| 1 | 11 | -1.00 🔥 | 0.60 🔥 | $81.64_{\pm3.07}$ | $83.33_{\pm3.43}$ | $79.29_{\pm3.24}$ | $80.05_{\pm3.46}$ | $91.73_{\pm2.16}$ | $91.77_{\pm2.16}$ |
| 1 | 11 | -1.00 ❄ | 0.70 ❄ | $82.84_{\pm1.59}$ | $85.24_{\pm1.04}$ | $80.22_{\pm2.58}$ | $81.13_{\pm2.37}$ | $92.97_{\pm0.53}$ | $93.03_{\pm0.47}$ |
| 1 | 11 | -1.00 🔥 | 0.70 🔥 | $83.37_{\pm1.53}$ | $85.39_{\pm1.02}$ | $80.92_{\pm2.38}$ | $81.83_{\pm2.19}$ | $93.12_{\pm0.44}$ | $93.17_{\pm0.41}$ |
| 1 | 11 | -1.00 ❄ | 0.80 ❄ | $83.21_{\pm1.26}$ | $85.12_{\pm1.01}$ | $80.78_{\pm2.07}$ | $81.69_{\pm1.82}$ | $92.67_{\pm0.56}$ | $92.72_{\pm0.50}$ |
| 1 | 11 | -1.00 🔥 | 0.80 🔥 | $81.71_{\pm2.85}$ | $84.43_{\pm1.93}$ | $78.82_{\pm3.93}$ | $79.66_{\pm3.81}$ | $91.85_{\pm1.61}$ | $91.91_{\pm1.67}$ |
| 1 | 11 | -1.00 ❄ | 0.90 ❄ | $81.12_{\pm1.64}$ | $83.78_{\pm1.27}$ | $78.20_{\pm2.68}$ | $79.06_{\pm2.50}$ | $91.78_{\pm0.87}$ | $91.84_{\pm0.80}$ |
| 1 | 11 | -1.00 🔥 | 0.90 🔥 | $82.00_{\pm1.53}$ | $83.96_{\pm1.20}$ | $79.41_{\pm2.35}$ | $80.28_{\pm2.18}$ | $91.47_{\pm0.66}$ | $91.56_{\pm0.56}$ |
| 1 | 11 | -1.00 ❄ | 1.00 ❄ | $81.36_{\pm1.50}$ | $83.66_{\pm1.26}$ | $78.52_{\pm2.24}$ | $79.43_{\pm2.21}$ | $91.25_{\pm0.61}$ | $91.32_{\pm0.55}$ |
| 1 | 11 | -1.00 🔥 | 1.00 🔥 | $80.85_{\pm1.66}$ | $83.29_{\pm1.40}$ | $77.91_{\pm2.43}$ | $78.80_{\pm2.42}$ | $91.21_{\pm0.51}$ | $91.29_{\pm0.49}$ |

| t | n | a | b | Accuracy | Precision | Recall | F1 score | AUROC | AUPRC |
|---|---|---|---|---|---|---|---|---|---|
| 1 | 11 | -0.90 ❄ | -1.00 ❄ | 80.14±1.22 | 80.37±1.18 | 78.14±1.45 | 78.78±1.41 | 87.82±2.40 | 87.65±2.41 |
| 1 | 11 | -0.90 🔥 | -1.00 🔥 | 79.36±2.28 | 79.55±2.03 | 77.39±2.81 | 77.92±2.81 | 87.28±2.79 | 87.05±2.90 |
| 1 | 11 | -0.90 🔥 | -0.90 ❄ | 79.34±1.81 | 80.12±2.30 | 77.02±2.02 | 77.70±2.04 | 87.42±3.02 | 87.18±3.21 |
| 1 | 11 | -0.90 🔥 | -0.90 🔥 | 79.64±1.46 | 80.28±2.29 | 77.48±1.31 | 78.13±1.38 | 87.80±2.46 | 87.62±2.54 |
| 1 | 11 | -0.90 ❄ | -0.80 ❄ | 79.99±1.70 | 80.35±2.32 | 78.03±1.33 | 78.65±1.54 | 88.22±2.41 | 88.02±2.55 |
| 1 | 11 | -0.90 🔥 | -0.80 🔥 | 80.08±1.43 | 80.51±2.03 | 78.05±1.27 | 78.70±1.37 | 88.32±2.07 | 88.16±2.20 |
| 1 | 11 | -0.90 ❄ | -0.70 ❄ | 80.13±1.81 | 80.51±2.14 | 78.09±2.04 | 78.72±2.02 | 88.30±3.05 | 88.19±3.26 |
| 1 | 11 | -0.90 🔥 | -0.70 🔥 | 80.46±1.64 | 81.08±1.97 | 78.33±1.94 | 79.01±1.88 | 88.73±2.75 | 88.60±2.93 |
| 1 | 11 | -0.90 ❄ | -0.60 ❄ | 80.95±2.47 | 81.47±2.49 | 78.90±2.85 | 79.56±2.88 | 89.18±2.81 | 89.13±2.98 |
| 1 | 11 | -0.90 🔥 | -0.60 🔥 | 81.15±1.86 | 81.78±1.94 | 79.03±2.12 | 79.75±2.13 | 89.45±1.97 | 89.39±2.06 |
| 1 | 11 | -0.90 ❄ | -0.50 ❄ | 81.48±1.97 | 81.81±2.02 | 80.02±2.69 | 80.37±2.37 | 89.74±2.37 | 89.70±2.46 |
| 1 | 11 | -0.90 🔥 | -0.50 🔥 | 81.48±2.50 | 82.06±2.28 | 79.74±3.32 | 80.20±3.09 | 89.57±2.98 | 89.54±3.09 |
| 1 | 11 | -0.90 ❄ | -0.40 ❄ | 81.22±3.24 | 81.77±2.51 | 79.45±4.52 | 79.83±4.12 | 89.10±3.93 | 89.12±3.95 |
| 1 | 11 | -0.90 🔥 | -0.40 🔥 | 81.10±3.01 | 81.01±2.73 | 79.59±3.72 | 79.98±3.54 | 88.97±3.37 | 89.03±3.47 |
| 1 | 11 | -0.90 ❄ | -0.30 ❄ | 82.32±1.46 | 82.67±1.14 | 80.71±2.18 | 81.21±1.87 | 90.17±1.98 | 90.32±1.97 |
| 1 | 11 | -0.90 🔥 | -0.30 🔥 | 82.77±2.66 | 82.67±2.04 | 81.85±3.76 | 81.91±3.29 | 90.29±3.45 | 90.29±3.53 |
| 1 | 11 | -0.90 ❄ | -0.20 ❄ | 82.33±1.09 | 83.49±1.87 | 80.24±1.52 | 80.98±1.30 | 90.80±1.66 | 90.96±1.71 |
| 1 | 11 | -0.90 🔥 | -0.20 🔥 | 82.77±1.68 | 83.45±1.65 | 80.87±2.20 | 81.55±2.01 | 90.88±1.98 | 91.04±2.08 |
| 1 | 11 | -0.90 ❄ | -0.10 ❄ | 84.37±1.46 | 84.89±1.69 | 82.85±1.56 | 83.43±1.53 | 92.36±1.32 | 92.45±1.49 |
| 1 | 11 | -0.90 🔥 | -0.10 🔥 | 84.14±1.71 | 84.40±2.14 | 82.80±1.63 | 83.28±1.71 | 92.31±1.42 | 92.40±1.58 |
| 1 | 11 | -0.90 ❄ | 0.10 ❄ | 85.19±1.31 | 86.08±1.29 | 83.39±1.68 | 84.16±1.53 | 93.98±0.84 | 94.08±0.91 |
| 1 | 11 | -0.90 🔥 | 0.10 🔥 | 84.25±2.52 | 85.39±2.50 | 82.41±2.92 | 83.10±2.90 | 93.15±2.03 | 93.18±2.23 |
| 1 | 11 | -0.90 ❄ | 0.20 ❄ | 83.24±1.93 | 84.15±2.38 | 81.61±2.19 | 82.15±2.14 | 92.52±1.61 | 92.61±1.56 |
| 1 | 11 | -0.90 🔥 | 0.20 🔥 | 82.68±1.90 | 83.89±2.25 | 80.90±2.49 | 81.43±2.29 | 92.36±1.49 | 92.42±1.43 |
| 1 | 11 | -0.90 ❄ | 0.30 ❄ | 81.52±1.86 | 83.80±2.72 | 79.08±2.15 | 79.83±2.12 | 92.12±1.77 | 92.19±1.72 |
| 1 | 11 | -0.90 🔥 | 0.30 🔥 | 81.72±2.91 | 84.83±3.05 | 79.04±3.60 | 79.77±3.77 | 93.15±1.85 | 93.18±1.85 |
| 1 | 11 | -0.90 ❄ | 0.40 ❄ | 82.68±2.38 | 84.76±3.01 | 80.42±2.63 | 81.19±2.65 | 92.73±1.87 | 92.79±1.89 |
| 1 | 11 | -0.90 🔥 | 0.40 🔥 | 82.33±2.22 | 84.56±2.94 | 79.99±2.52 | 80.75±2.53 | 92.73±1.87 | 92.79±1.89 |
| 1 | 11 | -0.90 ❄ | 0.50 ❄ | 80.22±2.26 | 83.74±3.01 | 77.06±2.85 | 77.86±3.00 | 91.89±1.90 | 91.96±1.91 |
| 1 | 11 | -0.90 🔥 | 0.50 🔥 | 82.25±1.63 | 84.45±2.98 | 79.83±1.60 | 80.67±1.67 | 92.55±1.83 | 92.59±1.80 |
| 1 | 11 | -0.90 ❄ | 0.60 ❄ | 81.94±3.27 | 85.51±1.67 | 78.75±4.37 | 79.66±4.48 | 93.08±1.12 | 93.08±1.12 |
| 1 | 11 | -0.90 🔥 | 0.60 🔥 | 80.59±3.45 | 83.82±3.66 | 77.69±4.10 | 78.38±4.36 | 91.83±2.16 | 91.87±2.14 |
| 1 | 11 | -0.90 ❄ | 0.70 ❄ | 81.48±3.27 | 84.64±1.36 | 78.39±4.45 | 79.21±4.53 | 92.24±1.32 | 92.25±1.22 |
| 1 | 11 | -0.90 🔥 | 0.70 🔥 | 81.65±2.80 | 85.06±1.38 | 78.49±3.90 | 79.40±3.87 | 92.44±1.43 | 92.47±1.41 |
| 1 | 11 | -0.90 ❄ | 0.80 ❄ | 82.43±2.13 | 84.70±1.00 | 79.76±3.07 | 80.65±2.99 | 92.18±0.69 | 92.20±0.61 |
| 1 | 11 | -0.90 🔥 | 0.80 🔥 | 82.38±1.89 | 84.62±1.17 | 79.77±2.88 | 80.63±2.75 | 92.13±0.82 | 92.19±0.77 |
| 1 | 11 | -0.90 ❄ | 0.90 ❄ | 80.80±1.61 | 83.74±1.34 | 77.75±2.67 | 78.60±2.53 | 91.39±1.01 | 91.46±0.95 |
| 1 | 11 | -0.90 🔥 | 0.90 🔥 | 80.20±1.74 | 83.09±2.11 | 77.10±2.67 | 77.93±2.53 | 90.86±1.74 | 90.93±1.78 |
| 1 | 11 | -0.90 ❄ | 1.00 ❄ | 80.77±1.59 | 83.06±1.41 | 77.92±2.45 | 78.76±2.39 | 90.62±0.65 | 90.70±0.66 |
| 1 | 11 | -0.90 🔥 | 1.00 🔥 | 80.71±1.75 | 83.07±1.14 | 77.87±2.78 | 78.67±2.76 | 90.46±0.67 | 90.55±0.62 |
| 1 | 11 | -0.80 ❄ | -1.00 ❄ | 80.31±0.95 | 80.70±1.70 | 78.46±1.16 | 79.01±1.05 | 88.49±1.64 | 88.43±1.68 |
| 1 | 11 | -0.80 🔥 | -1.00 🔥 | 80.07±0.97 | 80.54±1.85 | 78.10±0.62 | 78.71±0.75 | 88.22±1.85 | 88.17±1.95 |
| 1 | 11 | -0.80 ❄ | -0.90 ❄ | 79.85±1.16 | 80.44±1.54 | 77.69±1.39 | 78.34±1.35 | 88.06±2.13 | 87.98±2.23 |
| 1 | 11 | -0.80 🔥 | -0.90 🔥 | 79.93±1.22 | 80.24±1.97 | 78.05±1.05 | 78.62±1.11 | 88.02±2.16 | 87.92±2.29 |
| 1 | 11 | -0.80 ❄ | -0.80 ❄ | 80.31±1.34 | 81.35±1.74 | 77.82±1.43 | 78.64±1.47 | 88.31±2.37 | 88.23±2.52 |
| 1 | 11 | -0.80 🔥 | -0.80 🔥 | 80.10±1.45 | 80.86±1.73 | 77.87±1.73 | 78.55±1.69 | 88.30±2.41 | 88.17±2.58 |
| 1 | 11 | -0.80 ❄ | -0.70 ❄ | 80.22±2.11 | 80.68±2.41 | 78.13±2.34 | 78.80±2.32 | 88.09±3.14 | 87.99±3.35 |
| 1 | 11 | -0.80 🔥 | -0.70 🔥 | 80.42±2.33 | 80.79±2.78 | 78.45±2.48 | 79.08±2.49 | 88.28±3.28 | 88.18±3.51 |
| 1 | 11 | -0.80 ❄ | -0.60 ❄ | 80.50±2.06 | 81.00±2.64 | 78.55±2.19 | 79.16±2.15 | 88.78±3.01 | 88.71±3.22 |
| 1 | 11 | -0.80 🔥 | -0.60 🔥 | 80.75±2.27 | 81.63±2.17 | 78.52±2.94 | 79.20±2.85 | 88.94±2.83 | 88.92±3.00 |

| t | n | a | b | Accuracy | Precision | Recall | F1 score | AUROC | AUPRC |
|---|---|---|---|---|---|---|---|---|---|
| 1 | 11 | -0.80 ❄ | -0.50 ❄ | $80.99_{\pm 2.48}$ | $81.13_{\pm 2.94}$ | $79.36_{\pm 2.54}$ | $79.87_{\pm 2.56}$ | $88.87_{\pm 3.30}$ | $88.92_{\pm 3.45}$ |
| 1 | 11 | -0.80 🔥 | -0.50 🔥 | $81.01_{\pm 2.16}$ | $81.34_{\pm 2.34}$ | $79.36_{\pm 2.66}$ | $79.81_{\pm 2.52}$ | $89.09_{\pm 2.76}$ | $89.15_{\pm 2.86}$ |
| 1 | 11 | -0.80 ❄ | -0.40 ❄ | $81.15_{\pm 2.14}$ | $81.62_{\pm 2.19}$ | $79.36_{\pm 2.79}$ | $79.87_{\pm 2.60}$ | $88.92_{\pm 2.87}$ | $89.03_{\pm 2.99}$ |
| 1 | 11 | -0.80 🔥 | -0.40 🔥 | $81.19_{\pm 2.06}$ | $81.17_{\pm 1.63}$ | $79.70_{\pm 2.78}$ | $80.08_{\pm 2.59}$ | $88.93_{\pm 2.74}$ | $89.02_{\pm 2.80}$ |
| 1 | 11 | -0.80 ❄ | -0.30 ❄ | $82.22_{\pm 1.90}$ | $82.63_{\pm 1.73}$ | $80.62_{\pm 2.50}$ | $81.10_{\pm 2.32}$ | $89.95_{\pm 2.94}$ | $90.08_{\pm 3.07}$ |
| 1 | 11 | -0.80 🔥 | -0.30 🔥 | $82.14_{\pm 1.92}$ | $82.55_{\pm 1.57}$ | $80.49_{\pm 2.58}$ | $80.98_{\pm 2.40}$ | $89.90_{\pm 2.73}$ | $90.08_{\pm 2.81}$ |
| 1 | 11 | -0.80 ❄ | -0.20 ❄ | $82.18_{\pm 1.22}$ | $83.92_{\pm 1.86}$ | $79.72_{\pm 1.70}$ | $80.60_{\pm 1.58}$ | $90.70_{\pm 2.02}$ | $90.93_{\pm 2.10}$ |
| 1 | 11 | -0.80 🔥 | -0.20 🔥 | $82.71_{\pm 1.97}$ | $83.36_{\pm 2.01}$ | $80.95_{\pm 2.45}$ | $81.54_{\pm 2.33}$ | $90.79_{\pm 2.29}$ | $90.95_{\pm 2.37}$ |
| 1 | 11 | -0.80 ❄ | -0.10 ❄ | $83.70_{\pm 1.39}$ | $84.68_{\pm 1.85}$ | $81.92_{\pm 1.80}$ | $82.58_{\pm 1.61}$ | $92.13_{\pm 1.48}$ | $92.33_{\pm 1.54}$ |
| 1 | 11 | -0.80 🔥 | -0.10 🔥 | $83.87_{\pm 1.55}$ | $84.82_{\pm 1.89}$ | $82.23_{\pm 1.91}$ | $82.80_{\pm 1.74}$ | $92.14_{\pm 1.53}$ | $92.36_{\pm 1.56}$ |
| 1 | 11 | -0.80 ❄ | 0.10 ❄ | $84.17_{\pm 1.31}$ | $85.52_{\pm 2.56}$ | $82.50_{\pm 1.43}$ | $83.09_{\pm 1.31}$ | $93.34_{\pm 1.36}$ | $93.43_{\pm 1.40}$ |
| 1 | 11 | -0.80 🔥 | 0.10 🔥 | $84.50_{\pm 1.85}$ | $85.22_{\pm 2.27}$ | $83.16_{\pm 2.08}$ | $83.60_{\pm 2.00}$ | $93.11_{\pm 1.23}$ | $93.21_{\pm 1.27}$ |
| 1 | 11 | -0.80 ❄ | 0.20 ❄ | $84.18_{\pm 2.35}$ | $85.30_{\pm 2.13}$ | $82.11_{\pm 2.81}$ | $82.96_{\pm 2.71}$ | $93.22_{\pm 1.82}$ | $93.26_{\pm 1.85}$ |
| 1 | 11 | -0.80 🔥 | 0.20 🔥 | $83.61_{\pm 1.84}$ | $85.27_{\pm 2.07}$ | $81.21_{\pm 2.10}$ | $82.19_{\pm 2.08}$ | $93.07_{\pm 1.62}$ | $93.11_{\pm 1.65}$ |
| 1 | 11 | -0.80 ❄ | 0.30 ❄ | $81.97_{\pm 3.23}$ | $83.74_{\pm 2.62}$ | $79.74_{\pm 4.14}$ | $80.34_{\pm 4.16}$ | $92.14_{\pm 1.82}$ | $92.17_{\pm 1.83}$ |
| 1 | 11 | -0.80 🔥 | 0.30 🔥 | $82.22_{\pm 2.81}$ | $83.93_{\pm 2.42}$ | $80.01_{\pm 3.62}$ | $80.66_{\pm 3.54}$ | $92.15_{\pm 1.74}$ | $92.19_{\pm 1.75}$ |
| 1 | 11 | -0.80 ❄ | 0.40 ❄ | $82.52_{\pm 2.69}$ | $85.46_{\pm 1.58}$ | $79.52_{\pm 3.50}$ | $80.54_{\pm 3.50}$ | $93.40_{\pm 1.17}$ | $93.43_{\pm 1.26}$ |
| 1 | 11 | -0.80 🔥 | 0.40 🔥 | $82.52_{\pm 2.69}$ | $85.44_{\pm 1.65}$ | $79.53_{\pm 3.48}$ | $80.54_{\pm 3.56}$ | $93.48_{\pm 1.00}$ | $93.50_{\pm 1.10}$ |
| 1 | 11 | -0.80 ❄ | 0.50 ❄ | $82.04_{\pm 2.79}$ | $84.87_{\pm 1.69}$ | $79.13_{\pm 3.81}$ | $80.02_{\pm 3.81}$ | $92.83_{\pm 0.97}$ | $92.87_{\pm 1.01}$ |
| 1 | 11 | -0.80 🔥 | 0.50 🔥 | $82.19_{\pm 3.01}$ | $85.03_{\pm 1.60}$ | $79.33_{\pm 4.16}$ | $80.18_{\pm 4.12}$ | $92.96_{\pm 0.94}$ | $93.00_{\pm 0.99}$ |
| 1 | 12 | -1.00 ❄ | -1.00 ❄ | $84.18_{\pm 2.02}$ | $84.30_{\pm 2.64}$ | $83.31_{\pm 1.89}$ | $83.49_{\pm 1.94}$ | $92.87_{\pm 1.55}$ | $92.95_{\pm 1.61}$ |
| 1 | 12 | -1.00 🔥 | -1.00 🔥 | $83.59_{\pm 1.61}$ | $83.64_{\pm 1.85}$ | $83.17_{\pm 0.85}$ | $83.02_{\pm 1.36}$ | $92.53_{\pm 1.00}$ | $92.60_{\pm 1.04}$ |
| 1 | 12 | -1.00 ❄ | -0.90 ❄ | $84.30_{\pm 1.87}$ | $84.23_{\pm 2.35}$ | $83.67_{\pm 1.63}$ | $83.71_{\pm 1.77}$ | $92.93_{\pm 1.25}$ | $92.99_{\pm 1.35}$ |
| 1 | 12 | -1.00 🔥 | -0.90 🔥 | $84.46_{\pm 2.02}$ | $84.24_{\pm 2.35}$ | $83.84_{\pm 1.87}$ | $83.88_{\pm 1.99}$ | $92.75_{\pm 1.60}$ | $92.86_{\pm 1.59}$ |
| 1 | 12 | -1.00 ❄ | -0.80 ❄ | $84.88_{\pm 2.12}$ | $84.88_{\pm 2.73}$ | $84.08_{\pm 2.16}$ | $84.24_{\pm 2.13}$ | $93.34_{\pm 1.55}$ | $93.42_{\pm 1.62}$ |
| 1 | 12 | -1.00 🔥 | -0.80 🔥 | $84.58_{\pm 2.26}$ | $84.43_{\pm 2.63}$ | $83.82_{\pm 2.08}$ | $83.97_{\pm 2.22}$ | $92.96_{\pm 1.41}$ | $93.06_{\pm 1.46}$ |
| 1 | 12 | -1.00 ❄ | -0.70 ❄ | $85.19_{\pm 1.93}$ | $85.16_{\pm 2.35}$ | $84.34_{\pm 1.91}$ | $84.55_{\pm 1.93}$ | $93.39_{\pm 1.39}$ | $93.46_{\pm 1.47}$ |
| 1 | 12 | -1.00 🔥 | -0.70 🔥 | $85.19_{\pm 1.78}$ | $85.04_{\pm 2.09}$ | $84.32_{\pm 1.84}$ | $84.55_{\pm 1.83}$ | $93.24_{\pm 1.39}$ | $93.32_{\pm 1.47}$ |
| 1 | 12 | -1.00 ❄ | -0.60 ❄ | $85.35_{\pm 1.94}$ | $85.64_{\pm 2.67}$ | $84.55_{\pm 2.14}$ | $84.69_{\pm 1.99}$ | $93.76_{\pm 1.56}$ | $93.90_{\pm 1.56}$ |
| 1 | 12 | -1.00 🔥 | -0.60 🔥 | $85.41_{\pm 1.65}$ | $85.79_{\pm 2.22}$ | $84.48_{\pm 2.21}$ | $84.69_{\pm 1.83}$ | $93.77_{\pm 1.35}$ | $93.93_{\pm 1.34}$ |
| 1 | 12 | -1.00 ❄ | -0.50 ❄ | $85.14_{\pm 1.30}$ | $85.62_{\pm 1.70}$ | $83.78_{\pm 1.67}$ | $84.29_{\pm 1.43}$ | $93.52_{\pm 1.33}$ | $93.70_{\pm 1.27}$ |
| 1 | 12 | -1.00 🔥 | -0.50 🔥 | $85.52_{\pm 1.80}$ | $85.49_{\pm 1.87}$ | $84.52_{\pm 2.07}$ | $84.83_{\pm 1.94}$ | $93.54_{\pm 1.44}$ | $93.71_{\pm 1.39}$ |
| 1 | 12 | -1.00 ❄ | -0.40 ❄ | $84.03_{\pm 1.12}$ | $84.71_{\pm 2.00}$ | $82.58_{\pm 1.36}$ | $83.07_{\pm 1.19}$ | $93.14_{\pm 0.98}$ | $93.30_{\pm 0.95}$ |
| 1 | 12 | -1.00 🔥 | -0.40 🔥 | $83.98_{\pm 1.14}$ | $84.70_{\pm 2.22}$ | $82.42_{\pm 0.60}$ | $83.01_{\pm 0.88}$ | $92.91_{\pm 1.18}$ | $93.09_{\pm 1.13}$ |
| 1 | 12 | -1.00 ❄ | -0.30 ❄ | $84.05_{\pm 1.06}$ | $84.61_{\pm 1.59}$ | $82.58_{\pm 1.51}$ | $83.09_{\pm 1.28}$ | $92.92_{\pm 0.78}$ | $93.08_{\pm 0.78}$ |
| 1 | 12 | -1.00 🔥 | -0.30 🔥 | $83.97_{\pm 1.24}$ | $84.45_{\pm 1.65}$ | $82.61_{\pm 1.74}$ | $83.04_{\pm 1.51}$ | $92.91_{\pm 0.79}$ | $93.06_{\pm 0.79}$ |
| 1 | 12 | -1.00 ❄ | -0.20 ❄ | $85.24_{\pm 0.88}$ | $85.64_{\pm 1.45}$ | $83.77_{\pm 0.73}$ | $84.39_{\pm 0.83}$ | $93.51_{\pm 0.87}$ | $93.67_{\pm 0.82}$ |
| 1 | 12 | -1.00 🔥 | -0.20 🔥 | $85.19_{\pm 1.04}$ | $85.49_{\pm 1.37}$ | $83.76_{\pm 0.98}$ | $84.35_{\pm 1.05}$ | $93.45_{\pm 0.89}$ | $93.61_{\pm 0.84}$ |
| 1 | 12 | -1.00 ❄ | -0.10 ❄ | $83.69_{\pm 1.77}$ | $84.82_{\pm 1.96}$ | $81.70_{\pm 2.22}$ | $82.46_{\pm 2.15}$ | $92.87_{\pm 1.30}$ | $93.03_{\pm 1.28}$ |
| 1 | 12 | -1.00 🔥 | -0.10 🔥 | $83.62_{\pm 2.13}$ | $85.05_{\pm 1.89}$ | $81.46_{\pm 2.78}$ | $82.27_{\pm 2.63}$ | $92.81_{\pm 1.47}$ | $92.96_{\pm 1.46}$ |
| 1 | 12 | -1.00 ❄ | 0.10 ❄ | $84.25_{\pm 1.76}$ | $84.32_{\pm 2.16}$ | $82.92_{\pm 1.65}$ | $83.42_{\pm 1.79}$ | $92.53_{\pm 1.45}$ | $92.65_{\pm 1.46}$ |
| 1 | 12 | -1.00 🔥 | 0.10 🔥 | $84.33_{\pm 1.73}$ | $84.54_{\pm 2.03}$ | $82.88_{\pm 1.72}$ | $83.45_{\pm 1.80}$ | $92.63_{\pm 1.45}$ | $92.74_{\pm 1.46}$ |
| 1 | 12 | -1.00 ❄ | 0.20 ❄ | $82.74_{\pm 1.78}$ | $83.24_{\pm 1.66}$ | $80.88_{\pm 2.12}$ | $81.57_{\pm 2.03}$ | $91.24_{\pm 1.56}$ | $91.40_{\pm 1.55}$ |
| 1 | 12 | -1.00 🔥 | 0.20 🔥 | $82.21_{\pm 1.51}$ | $82.63_{\pm 1.77}$ | $80.39_{\pm 1.64}$ | $81.04_{\pm 1.63}$ | $90.98_{\pm 1.67}$ | $91.05_{\pm 1.76}$ |
| 1 | 12 | -1.00 ❄ | 0.30 ❄ | $80.52_{\pm 2.72}$ | $80.56_{\pm 3.10}$ | $79.14_{\pm 2.51}$ | $79.49_{\pm 2.72}$ | $89.65_{\pm 2.26}$ | $89.66_{\pm 2.42}$ |
| 1 | 12 | -1.00 🔥 | 0.30 🔥 | $80.64_{\pm 2.87}$ | $80.62_{\pm 3.13}$ | $79.39_{\pm 2.69}$ | $79.68_{\pm 2.89}$ | $89.80_{\pm 2.23}$ | $89.80_{\pm 2.44}$ |
| 1 | 12 | -1.00 ❄ | 0.40 ❄ | $79.37_{\pm 2.80}$ | $79.60_{\pm 2.74}$ | $78.00_{\pm 2.38}$ | $78.26_{\pm 2.69}$ | $88.59_{\pm 2.08}$ | $88.71_{\pm 2.21}$ |
| 1 | 12 | -1.00 🔥 | 0.40 🔥 | $79.55_{\pm 2.75}$ | $79.95_{\pm 3.15}$ | $77.86_{\pm 2.33}$ | $78.30_{\pm 2.64}$ | $88.76_{\pm 2.13}$ | $88.87_{\pm 2.26}$ |
| 1 | 12 | -1.00 ❄ | 0.50 ❄ | $77.61_{\pm 2.79}$ | $78.16_{\pm 3.04}$ | $75.85_{\pm 2.19}$ | $76.17_{\pm 2.56}$ | $86.92_{\pm 1.93}$ | $87.05_{\pm 2.14}$ |
| 1 | 12 | -1.00 🔥 | 0.50 🔥 | $77.75_{\pm 2.77}$ | $78.29_{\pm 2.93}$ | $76.01_{\pm 2.18}$ | $76.33_{\pm 2.56}$ | $87.06_{\pm 1.87}$ | $87.16_{\pm 2.12}$ |

| t | n | a | b | Accuracy | Precision | Recall | F1 score | AUROC | AUPRC |
|---|---|---|---|---|---|---|---|---|---|
| 1 | 12 | -1.00 ❄ | 0.60 ❄ | $76.59_{\pm2.74}$ | $76.89_{\pm2.96}$ | $74.58_{\pm2.54}$ | $74.97_{\pm2.77}$ | $85.51_{\pm1.86}$ | $85.68_{\pm2.08}$ |
| 1 | 12 | -1.00 🔥 | 0.60 🔥 | $76.80_{\pm2.77}$ | $77.22_{\pm3.10}$ | $74.67_{\pm2.60}$ | $75.11_{\pm2.83}$ | $85.64_{\pm1.94}$ | $85.82_{\pm2.16}$ |
| 1 | 12 | -1.00 ❄ | 0.70 ❄ | $74.58_{\pm1.66}$ | $75.08_{\pm1.90}$ | $72.82_{\pm1.73}$ | $72.87_{\pm1.76}$ | $83.55_{\pm0.69}$ | $83.69_{\pm0.95}$ |
| 1 | 12 | -1.00 🔥 | 0.70 🔥 | $74.14_{\pm2.41}$ | $75.06_{\pm1.99}$ | $72.17_{\pm3.45}$ | $72.02_{\pm3.77}$ | $83.26_{\pm1.57}$ | $83.38_{\pm1.80}$ |
| 1 | 12 | -1.00 ❄ | 0.80 ❄ | $72.86_{\pm0.93}$ | $73.07_{\pm1.82}$ | $70.88_{\pm1.55}$ | $70.94_{\pm1.47}$ | $81.88_{\pm0.69}$ | $81.97_{\pm0.94}$ |
| 1 | 12 | -1.00 🔥 | 0.80 🔥 | $72.38_{\pm1.91}$ | $72.81_{\pm1.18}$ | $70.22_{\pm3.39}$ | $70.05_{\pm3.82}$ | $81.26_{\pm1.44}$ | $81.39_{\pm1.58}$ |
| 1 | 12 | -1.00 ❄ | 0.90 ❄ | $72.69_{\pm1.54}$ | $72.52_{\pm1.88}$ | $70.35_{\pm1.86}$ | $70.63_{\pm1.91}$ | $80.94_{\pm1.12}$ | $81.00_{\pm1.17}$ |
| 1 | 12 | -1.00 🔥 | 0.90 🔥 | $72.30_{\pm1.19}$ | $72.15_{\pm1.49}$ | $69.91_{\pm1.78}$ | $70.14_{\pm1.79}$ | $80.65_{\pm1.03}$ | $80.73_{\pm1.06}$ |
| 1 | 12 | -1.00 ❄ | 1.00 ❄ | $72.82_{\pm1.82}$ | $72.76_{\pm1.85}$ | $70.16_{\pm2.39}$ | $70.48_{\pm2.47}$ | $80.34_{\pm1.35}$ | $80.52_{\pm1.41}$ |
| 1 | 12 | -1.00 🔥 | 1.00 🔥 | $72.30_{\pm2.14}$ | $72.34_{\pm2.32}$ | $69.73_{\pm2.70}$ | $69.93_{\pm2.98}$ | $79.28_{\pm2.79}$ | $79.61_{\pm2.69}$ |
| 1 | 12 | -0.90 ❄ | -1.00 ❄ | $82.81_{\pm2.69}$ | $83.16_{\pm3.15}$ | $81.99_{\pm2.68}$ | $82.04_{\pm2.64}$ | $91.96_{\pm2.05}$ | $92.09_{\pm2.05}$ |
| 1 | 12 | -0.90 🔥 | -1.00 🔥 | $84.29_{\pm1.64}$ | $83.94_{\pm1.78}$ | $83.60_{\pm1.85}$ | $83.68_{\pm1.73}$ | $92.44_{\pm1.44}$ | $92.58_{\pm1.41}$ |
| 1 | 12 | -0.90 ❄ | -0.90 ❄ | $84.19_{\pm1.82}$ | $84.06_{\pm2.20}$ | $83.58_{\pm1.73}$ | $83.60_{\pm1.78}$ | $92.65_{\pm1.14}$ | $92.73_{\pm1.19}$ |
| 1 | 12 | -0.90 🔥 | -0.90 🔥 | $84.12_{\pm1.68}$ | $84.03_{\pm2.13}$ | $83.37_{\pm1.63}$ | $83.48_{\pm1.64}$ | $92.39_{\pm1.34}$ | $92.52_{\pm1.33}$ |
| 1 | 12 | -0.90 ❄ | -0.80 ❄ | $84.10_{\pm1.29}$ | $84.81_{\pm2.47}$ | $82.86_{\pm1.63}$ | $83.21_{\pm1.28}$ | $93.24_{\pm1.59}$ | $93.30_{\pm1.67}$ |
| 1 | 12 | -0.90 🔥 | -0.80 🔥 | $84.47_{\pm1.62}$ | $84.60_{\pm2.31}$ | $83.65_{\pm1.61}$ | $83.80_{\pm1.57}$ | $93.11_{\pm1.30}$ | $93.18_{\pm1.39}$ |
| 1 | 12 | -0.90 ❄ | -0.70 ❄ | $84.56_{\pm1.58}$ | $85.14_{\pm2.61}$ | $83.39_{\pm1.79}$ | $83.74_{\pm1.57}$ | $92.90_{\pm1.45}$ | $93.07_{\pm1.46}$ |
| 1 | 12 | -0.90 🔥 | -0.70 🔥 | $84.51_{\pm2.21}$ | $84.74_{\pm2.93}$ | $83.60_{\pm2.04}$ | $83.82_{\pm2.14}$ | $92.89_{\pm1.70}$ | $93.04_{\pm1.69}$ |
| 1 | 12 | -0.90 ❄ | -0.60 ❄ | $84.65_{\pm2.23}$ | $85.00_{\pm2.71}$ | $83.52_{\pm2.45}$ | $83.86_{\pm2.34}$ | $93.03_{\pm1.75}$ | $93.21_{\pm1.71}$ |
| 1 | 12 | -0.90 🔥 | -0.60 🔥 | $85.00_{\pm1.91}$ | $85.23_{\pm2.24}$ | $83.81_{\pm2.17}$ | $84.22_{\pm2.04}$ | $93.19_{\pm1.58}$ | $93.34_{\pm1.56}$ |
| 1 | 12 | -0.90 ❄ | -0.50 ❄ | $85.51_{\pm1.90}$ | $85.74_{\pm1.89}$ | $84.36_{\pm2.36}$ | $84.74_{\pm2.14}$ | $93.45_{\pm1.53}$ | $93.61_{\pm1.50}$ |
| 1 | 12 | -0.90 🔥 | -0.50 🔥 | $85.13_{\pm1.85}$ | $85.42_{\pm1.88}$ | $83.84_{\pm2.27}$ | $84.31_{\pm2.07}$ | $93.20_{\pm1.51}$ | $93.33_{\pm1.42}$ |
| 1 | 12 | -0.90 ❄ | -0.40 ❄ | $85.13_{\pm2.39}$ | $85.22_{\pm2.55}$ | $83.93_{\pm2.63}$ | $84.36_{\pm2.56}$ | $93.20_{\pm1.56}$ | $93.33_{\pm1.46}$ |
| 1 | 12 | -0.90 🔥 | -0.40 🔥 | $85.55_{\pm2.00}$ | $85.49_{\pm2.36}$ | $84.69_{\pm2.00}$ | $84.92_{\pm2.02}$ | $93.40_{\pm1.36}$ | $93.52_{\pm1.28}$ |
| 1 | 12 | -0.90 ❄ | -0.30 ❄ | $84.57_{\pm1.39}$ | $85.00_{\pm2.08}$ | $83.07_{\pm1.20}$ | $83.67_{\pm1.35}$ | $93.11_{\pm1.20}$ | $93.28_{\pm1.14}$ |
| 1 | 12 | -0.90 🔥 | -0.30 🔥 | $83.90_{\pm1.33}$ | $85.12_{\pm1.39}$ | $81.87_{\pm1.84}$ | $82.67_{\pm1.66}$ | $92.70_{\pm0.87}$ | $92.85_{\pm0.85}$ |
| 1 | 12 | -0.90 ❄ | -0.20 ❄ | $84.49_{\pm1.17}$ | $84.66_{\pm1.76}$ | $83.14_{\pm0.85}$ | $83.66_{\pm1.08}$ | $92.85_{\pm0.96}$ | $93.03_{\pm0.90}$ |
| 1 | 12 | -0.90 🔥 | -0.20 🔥 | $84.72_{\pm1.32}$ | $84.68_{\pm1.71}$ | $83.55_{\pm1.11}$ | $83.97_{\pm1.29}$ | $93.13_{\pm1.09}$ | $93.25_{\pm1.04}$ |
| 1 | 12 | -0.90 ❄ | -0.10 ❄ | $83.91_{\pm1.50}$ | $84.79_{\pm1.54}$ | $82.10_{\pm1.97}$ | $82.79_{\pm1.81}$ | $92.70_{\pm1.23}$ | $92.86_{\pm1.19}$ |
| 1 | 12 | -0.90 🔥 | -0.10 🔥 | $84.15_{\pm1.58}$ | $84.99_{\pm1.58}$ | $82.26_{\pm1.89}$ | $83.03_{\pm1.82}$ | $92.89_{\pm1.28}$ | $93.00_{\pm1.24}$ |
| 1 | 12 | -0.90 ❄ | 0.10 ❄ | $84.04_{\pm1.85}$ | $84.19_{\pm1.99}$ | $82.61_{\pm1.99}$ | $83.14_{\pm1.99}$ | $92.08_{\pm1.52}$ | $92.21_{\pm1.55}$ |
| 1 | 12 | -0.90 🔥 | 0.10 🔥 | $83.41_{\pm2.31}$ | $83.47_{\pm2.29}$ | $81.95_{\pm2.59}$ | $82.47_{\pm2.51}$ | $91.81_{\pm1.88}$ | $91.93_{\pm1.93}$ |
| 1 | 12 | -0.90 ❄ | 0.20 ❄ | $81.77_{\pm2.66}$ | $81.92_{\pm2.74}$ | $80.48_{\pm2.58}$ | $80.81_{\pm2.71}$ | $90.51_{\pm2.00}$ | $90.50_{\pm2.17}$ |
| 1 | 12 | -0.90 🔥 | 0.20 🔥 | $80.66_{\pm1.85}$ | $80.78_{\pm1.60}$ | $79.51_{\pm2.69}$ | $79.64_{\pm2.31}$ | $89.94_{\pm1.79}$ | $89.97_{\pm1.88}$ |
| 1 | 12 | -0.90 ❄ | 0.30 ❄ | $80.22_{\pm2.29}$ | $80.42_{\pm2.57}$ | $78.60_{\pm2.23}$ | $79.04_{\pm2.34}$ | $89.09_{\pm2.03}$ | $89.06_{\pm2.27}$ |
| 1 | 12 | -0.90 🔥 | 0.30 🔥 | $80.24_{\pm2.39}$ | $80.57_{\pm2.60}$ | $78.68_{\pm2.22}$ | $79.07_{\pm2.39}$ | $89.24_{\pm2.15}$ | $89.19_{\pm2.41}$ |
| 1 | 12 | -0.90 ❄ | 0.40 ❄ | $78.23_{\pm3.06}$ | $78.77_{\pm2.11}$ | $76.39_{\pm4.18}$ | $76.58_{\pm4.21}$ | $87.60_{\pm2.29}$ | $87.70_{\pm2.39}$ |
| 1 | 12 | -0.90 🔥 | 0.40 🔥 | $78.21_{\pm3.00}$ | $78.94_{\pm2.08}$ | $76.13_{\pm4.20}$ | $76.41_{\pm4.14}$ | $87.42_{\pm2.65}$ | $87.53_{\pm2.74}$ |
| 1 | 12 | -0.90 ❄ | 0.50 ❄ | $77.44_{\pm2.18}$ | $77.78_{\pm2.37}$ | $75.35_{\pm2.23}$ | $75.82_{\pm2.33}$ | $86.03_{\pm1.63}$ | $86.14_{\pm1.89}$ |
| 1 | 12 | -0.90 🔥 | 0.50 🔥 | $76.73_{\pm2.99}$ | $77.57_{\pm2.51}$ | $74.30_{\pm3.78}$ | $74.64_{\pm4.06}$ | $85.69_{\pm2.08}$ | $85.84_{\pm2.31}$ |
| 1 | 12 | -0.90 ❄ | 0.60 ❄ | $76.27_{\pm2.64}$ | $76.83_{\pm2.90}$ | $74.00_{\pm2.56}$ | $74.42_{\pm2.76}$ | $84.85_{\pm1.71}$ | $84.99_{\pm1.96}$ |
| 1 | 12 | -0.90 🔥 | 0.60 🔥 | $75.74_{\pm2.75}$ | $76.63_{\pm2.59}$ | $73.23_{\pm3.24}$ | $73.55_{\pm3.53}$ | $84.34_{\pm1.82}$ | $84.46_{\pm2.11}$ |
| 1 | 12 | -0.90 ❄ | 0.70 ❄ | $75.22_{\pm2.41}$ | $75.97_{\pm2.68}$ | $72.67_{\pm2.36}$ | $73.07_{\pm2.61}$ | $83.65_{\pm1.55}$ | $83.67_{\pm1.87}$ |
| 1 | 12 | -0.90 🔥 | 0.70 🔥 | $74.03_{\pm1.66}$ | $74.65_{\pm2.27}$ | $71.90_{\pm1.85}$ | $72.03_{\pm1.81}$ | $83.04_{\pm0.82}$ | $82.99_{\pm1.05}$ |
| 1 | 12 | -0.90 ❄ | 0.80 ❄ | $73.00_{\pm1.76}$ | $73.29_{\pm2.36}$ | $70.68_{\pm1.86}$ | $70.87_{\pm1.92}$ | $81.77_{\pm0.94}$ | $81.75_{\pm1.20}$ |
| 1 | 12 | -0.90 🔥 | 0.80 🔥 | $73.11_{\pm1.55}$ | $73.26_{\pm2.04}$ | $70.86_{\pm1.66}$ | $71.07_{\pm1.69}$ | $81.76_{\pm0.93}$ | $81.69_{\pm1.23}$ |
| 1 | 12 | -0.90 ❄ | 0.90 ❄ | $73.24_{\pm1.86}$ | $73.84_{\pm2.05}$ | $70.25_{\pm2.31}$ | $70.57_{\pm2.50}$ | $81.18_{\pm1.01}$ | $81.02_{\pm1.42}$ |
| 1 | 12 | -0.90 🔥 | 0.90 🔥 | $72.12_{\pm1.04}$ | $72.84_{\pm2.07}$ | $69.16_{\pm1.45}$ | $69.33_{\pm1.54}$ | $80.58_{\pm0.57}$ | $80.50_{\pm0.91}$ |
| 1 | 12 | -0.90 ❄ | 1.00 ❄ | $72.33_{\pm1.92}$ | $73.19_{\pm1.90}$ | $68.99_{\pm2.68}$ | $69.16_{\pm2.95}$ | $79.75_{\pm1.64}$ | $79.69_{\pm1.81}$ |
| 1 | 12 | -0.90 🔥 | 1.00 🔥 | $71.73_{\pm2.72}$ | $72.00_{\pm3.35}$ | $68.88_{\pm2.88}$ | $69.10_{\pm3.16}$ | $79.28_{\pm2.16}$ | $79.35_{\pm2.25}$ |

| t | n | a | b | Accuracy | Precision | Recall | F1 score | AUROC | AUPRC |
|---|---|---|---|---|---|---|---|---|---|
| 1 | 12 | -0.80 ❄ | -1.00 ❄ | $84.43_{\pm1.79}$ | $84.12_{\pm2.00}$ | $83.83_{\pm1.94}$ | $83.85_{\pm1.83}$ | $92.42_{\pm1.46}$ | $92.51_{\pm1.44}$ |
| 1 | 12 | -0.80 🔥 | -1.00 🔥 | $84.28_{\pm1.52}$ | $83.87_{\pm1.75}$ | $83.75_{\pm1.53}$ | $83.73_{\pm1.52}$ | $92.46_{\pm1.43}$ | $92.51_{\pm1.39}$ |
| 1 | 12 | -0.80 ❄ | -0.90 ❄ | $83.47_{\pm1.22}$ | $83.33_{\pm1.42}$ | $83.08_{\pm0.90}$ | $82.91_{\pm1.05}$ | $92.20_{\pm0.98}$ | $92.28_{\pm1.00}$ |
| 1 | 12 | -0.80 🔥 | -0.90 🔥 | $83.87_{\pm1.72}$ | $83.65_{\pm2.13}$ | $83.45_{\pm1.51}$ | $83.34_{\pm1.64}$ | $92.49_{\pm1.40}$ | $92.54_{\pm1.40}$ |
| 1 | 12 | -0.80 ❄ | -0.80 ❄ | $84.07_{\pm1.47}$ | $83.83_{\pm1.68}$ | $83.54_{\pm1.65}$ | $83.49_{\pm1.51}$ | $92.49_{\pm1.19}$ | $92.61_{\pm1.26}$ |
| 1 | 12 | -0.80 🔥 | -0.80 🔥 | $84.08_{\pm1.57}$ | $84.21_{\pm2.13}$ | $83.25_{\pm1.69}$ | $83.38_{\pm1.56}$ | $92.63_{\pm1.37}$ | $92.71_{\pm1.43}$ |
| 1 | 12 | -0.80 ❄ | -0.70 ❄ | $84.35_{\pm1.59}$ | $84.69_{\pm2.35}$ | $83.18_{\pm1.55}$ | $83.54_{\pm1.54}$ | $92.81_{\pm1.58}$ | $92.87_{\pm1.68}$ |
| 1 | 12 | -0.80 🔥 | -0.70 🔥 | $84.22_{\pm2.03}$ | $84.39_{\pm2.67}$ | $83.07_{\pm2.07}$ | $83.43_{\pm2.04}$ | $92.61_{\pm1.85}$ | $92.76_{\pm1.81}$ |
| 1 | 12 | -0.80 ❄ | -0.60 ❄ | $84.91_{\pm2.08}$ | $85.43_{\pm2.01}$ | $83.42_{\pm2.56}$ | $83.98_{\pm2.35}$ | $92.77_{\pm1.68}$ | $92.97_{\pm1.59}$ |
| 1 | 12 | -0.80 🔥 | -0.60 🔥 | $84.65_{\pm1.82}$ | $84.98_{\pm2.11}$ | $83.30_{\pm2.03}$ | $83.80_{\pm1.96}$ | $92.74_{\pm1.63}$ | $92.90_{\pm1.58}$ |
| 1 | 12 | -0.80 ❄ | -0.50 ❄ | $84.33_{\pm2.12}$ | $85.09_{\pm1.94}$ | $82.86_{\pm2.92}$ | $83.33_{\pm2.52}$ | $92.82_{\pm1.78}$ | $92.99_{\pm1.71}$ |
| 1 | 12 | -0.80 🔥 | -0.50 🔥 | $84.57_{\pm1.91}$ | $85.18_{\pm1.95}$ | $83.12_{\pm2.52}$ | $83.62_{\pm2.23}$ | $93.02_{\pm1.69}$ | $93.19_{\pm1.63}$ |
| 1 | 12 | -0.80 ❄ | -0.40 ❄ | $84.30_{\pm1.52}$ | $85.33_{\pm2.20}$ | $82.42_{\pm1.70}$ | $83.20_{\pm1.64}$ | $92.90_{\pm1.53}$ | $93.01_{\pm1.44}$ |
| 1 | 12 | -0.80 🔥 | -0.40 🔥 | $84.05_{\pm1.41}$ | $84.94_{\pm2.21}$ | $82.42_{\pm1.82}$ | $83.01_{\pm1.57}$ | $92.92_{\pm1.57}$ | $93.05_{\pm1.46}$ |
| 1 | 12 | -0.80 ❄ | -0.30 ❄ | $84.23_{\pm1.66}$ | $85.00_{\pm1.84}$ | $82.54_{\pm1.99}$ | $83.19_{\pm1.89}$ | $92.87_{\pm1.22}$ | $93.03_{\pm1.15}$ |
| 1 | 12 | -0.80 🔥 | -0.30 🔥 | $84.30_{\pm1.38}$ | $85.64_{\pm1.53}$ | $82.33_{\pm2.16}$ | $83.10_{\pm1.72}$ | $93.14_{\pm1.40}$ | $93.27_{\pm1.32}$ |
| 1 | 12 | -0.80 ❄ | -0.20 ❄ | $84.46_{\pm1.62}$ | $85.23_{\pm1.53}$ | $82.63_{\pm1.94}$ | $83.39_{\pm1.87}$ | $92.85_{\pm1.15}$ | $92.98_{\pm1.07}$ |
| 1 | 12 | -0.80 🔥 | -0.20 🔥 | $84.26_{\pm1.58}$ | $85.56_{\pm1.83}$ | $82.18_{\pm1.88}$ | $83.05_{\pm1.82}$ | $92.96_{\pm1.30}$ | $93.11_{\pm1.25}$ |
| 1 | 12 | -0.80 ❄ | -0.10 ❄ | $83.69_{\pm0.93}$ | $84.36_{\pm1.98}$ | $81.99_{\pm0.32}$ | $82.65_{\pm0.67}$ | $92.33_{\pm1.16}$ | $92.44_{\pm1.28}$ |
| 1 | 12 | -0.80 🔥 | -0.10 🔥 | $83.42_{\pm1.18}$ | $84.76_{\pm1.61}$ | $81.21_{\pm1.44}$ | $82.10_{\pm1.39}$ | $92.74_{\pm0.96}$ | $92.85_{\pm0.96}$ |
| 1 | 12 | -0.80 ❄ | 0.10 ❄ | $81.87_{\pm1.80}$ | $82.34_{\pm2.45}$ | $80.32_{\pm1.48}$ | $80.80_{\pm1.69}$ | $90.48_{\pm2.20}$ | $90.59_{\pm2.21}$ |
| 1 | 12 | -0.80 🔥 | 0.10 🔥 | $82.39_{\pm1.66}$ | $82.48_{\pm1.35}$ | $80.87_{\pm2.25}$ | $81.35_{\pm1.99}$ | $90.82_{\pm1.77}$ | $90.87_{\pm1.86}$ |
| 1 | 12 | -0.80 ❄ | 0.20 ❄ | $81.33_{\pm2.44}$ | $81.99_{\pm2.04}$ | $79.28_{\pm3.06}$ | $79.92_{\pm3.00}$ | $90.08_{\pm1.91}$ | $90.21_{\pm1.93}$ |
| 1 | 12 | -0.80 🔥 | 0.20 🔥 | $81.08_{\pm2.25}$ | $81.50_{\pm2.14}$ | $79.16_{\pm2.70}$ | $79.76_{\pm2.64}$ | $89.78_{\pm1.94}$ | $89.90_{\pm1.97}$ |
| 1 | 12 | -0.80 ❄ | 0.30 ❄ | $80.32_{\pm1.87}$ | $80.76_{\pm2.18}$ | $78.45_{\pm1.95}$ | $79.00_{\pm1.98}$ | $88.72_{\pm1.82}$ | $88.61_{\pm2.11}$ |
| 1 | 12 | -0.80 🔥 | 0.30 🔥 | $79.55_{\pm1.02}$ | $79.72_{\pm1.23}$ | $78.22_{\pm1.77}$ | $78.42_{\pm1.32}$ | $88.34_{\pm1.65}$ | $88.22_{\pm1.85}$ |
| 1 | 12 | -0.80 ❄ | 0.40 ❄ | $77.65_{\pm3.37}$ | $78.93_{\pm1.57}$ | $75.15_{\pm4.83}$ | $75.37_{\pm5.30}$ | $86.89_{\pm1.51}$ | $87.04_{\pm1.63}$ |
| 1 | 12 | -0.80 🔥 | 0.40 🔥 | $78.07_{\pm2.81}$ | $79.33_{\pm1.61}$ | $75.53_{\pm4.00}$ | $75.93_{\pm4.22}$ | $87.00_{\pm1.53}$ | $87.14_{\pm1.61}$ |
| 1 | 12 | -0.80 ❄ | 0.50 ❄ | $77.51_{\pm2.91}$ | $78.32_{\pm3.86}$ | $75.04_{\pm2.90}$ | $75.65_{\pm3.12}$ | $86.13_{\pm3.08}$ | $86.21_{\pm3.24}$ |
| 1 | 12 | -0.80 🔥 | 0.50 🔥 | $77.33_{\pm2.74}$ | $77.67_{\pm2.78}$ | $75.09_{\pm3.15}$ | $75.59_{\pm3.17}$ | $85.76_{\pm2.27}$ | $85.88_{\pm2.52}$ |

## J.2. Generalizability of Hyperparameters

We transfer the key hyperparameters obtained from HM-BiTCN—such as the fusion coefficients $(a, b)$, and segment length $(n)$—to other baseline models. This experiment evaluates whether these hyperparameters retain effectiveness when applied to different architectures, indicating their robustness and broader applicability.

*Table 13.* **Results of Subject-Independent Setup.** Green cells indicate performance improvement with CIF.

| Datasets | Models | Accuracy | Precision | Recall | F1 score | AUROC | AUPRC |
|---|---|---|---|---|---|---|---|
| | Autoformer | $73.18_{\pm7.33}$ | $73.87_{\pm6.72}$ | $73.01_{\pm6.10}$ | $72.40_{\pm7.03}$ | $81.64_{\pm7.24}$ | $81.10_{\pm7.75}$ |
| | Autoformer + CIF | $75.96_{\pm2.68}$ | $76.33_{\pm3.47}$ | $75.11_{\pm1.10}$ | $74.97_{\pm1.90}$ | $83.42_{\pm1.99}$ | $83.15_{\pm2.63}$ |
| | Crossformer | $72.76_{\pm2.04}$ | $79.64_{\pm2.45}$ | $67.41_{\pm2.62}$ | $66.88_{\pm3.61}$ | $71.81_{\pm4.06}$ | $71.64_{\pm3.74}$ |
| | Crossformer + CIF | $82.32_{\pm2.60}$ | $85.35_{\pm1.83}$ | $79.21_{\pm3.17}$ | $80.29_{\pm3.30}$ | $90.39_{\pm1.58}$ | $90.02_{\pm1.47}$ |
| | FEDformer | $75.16_{\pm1.67}$ | $74.98_{\pm0.69}$ | $73.34_{\pm2.97}$ | $73.50_{\pm2.90}$ | $83.89_{\pm1.54}$ | $83.27_{\pm1.62}$ |
| | FEDformer + CIF | $77.20_{\pm2.17}$ | $76.97_{\pm2.07}$ | $77.02_{\pm2.60}$ | $76.55_{\pm2.40}$ | $86.70_{\pm1.73}$ | $86.53_{\pm1.76}$ |
| | Informer | $72.20_{\pm2.78}$ | $73.92_{\pm4.80}$ | $68.48_{\pm2.51}$ | $68.74_{\pm2.70}$ | $70.14_{\pm3.43}$ | $70.84_{\pm3.80}$ |
| | Informer + CIF | $79.78_{\pm2.07}$ | $82.29_{\pm3.03}$ | $76.55_{\pm2.05}$ | $77.53_{\pm2.23}$ | $78.56_{\pm1.33}$ | $78.58_{\pm1.14}$ |
| | iTransformer | $74.55_{\pm1.66}$ | $74.77_{\pm2.10}$ | $71.76_{\pm1.72}$ | $72.30_{\pm1.79}$ | $85.59_{\pm1.55}$ | $84.39_{\pm1.57}$ |
| | iTransformer + CIF | $74.95_{\pm0.87}$ | $74.40_{\pm0.75}$ | $73.79_{\pm1.78}$ | $73.81_{\pm1.43}$ | $84.30_{\pm0.88}$ | $82.49_{\pm0.96}$ |
| | MTST | $69.24_{\pm1.24}$ | $75.87_{\pm2.80}$ | $63.28_{\pm1.81}$ | $61.62_{\pm2.75}$ | $66.09_{\pm3.27}$ | $68.08_{\pm2.93}$ |
| **APAVA** | MTST + CIF | $76.20_{\pm2.39}$ | $81.46_{\pm1.11}$ | $71.67_{\pm3.06}$ | $72.06_{\pm3.73}$ | $77.65_{\pm3.37}$ | $77.98_{\pm2.73}$ |
| (2-Classes) | Nonformer | $71.81_{\pm4.20}$ | $71.31_{\pm4.40}$ | $70.15_{\pm3.38}$ | $70.38_{\pm3.74}$ | $71.54_{\pm2.73}$ | $72.79_{\pm2.50}$ |
| Reproduced | Nonformer + CIF | $71.89_{\pm3.81}$ | $71.80_{\pm4.58}$ | $69.44_{\pm3.56}$ | $69.74_{\pm3.84}$ | $70.55_{\pm2.96}$ | $70.78_{\pm4.08}$ |
| | Reformer | $78.42_{\pm2.85}$ | $80.89_{\pm4.52}$ | $75.20_{\pm2.28}$ | $76.09_{\pm2.54}$ | $75.48_{\pm2.79}$ | $77.52_{\pm2.64}$ |
| | Reformer + CIF | $81.51_{\pm0.57}$ | $\mathbf{84.90_{\pm0.86}}$ | $78.18_{\pm0.60}$ | $79.32_{\pm0.64}$ | $79.10_{\pm2.50}$ | $80.77_{\pm2.21}$ |
| | Transformer | $75.53_{\pm4.28}$ | $76.90_{\pm5.05}$ | $72.14_{\pm4.87}$ | $72.64_{\pm5.44}$ | $72.30_{\pm6.04}$ | $73.04_{\pm7.15}$ |
| | Transformer + CIF | $77.96_{\pm2.82}$ | $79.34_{\pm3.86}$ | $75.07_{\pm2.52}$ | $75.87_{\pm2.73}$ | $74.75_{\pm1.62}$ | $74.76_{\pm2.33}$ |
| | Medformer | $77.85_{\pm2.42}$ | $80.31_{\pm3.21}$ | $74.38_{\pm2.49}$ | $75.21_{\pm2.67}$ | $80.85_{\pm3.80}$ | $81.62_{\pm3.24}$ |
| | Medformer + CIF | $81.06_{\pm1.58}$ | $82.97_{\pm2.23}$ | $78.26_{\pm1.52}$ | $79.23_{\pm1.65}$ | $85.74_{\pm1.85}$ | $86.32_{\pm1.48}$ |
| | MedGNN | $77.40_{\pm5.77}$ | $82.77_{\pm4.46}$ | $73.24_{\pm7.06}$ | $73.29_{\pm9.01}$ | $81.31_{\pm2.94}$ | $82.80_{\pm2.91}$ |
| | MedGNN + CIF | $81.02_{\pm1.51}$ | $84.21_{\pm2.62}$ | $77.76_{\pm1.52}$ | $78.83_{\pm1.64}$ | $86.27_{\pm2.79}$ | $87.45_{\pm2.40}$ |
| | HM-BiTCN | $82.49_{\pm1.40}$ | $82.38_{\pm1.79}$ | $81.20_{\pm1.32}$ | $81.60_{\pm1.39}$ | $91.10_{\pm1.63}$ | $91.30_{\pm1.71}$ |
| | HM-BiTCN + CIF | $\mathbf{85.16_{\pm1.55}}$ | $84.76_{\pm1.62}$ | $\mathbf{85.33_{\pm1.27}}$ | $\mathbf{84.82_{\pm1.49}}$ | $\mathbf{94.06_{\pm1.07}}$ | $\mathbf{94.21_{\pm0.99}}$ |

*Table 14.* **Results of Subject-Independent Setup.** Green cells indicate performance improvement with CIF.

| Datasets | Models | Accuracy | Precision | Recall | F1 score | AUROC | AUPRC |
|---|---|---|---|---|---|---|---|
| **TDBrain** (2-Classes) Reproduced | Autoformer | 90.38$_{\pm3.03}$ | 91.16$_{\pm2.42}$ | 90.38$_{\pm3.03}$ | 90.31$_{\pm3.09}$ | 95.83$_{\pm2.14}$ | 95.43$_{\pm2.31}$ |
| | Autoformer + CIF | **93.42$_{\pm2.49}$** | **93.71$_{\pm2.27}$** | **93.42$_{\pm2.49}$** | **93.40$_{\pm2.51}$** | 97.46$_{\pm1.08}$ | 97.21$_{\pm1.15}$ |
| | Crossformer | 82.15$_{\pm2.60}$ | 82.81$_{\pm2.11}$ | 82.15$_{\pm2.60}$ | 82.04$_{\pm2.70}$ | 91.20$_{\pm2.23}$ | 91.47$_{\pm2.16}$ |
| | Crossformer + CIF | 89.40$_{\pm1.26}$ | 89.83$_{\pm1.18}$ | 89.40$_{\pm1.26}$ | 89.37$_{\pm1.27}$ | 96.76$_{\pm0.62}$ | 96.80$_{\pm0.62}$ |
| | FEDformer | 77.60$_{\pm1.23}$ | 78.25$_{\pm1.52}$ | 77.60$_{\pm1.23}$ | 77.48$_{\pm1.19}$ | 86.31$_{\pm1.23}$ | 86.48$_{\pm1.36}$ |
| | FEDformer + CIF | 78.87$_{\pm1.94}$ | 79.19$_{\pm2.00}$ | 78.88$_{\pm1.94}$ | 78.82$_{\pm1.94}$ | 88.12$_{\pm1.87}$ | 88.39$_{\pm1.92}$ |
| | Informer | 88.42$_{\pm2.99}$ | 89.01$_{\pm2.45}$ | 88.42$_{\pm2.99}$ | 88.36$_{\pm3.05}$ | 96.54$_{\pm0.90}$ | 96.66$_{\pm0.85}$ |
| | Informer + CIF | 89.38$_{\pm1.66}$ | 89.71$_{\pm1.37}$ | 89.38$_{\pm1.66}$ | 89.35$_{\pm1.68}$ | 96.92$_{\pm0.66}$ | 97.03$_{\pm0.64}$ |
| | iTransformer | 74.69$_{\pm1.02}$ | 74.76$_{\pm1.04}$ | 74.69$_{\pm1.02}$ | 74.67$_{\pm1.02}$ | 83.35$_{\pm1.24}$ | 83.65$_{\pm1.41}$ |
| | iTransformer + CIF | 72.79$_{\pm1.12}$ | 72.89$_{\pm1.16}$ | 72.79$_{\pm1.12}$ | 72.76$_{\pm1.11}$ | 81.08$_{\pm1.03}$ | 81.31$_{\pm0.93}$ |
| | MTST | 77.67$_{\pm3.58}$ | 78.97$_{\pm4.37}$ | 77.67$_{\pm3.58}$ | 77.45$_{\pm3.55}$ | 86.47$_{\pm4.84}$ | 84.99$_{\pm6.43}$ |
| | MTST + CIF | 87.21$_{\pm1.99}$ | 87.30$_{\pm2.06}$ | 87.21$_{\pm1.99}$ | 87.20$_{\pm1.99}$ | 94.14$_{\pm1.83}$ | 93.24$_{\pm2.63}$ |
| | Nonformer | 88.10$_{\pm2.39}$ | 88.76$_{\pm1.74}$ | 88.10$_{\pm2.39}$ | 88.04$_{\pm2.47}$ | 96.56$_{\pm0.91}$ | 96.36$_{\pm1.21}$ |
| | Nonformer + CIF | 88.00$_{\pm2.06}$ | 88.91$_{\pm1.68}$ | 88.00$_{\pm2.06}$ | 87.92$_{\pm2.12}$ | 97.10$_{\pm1.10}$ | 97.17$_{\pm1.09}$ |
| | PatchTST | 77.98$_{\pm2.64}$ | 79.30$_{\pm3.73}$ | 77.98$_{\pm2.64}$ | 77.76$_{\pm2.65}$ | 86.67$_{\pm4.03}$ | 84.93$_{\pm5.47}$ |
| | PatchTST + CIF | 79.58$_{\pm0.86}$ | 80.22$_{\pm0.82}$ | 79.58$_{\pm0.86}$ | 79.47$_{\pm0.88}$ | 87.20$_{\pm1.57}$ | 85.81$_{\pm1.17}$ |
| | Reformer | 88.50$_{\pm2.30}$ | 89.01$_{\pm1.80}$ | 88.50$_{\pm2.30}$ | 88.45$_{\pm2.35}$ | 96.10$_{\pm0.63}$ | 96.19$_{\pm0.55}$ |
| | Reformer + CIF | 89.08$_{\pm1.19}$ | 89.47$_{\pm0.70}$ | 89.08$_{\pm1.19}$ | 89.05$_{\pm1.23}$ | 96.88$_{\pm0.35}$ | 97.00$_{\pm0.35}$ |
| | Transformer | 85.13$_{\pm1.86}$ | 86.39$_{\pm1.56}$ | 85.13$_{\pm1.86}$ | 84.99$_{\pm1.93}$ | 95.61$_{\pm1.05}$ | 95.63$_{\pm0.91}$ |
| | Transformer + CIF | 89.96$_{\pm1.57}$ | 90.53$_{\pm1.26}$ | 89.96$_{\pm1.57}$ | 89.92$_{\pm1.60}$ | 97.73$_{\pm0.45}$ | 97.77$_{\pm0.46}$ |
| | Medformer | 88.77$_{\pm1.24}$ | 88.91$_{\pm1.11}$ | 88.77$_{\pm1.24}$ | 88.76$_{\pm1.25}$ | 96.38$_{\pm0.34}$ | 96.44$_{\pm0.30}$ |
| | Medformer + CIF | 90.88$_{\pm0.87}$ | 90.94$_{\pm0.81}$ | 90.88$_{\pm0.87}$ | 90.87$_{\pm0.87}$ | 97.39$_{\pm0.30}$ | 97.46$_{\pm0.28}$ |
| | HM-BiTCN | 84.90$_{\pm2.60}$ | 86.02$_{\pm2.00}$ | 84.90$_{\pm2.60}$ | 84.76$_{\pm2.74}$ | 93.94$_{\pm1.92}$ | 94.20$_{\pm1.85}$ |
| | HM-BiTCN + CIF | 93.13$_{\pm1.41}$ | 93.33$_{\pm1.37}$ | 93.13$_{\pm1.41}$ | 93.12$_{\pm1.42}$ | **98.62$_{\pm0.66}$** | **98.68$_{\pm0.63}$** |

*Table 15.* **Results of Subject-Independent Setup.** Green cells indicate performance improvement with CIF.

| Datasets | Models | Accuracy | Precision | Recall | F1 score | AUROC | AUPRC |
|---|---|---|---|---|---|---|---|
| **ADFTD** (3-Classes) Reproduced | Autoformer | $46.90_{\pm2.89}$ | $45.59_{\pm2.37}$ | $44.91_{\pm2.23}$ | $44.34_{\pm2.52}$ | $63.49_{\pm2.44}$ | $45.63_{\pm2.29}$ |
| | Autoformer + CIF | $45.92_{\pm2.78}$ | $44.81_{\pm2.44}$ | $44.66_{\pm2.25}$ | $44.26_{\pm2.38}$ | $62.69_{\pm2.21}$ | $44.93_{\pm2.68}$ |
| | Crossformer | $50.18_{\pm1.97}$ | $45.97_{\pm1.84}$ | $46.30_{\pm1.73}$ | $45.90_{\pm1.84}$ | $66.68_{\pm1.67}$ | $48.65_{\pm1.89}$ |
| | Crossformer + CIF | $54.58_{\pm1.22}$ | $47.85_{\pm0.67}$ | $48.96_{\pm0.89}$ | $48.22_{\pm0.72}$ | $69.10_{\pm1.05}$ | $52.23_{\pm0.98}$ |
| | FEDformer | $45.75_{\pm0.78}$ | $45.71_{\pm1.29}$ | $44.27_{\pm1.28}$ | $43.51_{\pm1.00}$ | $62.64_{\pm1.64}$ | $45.88_{\pm1.35}$ |
| | FEDformer + CIF | $48.63_{\pm1.99}$ | $46.97_{\pm1.58}$ | $46.87_{\pm1.63}$ | $46.69_{\pm1.54}$ | $65.56_{\pm2.14}$ | $48.30_{\pm2.30}$ |
| | Informer | $48.42_{\pm1.99}$ | $46.94_{\pm1.60}$ | $46.41_{\pm0.99}$ | $45.76_{\pm0.43}$ | $65.99_{\pm1.14}$ | $47.49_{\pm1.07}$ |
| | Informer + CIF | $50.12_{\pm0.66}$ | $47.23_{\pm0.47}$ | $46.77_{\pm0.39}$ | $46.62_{\pm0.38}$ | $65.13_{\pm0.52}$ | $46.84_{\pm0.54}$ |
| | iTransformer | $52.85_{\pm1.36}$ | $46.97_{\pm1.05}$ | $47.31_{\pm1.03}$ | $46.84_{\pm0.78}$ | $67.46_{\pm0.96}$ | $49.90_{\pm0.89}$ |
| | iTransformer + CIF | $50.76_{\pm0.50}$ | $47.11_{\pm0.67}$ | $47.29_{\pm0.59}$ | $47.10_{\pm0.66}$ | $67.00_{\pm0.55}$ | $49.60_{\pm0.69}$ |
| | MTST | $45.77_{\pm1.70}$ | $44.39_{\pm1.73}$ | $43.70_{\pm1.82}$ | $43.36_{\pm1.98}$ | $61.38_{\pm1.57}$ | $44.01_{\pm1.60}$ |
| | MTST + CIF | $46.36_{\pm0.93}$ | $45.04_{\pm0.65}$ | $45.47_{\pm0.88}$ | $44.87_{\pm0.81}$ | $63.57_{\pm1.24}$ | $47.35_{\pm1.36}$ |
| | Nonformer | $50.81_{\pm1.06}$ | $48.71_{\pm1.40}$ | $48.55_{\pm1.47}$ | $48.36_{\pm1.38}$ | $66.95_{\pm1.54}$ | $48.08_{\pm1.82}$ |
| | Nonformer + CIF | $51.81_{\pm2.11}$ | $49.93_{\pm0.81}$ | $49.66_{\pm0.36}$ | $49.10_{\pm0.57}$ | $68.59_{\pm0.60}$ | $50.44_{\pm0.94}$ |
| | Reformer | $51.28_{\pm2.60}$ | $49.68_{\pm2.75}$ | $49.64_{\pm2.02}$ | $48.45_{\pm2.06}$ | $69.20_{\pm2.53}$ | $51.74_{\pm3.24}$ |
| | Reformer + CIF | $52.50_{\pm0.91}$ | $52.10_{\pm2.91}$ | $49.01_{\pm2.74}$ | $47.53_{\pm4.23}$ | $68.25_{\pm1.79}$ | $50.44_{\pm2.59}$ |
| | Transformer | $50.53_{\pm0.94}$ | $49.31_{\pm0.87}$ | $48.57_{\pm1.23}$ | $48.42_{\pm1.28}$ | $67.98_{\pm0.90}$ | $49.07_{\pm1.35}$ |
| | Transformer + CIF | $52.84_{\pm2.50}$ | $51.27_{\pm2.33}$ | $51.53_{\pm2.37}$ | $51.10_{\pm2.21}$ | $70.25_{\pm2.42}$ | $52.35_{\pm2.94}$ |
| | Medformer | $53.70_{\pm1.18}$ | $51.51_{\pm1.32}$ | $50.49_{\pm1.48}$ | $50.35_{\pm1.53}$ | $70.48_{\pm1.17}$ | $50.91_{\pm1.13}$ |
| | Medformer + CIF | $55.88_{\pm0.82}$ | $51.91_{\pm1.90}$ | $50.80_{\pm1.63}$ | $50.29_{\pm1.92}$ | $70.45_{\pm1.33}$ | $53.73_{\pm1.70}$ |
| | MedGNN | $50.22_{\pm3.21}$ | $48.65_{\pm3.72}$ | $47.50_{\pm4.57}$ | $47.33_{\pm4.40}$ | $67.18_{\pm4.39}$ | $48.84_{\pm4.11}$ |
| | MedGNN + CIF | $54.89_{\pm1.23}$ | $51.57_{\pm1.57}$ | $51.46_{\pm1.68}$ | $50.85_{\pm2.05}$ | $71.98_{\pm1.59}$ | $53.89_{\pm1.73}$ |
| | HM-BiTCN | $52.05_{\pm2.22}$ | $50.45_{\pm3.00}$ | $50.40_{\pm2.55}$ | $49.48_{\pm2.70}$ | $69.43_{\pm2.84}$ | $50.99_{\pm3.15}$ |
| | HM-BiTCN + CIF | $\mathbf{58.56_{\pm0.93}}$ | $\mathbf{55.65_{\pm0.81}}$ | $\mathbf{55.86_{\pm0.79}}$ | $\mathbf{55.42_{\pm0.82}}$ | $\mathbf{76.07_{\pm0.59}}$ | $\mathbf{59.75_{\pm0.67}}$ |

During the process of transferring the optimized CIF hyperparameter configurations from the HM-BiTCN architecture to other model structures, we observed significant improvements in performance across most models and evaluation metrics. This result strongly demonstrates that the CIF module not only exhibits high generalizability but also possesses well-transferable hyperparameters that can be effectively adapted to various architectures. These hyperparameters enhance feature extraction and representation capabilities across different model designs. This finding further highlights the potential of the CIF architecture in diverse tasks and provides valuable insights for future model design and parameter sharing.

*Table 16.* **Results of Subject-Independent Setup.** Green cells indicate performance improvement with CIF.

| Datasets | Models | Accuracy | Precision | Recall | F1 score | AUROC | AUPRC |
|---|---|---|---|---|---|---|---|
| | **Autoformer** | $71.99_{\pm2.74}$ | $69.60_{\pm3.85}$ | $61.50_{\pm4.23}$ | $61.43_{\pm5.07}$ | $74.29_{\pm1.89}$ | $70.26_{\pm2.00}$ |
| | **Autoformer + CIF** | $77.71_{\pm0.63}$ | $77.15_{\pm1.29}$ | $70.32_{\pm1.62}$ | $71.77_{\pm1.60}$ | $81.20_{\pm4.15}$ | $78.13_{\pm4.80}$ |
| | **Crossformer** | $78.06_{\pm3.44}$ | $81.53_{\pm3.13}$ | $68.62_{\pm5.63}$ | $69.76_{\pm6.53}$ | $88.31_{\pm2.07}$ | $85.81_{\pm2.43}$ |
| | **Crossformer + CIF** | $84.55_{\pm4.80}$ | $85.96_{\pm3.63}$ | $78.64_{\pm7.49}$ | $80.25_{\pm7.89}$ | $92.17_{\pm2.99}$ | $91.16_{\pm3.70}$ |
| | **FEDformer** | $74.54_{\pm2.27}$ | $77.99_{\pm4.10}$ | $63.14_{\pm3.29}$ | $63.28_{\pm4.36}$ | $84.63_{\pm4.27}$ | $80.91_{\pm5.55}$ |
| | **FEDformer + CIF** | $77.93_{\pm1.86}$ | $80.65_{\pm3.50}$ | $68.59_{\pm2.20}$ | $70.23_{\pm2.63}$ | $86.00_{\pm3.51}$ | $83.42_{\pm4.76}$ |
| | **Informer** | $79.59_{\pm0.65}$ | $83.33_{\pm0.77}$ | $70.58_{\pm0.95}$ | $72.58_{\pm1.08}$ | $92.77_{\pm0.48}$ | $90.89_{\pm0.57}$ |
| | **Informer + CIF** | $83.63_{\pm2.32}$ | $85.31_{\pm1.80}$ | $77.28_{\pm3.67}$ | $79.37_{\pm3.45}$ | $93.87_{\pm0.48}$ | $92.29_{\pm0.63}$ |
| | **iTransformer** | $83.43_{\pm1.19}$ | $88.06_{\pm1.47}$ | $75.64_{\pm1.55}$ | $78.29_{\pm1.70}$ | $91.38_{\pm1.41}$ | $91.08_{\pm1.30}$ |
| | **iTransformer + CIF** | $81.62_{\pm2.96}$ | $87.48_{\pm2.03}$ | $72.74_{\pm4.41}$ | $74.99_{\pm4.91}$ | $90.25_{\pm3.29}$ | $89.80_{\pm3.28}$ |
| | **MTST** | $75.53_{\pm2.45}$ | $78.72_{\pm1.87}$ | $64.78_{\pm4.06}$ | $65.30_{\pm4.81}$ | $87.76_{\pm4.09}$ | $83.60_{\pm3.92}$ |
| | **MTST + CIF** | $81.80_{\pm1.60}$ | $84.19_{\pm2.05}$ | $74.28_{\pm2.05}$ | $76.52_{\pm2.25}$ | $91.30_{\pm2.48}$ | $88.41_{\pm3.30}$ |
| **PTB** | **Nonformer** | $78.93_{\pm1.46}$ | $82.48_{\pm1.53}$ | $69.68_{\pm2.15}$ | $71.50_{\pm2.56}$ | $90.54_{\pm0.59}$ | $87.78_{\pm1.46}$ |
| **(2-Classes)** | **Nonformer + CIF** | $71.89_{\pm3.81}$ | $71.80_{\pm4.58}$ | $69.44_{\pm3.56}$ | $69.74_{\pm3.84}$ | $70.55_{\pm2.96}$ | $70.78_{\pm4.08}$ |
| | **PatchTST** | $75.28_{\pm2.44}$ | $77.05_{\pm2.44}$ | $64.86_{\pm4.05}$ | $65.41_{\pm5.28}$ | $88.11_{\pm2.59}$ | $82.65_{\pm2.87}$ |
| | **PatchTST + CIF** | $80.80_{\pm1.90}$ | $82.46_{\pm1.75}$ | $73.23_{\pm2.81}$ | $75.24_{\pm2.95}$ | $92.45_{\pm0.91}$ | $88.23_{\pm1.61}$ |
| | **Reformer** | $78.11_{\pm1.65}$ | $82.70_{\pm0.80}$ | $68.17_{\pm2.68}$ | $69.68_{\pm3.34}$ | $90.77_{\pm1.56}$ | $88.14_{\pm1.20}$ |
| | **Reformer + CIF** | $81.95_{\pm2.09}$ | $84.63_{\pm1.07}$ | $74.44_{\pm3.53}$ | $76.55_{\pm3.64}$ | $93.36_{\pm0.87}$ | $91.53_{\pm1.05}$ |
| | **Transformer** | $76.43_{\pm1.98}$ | $81.25_{\pm1.15}$ | $65.64_{\pm3.29}$ | $66.44_{\pm4.39}$ | $90.21_{\pm1.24}$ | $87.28_{\pm1.49}$ |
| | **Transformer + CIF** | $78.80_{\pm2.44}$ | $82.45_{\pm2.36}$ | $69.42_{\pm3.62}$ | $71.10_{\pm4.33}$ | $92.50_{\pm1.17}$ | $89.07_{\pm1.39}$ |
| | **Medformer** | $80.99_{\pm0.75}$ | $83.01_{\pm0.72}$ | $73.35_{\pm1.16}$ | $75.47_{\pm1.21}$ | $93.10_{\pm1.18}$ | $90.69_{\pm1.04}$ |
| | **Medformer + CIF** | $78.74_{\pm0.64}$ | $81.11_{\pm0.84}$ | $75.40_{\pm0.66}$ | $76.31_{\pm0.71}$ | $83.20_{\pm0.91}$ | $83.66_{\pm0.92}$ |
| | **HM-BiTCN** | $81.87_{\pm1.87}$ | $86.50_{\pm1.24}$ | $73.49_{\pm2.90}$ | $75.84_{\pm3.20}$ | $94.20_{\pm0.29}$ | $93.04_{\pm0.45}$ |
| | **HM-BiTCN + CIF** | $88.29_{\pm1.45}$ | $90.66_{\pm1.48}$ | $83.21_{\pm2.02}$ | $85.59_{\pm1.96}$ | $94.28_{\pm0.93}$ | $93.78_{\pm1.11}$ |

## J.3. Fine-tuning Transferred Parameters

For models where the transferred hyperparameters from HM-BiTCN do not yield optimal results, we perform additional fine-tuning. This step investigates the adaptability of CIF-related parameters and explores how they can be optimized for other model structures.

*Table 17.* **Results of Subject-Independent Setup.** Green cells indicate performance improvement with CIF.

| Datasets | Models | Accuracy | Precision | Recall | F1 score | AUROC | AUPRC |
|---|---|---|---|---|---|---|---|
| **TDBrain** (2-Classes) Reproduced | iTransformer | $74.69_{\pm1.02}$ | $74.76_{\pm1.04}$ | $74.69_{\pm1.02}$ | $74.67_{\pm1.02}$ | $83.35_{\pm1.24}$ | $83.65_{\pm1.41}$ |
| | iTransformer + CIF(HM-BiTCN) | $72.79_{\pm1.12}$ | $72.89_{\pm1.16}$ | $72.79_{\pm1.12}$ | $72.76_{\pm1.11}$ | $81.08_{\pm1.03}$ | $81.31_{\pm0.93}$ |
| | iTransformer + CIF(New) | $76.10_{\pm0.76}$ | $76.16_{\pm0.75}$ | $76.10_{\pm0.76}$ | $76.09_{\pm0.76}$ | $84.79_{\pm1.15}$ | $85.13_{\pm1.11}$ |
| | Nonformer | $88.10_{\pm2.39}$ | $88.76_{\pm1.74}$ | $88.10_{\pm2.39}$ | $88.04_{\pm2.47}$ | $96.56_{\pm0.91}$ | $96.36_{\pm1.21}$ |
| | Nonformer + CIF(HM-BiTCN) | $88.00_{\pm2.06}$ | $88.91_{\pm1.68}$ | $88.00_{\pm2.06}$ | $87.92_{\pm2.12}$ | $97.10_{\pm1.10}$ | $97.17_{\pm1.09}$ |
| | Nonformer + CIF (New) | $88.67_{\pm1.30}$ | $89.34_{\pm0.93}$ | $88.67_{\pm1.30}$ | $88.61_{\pm1.33}$ | $96.89_{\pm0.31}$ | $96.91_{\pm0.37}$ |

*Table 18.* **Results of Subject-Independent Setup.** Green cells indicate performance improvement with CIF.

| Datasets | Models | Accuracy | Precision | Recall | F1 score | AUROC | AUPRC |
|---|---|---|---|---|---|---|---|
| **ADFTD** (3-Classes) Reproduced | Autoformer | $46.90_{\pm2.89}$ | $45.59_{\pm2.37}$ | $44.91_{\pm2.23}$ | $44.34_{\pm2.52}$ | $63.49_{\pm2.44}$ | $45.63_{\pm2.29}$ |
| | Autoformer + CIF((HM-BiTCN)) | $45.92_{\pm2.78}$ | $44.81_{\pm2.44}$ | $44.66_{\pm2.25}$ | $44.26_{\pm2.38}$ | $62.69_{\pm2.21}$ | $44.93_{\pm2.68}$ |
| | Autoformer + CIF(New) | $48.49_{\pm1.31}$ | $46.42_{\pm1.07}$ | $45.39_{\pm1.80}$ | $44.94_{\pm1.97}$ | $63.75_{\pm1.46}$ | $46.10_{\pm1.84}$ |
| | iTransformer | $52.85_{\pm1.36}$ | $46.97_{\pm1.05}$ | $47.31_{\pm1.03}$ | $46.84_{\pm0.78}$ | $67.46_{\pm0.96}$ | $49.90_{\pm0.89}$ |
| | iTransformer + CIF(HM-BiTCN) | $50.76_{\pm0.50}$ | $47.11_{\pm0.67}$ | $47.29_{\pm0.59}$ | $47.10_{\pm0.66}$ | $67.00_{\pm0.55}$ | $49.60_{\pm0.69}$ |
| | iTransformer + CIF(New) | $53.93_{\pm1.52}$ | $48.44_{\pm0.89}$ | $49.13_{\pm1.17}$ | $48.56_{\pm0.84}$ | $68.70_{\pm0.94}$ | $50.88_{\pm1.20}$ |
| | Reformer | $51.28_{\pm2.60}$ | $49.68_{\pm2.75}$ | $49.64_{\pm2.02}$ | $48.45_{\pm2.06}$ | $69.20_{\pm2.53}$ | $51.74_{\pm3.24}$ |
| | Reformer + CIF(HM-BiTCN) | $52.50_{\pm0.91}$ | $52.10_{\pm2.91}$ | $49.01_{\pm2.74}$ | $47.53_{\pm4.23}$ | $68.25_{\pm1.79}$ | $50.44_{\pm2.59}$ |
| | Reformer + CIF(New) | $53.11_{\pm1.71}$ | $52.14_{\pm1.42}$ | $52.42_{\pm1.90}$ | $51.73_{\pm2.01}$ | $70.47_{\pm1.64}$ | $51.88_{\pm1.92}$ |
| | Medformer | $53.70_{\pm1.18}$ | $51.51_{\pm1.32}$ | $50.49_{\pm1.48}$ | $50.35_{\pm1.53}$ | $70.48_{\pm1.17}$ | $50.91_{\pm1.13}$ |
| | Medformer + CIF(HM-BiTCN) | $55.88_{\pm0.82}$ | $51.91_{\pm1.90}$ | $50.80_{\pm1.63}$ | $50.29_{\pm1.92}$ | $70.45_{\pm1.33}$ | $53.73_{\pm1.70}$ |
| | Medformer + CIF(New) | $55.21_{\pm1.42}$ | $52.84_{\pm2.36}$ | $52.60_{\pm2.26}$ | $52.43_{\pm2.65}$ | $72.11_{\pm2.42}$ | $54.10_{\pm3.09}$ |

*Table 19.* **Results of Subject-Independent Setup.** Green cells indicate performance improvement with CIF.

| Datasets | Models | Accuracy | Precision | Recall | F1 score | AUROC | AUPRC |
|---|---|---|---|---|---|---|---|
| **PTB** (2-Classes) Reproduced | iTransformer | $83.43_{\pm1.19}$ | $88.06_{\pm1.47}$ | $75.64_{\pm1.55}$ | $78.29_{\pm1.70}$ | $91.38_{\pm1.41}$ | $91.08_{\pm1.30}$ |
| | iTransformer + CIF(HM-BiTCN) | $81.62_{\pm2.96}$ | $87.48_{\pm2.03}$ | $72.74_{\pm4.41}$ | $74.99_{\pm4.91}$ | $90.25_{\pm3.29}$ | $89.80_{\pm3.28}$ |
| | iTransformer + CIF(New) | $83.56_{\pm1.15}$ | $86.00_{\pm1.03}$ | $76.78_{\pm1.94}$ | $79.10_{\pm1.87}$ | $90.40_{\pm0.75}$ | $89.80_{\pm0.93}$ |
| | Nonformer | $78.93_{\pm1.46}$ | $82.48_{\pm1.53}$ | $69.68_{\pm2.15}$ | $71.50_{\pm2.56}$ | $90.54_{\pm0.59}$ | $87.78_{\pm1.46}$ |
| | Nonformer + CIF(HM-BiTCN) | $71.89_{\pm3.81}$ | $71.80_{\pm4.58}$ | $69.44_{\pm3.56}$ | $69.74_{\pm3.84}$ | $70.55_{\pm2.96}$ | $70.78_{\pm4.08}$ |
| | Nonformer + CIF(New) | $80.46_{\pm2.20}$ | $84.22_{\pm2.44}$ | $71.79_{\pm3.10}$ | $73.89_{\pm3.50}$ | $92.74_{\pm0.97}$ | $90.30_{\pm1.26}$ |
| | Medformer | $80.99_{\pm0.75}$ | $83.01_{\pm0.72}$ | $73.35_{\pm1.16}$ | $75.47_{\pm1.21}$ | $93.10_{\pm1.18}$ | $90.69_{\pm1.04}$ |
| | Medformer + CIF(HM-BiTCN) | $78.74_{\pm0.64}$ | $81.11_{\pm0.84}$ | $75.40_{\pm0.66}$ | $76.31_{\pm0.71}$ | $83.20_{\pm0.91}$ | $83.66_{\pm0.92}$ |
| | Medformer + CIF(New) | $83.98_{\pm0.81}$ | $85.21_{\pm0.62}$ | $78.00_{\pm1.30}$ | $80.11_{\pm1.22}$ | $93.32_{\pm0.97}$ | $90.97_{\pm1.69}$ |

For models that did not initially benefit from the direct transfer of CIF hyperparameters optimized for HM-BiTCN, we conducted further fine-tuning of the key CIF parameters. The results demonstrate that, after targeted adjustment, these models also achieved notable performance improvements. This process not only reinforces the adaptability of the CIF architecture across diverse models but also highlights the high tunability and flexibility of its hyperparameters. By appropriately modifying the number of selected channels, fusion direction, and scaling factors, as shown in Tables 13, 14, 15, 16 and Figures 11, 12, 13, 14, CIF can be effectively tailored to different network structures, thereby maximizing its strengths in feature modeling and discriminative capability.

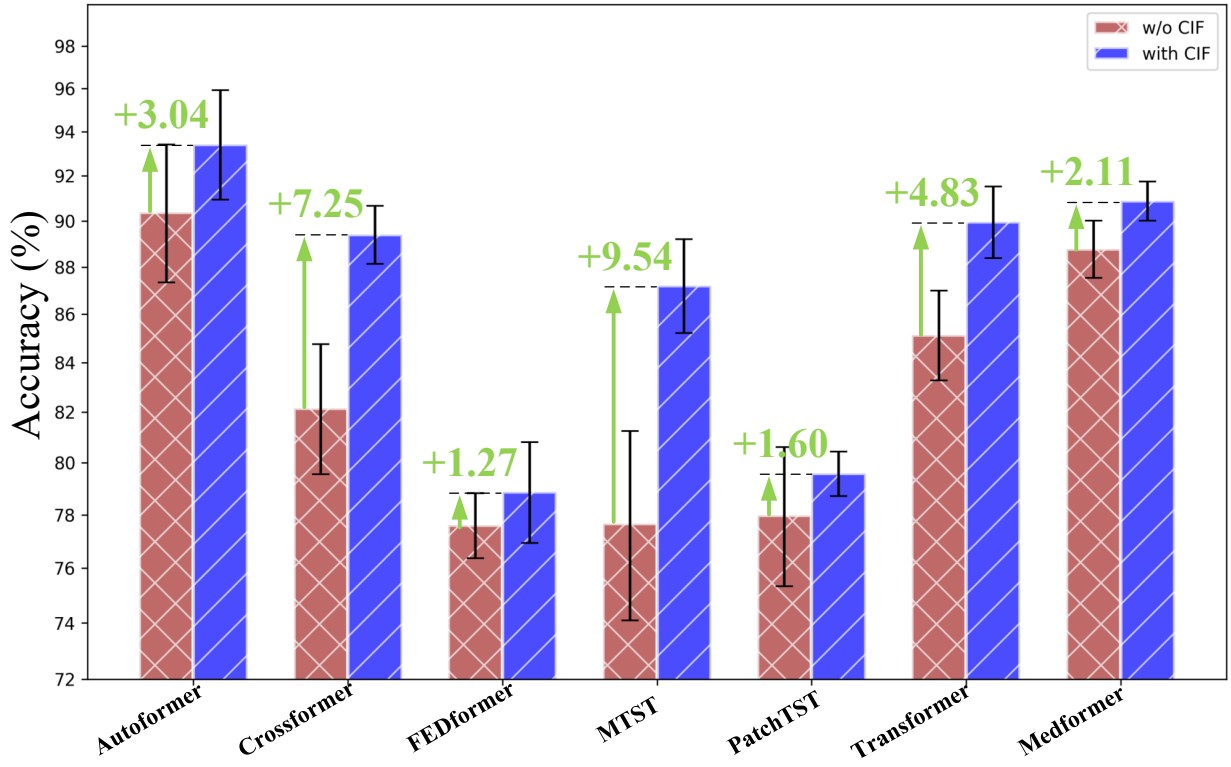

*Figure 11.* Performance improvement on the APAVA dataset using the CIF method.

*Figure 12.* Performance improvement on the TDBRAIN dataset using the CIF method.

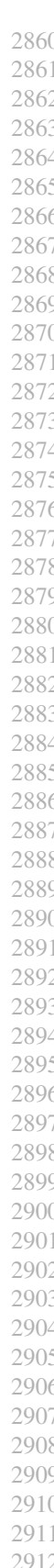

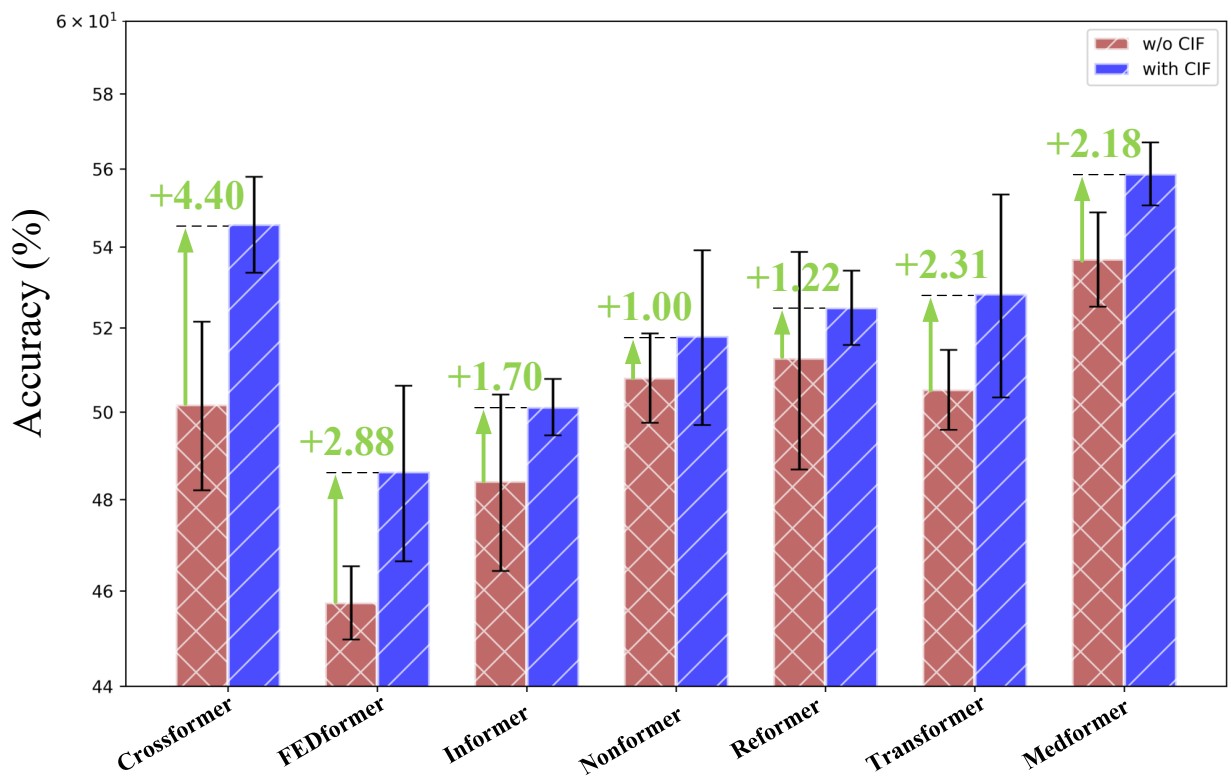

*Figure 13.* Performance improvement on the PTB dataset using the CIF method.

*Figure 14.* Performance improvement on the ADFTD dataset using the CIF method.

## K. Limitations and Future Work

**Limitations:**

Biomedical time series exhibit complex modal characteristics, which lead to significant efficiency bottlenecks when manually adjusting the prior parameters (t, n, a, b) with clear medical interpretations in the CIF model based on empirical experience. This limitation highlights the urgent need for developing novel automated hyperparameter optimization frameworks.

**Future Work:**

We plan to explore a more universal and generalizable time-series analysis approach, incorporating domain knowledge, structural modeling, and automated hyperparameter optimization. This integration should foster both deeper theoretical insights and stronger practical applicability, providing robust solutions for real-world medical problems.

Furthermore, incorporating domain-specific prior knowledge into medical time-series analysis can more precisely reveal and model relationships between channels. By integrating medical expertise, clinical experience, and existing pathological data, the interpretability and predictive performance of models can be enhanced, thereby supporting clinical decision-making and interventions. On this basis, frequency-domain analysis (Hu et al., 2025; Nason & Sachs, 1999; Yi et al., 2025) offers an additional perspective: by applying Fourier transform or wavelet decomposition to the signals, physiological features at different frequency components can be identified, revealing patterns that are difficult to capture in the time domain. This is particularly valuable for noise reduction, extraction of periodic signals, and detection of pathological events, and can also provide richer feature representations for model inputs. Future research could further explore how to combine time-domain and frequency-domain information, integrating domain priors to improve the accuracy and robustness of intelligent medical analytics.

Finally, we must acknowledge that the development trends in the field of artificial intelligence highlight the importance of architectural innovation. Future research should focus on designing novel architectures that align more closely with the CIF method, combining the strengths of existing models. For example, the local feature extraction capabilities of CNNs (LeCun et al., 1989), the temporal stability of TCNs (Bai et al., 2018) for long sequences, the long-term dependency modeling of RNNs (Rumelhart et al., 1986) and LSTMs (Hochreiter & Schmidhuber, 1997), the global modeling efficiency of Transformers (Vaswani et al., 2017), the resource-efficient computation of Mamba (Gu & Dao, 2023), and the hybrid recurrence-attention structure of RWKV (Peng et al., 2023). By adapting and integrating these methods, we aim to build a powerful model that is not only deeply compatible with the CIF framework but also capable of efficiently handling complex medical time-series data.

## L. Discussion

We sincerely appreciate the readers for reviewing this section. To address potential concerns, we provide concise responses to a few key questions:

**Q1: What did we do?**
We approached medical time series classification from a *data-centric* perspective, leveraging the intrinsic physiological characteristics of medical signals and inspired by the idea that the signal-to-noise ratio can ideally be improved through linear combinations across channels. Based on this, we proposed the CIF data processing method. Combined with a simple TCN architecture, our approach outperforms multiple heavily modified advanced models, demonstrating the clear advantage of a data-driven paradigm in medical time series tasks. Importantly, our method is also applicable to general time series classification, highlighting its broad applicability and transferability. Moreover, our data-centric strategy stands in sharp contrast to the mainstream model-centric approaches.

**Q2: What are the key innovations of this work?**
The main innovation lies in adopting a *data-centric* design philosophy that leverages the intrinsic physiological characteristics of medical time series. We introduce *Channel-Imposed Fusion (CIF)* and a simplified HM-BiTCN architecture, enabling the model to fully exploit physiological priors without relying on complex model structures. Moreover, our data-centric strategy stands in sharp contrast to the mainstream model-centric approaches.

**Q3: What new insights does this work provide?**
Our study emphasizes that physiological relationships among channels in medical time series are critically important. Structured priors derived from these relationships can significantly enhance feature representations. We also show that

merely increasing model complexity is insufficient to address the low signal-to-noise ratio (SNR) inherent in medical time series. A data-centric strategy provides a more robust solution and offers new perspectives for future research. In our experiments, we explored several ways to integrate physiological information, but aside from CIF (via CCA), most approaches were simple additive combinations. Future work could investigate more sophisticated data-level fusion strategies to further enhance performance.

**Q4: Why did we adopt this approach?**

Current mainstream research often focuses on designing highly modified architectures that compete independently, but they fail to address the fundamental nature of medical time series. Low SNR is an intrinsic property of the data, which cannot be fundamentally mitigated through model design alone. Therefore, we take a data-driven approach that combines physiological priors with raw signals to optimize feature representations, improving both performance and robustness at a fundamental level.

Specifically, our approach effectively utilizes two types of redundancy present in medical time series data: First, **inter-channel redundancy**, where different electrodes or leads capture the same physiological activity and hence exhibit overlapping information, which is addressed using the CIF method; second, **temporal redundancy**, where adjacent time points have highly correlated signals, which is handled using dilated convolutions within the TCN structure. By exploiting these redundancies across both channels and time, we enhance the effective signal and suppress noise, thus improving performance while increasing the robustness of the model.

**Additional clarification:**

It is important to note that CIF is inspired by the theoretical observation that, under ideal conditions, the linear combination of two channels can enhance SNR. However, given the complexity of medical time series, we *do not claim* that CIF reliably improves SNR in real-world scenarios.

Since CIF is specifically designed for medical time series tasks, it is not possible to directly demonstrate an improvement in SNR. However, in tasks explicitly focused on SNR, CIF may offer a clearer theoretical explanation and more interpretable effects.

We plan to explore a more universal and generalizable time-series analysis approach, incorporating domain knowledge, structural modeling, and automated hyperparameter optimization. This integration should foster both deeper theoretical insights and stronger practical applicability, providing robust solutions for real-world medical problems.

We strongly believe that simple yet effective methods can inspire the community to think more deeply about underlying principles. While our approach may be subject to debate, we view the process of exploration itself as inherently valuable—a necessary step toward meaningful progress.

