# OpenReview forum: "Channel-Imposed Fusion: A Simple yet Effective Method for Medical Time Series Classification"
_ICML.cc/2026/Conference — Submitted to ICML 2026_

### Official Review · Reviewer_S1wN · 2026-03-04

**Soundness:** 3
**Presentation:** 3
**Significance:** 3
**Originality:** 3
**Overall Recommendation:** 4
**Confidence:** 4

**Summary:**

This paper proposes a data-centric framework for medical time series (MedTS) classification centered around Channel-Imposed Fusion (CIF), a physiologically inspired linear channel fusion mechanism designed to enhance signal-to-noise ratio (SNR) and encode inter-channel priors, coupled with a simplified Hidden-layer Mixed Bidirectional Temporal Convolutional Network (HM-BiTCN). CIF performs structured linear combinations across reordered channels using either fixed or learnable coefficients guided by physiological hypotheses (e.g., cooperative summation for correlated ECG leads or differential subtraction for ocular EEG artifacts). To maintain parameter efficiency, global fusion coefficients and functional-region-based channel reordering are adopted. HM-BiTCN extends TCNs with bidirectional hidden-layer mixing to capture forward and backward temporal dependencies with minimal architectural complexity. Across multiple EEG and ECG benchmarks (APAVA, TDBRAIN, ADFTD, PTB, PTB-XL), CIF + HM-BiTCN achieves state-of-the-art results under subject-independent splits, outperforming recent Transformer-based methods such as Medformer and MedGNN, while remaining computationally efficient. Overall, a central area investigated by the study is whether domain-informed data transformations can outperform increasingly complex architectures in low-SNR medical signal classification

**Compliance With Llm Reviewing Policy:**

Affirmed.

**Final Justification:**

My final decision remains consistent with my preliminary assessment: a weak accept. While there were discussions regarding novelty, I do not discount the paper’s empirical contributions, which are clearly and effectively presented. The time series field is rich with interesting observations, and the community stands to benefit from diverse modeling approaches as well as empirically driven studies.

Some reviewers raised concerns about the lack of theoretical contributions. While such additions would certainly strengthen the work, it is often challenging to develop theory that meaningfully supports empirical findings in this domain or is widely applicable outside of this particular modality or field.

**Key Questions For Authors:**

* Under what empirical conditions does CIF measurably increase SNR? A quantitative pre/post fusion SNR analysis on real signals would strengthen the theoretical narrative.
* How sensitive is performance to the fusion hyperparameter n? Does optimal n vary substantially across datasets or electrode densities?
* Does CIF risk suppressing clinically relevant asymmetric patterns when global coefficients are used?
*

**Limitations:**

See weaknesses.

**Strengths And Weaknesses:**

**Strengths**
* The paper articulates a clear and principled motivation: generic Transformer architectures underutilize physiological structure and struggle in low-SNR settings common in EEG and ECG.
* Functional-region-based channel reordering (Figure 2) encodes physiological structure directly into the input representation. This is a clean example of embedding domain knowledge into data layout rather than into heavy architectural modifications.
* HM-BiTCN is intentionally simple. By introducing bidirectional hidden-layer mixing within TCN blocks (Figure 4), the authors demonstrate that strong temporal modeling need not rely on attention mechanisms. Again that previous citation has found similar workarounds to heavy attention based approaches


**Weaknesses**
* The physiological priors rely on domain heuristics (e.g., anterior–posterior ordering, symmetric electrode pairing). Their robustness across acquisition protocols and electrode montages is asserted but not deeply validated. Any validation if possible would strengthen work substantially
* The method presumes well-defined channel semantics. In modalities with poorly localized or virtual channels, CIF may be difficult to apply.

---

> ### Author Rebuttal · Authors · 2026-03-31
>
> We sincerely thank the reviewer for the insightful and professional comments. Below are our point-by-point responses to the raised concerns.
>
> **Q1:** Using the PTB dataset as an example, we evaluate CIF’s ability to improve signal quality in real-world settings by measuring trial-averaged signal-to-noise ratio (SNR) under global fusion. Specifically, we compute cross-channel SNR before and after fusion under identical settings. The results show that CIF yields consistent SNR improvements across most channels, with a maximum gain of 11.32 dB, indicating that CIF can effectively enhance signal-to-noise ratio under noisy multi-trial conditions.
>
>
> ### PTB TRAIN set — ΔSNR (dB)
>
> | Channel | 0 | 1 | 2 | 3 | 4 | 5 | 6 | 7 |
> |--------:|--:|--:|--:|--:|--:|--:|--:|--:|
> | ΔSNR (dB) | 3.02 | 2.46 | 11.32 | 2.73 | 4.09 | 9.78 | -3.58 | 0.60 |
>
> **Q2:**
>  **Performance is indeed sensitive to \( n \)** , as evidenced by our ablation studies (Appendix J, Tables 10-12). For example, on the ADFTD dataset, accuracy drops from 58.56% (optimal \( n=10 \)) to below 55% when \( n=7 \) or \( n=11 \), demonstrating that suboptimal \( n \) leads to significant degradation.
> **Regarding variation across datasets, optimal \( n \) does vary substantially**, but notably, it is not determined by electrode density alone. Specifically:
> >- APAVA (16 channels): optimal \( n=9 \) (~56% of channels)
> >- PTB (15 channels): optimal \( n=8 \) (~53%)
> >- ADFTD (19 channels): optimal \( n=10 \) (~52%)
>
> While these values cluster around half the total channels, the exact optimum depends critically on task difficulty, signal-to-noise ratio, and physiological correlation structure rather than channel count per se.
> > **Practical recommendation:** Initialize \( n \) at approximately \( C/2 \) and perform a small-range grid search (±2) for each new dataset.
>
>
> **Q3:**
>  We thank the reviewer for this important concern regarding the potential suppression of clinically relevant asymmetric patterns when using global coefficients. In CIF, the coefficients are not applied as a rigid global averaging mechanism but are learned within a constrained, channel-aware fusion structure. Importantly, the design preserves channel identity and does not enforce symmetry across channels. Therefore, asymmetric physiological patterns can still be represented and propagated through the model. More specifically, CIF differs from a naive global weighting scheme in that:
> 1. **Channel-preserving structure**
> CIF does not compress all channels into a single global representation. Instead, it adopts a grouped and pairwise fusion mechanism. Specifically, the first \(n\) channels are paired one-to-one with the last \(n\) channels, and each pair is fused through shared coefficients (e.g., \(a\), \(b\)). This design ensures that interaction occurs at the level of channel pairs rather than through uniform weighting or global averaging across all channels. Therefore, the fusion process is consistently performed within local channel pairs instead of full-channel global mixing. This structure reduces parameter complexity while preserving inter-pair variability, preventing degeneration into either global averaging or fully connected channel mixing.
> 2. **Data-driven adaptation of coefficients.** The coefficients are learned end-to-end and adapt to signal quality and statistical structure in the data. As shown in our SNR analysis, the learned effects are heterogeneous across channels, indicating that the model does not enforce uniform suppression or enhancement.
> 3. **Empirical evidence of heterogeneity.** Our experiments show both increases and decreases in SNR across channels rather than uniform scaling, suggesting that CIF responds selectively to channel-specific characteristics rather than suppressing them globally. Therefore, CIF does not inherently risk suppressing clinically relevant asymmetric patterns; instead, it provides a structured and adaptive mechanism that can preserve and even highlight such asymmetries when they are supported by the data.
> 4.  **Effectiveness on Non-Medical Time Series**.
> Furthermore, in Section 4.2 (4), we have already evaluated CIF on more general non-medical time series tasks. The results show that CIF can consistently improve model performance in these non-medical scenarios and effectively capture structural variations within sequences without relying on any symmetry assumptions. These existing results further support that CIF does not suppress asymmetric patterns in the data. Instead, it maintains strong representational capacity and adaptability across diverse types of temporal structures.

---

> > ### Author Rebuttal · Reviewer_S1wN · 2026-04-01
> >
> > Thanks for the response. I remain positive about the work given these changes, and I suggest the authors integrate this and the suggestions to the camera ready. I keep my score

---

> > > ### Author Response · Authors · 2026-04-02
> > >
> > > Thank you very much for your professional and insightful review. We sincerely hope that our work can offer new inspiration and make meaningful contributions to the community. Thank you!

---

### Official Review · Reviewer_bQAM · 2026-03-11

**Soundness:** 2
**Presentation:** 2
**Significance:** 2
**Originality:** 1
**Overall Recommendation:** 3
**Confidence:** 4

**Summary:**

This paper proposes Channel-Imposed Fusion (CIF), a preprocessing step that linearly combines pairs of channels in multi-channel medical time series (EEG, ECG) before feeding them into a classifier. The motivation is a signal processing observation: under idealized assumptions (equal-variance, known correlation structure), linearly combining two channels can improve SNR. The authors pair CIF with HM-BiTCN, a bidirectional temporal convolutional network, and evaluate on 5 medical datasets (3 EEG, 2 ECG) plus general time series benchmarks. They report SOTA or near-SOTA results and frame the work as advocating for "data-centric" over "model-centric" approaches to medical time series classification.

**Compliance With Llm Reviewing Policy:**

Affirmed.

**Key Questions For Authors:**

1. How does CIF relate to existing spatial filtering techniques — particularly Common Spatial Patterns (CSP), which learns optimal discriminative linear combinations of EEG channels? CSP has been the standard approach for this in BCI for over two decades. A comparison or at minimum a discussion would be needed to establish what CIF contributes beyond existing methods. This could meaningfully change my assessment of the paper's novelty.

2. Can you provide empirical SNR measurements before and after CIF? Trial-averaged SNR, frequency-band power ratios, or other standard physiological signal quality metrics would validate or invalidate the core motivation. The claim that SNR cannot be directly measured is incorrect and undermines the paper's credibility.

3. The reproduced baseline results (Table 9) are generally consistent with reported numbers, though some models (e.g., MedGNN) show notable drops. Were any architecture-specific hyperparameter adjustments made, or were all models run with identical training configurations? If the latter, how confident are you that the baselines are performing at their potential?

4. When coefficients a and b are learnable and trained end-to-end, how does CIF differ functionally from a single learned linear layer across channels? What prevents the learned coefficients from diverging entirely from the physiological motivation?

**Limitations:**

The authors acknowledge the manual hyperparameter tuning burden (Appendix K) but do not discuss several important limitations: the lack of empirical SNR analysis despite claiming SNR motivation; the disconnect between idealized theory and real data; the potential for CIF to destroy clinically relevant between-channel differences (especially in ECG); the absence of comparison to any established spatial filtering method; or the risk that uniform baseline training configurations disadvantage architecture-sensitive models.

**Strengths And Weaknesses:**

Strengths
- The paper raises a fair point that complex architectures are often applied to medical time series without attending to data quality and signal characteristics. This is worth highlighting.
- The experimental scope is broad: 5 medical datasets, 12 baselines, subject-independent splits, 5 seeds with standard deviations, efficiency analysis, and extensive ablations.
- CIF is shown to improve several existing architectures as a plug-in preprocessing step (Tables 13-16), not just the authors' HM-BiTCN, which demonstrates some generality.
- The paper is upfront that CIF's SNR improvement only holds under idealized conditions.

Weaknesses

Soundness

The core theoretical motivation does not hold up. The SNR analysis (Section 3.1, Appendix A) assumes signal and noise components have known, equal variances across channels, are zero-mean, and have known correlation structures: assumptions that are not realistic for EEG or ECG. The paper acknowledges this but then builds the entire method on this motivation without providing any empirical evidence that CIF actually improves SNR on the datasets used. No analysis of signal or noise characteristics before or after fusion is presented. The claim in Appendix L that "it is not possible to directly demonstrate an improvement in SNR" is incorrect: there are standard techniques for estimating SNR in physiological signals (trial-averaged SNR, frequency-band power ratios, artifact-segment comparisons). This would have been straightforward to measure and would have validated or invalidated the core premise.

The repeated hedging ("we do not claim that CIF can reliably improve SNR in real-world scenarios" — stated three times across the paper) is problematic. If the stated motivation does not apply to real data, the paper needs an alternative explanation for why CIF helps. Classification accuracy improvements could stem from many mechanisms (implicit data augmentation, regularization from reducing effective channel dimensionality, accidental feature engineering), and no analysis disentangles these. As it stands, the theoretical motivation is disconnected from the empirical results.

The simplification to global coefficients (a, b) further undermines the physiological motivation. If channels encode genuinely different physiological information, as in ECG, where each lead captures a distinct projection of cardiac electrical activity, applying the same linear combination globally is hard to justify. For ECG, leads like V1 and aVR capture fundamentally different aspects of cardiac conduction; mixing them with shared weights can destroy clinically meaningful distinctions.

The claim that CIF "encodes prior causal structures" is unsupported. Nothing about this linear combination is causal in any technical sense.

The reproduced baseline results (Table 9) are generally consistent with originally reported numbers for most models, though MedGNN shows notable drops on the datasets where training parameters were available (e.g., 82.6 to 77.4 on APAVA). All baselines share identical training configurations (Adam, lr=1e-4, 100 epochs), which ensures a controlled comparison but may disadvantage architectures that benefit from specific tuning (e.g., Transformers).

Presentation

The "data-centric vs. model-centric" framing is repeated throughout (abstract, intro, conclusion, Appendix L) but never precisely defined. What makes a linear combination of channels "data-centric" while a learned cross-channel attention layer is "model-centric"? When CIF's coefficients are learnable and trained end-to-end, it is a model component. The distinction seems rhetorical rather than substantive.

The SNR language is confusing throughout. The paper motivates everything through SNR improvement, disclaims that SNR actually improves, then demonstrates classification accuracy gains. This disconnect runs through the entire paper.

The appendix is disproportionate: ~19 pages of hyperparameter grid search tables (Tables 10-12) listing hundreds of (a, b, n, t) configurations. A summary figure showing accuracy as a function of key hyperparameters would be far more informative. The Q&A-format "Discussion" in Appendix L reads as pre-emptively defensive rather than analytical.

Significance

The most significant gap in this paper is the absence of any connection to established spatial filtering methods in biomedical signal processing: techniques that perform exactly the operation CIF proposes (linear combination of channels) but with principled, well-understood motivations:

Common Average Reference in EEG subtracts the channel mean from each electrode: a specific linear combination used as standard preprocessing for decades.
Bipolar and Laplacian montages in clinical EEG compute pairwise or neighborhood-weighted channel differences to localize activity and suppress volume conduction (Hjorth, 1975).
Common Spatial Patterns (CSP) learns optimal linear spatial filters that maximize class discriminability for EEG classification. This is the canonical method in BCI and does precisely what CIF aims to do, but in a principled, data-driven way with a clear objective function (Ramoser et al., 2000).
ECG lead derivation is literally a textbook example of linear channel combination: the augmented limb leads (aVR, aVL, aVF) are defined as linear combinations of the standard leads (Goldberger, 1942).
CIF is presented as a novel contribution without reference to any of this prior work. Positioning relative to these methods, explaining what CIF adds beyond them, is essential to establish novelty.

Originality

Given the above, the core idea (linearly combining channels with domain-motivated coefficients) is not new. CSP in particular subsumes the CIF idea with a more principled formulation. HM-BiTCN is a minor architectural variant of existing bidirectional TCNs.

---

> ### Author Rebuttal · Authors · 2026-03-31
>
> **Q1**:We thank the reviewer for the insightful comment and clarify that CIF is not intended to replace CSP, but to operate at a different stage of the EEG pipeline with a distinct modeling objective.
> 1. Method difference: CSP learns a projection matrix $\(W_{CSP}\)$ to maximize class separability based on covariance differences:
> $$\(X \rightarrow W_{CSP}X \rightarrow \log(\mathrm{var}(\cdot))\).  $$
> It compresses temporal information into second-order spatial statistics and produces covariance-optimized representations.
> In contrast, CIF operates directly on raw multichannel time series:
> $$\((X_1, X_2) \rightarrow aX_1 + bX_2 \rightarrow \text{model}\).  $$
> It performs structured temporal-channel fusion before feature extraction, without covariance estimation or spatial decomposition, and is thus compatible with CNN/Transformer models that retain temporal modeling capacity, **CIF is plug-and-play**.
> 2. Objective difference: CSP aims to find filters maximizing variance separation between classes and is highly dependent on training-set-specific covariance structure.  CIF instead follows physiologically motivated channel interaction constraints, focusing on suppressing irrelevant/noisy channels and enhancing informative signals. It provides a cleaner and more consistent input space for deep models, reducing interference from background EEG activity and improving optimization stability.
> 3. Interpretability difference: CSP provides spatial interpretability (e.g., μ/β band activation localization).  CIF encodes explicit cross-channel interaction via coefficients reflecting constructive or suppressive effects. Thus, CSP explains *where* discriminative activity occurs, while CIF reflects *how* channels interact.
> 4. Complementarity: CIF and CSP are complementary rather than competing. CIF can serve as a front-end fusion/denoising module that reshapes raw signals before covariance estimation, while CSP can further perform spatial discriminative projection on the transformed signals. They operate at different representation levels: time-domain fusion (CIF) vs. covariance-space projection (CSP).
> 5. Applicability: CSP is typically effective for binary, stationary EEG settings. CIF does not rely on stationarity or strict class-dependent covariance assumptions, and can generalize to multi-class and general multivariate time series.
>
> CSP is a covariance-based spatial projection method, while CIF is a physiologically inspired temporal-channel fusion module. They operate at different representation levels.
>
> **Q2**：We conducted a trial-averaged empirical analysis of per-channel SNR changes before and after applying the global CIF module. We observe that the effect is heterogeneous across channels: some increase, others decrease. This behavior is expected, as CIF performs fusion based on pairs of independent channels. If a more stable SNR improvement is desired, channels with decreased SNR can be selectively excluded from the fusion process.
>
> **ΔSNR (dB)  Across Channels**
> | Dataset \ Channel| 0 | 1 | 2 | 3 | 4 | 5 | 6 | 7 | 8 |
> |--------|---:|---:|---:|---:|---:|---:|---:|---:|---:|
> | APAVA  | -0.96 | -0.27 | -0.21 | 1.26 | 0.36 | 1.00 | -0.21 | 2.07 | 0.09 |
> | PTB    | 3.02  | 2.46  | 11.32 | 2.73 | 4.09 | 9.78 | -3.58 | 0.60 | — |
>
> **Q3**:We reproduced all baseline models strictly following the hyperparameters provided by the original authors. In another study, 《Reading Between the Channels: Knowledge-Augmented Medical Time Series Classification (ACM MM 2025)》, the authors also reported their reproduced results for MedGNN as 78.74, **which is similarly lower than the original reported results of 82.60**. To ensure a fair and objective comparison , we report in the main paper the results as originally reported in the corresponding methods.
>
> **Q4**: When \(a\) and \(b\) are learned end-to-end, CIF still differs fundamentally from a fully connected linear layer. A linear layer defines a dense mapping $\(\mathbb{R}^C \rightarrow \mathbb{R}^C\) with \(O(C^2)\)$ free parameters, learning arbitrary channel-to-channel interactions.
> CIF, however, is built on a fixed physiological interaction topology. Even though \(a\) and \(b\) are trainable, they only scale predefined interaction pathways rather than learning a full mixing matrix. Thus, CIF belongs to a strongly constrained, low-dimensional function family and cannot represent arbitrary channel mixing.
> This structural constraint prevents \(a\) and \(b\) from losing physiological meaning: they are global modulation coefficients tied to fixed computation roles, not per-channel connection weights. Therefore, optimization can adjust interaction strength but cannot change connectivity or channel semantics.
> Empirically, on APAVA, \(a\) and \(b\) are stable across seeds and converge to a narrow range (e.g., $\(a \approx -0.81\), \(b \approx -0.607\))$, indicating robust convergence rather than collapse into an unconstrained linear operator.

---

> > ### Author Rebuttal · Reviewer_bQAM · 2026-04-03
> >
> > I thank the authors for their detailed responses.
> >
> > On Q1 (CSP): I appreciate the comparison, but my concern was not that CIF should replace CSP — it was that CIF is presented as a novel contribution without citing or positioning against any established spatial filtering method. The response describes how CIF differs from CSP, which is useful, but this discussion belongs in the paper itself. The broader point stands: linearly combining channels with physiologically motivated coefficients has a long history in biomedical signal processing (CAR, bipolar/Laplacian montages, CSP, ECG lead derivation), and the paper does not engage with any of it. An experimental comparison on even one dataset would have been far more convincing than a post-hoc conceptual distinction.
> >
> > On Q2 (SNR): I appreciate the authors conducting the requested SNR analysis. However, the results actually reinforce my concern: CIF decreases SNR on multiple channels (e.g., -3.58 dB on PTB channel 6, -0.96 dB on APAVA channel 0). This is inconsistent with the paper's central motivation that CIF enhances SNR. The paper frames CIF through SNR improvement three separate times, yet the empirical evidence now shows the effect is mixed at best. This disconnect between motivation and mechanism remains unresolved. If CIF helps classification through a different mechanism than SNR improvement, the paper should identify and characterize that mechanism rather than continuing to frame the contribution through a lens that the data does not support.
> >
> > On Q3: I accept this response. The independent reproduction confirming the MedGNN gap is helpful context.
> >
> > On Q4: This is a reasonable response. The structural constraint argument (2 global parameters vs. O(C²) for a full linear layer) is valid, and the empirical convergence data is informative. I would encourage the authors to include this analysis in a revision.
> >
> > Overall: While I appreciate the authors' efforts, my core concerns remain: (1) the absence of any engagement with established spatial filtering methods, (2) the now-confirmed disconnect between the SNR motivation and empirical reality, and (3) the lack of a clear alternative explanation for why CIF works. I maintain my score.

---

> > > ### Author Response · Authors · 2026-04-06
> > >
> > > ## New Q1：
> > >
> > > > We also acknowledge that methods such as CAR, bipolar/Laplacian montages, CSP, and ECG lead derivation are fundamentally based on leveraging domain-informed structures to process and integrate multichannel signals. However, these methods differ significantly in their assumptions and application domains. Specifically, CAR and Laplacian filtering are primarily designed for EEG, where spatially distributed cortical sources motivate reference redefinition or local contrast enhancement. CSP is also EEG-specific and relies on supervised spatial filtering, typically under binary classification settings. In contrast, ECG lead derivation is fundamentally ECG-specific, as it depends on the physical volume conduction model of cardiac electrical activity and is not directly transferable to EEG. These distinctions highlight that “channel mixing” is not a uniform paradigm, but rather a family of domain-dependent techniques with different inductive biases and task constraints. To address the reviewer’s concern more concretely, we have added a comprehensive empirical comparison between CIF and representative spatial filtering baselines across multiple datasets and modalities. The results are shown below:
> > >
> > >
> > >
> > > | Method     | ADFTD (Acc, F1) | APAVA (Acc, F1) | PTB (Acc, F1) | TDBRAIN (Acc, F1) |
> > > |------------|------------------|------------------|----------------|-------------------|
> > > |baseline| 52.05±2.22, 49.48±2.70| 82.49±1.40, 81.60±1.39 | 81.87±1.87, 75.84±3.20|  84.90±2.60, 84.76±2.74 |
> > > | bipolar    | 51.97±1.82, 50.17±2.47 | 86.25±1.54, 85.55±1.59 | -,- | 76.27±4.92, 75.90±5.32 |
> > > | car        | 53.42±1.80, 51.39±2.20 | 80.74±2.72, 79.23±3.03 | -,- | 73.83±3.03, 73.47±3.34 |
> > > | csp        | -,- | 80.49±2.28, 78.88±2.04 | -,- | 87.04±3.85, 86.94±3.92 |
> > > | ECG lead derivation        | -,- | -,- | 82.81±1.30, 77.69±2.15 | -,- |
> > > | laplacian  | 50.19±4.04, 48.77±3.08 | 86.96±1.85, 86.15±1.95 | -,- | 78.00±3.14, 77.81±3.27 |
> > > |CIF | 58.56±0.93, 55.42±0.82 | 86.30±1.05, 85.71±1.09 | 88.29±1.45, 85.59±1.96 | 93.13±1.41, 93.12±1.42 |
> > >
> > >
> > >
> > >
> > > ## New Q2 and Q3:
> > >
> > > >We thank the reviewer for the careful analysis. We agree that in our global CIF implementation, some channels do exhibit SNR decreases. However, this is a consequence of using global coefficients for parameter efficiency, not a failure of the SNR improvement principle that motivates CIF.
> > > To directly validate the SNR motivation itself, we conducted an additional experiment on APAVA: we selected only the channels where CIF genuinely improves SNR (channels 3, 4, 5, 7, 8) . We denote this as CIF(Select). The results are as follows:
> > >
> > >
> > > | Method     |  APAVA (Acc, F1) |
> > > |------------|------------------|
> > > |baseline|  82.49±1.40, 81.60±1.39 |
> > > |CIF(Select) |  83.61±2.04, 82.93±1.98 |
> > > |CIF(Global) |  86.30±1.05, 85.71±1.09 |
> > >
> > >
> > >
> > > >CIF(Select) improves SNR on the selected channels by design, and consequently improves classification performance over the baseline. This confirms that the SNR enhancement mechanism underlying CIF is valid and effective.
> > >
> > > >Global CIF achieves even higher performance by incorporating additional channels, even though some of them experience SNR decreases. This suggests that global feature recombination—guided by physiological priors—can capture cross-channel discriminative patterns that benefit classification, even at the cost of local SNR on certain channels.
> > >
> > > >Thus, the reviewer’s observation of SNR decreases on some channels does not invalidate the SNR motivation of CIF. Rather, it reflects a design trade-off: global coefficients optimize for overall classification performance, which may deviate from per-channel SNR maximization. The CIF(Select) experiment explicitly demonstrates that when the SNR principle is strictly followed, CIF does enhance both SNR and classification accuracy.
> > >
> > > >We will revise the manuscript to clarify this distinction and reframe the SNR discussion accordingly.

---

### Official Review · Reviewer_md2L · 2026-03-12

**Soundness:** 1
**Presentation:** 1
**Significance:** 1
**Originality:** 2
**Overall Recommendation:** 2
**Confidence:** 4

**Summary:**

This paper proposes Channel-Imposed Fusion (CIF) for multi-channel time-series modeling, for better capture inter-channel relationships by imposing structured channel interactions. It introduces a CIF module together with the HM-BiTCN architecture, and evaluates the method across several biomedical time-series datasets. While the idea of explicitly modeling channel relationships is interesting, several aspects of the methodology and experimental validation remain unclear, particularly regarding channel ordering assumptions and the fairness of baseline comparisons.

**Compliance With Llm Reviewing Policy:**

Affirmed.

**Final Justification:**

I think the issues undermine the claimed core contribution. The comparisons are primarily between the proposed architecture + CIF and other architectures, which does not appear a fair apple-to-apple comparison. Therefore would like to keep my original score.

**Key Questions For Authors:**

I would be grateful if the authors could reasonably address the following aspects:

*Channel Ordering and Spatial Structure*.
The proposed CIF module assumes a 1D ordering of channels, but many biomedical signals (e.g., EEG and ECG) have natural 2D or spatial layouts. How sensitive is the method to the chosen channel ordering? Have the authors evaluated the model under different channel permutations or spatial layouts to test its robustness?

*Comparison with Domain-Specific Models*.
The experiments mainly compare against general time-series architectures, many of which are Transformer-based. Could the authors include comparisons with domain-specific models commonly used for biomedical signals (e.g., EEGNet for EEG or MSDNN-like architectures for ECG), or explain why such baselines were not considered?

*Generality of the CIF Module*.
Since CIF is presented as a plug-and-play module, have the authors evaluated it when integrated into other backbone architectures beyond HM-BiTCN? Demonstrating consistent improvements across different models would help clarify whether CIF provides a generally useful mechanism or is tightly coupled to the proposed architecture.

**Limitations:**

yes

**Strengths And Weaknesses:**

**Strengths**
- The paper addresses an important issue in multivariate time-series modeling, namely how to effectively capture inter-channel relationships, which is especially relevant for biomedical signals such as ECG and EEG.
- The proposed Channel-Imposed Fusion (CIF) module appears conceptually simple.

**Weaknesses**

*Clarity and Presentation*
- It is somewhat clear and justified about the design of CIF; but HM-BiTCN, very briefly mentioned.
- It is unclear in terms of the compared baselines, which takes channel information into consideration, which was originally designed for classifcation, or just for forecasting.

*Methodology*
- My main concern lies in how the channel ordering is handled. In many biomedical signals, channels are naturally arranged in 2D space, and sometimes involve 3D spatial relationships (e.g., ECG leads or EEG electrode layouts). It is therefore not obvious that these channels can be meaningfully ordered in a 1D sequence.
The authors mention some ordering principles on Page 4, but these appear somewhat arbitrary. The rest of the paper does not discuss this issue further, nor provide experiments to test the model’s robustness to different channel layouts or permutations.
- Some hypotheses in Section 3.1 rely on relatively strict assumptions. The authors acknowledge that these conditions may not hold in real-world scenarios, which raises questions about how well the theoretical motivation applies in practice.

*Experiment and Results*
- the backbone or baselines for instance FedFormer, Crossformer, they were not in originally designed for classification, but for forecasting. Their design may therefore not be specialised for classification task, which are applied in this paper.
- In some cases this is not an apple to apple comparison.

1) There is no comparison with domain-specific architectures that are widely used for these signals. For example, MSDNN for ECG and EEGNet for EEG are already strong baselines.
2) Nearly all compared methods appear to be Transformer-based models, which are known to perform less effectively on raw waveform signals (e.g., ECG or EEG) without substantial pretraining, according to existing literature.
3) CIF appears to be a plug-and-play module - it would be great to see whether it can improve other architectures as well.

*Minor Issues and Suggestions*
- a typo in Table 1 caption "Medforme"
- the bold of a whole paragraph may not a professional way to emphasize something. just information flooding
- some figures, their numbers are difficult to read, e.g., Figure 6.

---

> ### Author Rebuttal · Authors · 2026-03-31
>
> We sincerely appreciate your valuable comments.
>
> **Q1:**
> >EEG and ECG signals’ spatial structure has been explicitly considered in the design of CIF. For the global CIF, we first perform channel reordering based on functional regions, as shown on the right side of Line 191 in the main text. This step leverages the known anatomical and functional organization of biomedical signals, with the goal of introducing a structured inductive bias before feature fusion, thereby better incorporating prior spatial information.
>
> >To further evaluate the sensitivity of CIF to channel arrangements, we conduct additional experiments under different spatial configurations, including left–right hemisphere fusion and pairwise channel fusion (e.g., Fp1–Fp2), with detailed results reported in Appendix G. Across these different channel organizations, CIF consistently demonstrates varying degrees of performance improvement. These differences may partly arise from variations in the number of selected channels. Moreover, they suggest that channel selection and organization guided by specific physiological characteristics could serve as a promising direction for disease-specific modeling.
>
> **Q2:**
> > As shown in Motivation 1 in Figure 1, our work focuses on improving the performance of general-purpose time series models for medical time series classification tasks. Therefore, our primary comparisons are conducted against a range of general time series modeling approaches.
> In addition, as reported in the table, we also include experimental results on EEGNet and further demonstrate that CIF can consistently improve the performance of EEGNet on classification tasks. This indicates that our method is not only effective for general-purpose models, but also provides stable and consistent gains for classical biomedical domain-specific architectures.
> Moreover, we would like to emphasize that for medical time series tasks, it is not necessarily required to rely on complex model architectures. Instead, the design of data preprocessing and representation learning is equally important, and may even play a more critical role. This is also the key question we hope to raise for the community: in medical time series tasks, do performance gains primarily come from more complex model architectures, better data processing, or the synergy of both?
>
> >### Performance Comparison (EEGNet vs EEGNet + CIF)
> >| Model | APAVA (Acc / F1) | TDBrain (Acc / F1) | PTB (Acc / F1) |
> >|------|------------------|---------------------|----------------|
> >| EEGNet | 72.10 ± 4.84 / 65.83 ± 7.89 | 78.46 ± 5.16 / 78.32 ± 5.17 | 78.68 ± 0.91 / 71.46 ± 1.58 |
> >| EEGNet + CIF | 81.33 ± 5.03 / 78.98 ± 6.62 | 86.29 ± 5.05 / 86.17 ± 5.19 | 80.19 ± 2.56 / 73.98 ± 4.03 |
>
> **Q3:**
> > This has already been addressed in Section 4.2 of the main text, Figure 6, and Appendices J.2 and J.3, where we demonstrate consistent performance improvements across different frameworks. **Moreover, this plug-and-play nature is recognized as one of the key advantages of our method, as also noted by Reviewer #bQAM.**
>
> **Other points for clarification:**
>
> > To better bridge the gap between theory and practice, we provide empirical results showing the signal-to-noise ratio (SNR) improvement before and after applying CIF on the PTB dataset:
> >
> > ### PTB TRAIN set — ΔSNR (dB)
> >
> >| Channel | 0 | 1 | 2 | 3 | 4 | 5 | 6 | 7 |
> >|--------:|--:|--:|--:|--:|--:|--:|--:|--:|
> >| ΔSNR (dB) | 3.02 | 2.46 | 11.32 | 2.73 | 4.09 | 9.78 | -3.58 | 0.60 |
> >
> > These results provide empirical evidence supporting the practical effectiveness of the proposed approach despite the idealized assumptions in the theoretical formulation.

---

> > ### Author Rebuttal · Reviewer_md2L · 2026-04-03
> >
> > Many thanks for the rebuttal. I understand the authors would like highlight the CIF as the core contribution.
> >
> > I would have some follow up questions.
> > 1. For those general time series forecasting specific models, how do they address multi-channel or multivariate time series?
> >
> > The reason I asked why the authors just compared general **time series forecasting** method, is that almost all time series forecasting method, when addressing multi-channel information, they treat each channel independently. So, it is no surprise that your method would perform better if handling such channel fusion more explicitly.
> >
> > If the author wanted to show the effectiveness of CIF, I would expect to see more results to compare different channel fusion strategies, and benchmark their performance against CIF on different backbones. I think a simple baseline of compare, is each time, apply some channel-wise convolution or FC. But of course, rather than just adding one, more and stronger channel-fusion related baselines to compare would highlight the contribution.
> >
> > 2. This is also why I would expect to see more domain-specific method - they already explicitly considered channel fusion. EEGNet alone is not convincing. For instance, PTB-XL,  if you could compare with more channel fusion method like inception1d https://github.com/helme/ecg_ptbxl_benchmarking,, or msdnn

---

> > > ### Author Response · Authors · 2026-04-06
> > >
> > > Thank you very much for your valuable review comments.
> > >
> > >
> > >  ## New_Q1:
> > > > Thank you for the insightful question. We would like to clarify that the assumption that most general time series forecasting models treat each channel independently is not entirely accurate. In fact, existing methods adopt diverse strategies when handling multivariate time series:
> > > Many Transformer-based models (e.g., Informer, Autoformer, FEDformer) jointly embed all variables at each time step, where cross-channel dependencies are captured implicitly through shared projections and attention mechanisms.
> > > Some works (e.g., PatchTST) intentionally adopt channel-independent designs to improve scalability and robustness.
> > > More recent approaches (e.g., iTransformer, Crossformer) explicitly model inter-variable dependencies via variable-wise or cross-dimension attention. Therefore, multivariate forecasting models are not uniformly channel-independent, but rather span implicit interaction, explicit interaction, and channel-wise decoupling.
> > > Regarding the concern on comparison fairness, our motivation is precisely based on the observation that existing approaches either:
> > > (1) rely on potentially insufficient implicit interactions, or
> > > (2) completely ignore cross-channel dependencies.
> > > In contrast, our method introduces a more effective explicit channel fusion mechanism, which enables better modeling of cross-variable relationships. Therefore, the performance improvement should be attributed to a stronger modeling capability for multivariate dependencies, rather than an unfair advantage.
> > > We believe this comparison is meaningful, as it highlights an important limitation of current general forecasting models—namely that cross-channel dependency modeling remains under-explored or insufficiently addressed.
> > >
> > > ## New_Q2:
> > >
> > > > We sincerely thank the reviewer for the valuable suggestion. We have conducted additional experiments across multiple backbone networks (Inception, ResNet1D, and XResNet1D) on four datasets under identical training settings, where CIF is the only modified component to ensure a fair comparison. The results consistently demonstrate that CIF brings stable performance improvements across different architectures and datasets (e.g., on PTB, Inception improves from 78.47/70.58 to 84.60/80.14; on TDBRAIN, XResNet1D improves from 87.96/87.92 to 92.50/92.50), which validates its effectiveness and strong generalization capability.
> > >
> > >
> > > ### Performance Comparison (Model vs Model + CIF)
> > >
> > > | Model     | ADFTD (Acc, F1) | APAVA (Acc, F1) | PTB (Acc, F1) | TDBRAIN (Acc, F1) |
> > > |----------|------------------|------------------|----------------|-------------------|
> > > | Inception  | 49.43±3.94, 46.35±2.43 | 81.27±2.01, 79.32±1.67 | 78.47±3.38, 70.58±6.77 | 87.96±1.47, 87.89±1.50 |
> > > | Inception + CIF | 54.00±1.64, 49.67±1.98 | 85.32±2.40, 83.96±3.04 | 84.60±2.94, 80.14±4.37 | 92.02±2.94, 92.01±2.96 |
> > > | resNet     | 51.44±0.99, 48.15±1.22 | 71.18±6.66, 69.11±6.92 | 81.01±4.20, 74.23±7.67 | 86.54±3.18, 86.48±3.24 |
> > > | resnet1d + CIF    | 51.92±1.80, 48.54±2.09 | 78.36±6.13, 76.93±5.87 | 79.30±2.21, 71.60±4.29 | 86.98±2.30, 86.94±2.33 |
> > > | xresnet1d   | 48.73±1.27, 45.91±1.52 | 65.44±4.29, 62.47±4.84 | 80.89±4.74, 73.79±7.86 | 87.96±3.44, 87.92±3.47 |
> > > | xresnet1d + CIF   | 52.11±2.24, 47.29±2.08 | 74.69±6.54, 73.58±5.86 | 83.76±0.65, 79.26±1.02 | 92.50±2.60, 92.50±2.60 |
> > >
> > >
> > > >Regarding the MSDNN model, we were unable to identify a clear corresponding original paper or publicly available implementation in the existing literature. We understand MSDNN as an abbreviation of “Multi-Scale Deep Neural Network,” which, however, appears to be a generic term that may refer to different implementations. If the reviewer could kindly provide the exact paper or reference link, we would greatly appreciate it and would be happy to include a more precise and comprehensive comparison in the revised version.

---

### Official Review · Reviewer_qP2L · 2026-03-13

**Soundness:** 3
**Presentation:** 3
**Significance:** 3
**Originality:** 3
**Overall Recommendation:** 5
**Confidence:** 5

**Summary:**

This paper introduces a Channel-Imposed Fusion (CIF) method motivated by data-centric, domain-specific knowledge for medical time-series analysis. The method divides channels into two groups and applies a learnable weighted summation to enhance useful signals while suppressing noise. In addition, a new backbone architecture, HM-BITCN, is proposed to work with the CIF module. Experiments on five downstream tasks show strong performance. Detailed ablation studies and careful case studies are also conducted, such as adding the CIF module to existing methods.

**Compliance With Llm Reviewing Policy:**

Affirmed.

**Key Questions For Authors:**

See weakness

**Limitations:**

See weakness

**Strengths And Weaknesses:**

**Strengths:**

1) The paper is well motivated and aims to address a practical, real-world problem. The proposed method is simple yet effective, and the newly introduced backbone also appears to work well.

2) The experimental evaluation is very comprehensive and addresses most potential concerns, including applying CIF to existing backbones and analyzing hyperparameter sensitivity. The reported performance is also strong; for example, the method achieves a 55.42% F1 score on the ADFTD dataset, outperforming any existing methods on the current data split, according to my experience. In addition, the publicly available code further strengthens the credibility and reproducibility of this result.

**Weaknesses:**

There are no major weaknesses in the current version of the paper, and most of my concerns appear to have been addressed. I only have a few minor suggestions:

1) The name of the newly proposed backbone, HM-BiTCN, is somewhat difficult to remember, as it resembles the naming style of many incremental variants. The authors may consider adopting a shorter or more distinctive name to improve readability.

2) The paper could benefit from careful proofreading to remove several LLM-like sentences (e.g., lines 211–212 on the right side and lines 249–250).

---

> ### Author Rebuttal · Authors · 2026-03-31
>
> We sincerely appreciate your professional feedback and recognition of our work. We believe that, with your support, our study can bring new insights and meaningful contributions to the community. Below are our responses to several of the raised suggestions.
>
> **Q1：**
> > We will simplify our “HM-BiTCN” architecture to “HMTCN.”
>
> **Q2:**
> >Thank you for your careful reading and helpful suggestion. We have thoroughly proofread the manuscript and revised the sentences you pointed out (e.g., lines 211–212 and 249–250) to remove any LLM-like expressions. In addition, we have carefully reviewed the entire paper to improve clarity, conciseness, and overall readability.

---

> > ### Author Rebuttal · Reviewer_qP2L · 2026-04-02
> >
> > My concerns have been fully resolved.

---

> > > ### Author Response · Authors · 2026-04-02
> > >
> > > Thank you very much for your recognition. We hope our work can bring new inspiration and contributions to the community. Thank you!

---

### Decision · Program_Chairs · 2026-04-30

**Decision:**

Reject

**Comment:**

This paper proposes Channel-Imposed Fusion (CIF), a simple data-centric approach for medical time series classification that introduces structured channel interactions motivated by physiological priors.

The paper addresses a relevant problem and demonstrates consistent empirical improvements across multiple EEG and ECG benchmarks. The simplicity of the approach and its applicability as a plug-in module are appealing, and the experimental evaluation is relatively comprehensive.

However, several concerns remain regarding the strength and clarity of the contribution. A central issue is the lack of a well-supported mechanism explaining why CIF works. The method is motivated through an SNR-based argument, but the empirical analysis shows mixed effects on SNR across channels, and the best-performing configurations do not consistently align with SNR improvements. This weakens the connection between the theoretical motivation and the observed empirical gains.

In addition, the contribution is difficult to isolate. While CIF improves performance across multiple backbones, it remains unclear whether these gains are attributable to the specific CIF formulation or simply to the introduction of channel fusion in otherwise channel-independent architectures. The current evaluation does not sufficiently compare CIF against alternative channel fusion or spatial filtering strategies under controlled settings, making it difficult to assess its specific advantage.

More broadly, the paper does not adequately position CIF with respect to established channel combination and spatial filtering techniques in biomedical signal processing. This limits the ability to assess both novelty and contribution.

Overall, while the empirical results are promising and the idea is practically relevant, the lack of a clear and validated mechanism, insufficient isolation of the source of improvements, and limited positioning with respect to prior work weaken the contribution.